# (2+1)d Lattice Models and Tensor Networks for Gapped Phases with Categorical Symmetry

Kansei Inamura[1,2], Sheng-Jie Huang[1] Apoorv Tiwari[3], Sakura Schäfer-Nameki[1]

[1] *Mathematical Institute, University of Oxford,*
*Andrew Wiles Building, Woodstock Road, Oxford, OX2 6GG, UK*
[2] *Rudolf Peierls Centre for Theoretical Physics, University of Oxford,*
*Parks Road, Oxford, OX1 3PU, UK*
[3] *Center for Quantum Mathematics at IMADA, Southern Denmark University,*
*Campusvej 55, 5230 Odense, Denmark*

Gapped phases in 2+1 dimensional quantum field theories with fusion 2-categorical symmetries were recently classified and characterized using the Symmetry Topological Field Theory (SymTFT) approach [1,2]. In this paper, we provide a systematic lattice model construction for all such gapped phases. Specifically, we consider "All boson type" fusion 2-category symmetries, all of which are obtainable from 0-form symmetry groups $G$ (possibly with an 't Hooft anomaly) via generalized gauging—that is, by stacking with an $H$-symmetric TFT and gauging a subgroup $H$. The continuum classification directly informs the lattice data, such as the generalized gauging that determines the symmetry category, and the data that specifies the gapped phase. We construct commuting projector Hamiltonians and ground states applicable to any non-chiral gapped phase with such symmetries. We also describe the ground states in terms of tensor networks. In light of the length of the paper, we include a self-contained summary section presenting the main results and examples.

# 1   Introduction

Three spacetime dimensions (2+1d) are a particularly interesting setting to study gapped or topological phases, due to the existence of a rich landscape of non-trivial topological orders (TO), a full classification for which is yet to be achieved. It has recently become apparent that a symmetry-based principle can be formulated to organize this landscape of gapped phases, in particular TOs, as well as to discover several new kinds of quantum orders [1, 2] characterized in terms of fusion 2-category [3] symmetries[1], which are the most general finite internal symmetries in 2+1d. The recent classification of fusion 2-categories in [19] provides a well-developed setting to not only study examples, but develop a comprehensive structure to systematize the space of gapped phases in 2+1d.

---

[1]In Quantum Field Theories, such non-invertible categorical symmetries were subsequently studied in [4–16] (see [17, 18] for reviews on non-invertible symmetries in higher dimensions).

Recent developments in the classification of gapped phases with fusion 2-category symmetry [1, 2] and advances towards a systematic study of gapless phases [20–22] based in the Symmetry Topological Field Theory (SymTFT) [23–32] is an important step in this direction. These works are part of an on-going program, dubbed the categorical Landau paradigm [33][2] to develop an understanding of quantum matter in terms of its generalized symmetry properties. This has been successfully applied in several setups in various dimensions [12, 31, 32, 35–59].

The proposal in [33] posits the following: given a finite, categorical symmetry $\mathcal{C}$, acting on a quantum system in $d$ spacetime dimensions, the SymTFT is a $d+1$ dimensional TQFT, which has the only purpose to gauge, with flat background fields, the symmetry $\mathcal{C}$. The SymTFT has topological defects which form the Drinfeld center $\mathcal{Z}(\mathcal{C})$ of the categorical symmetry $\mathcal{C}$. The SymTFT allows separating the symmetry aspects from the dynamics in the following way: there are two $d$-dimensional boundary conditions, which have two very different purposes in the SymTFT construction:

- The symmetry boundary $\mathfrak{B}^{\text{sym}}$ is a topological boundary condition, which is specified in terms of a set of mutually local topological defects in $\mathcal{Z}(\mathcal{C})$ (forming a maximal or Lagrangian, condensable algebra). The topological defects in $\mathcal{Z}(\mathcal{C})$ that cannot end on the symmetry boundary encode the symmetry category $\mathcal{C}$.

- The physical boundary $\mathfrak{B}^{\text{phys}}$, which may or may not be topological, encodes all the dynamical information of the original quantum theory.

The key insight is to note that this setup allows a complete classification of all gapped phases as follows: fix the symmetry boundary to realize $\mathcal{C}$. Then any topological physical boundary condition of the SymTFT is in 1-1 correspondence with a gapped phase with symmetry $\mathcal{C}$.[3] Its generalized charges that are order parameters are encoded in the topological defects that can end on the physical and symmetry boundaries [12].

This SymTFT approach to gapped phases has several advantages: it is completely systematic in any dimension, and encodes all the salient features such as order parameters of gapped phases, and even organizes phase transitions (i.e. gapless phases with categorical symmetries [20–22, 35, 39, 40]). The only ingredients in the program of classifying and characterizing gapped phases, once $\mathcal{C}$ is fixed, is a construction of the Drinfeld center, and a classification of gapped boundary conditions.

Thus, a key ingredient to apply this approach to 2+1d gapped phases with categorical symmetries is the classification of gapped boundary conditions in the Drinfeld center of the

---

[2]For a review of earlier generalizations of the Landau paradigm to invertible higher-form symmetries see [34].
[3]In 1+1d, the correspondence between gapped phases and topological boundary conditions of the SymTFT was noted in [26].

fusion 2-category [1, 2, 20–22, 60, 61]. Unlike the 1+1d situation, in 2+1d given a symmetry category, there are infinitely many gapped phases with this symmetry. To parse this statement let us briefly recall the classification of symmetry categories in 2+1d.

**Categorical Landau for Fusion 2-Categories.** Our focus is on the so-called fusion 2-categories of all-boson type. These have the key property that they are related to 0-form symmetries that are finite groups $G$, i.e. $2\mathsf{Vec}_G^\omega$, possibly with a 't Hooft anomaly $[\omega] \in H^4(G, U(1))$. The SymTFT for this symmetry is the 3+1d $G$ Dijkgraaf-Witten (DW) theory with twist (topological action) $\omega$. It has a canonical, so-called Dirichlet, gapped boundary condition, on which the category of defects is $2\mathsf{Vec}_G^\omega$. We can obtain all other gapped boundaries and correspondingly a rich class of fusion 2-categories by gauging a finite group $H$ that is anomaly free, resulting in partial Neumann boundary conditions. However, the most general gauging is in fact a substantial enrichment of this construction: in 2+1d, we can stack a (possibly anomalous) $H$-symmetric TQFT on top of the Dirichlet boundary condition, and then gauge the (non-anomalous) diagonal $H$ of the combined system. This is what we will refer to as **generalized gauging**. In particular, this means that there are an infinite number of gapped BCs for any 3+1d $G$ DW theory! If the TQFT that we stack is invertible, we call this a **minimal boundary condition**, if it is not, a **non-minimal boundary condition**.[4] The minimal boundaries include all the standard Dirichlet, Neumann, and mixed boundary conditions.

For the construction of gapped phases, we choose the symmetry boundary and the physical boundary as follows:

- Symmetry boundary: $\mathfrak{B}^{\mathrm{sym}}$ can be minimal or non-minimal. When $\mathfrak{B}^{\mathrm{sym}}$ is minimal, the corresponding symmetry is called a minimal symmetry, and otherwise, it is a non-minimal symmetry. The non-minimal symmetries are generically non-invertible. Even for the minimal cases, the instances when the initial group is non-abelian yield generically non-invertible symmetry categories.

- Physical boundary: $\mathfrak{B}^{\mathrm{phys}}$ can also be minimal or non-minimal. For a fixed symmetry boundary (minimal or not), we define a minimal gapped phase as one where the physical boundary is a minimal gapped boundary condition, else it is a non-minimal phase.

The advantage of the SymTFT is that these different choices of gaugings (or symmetries) and phases are simply encoded in choices of gapped BCs. We note that we will focus on the case

---

[4]Here, we suppose that the symmetry $H$ of the stacked TQFT is not spontaneously broken. Symmetry-broken TQFTs do not produce new boundary conditions.

where the stacked TQFT is non-chiral throughout this paper.

**Lattice Models.** The analysis of gapped phases and gapless phases using fusion 2-categories as an organizational principle in 2+1d has been systematically carried out in the continuum. The present paper connects this to concrete lattice realizations, building on the fusion surface model of [62]. The main advance in comparison to that paper is the construction of a fusion surface model that realizes any given minimal or non-minimal gapped phase – with minimal or non-minimal symmetry category.[5] We should emphasize that there is a huge literature on 2+1d lattice models with gapped phases, see for a selection [63–82].

Apart from the appeal that this framework has in terms of generality and systematicness, one may ask whether this is perhaps a mathematical overkill in that these symmetries and phases are related by generalized gauging to standard gapped phases with ordinary group symmetry $G$. Let us give several points to motivate this approach further:

- From a SymTFT point of view, we know that the gapped phases can be classified in terms of stacking and gauging starting with the Dirichlet boundary condition, yielding (non-)minimal symmetries and/or phases. However, implementing this on the lattice is a priori not obvious. The approach taken in this paper, based on (higher) category theory, manifests this classification directly in the construction of the lattice model.

- Minimal symmetry boundaries can indeed be implemented by standard textbook gaugings, imposing Gauss law constraints (see section 2.2.3 for a comparison to the present approach using category theory). However, the symmetry of the gauged model is a priori not obvious from the standard gauging prescription. The categorical formulation makes the symmetry structure of the gauged model manifest – by construction – even for the generalized gauging.[6]

- Our construction is particularly nice in that it does not require the full data (e.g., the 10-j symbols) of the symmetry category. Specifically, all we need is the $F$-symbols of $G$-graded fusion categories, when the initial symmetry $G$ is non-anomalous. The classification of $G$-extensions of a given fusion category is already known in the literature [83]. On the other hand, when $G$ has an anomaly $\omega$, all data we need is the $F$-symbols of $G$-graded $\omega$-twisted fusion categories.[7]

---

[5]In [62], the construction of gapped phases in the fusion surface model was outlined only briefly.
[6]See [76] for the categorical formulation of ordinary (twisted) gauging in 2+1d lattice models.
[7]The classification of $G$-graded $\omega$-twisted fusion categories is less developed to the best of our knowledge.

- Ultimately, the main challenge in 2+1d are gapless phases. In [20, 22] some initial steps towards the second order phase transitions that connect two gapped phases with fusion 2-category symmetries were made. Much remains to be explored, and providing a lattice formulation for the transitions will be an important step forward in the exploration of 2+1d gapless phases. This will be addressed in a future paper.

**Tensor Networks.** The ground states of the gapped lattice models constructed in this paper are efficiently expressed using tensor networks, which are useful tools to represent entangled quantum states on the lattice [84]. Tensor networks have been used to classify and characterize gapped phases with finite group symmetries in both 1+1d [85–89] and 2+1d [89–93]. In recent years, it has become clear that tensor networks are also suitable for representing non-invertible symmetry operators on the lattice [52,55,94–99]. In 1+1d, various properties of gapped phases with finite non-invertible symmetries have been studied using tensor networks [95, 100–106]. On the other hand, in 2+1d, there has not been a systematic study of gapped phases with non-invertible symmetries based on tensor networks. This paper provides a first step towards this direction. In particular, we present tensor network states for all 2+1d non-chiral gapped phases with all-boson fusion 2-category symmetries,[8] including non-invertible SPT and SET phases. Our tensor network states would enable us to investigate the properties of these gapped phases in detail. An in-depth study of these gapped phases will be left for the future.

**Structure of the Paper.** The structure of this paper is as follows: There are two parts to the paper. **Part I** is a summary of all the main results, which are illustrated with examples, both familiar ones and new ones. The impatient reader should focus on this summary section 2. A glossary of notation is provided in appendix A.

The main body of the work is **Part II**: This contains the main results of the paper, including their detailed derivations. The remaining part of the paper provides an in-depth analysis. We start with section 3 providing background on fusion 2-categories and tensor networks. The first result is the formulation of generalized gauging in the fusion surface models in section 4. This is then applied to the model with $G$ 0-form symmetry and trivial 't Hooft anomaly in section 5. We carry out the generalized gauging in the lattice model (both minimal and non-minimal), and discuss the symmetries of the gauged models. This is then extended to the case with 't Hooft anomaly in section 6. Finally in section 7 we propose the minimal and non-minimal gapped phases for minimal and non-minimal symmetries. This is the

---

[8]More precisely, we will focus on all-boson fusion 2-categories such that the TQFT stacked before gauging is non-chiral.

main result of the current work. As an example, in section 2.3.5, we illustrate the construction with lattice models for gapped phases with fusion 2-category symmetries that are related by generalized gauging to $S_3$, e.g. $2\mathsf{Rep}(\mathbb{Z}_3^{(1)} \rtimes \mathbb{Z}_2^{(0)})$ two-representations of the two-group non-invertible symmetries obtained by gauging $\mathbb{Z}_2$. Several appendices detail background and technical supplementary material.

# Part I
# Standalone Summary of Key Results

This first part of the paper acts as an independently self-contained summary of the main setup, results and examples. For the reader not interested in the detailed proofs and technical derivations, it suffices to read this part.

## 2 Summary of General Construction and Examples

In this paper, we will study 2+1d lattice models with categorical symmetries, so-called All boson type fusion 2-category symmetries. These fusion 2-categories, denoted by $\mathcal{S}$, comprise a large class of fusion 2-categories (in fact, the only finite internal symmetries in 2+1d not of this type are so-called emergent fermion fusion 2-categories [19]), and have the property that their Drinfeld center is given by

$$\mathcal{Z}(\mathcal{S}) = \mathcal{Z}(2\mathsf{Vec}_G^\omega) \tag{2.1}$$

for some $G$ a finite group and $\omega \in Z^4(G, \mathrm{U}(1))$ a 't Hooft anomaly for $G$. This means that any such fusion 2-category is Morita equivalent to [107]

$$2\mathsf{Vec}_G^\omega \boxtimes \Sigma\mathcal{M} \,, \tag{2.2}$$

where $\Sigma\mathcal{M}$ denotes the fusion 2-category obtained by the condensation completion of a non-degenerate braided fusion 1-category $\mathcal{M}$ [108].

Concretely, this class of symmetries in 2+1d encompasses finite group 0-form symmetries with 't Hooft anomaly $2\mathsf{Vec}_G^\omega$, abelian 1-form symmetry groups $2\mathsf{Rep}(\mathbb{A})$, and 2-group symmetries $\mathbb{G}^{(2)}$ that mix these two, and finally, non-invertible symmetries such as various non-invertible 1-form symmetries and representations of 2-groups $2\mathsf{Rep}(\mathbb{G}^{(2)})$.

## 2.1 Brief Review: Gapped Phases in 2+1d with Categorical Symmetries

Let us begin with a brief review of the SymTFT classification of gapped phases with all-boson fusion 2-category symmetries [1]. We will use this classification data as an input for

constructing lattice models for these gapped phases.

Given a all-boson fusion 2-category $\mathcal{S}$, one can construct $\mathcal{S}$-symmetric gapped phases in 2+1d by the interval compactification of the 3+1d SymTFT $\mathcal{Z}(2\mathsf{Vec}_G^\omega)$ [2, 20]. The symmetry boundary $\mathfrak{B}^{\mathrm{sym}}$ is chosen so that the topological defects on $\mathfrak{B}^{\mathrm{sym}}$ form the fusion 2-category $\mathcal{S}$.[9] For instance, for $\mathcal{S} = 2\mathsf{Vec}_G^\omega$, we can choose $\mathfrak{B}^{\mathrm{sym}}$ to be the standard Dirichlet boundary $\mathfrak{B}_{\mathrm{Dir}}$. On the other hand, the physical boundary $\mathfrak{B}^{\mathrm{phys}}$ is an arbitrary topological boundary condition of $\mathcal{Z}(2\mathsf{Vec}_G^\omega)$. By compactifying the 3+1d bulk TFT, we obtain a 2+1d gapped phase with symmetry $\mathcal{S}$. This construction implies that the classification of $\mathcal{S}$-symmetric gapped phases reduces to the classification of topological boundary conditions of $\mathcal{Z}(2\mathsf{Vec}_G^\omega)$.

The topological boundary conditions of $\mathcal{Z}(2\mathsf{Vec}_G^\omega)$ can be classified into the following two types:

- **Minimal BCs**: these are obtained from the standard Dirichlet BC $\mathfrak{B}_{\mathrm{Dir}}$ by stacking with an SPT $[\nu] \in H^3(H, \mathrm{U}(1))$ and gauging the diagonal $H$ subgroup. These boundary conditions will be denoted by $\mathfrak{B}_{(\mathrm{Neu}(H),\nu)}$ with $(\mathrm{Neu}(1), 1) = \mathrm{Dir}$ is the Dirichlet BC. This construction, called minimal gauging, works only for a non-anomalous subgroup $H \subset G$.

- **Non-minimal BCs**: these are obtained by stacking a general $H$-symmetric TFT on the Dirichlet boundary and gauging the diagonal $H$ subgroup. There are infinitely many such boundary conditions. This construction, dubbed non-minimal gauging, works for any subgroup $H$, which may or may not be anomalous. When $H$ is anomalous, the stacked TFT must have an anomaly $\omega|_H^{-1}$ so that the diagonal $H$ symmetry is non-anomalous. Here, $\omega|_H$ denotes the restriction of $\omega$ to $H$.

When the symmetry boundary $\mathfrak{B}^{\mathrm{sym}}$ is minimal, the corresponding symmetry $\mathcal{S}$ is called a minimal symmetry. On the other hand, when $\mathfrak{B}^{\mathrm{sym}}$ is non-minimal, $\mathcal{S}$ is referred to as a non-minimal symmetry. Similarly, when the physical boundary $\mathfrak{B}^{\mathrm{phys}}$ is minimal, the corresponding phase is called a minimal gapped phase, and otherwise it is a non-minimal gapped phase. These notions can be combined into four possible types of gapped phases: (non-)minimal gapped phases with (non-)minimal symmetries. See figure 1 for the SymTFT construction of minimal gapped phases with minimal symmetries.

For more detailed discussions of the continuum aspects of the SymTFT construction of gapped phases in 2+1d, see [1, 2]. Particularly noteworthy phases are those where non-invertible symmetries can be spontaneously broken[10]. E.g. if the symmetry is the $2\mathsf{Rep}(\mathbb{G}^{(2)})$

---

[9]In general, there can be multiple topological boundary conditions that realize the same symmetry category $\mathcal{S}$. We fix one of such boundary conditions in the SymTFT construction of $\mathcal{S}$-symmetric gapped phases.

[10]These were referred to as 'superstar' phases.

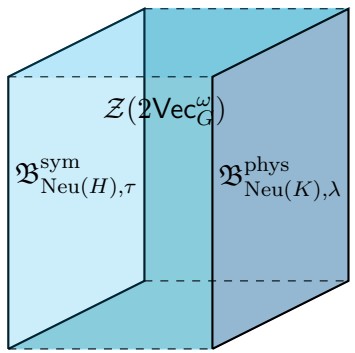

Figure 1: SymTFT description of a minimal gapped phase with minimal symmetry: The SymTFT is the DW theory in 3+1d for $G$ with twist $\omega$, and has topological defects given by the Drinfeld center $\mathcal{Z}(2\mathsf{Vec}_G^\omega)$. The gapped phase is specified by two gapped boundary conditions: the symmetry boundary $\mathfrak{B}^{\mathrm{sym}}$ determined by stacking an SPT $[\nu] \in H^3(H, \mathrm{U}(1))$ on the Dirichlet boundary and gauging the subgroup $H$ of $G$, and the physical boundary $\mathfrak{B}^{\mathrm{phys}}$ given by choosing instead a subgroup $K$ and $[\lambda] \in H^3(K, \mathrm{U}(1))$. For non-minimal boundary conditions, we stack with an $H$ (or $K$)-symmetric TFT before gauging $H$ (or $K$). The former case corresponds to a non-minimal symmetry, the latter to a non-minimal phase. Note that the stacked TFT can have an anomaly when $\omega$ is non-trivial.

(two-representations of a two-group), there are phases with multiple vacua (due to spontaneous breaking of the 0-form symmetry), but the topological order in each vacuum is distinct, and can be a non-trivial TO or an SPT. We will call such phases **Spontaneously Nonuniform Entangled Phase (SNEP)** as a phase exhibiting spontaneous symmetry breaking in which distinct ground states possess inequivalent entanglement structures. This contrasts with conventional topological phases, where the ground state manifold is uniformly entangled.

We will construct lattice models for (non-)minimal gapped phases with (non-)minimal symmetries in the current paper. Throughout the paper, we will assume that the stacked TFT is non-chiral.

## 2.2 Generalized Gauging on the Lattice

In this subsection, we describe our generalized (non-minimal) gauging prescription on the lattice. We will also provide the generalized gauging operators, i.e., operators that implement the generalized gauging and ungauging. For simplicity, we will focus on the generalized gauging of a non-anomalous finite group symmetry $G$ for the moment. Details of this will be discussed in section 5, with the generalization to anomalous $G$ discussed in detail in section 6.

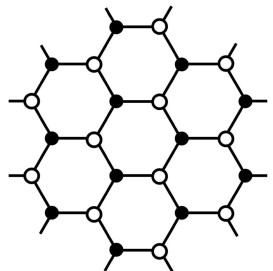

Figure 2: A honeycomb lattice. The black dots constitute the $A$-sublattice, while the white dots constitute the $B$-sublattice.

### 2.2.1   $G$-Symmetric Model

We first define a $G$-symmetric model before gauging. For concreteness, we consider a model defined on a honeycomb lattice. The sets of plaquettes, edges, and vertices of the lattice are denoted by $P$, $E$, and $V$, respectively. The set $V$ can be decomposed into two disjoint sets

$$V = V_A \cup V_B, \qquad V_A \cap V_B = \varnothing, \tag{2.3}$$

where $V_A$ is the set of vertices in the $A$-sublattice, while $V_B$ is the set of vertices in the $B$-sublattice, see figure 2.

The state space of the $G$-symmetric model is given by the tensor product

$$\mathcal{H}_{\text{original}} := \bigotimes_{i \in P} \mathbb{C}[G]. \tag{2.4}$$

Each state in $\mathcal{H}_{\text{original}}$ is denoted by $|\{g_i\}\rangle$, where $g_i \in G$ is the dynamical variable on the plaquette $i$. In what follows, we will employ the following diagrammatic representation of $|\{g_i\}\rangle$:

$$|\{g_i\}\rangle = \left| \begin{array}{c} g_l \\ g_j \quad g_k \\ g_i \end{array} \right\rangle. \tag{2.5}$$

The Hamiltonian of the model is then given by the sum

$$H_{\text{original}} = -\sum_{i \in P} \widehat{\mathsf{h}}_i, \tag{2.6}$$

where each local term $\widehat{\mathsf{h}}_i$ acts on the dynamical variables around a single plaquette as

$$\widehat{\mathsf{h}}_i \left| \begin{array}{c} g_8 \\ g_6 \quad g_7 \\ g_4 \\ g_2 \quad g_3 \\ g_1 \end{array} \right\rangle = \sum_{g_5 \in G} \chi(g_4^{-1} g_5; \{g_i^{-1} g_j\}) \left| \begin{array}{c} g_8 \\ g_6 \quad g_7 \\ g_5 \\ g_2 \quad g_3 \\ g_1 \end{array} \right\rangle. \tag{2.7}$$

Here, $\chi(g_4^{-1}g_5; \{g_i^{-1}g_j\})$ is an arbitrary complex number that depends on $g_4^{-1}g_5$ and $g_i^{-1}g_j$ for $i, j \neq 5$. The above Hamiltonian commutes with the symmetry operator defined by

$$U_g \left|\{g_i\}\right\rangle = \left|\{gg_i\}\right\rangle \tag{2.8}$$

for all $g \in G$.

We note that the generalized gauging that we will describe below can also be applied to more general $G$-symmetric models. Indeed, when constructing gapped phases with all-boson fusion 2-category symmetries, we will apply the generalized gauging to slightly more general models. The general prescription of the generalized gauging will be explained in section 4.2

### 2.2.2 Gauged Model

**Input Data.** To describe the generalized gauging, we first specify an $H$-symmetric TFT that we stack before gauging, where $H$ is a subgroup of $G$. Without loss of generality, we assume that the $H$ symmetry of the stacked TFT is not spontaneously broken. In general, 2+1d TFTs with unbroken $H$-symmetry are classified by $H$-crossed extensions of modular tensor categories [109]. In this paper, we focus on the case where the stacked TFT is non-chiral.

An $H$-symmetric non-chiral TFT in 2+1d is specified by the choice of an $H$-graded fusion category

$$A = \bigoplus_{h \in H} A_h. \tag{2.9}$$

We suppose that the $H$-grading on $A$ is faithful, i.e.,

$$A_h \neq 0, \qquad \forall \, h \in H. \tag{2.10}$$

The corresponding $H$-symmetric TFT, which we denote by $\mathfrak{T}_A^H$, is realized by the low-energy limit of the symmetry-enriched string-net model based on $A$ [70,71].[11] The continuum description of this TFT is discussed in [112]. Mathematically, this TFT is described by the relative center $\mathcal{Z}_{A_e}(A)$, which is an $H$-crossed extension of the Drinfeld center $\mathcal{Z}(A_e)$ of the trivially graded component $A_e$ [113]. Any $H$-symmetric non-chiral TFT in 2+1d can be obtained in this way due to the one-to-one correspondence between $H$-crossed extensions of $\mathcal{Z}_{(}A_e)$ and $H$-graded extensions of $A_e$ [83].

Schematically, the generalized gauging that we will discuss below can be written as

$$(X_{\text{original}} \boxtimes \mathfrak{T}_{A^{\text{rev}}}^H)/H^{\text{diag}}, \tag{2.11}$$

---

[11]The symmetry-enriched string-net models in [70, 71] should not be confused with the enriched string-net models in [110, 111]. The former is related to the fusion surface model based on finite groups, while the latter is related to the fusion surface model based on braided fusion 1-categories [78, 81].

where $X_{\text{original}}$ is the original $G$-symmetric model and $A^{\text{rev}}$ is the reverse category of $A$, i.e., the $H$-graded fusion category obtained by reversing all morphisms of $A$.[12] For simplicity, in this subsection, we assume that $A$ is multiplicity-free, i.e., the fusion coefficient of $A$ is either zero or one. The case of a more general $H$-graded fusion category will be discussed in section 5. The above generalized gauging reduces to the ordinary twisted gauging when $A = \text{Vec}_H^\nu$. We will comment more on the relation to the ordinary gauging in section 2.2.3.

Now, let us describe the generalized gauging on the lattice using the data of an $H$-graded fusion category $A$. We will first define the state space of the gauged model and then write down the Hamiltonian on the lattice.

**State Space.** The state space of the gauged model is given by

$$\mathcal{H}_{\text{gauged}} := \widehat{\pi}_{\text{LW}} \mathcal{H}'_{\text{gauged}}, \tag{2.12}$$

where $\widehat{\pi}_{\text{LW}}$ is a projector that we will define shortly, and $\mathcal{H}'_{\text{gauged}}$ is the vector space spanned by all possible configurations of the following dynamical variables:

- The dynamical variables on the plaquettes are labeled by the representatives of right $H$-cosets in $G$. The set of representatives will be denoted by $S_{H\backslash G}$.

- The dynamical variables on the edges are labeled by simple objects of $A^{\text{rev}}$.

We note that there are no dynamical variables on the vertices.[13] The state corresponding to each configuration of the dynamical variables is denoted by $|\{g_i, a_{ij}\}\rangle$, where $g_i \in S_{H\backslash G}$ is the representative of the right $H$-coset $Hg_i$ and $a_{ij} \in A^{\text{rev}}$ is a simple object of $A^{\text{rev}}$. The state $|\{g_i, a_{ij}\}\rangle$ is represented diagrammatically as

$$|\{g_i, a_{ij}\}\rangle = \tag{2.13}$$

---

[12]The reason why we stack $\mathfrak{T}_{A^{\text{rev}}}^H$ rather than $\mathfrak{T}_A^H$ becomes clear if we consider the generalized gauging of an anomalous symmetry. When the symmetry $G$ has an anomaly $\omega \in Z^4(G, \text{U}(1))$, the input datum of the generalized gauging is an $H$-graded $\omega|_H$-twisted fusion category $A$, where $\omega|_H$ is the restriction of $\omega$ to $H$. The symmetry-enriched string-net model based on $A$ now has an anomaly $\omega|_H$, see section 7.2. On the other hand, the symmetry-enriched string-net model based on $A^{\text{rev}}$ has an anomaly $\omega|_H^{-1}$ because $A^{\text{rev}}$ is an $H$-graded $\omega|_H^{-1}$-twisted fusion category. Thus, for the diagonal $H$ symmetry to be gaugeable, the stacked TFT must be $\mathfrak{T}_{A^{\text{rev}}}^H$. See section 6 for more details on the generalized gauging of anomalous symmetries.

[13]When $A$ is not multiplicity-free, the vertices also have dynamical variables, which are labeled by morphisms of $A^{\text{rev}}$.

The configuration of the dynamical variables on the edges must be compatible with the fusion rules. Namely, for every vertex $[ijk] \in V$, we require that $a_{ik}$ is a fusion channel of $a_{ij} \otimes a_{jk}$. On the other hand, there is no constraint on the dynamical variables on the plaquettes.

The projector $\widehat{\pi}_{\mathrm{LW}}$ in (2.12) is given by the product of the local commuting projectors

$$\widehat{\pi}_{\mathrm{LW}} := \prod_{i \in P} \widehat{B}_i, \tag{2.14}$$

where $\widehat{B}_i$ the Levin-Wen plaquette operator defined by [64]

$$\widehat{B}_i \left| \begin{array}{c} \end{array} \right\rangle = \sum_{a \in A_e^{\mathrm{rev}}} \frac{\dim(a)}{\mathcal{D}_{A_e}} \left| \begin{array}{c} a \end{array} \right\rangle. \tag{2.15}$$

Here, $\dim(a)$ is the quantum dimension of $a$, $\mathcal{D}_{A_e} = \sum_{a \in A_e} \dim(a)^2$ is the total dimension of $A_e$, and the summation is taken over all simple objects of $A_e^{\mathrm{rev}}$. The diagram on the right-hand side is evaluated by fusing the loop into the nearby edges using the $F$-move of $A^{\mathrm{rev}}$.[14] We note that $\widehat{B}_i$ does not act on the dynamical variables on the plaquettes. Intuitively, the above Levin-Wen constraint ensures that the edge degrees of freedom are in the ground state subspace of the stacked topological order.

By slight abuse of notation, the state $\widehat{\pi}_{\mathrm{LW}} |\{g_i, a_{ij}\}\rangle \in \mathcal{H}_{\mathrm{gauged}}$ will be denoted simply by $|\{g_i, a_{ij}\}\rangle$ in what follows. The same comment also applies to the later sections.

**Hamiltonian.** The Hamiltonian of the gauged model is given by the sum

$$H_{\mathrm{gauged}} = -\sum_{i \in P} \widehat{h}_i, \tag{2.16}$$

where each term $\widehat{h}_i$ is given by

$$\widehat{h}_i \left| \begin{array}{c} M^{g_4} \end{array} \right\rangle = \sum_{g_5 \in S_{H \backslash G}} \sum_{h_{45} \in H} \chi(g_4^{-1} h_{45} g_5; \{g_i^{-1} h_{ij} g_j\}) \sum_{a_{45} \in A_{h_{45}}^{\mathrm{rev}}} \frac{d_{45}^a}{\mathcal{D}_{A_{h_{45}}}} \left| \begin{array}{c} M^{g_5} \\ a_{45} \end{array} \right\rangle. \tag{2.17}$$

Here, $h_{ij} \in H$ is the grading of $a_{ij} \in A_{h_{ij}}^{\mathrm{rev}}$, and $d_{ij}^a$ is the quantum dimension of $a_{ij}$. The diagram on the right-hand side is again evaluated by using the $F$-move of $A^{\mathrm{rev}}$. More explicitly,

---

[14]The $F$-symbol of $A^{\mathrm{rev}}$ is given by the inverse of the $F$-symbol of $A$. In the diagrammatic calculus in this section (except for (2.19)), we will use the $F$-move of $A^{\mathrm{rev}}$ rather than that of $A$.

the action of $\widehat{\mathsf{h}}_i$ can be computed as

$$
\widehat{\mathsf{h}}_i \left| \begin{array}{c} a_{68} \quad a_{78} \\ a_{48} \\ a_{46} \quad a_{47} \; a_{37} \\ a_{26} \; \overleftarrow{\phantom{M}} M^{g_4} \overleftarrow{\phantom{a}} \\ a_{24} \quad a_{34} \\ a_{14} \\ a_{12} \quad a_{13} \end{array} \right\rangle = \sum_{g_5 \in S_{H \backslash G}} \sum_{h_{45} \in H} \chi(g_4^{-1} h_{45} g_5; \{g_i^{-1} h_{ij} g_j\}) \sum_{a_{45} \in A_{h_{45}}^{\mathrm{rev}}} \frac{d_{45}^a}{\mathcal{D}_{A_{h_{45}}}}
$$

$$
\sum_{a_{15}, \cdots, a_{58}} \overline{F}_{1245}^a \overline{F}_{2456}^a \overline{F}_{4568}^a F_{1345}^a F_{3457}^a F_{4578}^a \tag{2.18}
$$

$$
\sqrt{\frac{d_{15}^a d_{56}^a d_{57}^a}{d_{14}^a d_{46}^a d_{47}^a}} \sqrt{\frac{d_{24}^a d_{34}^a d_{48}^a}{d_{25}^a d_{35}^a d_{58}^a}} \left| \begin{array}{c} a_{68} \quad a_{78} \\ a_{58} \\ a_{56} \quad a_{57} \; a_{37} \\ a_{26} \; \overleftarrow{\phantom{M}} M^{g_5} \overleftarrow{\phantom{a}} \\ a_{25} \quad a_{35} \\ a_{15} \\ a_{12} \quad a_{13} \end{array} \right\rangle ,
$$

where $F_{ijkl}^a$ and $\overline{F}_{ijkl}^a$ are the $F$-symbols of $A$ (not $A^{\mathrm{rev}}$) defined by

$$
\begin{array}{ccc} a_{ij} \; a_{jk} \; a_{kl} & & a_{ij} \; a_{jk} \; a_{kl} \\ a_{ik} \diagdown \diagup & = \sum_{a_{jl}} F_{ijkl}^a & \diagdown \diagup a_{jl} \\ a_{il} & & a_{il} \end{array} , \qquad \begin{array}{ccc} a_{ij} \; a_{jk} \; a_{kl} & & a_{ij} \; a_{jk} \; a_{kl} \\ \diagdown \diagup a_{jl} & = \sum_{a_{ik}} \overline{F}_{ijkl}^a & a_{ik} \diagdown \diagup \\ a_{il} & & a_{il} \end{array} . \tag{2.19}
$$

We suppose that $F$ is unitary, i.e., $\overline{F}_{ijkl}^a = (F_{ijkl}^a)^*$. The morphisms at the vertices of (2.19) are normalized as in (5.9).

The symmetry of the gauged model is described by the fusion 2-category $_A(2\mathsf{Vec}_G)_A$. See section 5.3 for more details on this symmetry category. The symmetry operators on the lattice will be briefly summarized in section 2.2.5.

**A Remark on the Choice of Representatives.**  We note that the gauged model defined above depends on the choice of representatives of right $H$-cosets in $G$. From a gauge-theoretical perspective, this dependence originates from the gauge fixing; see the next subsection for more details. Similarly, the explicit forms of the symmetry operators also depend on the choice of representatives, although the symmetry category does not. Nevertheless, we expect that the phase of the gauged model does not depend on the choice of representatives as long as the symmetry is taken into account.[15] The isomorphism between two models with different choices of representatives is discussed around (4.5) in the context of general fusion surface models.

---

[15]This is possible because both the model and the symmetry operators depend on the choice of representatives.

### 2.2.3 Gauge Theoretical Description

In the above description of the gauged model, the relation to the standard gauging procedure may not be transparent. In this subsection, we provide a more gauge-theoretical description of our model. Specifically, we will define a gauged model by generalizing the standard gauging procedure and then show that it reduces to the above model after gauge fixing.[16]

**State Space.** We consider a model whose state space is given by

$$\widetilde{\mathcal{H}}_{\text{gauged}} := \widehat{\pi}_{\text{Gauss}} \widetilde{\mathcal{H}}'_{\text{gauged}}, \tag{2.20}$$

where $\widehat{\pi}_{\text{Gauss}}$ is a projector that we will define shortly, and $\widetilde{\mathcal{H}}'_{\text{gauged}}$ is the vector space spanned by all possible configurations of the following dynamical variables:

- The dynamical variables on the plaquettes are labeled by elements of $G$.

- The dynamical variables on the edges are labeled by simple objects of $A^{\text{rev}}$.

The state corresponding to each configuration of the dynamical variables is denoted by

$$|\{g_i, a_{ij}\}\rangle = \left| \begin{array}{c} \end{array} \right\rangle, \tag{2.21}$$

where $g_i \in G$ and $a_{ij} \in A^{\text{rev}}$. The dynamical variables on the edges must be compatible with the fusion rules, i.e., they are subject to the constraint

$$a_{ik} \in a_{ij} \otimes a_{jk}, \qquad \forall [ijk] \in V. \tag{2.22}$$

On the other hand, there is no constraint on the dynamical variables on the plaquettes. We note that the edge degrees of freedom can be regarded as generalized gauge fields, which reduce to the ordinary $H$-gauge fields when $A = \mathsf{Vec}_H^\nu$.

The projector $\widehat{\pi}_{\text{Gauss}}$ is the product of local commuting projectors

$$\widehat{\pi}_{\text{Gauss}} = \prod_{i \in P} \widehat{G}_i, \tag{2.23}$$

---

[16]We emphasize that the derivation of our gauged model in section 5.2.2 is based on (higher) category theory. One advantage of the category-theoretical approach is that it makes the symmetry structure of the gauged model manifest.

where $\widehat{G}_i$ is the generalized Gauss law operator defined by

$$\widehat{G}_i \left| \vcenter{\hbox{}} \right\rangle = \frac{1}{|H|} \sum_{h \in H} \sum_{a \in A_h^{\mathrm{rev}}} \frac{\dim(a)}{\mathcal{D}_{A_h}} \left| \vcenter{\hbox{}} \right\rangle . \tag{2.24}$$

The diagram on the right-hand side is evaluated by fusing the middle loop into the nearby edges using the $F$-move of $A^{\mathrm{rev}}$. We note that $\widehat{\pi}_{\mathrm{Gauss}}$ is a generalization of the Gauss law constraint. In particular, it reduces to the ordinary Gauss law constraint when $A = \mathsf{Vec}_H$.

**Hamiltonian.** The Hamiltonian of the model is defined by

$$\widetilde{H}_{\mathrm{gauged}} = - \sum_{i \in P} \widehat{\mathsf{h}}_i, \tag{2.25}$$

where each term $\widehat{\mathsf{h}}_i$ is obtained by modifying the original Hamiltonian (2.7) by the minimal coupling

$$\widehat{\mathsf{h}}_i \left| \vcenter{\hbox{}} \right\rangle = \sum_{g_5 \in G} \chi(g_4^{-1} g_5 ; \{ g_i^{-1} h_{ij} g_j \}) \left| \vcenter{\hbox{}} \right\rangle . \tag{2.26}$$

Here, $h_{ij} \in H$ is the grading of $a_{ij} \in A_{h_{ij}}$. We note that $\widehat{\mathsf{h}}_i$ changes the dynamical variable only on the middle plaquette.

**Gauge Fixing.** Now, we argue that the above model is equivalent to the gauged model defined in section 2.2.2. To this end, we first notice that the Gauss law operator (2.24) satisfies

$$\widehat{G}_i \left| \vcenter{\hbox{}} \right\rangle = \widehat{G}_i \sum_{a \in A_h^{\mathrm{rev}}} \frac{\dim(a)}{\mathcal{D}_{A_h}} \left| \vcenter{\hbox{}} \right\rangle . \tag{2.27}$$

This equality implies that the following two states are identified in $\widetilde{\mathcal{H}}_{\mathrm{gauged}}$:

$$\left| \vcenter{\hbox{}} \right\rangle \sim \sum_{a \in A_h^{\mathrm{rev}}} \frac{\dim(a)}{\mathcal{D}_{A_h}} \left| \vcenter{\hbox{}} \right\rangle . \tag{2.28}$$

Due to this identification, we can fix the plaquette degrees of freedom to the representatives of right $H$-cosets in $G$. This fixing procedure, known as the gauge fixing, partially solves the Gauss law constraint $\widehat{G}_i = 1$. More specifically, the gauge fixing reduces the Gauss law constraint to the Levin-Wen constraint $\widehat{B}_i = 1$. Therefore, after the gauge fixing, the state space $\widetilde{\mathcal{H}}_{\text{gauged}}$ agrees with $\mathcal{H}_{\text{gauged}}$ defined by (2.12). Similarly, the gauge fixing also reduces the Hamiltonian (2.26) to

$$
\widehat{\mathsf{h}}_i \left| \begin{array}{c} \text{hexagon with } g_4 \end{array} \right\rangle = \sum_{g_5 \in S_{H\backslash G}} \sum_{h_{45} \in H} \chi(g_4^{-1} h_{45} g_5; \{g_i^{-1} h_{ij} g_j\}) \left| \begin{array}{c} \text{hexagon with } h_{45} g_5 \end{array} \right\rangle
$$

$$
\sim \sum_{g_5 \in S_{H\backslash G}} \sum_{h_{45} \in H} \chi(g_4^{-1} h_{45} g_5; \{g_i^{-1} h_{ij} g_j\}) \sum_{a_{45} \in A^{\text{rev}}_{h_{45}}} \frac{d^a_{45}}{\mathcal{D}_{A_{h_{45}}}} \left| \begin{array}{c} \text{hexagon with } g_5 \text{ and } a_{45} \end{array} \right\rangle. \tag{2.29}
$$

where $g_i \in S_{H\backslash G}$ is the representative of the right $H$-coset $Hg_i$. On the second line, we used the identification (2.28). The above Hamiltonian agrees with the Hamiltonian (2.17).

We note that the above gauging procedure reduces to the ordinary gauging of a subgroup symmetry $H \subset G$ when $A = \mathsf{Vec}_H$. More generally, when $A = \mathsf{Vec}^\nu_H$ where $\nu \in Z^3(H, \mathrm{U}(1))$ is a 3-cocycle on $H$, our prescription reduces to the twisted gauging of $H$.

### 2.2.4 Generalized Gauging Operators

The operator that implements the generalized gauging will be called *generalized gauging operator*. We should remark that in some literature, this is also often referred to as the 'duality' operator. However, since this can be a confusing naming (in view of the meaning of exact dualities or IR dualities), we will generally refer to it as the generalized gauging operator. It is given by a linear map

$$
\mathsf{D}_A : \mathcal{H}_{\text{original}} \to \mathcal{H}_{\text{gauged}}, \tag{2.30}
$$

which intertwines the Hamiltonian of the original model and that of the gauged model:

$$
\mathsf{D}_A H_{\text{original}} = H_{\text{gauged}} \mathsf{D}_A. \tag{2.31}
$$

As we will see in section 5.2.3, the explicit form of $\mathsf{D}_A$ is given by

$$\mathsf{D}_A \left|\{h_i g_i\}\right\rangle = \sum_{\{a_i \in A_{h_i}^{\mathrm{rev}}\}} \prod_{i \in P} \frac{\dim(a_i)}{\mathcal{D}_{A_{h_i}}}$$

$$= \sum_{\{a_i \in A_{h_i}^{\mathrm{rev}}\}} \sum_{\{a_{ij} \in \overline{a_i} \otimes a_j\}} \prod_{i \in P} \frac{1}{\mathcal{D}_{A_{h_i}}} \prod_{[ijk] \in V} \left(\frac{d_{ij}^a d_{jk}^a}{d_{ik}^a}\right)^{\frac{1}{4}} \prod_{[ijk] \in V} (\overline{F}_{ijk}^a)^{s_{ijk}} \left|\{g_i, a_{ij}\}\right\rangle,$$

(2.32)

where $h_i \in H$, $g_i \in S_{H \backslash G}$,[17] $\overline{a_i}$ is the dual of $a_i$, $s_{ijk}$ is the sign defined by

$$s_{ijk} := \begin{cases} +1 & \text{if } [ijk] \in V_A, \\ -1 & \text{if } [ijk] \in V_B, \end{cases}$$

(2.33)

and we used the following notation of the $F$-symbols:

$$(\overline{F}_{ijk}^a)^{+1} := \overline{F}_{0ijk}^a, \qquad (\overline{F}_{ijk}^a)^{-1} := F_{0ijk}^a,$$

(2.34)

Here, we defined $a_{0i} := a_i$, $a_{0j} := a_j$, and $a_{0k} := a_k$.

Conversely, the generalized gauging operator that implements the ungauging procedure is a linear map

$$\overline{\mathsf{D}}_A : \mathcal{H}_{\mathrm{gauged}} \to \mathcal{H}_{\mathrm{original}},$$

(2.35)

which intertwines $H_{\mathrm{gauged}}$ and $H_{\mathrm{original}}$, that is,

$$\overline{\mathsf{D}}_A H_{\mathrm{gauged}} = H_{\mathrm{original}} \overline{\mathsf{D}}_A.$$

(2.36)

The explicit form of $\overline{\mathsf{D}}_A$ is given by

$$\overline{\mathsf{D}}_A \left|g_i, a_{ij}\right\rangle = \sum_{\{h_i \in H\}} \sum_{\{a_i \in A_{h_i}^{\mathrm{rev}}\}} \prod_{[ijk] \in V} \left(\frac{d_{ij}^a d_{jk}^a}{d_{ik}^a}\right)^{\frac{1}{4}} \prod_{[ijk] \in V} (F_{ijk}^a)^{s_{ijk}} \left|\{h_i g_i\}\right\rangle,$$

(2.37)

where $g_i \in S_{H \backslash G}$ and $a_{ij} \in A^{\mathrm{rev}}$. Here, we used the following notation of the $F$-symbols

$$(F_{ijk}^a)^{+1} := F_{0ijk}^a, \qquad (F_{ijk}^a)^{-1} := \overline{F}_{0ijk}^a,$$

(2.38)

where $a_{0i} := a_i$, $a_{0j} := a_j$, and $a_{0k} := a_k$.

---

[17]We note that $h_i g_i$ is the unique decomposition of an element of $G$ into the product of an element of $H$ and that of $S_{H \backslash G}$.

**Ordinary Twisted Gauging.** As we mentioned in section 2.2.3, the generalized gauging reduces to the ordinary twisted gauging when $A = \mathsf{Vec}_H^\nu$. In this case, the gauging operator $\mathsf{D}_A$ and the ungauging operator $\overline{\mathsf{D}}_A$ are given by

$$\mathsf{D}_A \left| \{h_i g_i\} \right\rangle = \prod_{[ijk] \in V} \nu(h_i, h_i^{-1} h_j, h_j^{-1} h_k)^{-s_{ijk}} \left| \{g_i, h_i^{-1} h_j\} \right\rangle, \tag{2.39}$$

$$\overline{\mathsf{D}}_A \left| g_i, h_{ij} \right\rangle = \sum_{\{h_i \in H\}} \prod_{[ij] \in E} \delta_{h_{ij}, h_i^{-1} h_j} \prod_{[ijk] \in V} \nu(h_i, h_i^{-1} h_j, h_j^{-1} h_k)^{s_{ijk}} \left| \{h_i g_i\} \right\rangle. \tag{2.40}$$

Here, the simple objects of $A = \mathsf{Vec}_H^\nu$ are denoted simply by the elements of $H$. The gauging operators for the ordinary twisted gauging were originally studied in [76].

**Gauge Theoretical Derivation.** Let us provide a gauge-theoretical derivation of the generalized gauging operator $\mathsf{D}_A$. See section 5.2.3 for another derivation based on (higher) category theory.

We first recall that the state space of the gauged model is given by $\widetilde{\mathcal{H}}_{\text{gauged}}$ before gauge fixing. Therefore, the target vector space of $\mathsf{D}_A$ can also be regarded as $\widetilde{\mathcal{H}}_{\text{gauged}}$. As a linear map from $\mathcal{H}_{\text{original}}$ to $\widetilde{\mathcal{H}}_{\text{gauged}}$, the generalized gauging operator $\mathsf{D}_A$ is defined by

$$\mathsf{D}_A \left| \{g_i\} \right\rangle = \widehat{\pi}_{\text{Gauss}} \left| \{g_i, \mathbb{1}_A\} \right\rangle, \tag{2.41}$$

where $\left| \{g_i, \mathbb{1}_A\} \right\rangle$ is a state such that all the edges are labeled by the unit object $\mathbb{1}_A \in A$. Due to (2.27) and (2.28), we find

$$\mathsf{D}_A \left| \{h_i g_i\} \right\rangle \sim \sum_{\{a_i \in A_{h_i}^{\text{rev}}\}} \prod_{i \in P} \frac{\dim(a_i)}{\mathcal{D}_{A_{h_i}}} \quad \cdots \quad , \tag{2.42}$$

where $h_i \in H$ and $g_i \in S_{H \backslash G}$. This result agrees with (2.32)

### 2.2.5 Symmetry Operators

By construction, the gauged model defined in section 2.2.2 has the fusion 2-category symmetry described by ${}_A(2\mathsf{Vec}_G)_A$, the 2-category of $(A, A)$-bimodules in $2\mathsf{Vec}_G$.[18] In particular, the 0-form and 1-form symmetry operators are labeled by objects and 1-morphisms of ${}_A(2\mathsf{Vec}_G)_A$. In what follows, we explicitly write down some of the 0-form symmetry operators on the lattice.

---

[18]This follows from the general discussion in section 4.2.

See section 5.3 for more details of this symmetry 2-category as well as the corresponding symmetry operators on the lattice.

Simple examples of the 0-form symmetry operators are obtained as

$$\mathsf{D}_A U_g \overline{\mathsf{D}}_A : \quad \mathcal{H}_{\text{gauged}} \to \mathcal{H}_{\text{gauged}}, \qquad g \in G, \tag{2.43}$$

where $\mathsf{D}_A$ and $\overline{\mathsf{D}}_A$ are the generalized gauging operators defined by (2.32) and (2.37), and $U_g$ is the symmetry operator (2.8) of the $G$-symmetric model before gauging. Similar construction of the symmetry operators was discussed in [114] for specific 1+1d lattice models. The action of the above symmetry operator can be computed as

$$\mathsf{D}_A U_g \overline{\mathsf{D}}_A \left|\{g_i, a_{ij}\}\right\rangle = \sum_{\{h_i \in H\}} \sum_{\{a_i \in A_{h_i}^{\text{rev}}\}} \sum_{\{a'_i \in A_{h'_i}^{\text{rev}}\}} \sum_{\{a'_{ij} \in \overline{a'_i} \otimes a'_j\}} \prod_{[ijk] \in V} \left(\frac{d_{ij}^a d_{jk}^a}{d_{ik}^a}\right)^{\frac{1}{4}} \prod_{[ijk] \in V} (F_{ijk}^a)^{s_{ijk}}$$

$$\prod_{i \in P} \frac{1}{\mathcal{D}_{A_{h'_i}}} \prod_{[ijk] \in V} \left(\frac{d_{ij}^{a'} d_{jk}^{a'}}{d_{ik}^{a'}}\right)^{\frac{1}{4}} \prod_{[ijk] \in V} (\overline{F}_{ijk}^{a'})^{s_{ijk}} \left|\{g'_i, a'_{ij}\}\right\rangle. \tag{2.44}$$

Here, for given $g \in G$, $h_i \in H$, and $g_i \in S_{H \backslash G}$, the elements $h'_i$ and $g'_i$ are defined by

$$h'_i g'_i = g h_i g_i, \qquad h'_i \in H, \quad g'_i \in S_{H \backslash G}. \tag{2.45}$$

We note that (2.45) uniquely determines both $h'_i$ and $g'_i$. In the case of ordinary twisted gauging $A = \mathsf{Vec}_H^\nu$, we have

$$\mathsf{D}_A U_g \overline{\mathsf{D}}_A \left|\{g_i, h_{ij}\}\right\rangle = \sum_{\{h_i \in H\}} \prod_{[ij] \in E} \delta_{h_{ij}, h_i^{-1} h_j} \prod_{[ijk] \in V} \frac{\nu(h_i, h_{ij}, h_{jk})}{\nu(h'_i, h'_{ij}, h'_{jk})} \left|\{g'_i, h'_{ij}\}\right\rangle, \tag{2.46}$$

where $h'_{ij} := (h'_i)^{-1} h'_j$.

The symmetry operators of the form (2.43) exhaust all the 0-form symmetry operators up to condensation. Namely, any 0-form symmetry operators of the gauged model can be obtained by condensing some line operators on these symmetry operators. See section 5.3.2 for more details on this point. In particular, $\mathsf{D}_A \overline{\mathsf{D}}_A$ is a pure condensation operator, from which we can obtain the identity operator by condensation.

## 2.3 Gapped Phases and Examples

One of the key results in this paper is the construction of lattice models for non-chiral gapped phases with all-boson fusion 2-category symmetries. These are obtained by starting from lattice models for $G$-symmetric gapped phases and then performing the generalized gauging. In this subsection, we will summarize these lattice models and write down their ground states. For simplicity, we will focus on the case where $G$ is non-anomalous for the moment. See section 7 for more details, including the case where $G$ is anomalous.

**Tensor Product Decomposition.** We note that the state space of our model admits a tensor product decomposition only when the symmetry is an ordinary group $G$, possibly with an anomaly. In particular, the state space of the gauged model cannot be decomposed into a tensor product of on-site state spaces due to the fusion constraint and the Levin-Wen constraint on the edge degrees of freedom. One may impose these constraints energetically so that the state space admits a tensor product decomposition. However, doing so makes some of the symmetry operators non-topological. In our model, we impose the above constraints exactly at the level of the state space so that the symmetry operators become topological.

In what follows, we denote by $A$ and $B$ the following data:

- $A$ is the graded fusion category that specifies the generalized gauging (and so the symmetry)

- $B$ is the graded fusion category that specifies the gapped phase.

In particular, we then get the following identification with the minimal and non-minimal classification in section 2.1:

- A minimal symmetry is obtained by choosing $A = \mathsf{Vec}_H^\nu$, where $H$ is a subgroup of $G$ and $\nu \in Z^3(H, \mathrm{U}(1))$ is a 3-cocycle on $H$. A non-minimal symmetry is the generalization to any $H$-graded fusion category $A$.

- A minimal phase is obtained by $B = \mathsf{Vec}_K^\lambda$, where $K$ is a subgroup of $G$ and $\lambda \in Z^3(K, \mathrm{U}(1))$ is a 3-cocycle on $K$. Again, a non-minimal phase is the generalization to an arbitrary $K$-graded fusion category $B$.

We will now summarize the gapped phase lattice models for all four combinations of these minimal/non-minimal choices. As in section 2.2, we will denote by $S_{H \backslash G}$ the set of representatives of right $H$-cosets in $G$.

### 2.3.1 Minimal Gapped Phases with Minimal Symmetry

Minimal gapped phases with minimal symmetries are obtained by the ordinary (twisted) gauging of $G$-symmetric gapped phases without topological orders. The corresponding choice of $A$ and $B$ is

$$A = \mathsf{Vec}_H^\nu, \qquad B = \mathsf{Vec}_K^\lambda. \tag{2.47}$$

The physical interpretation of the data $(H, \nu, K, \lambda)$ is as follows: $H$ is the gauged subgroup, $\nu \in Z^3(H, \mathrm{U}(1))$ is the discrete torsion, $K$ is the unbroken subgroup before gauging, and

$\lambda \in Z^3(K, \mathrm{U}(1))$ specifies the SPT order of each vacuum before gauging. In what follows, we will focus on the case where $\nu$ and $\lambda$ are trivial. The case of non-trivial $\nu$ and $\lambda$ will be briefly mentioned at the end and will be detailed in section 7.3.1.

**$G$-Symmetric Model.** Let us first define the $G$-symmetric model before gauging. The state space and the Hamiltonian of the model are given by

$$\mathcal{H}_{\mathrm{original}} := \bigotimes_{i \in P} \mathbb{C}[G], \tag{2.48}$$

$$H_{\mathrm{original}} = -\sum_{i \in P} \widehat{\mathsf{h}}_i - \sum_{[ij] \in E} \widehat{\mathsf{h}}_{ij}, \tag{2.49}$$

where $\widehat{\mathsf{h}}_i$ and $\widehat{\mathsf{h}}_{ij}$ are defined by[19]

$$\widehat{\mathsf{h}}_i \left| \vcenter{\hbox{\includegraphics{g4}}} \right\rangle = \frac{1}{|K|} \sum_{g_5 \in G} \delta_{g_4^{-1} g_5 \in K} \left| \vcenter{\hbox{\includegraphics{g5}}} \right\rangle, \tag{2.50}$$

$$\widehat{\mathsf{h}}_{ij} \left| \begin{matrix} g_j \\ \rule{1cm}{0.4pt} \\ g_i \end{matrix} \right\rangle = \delta_{g_i^{-1} g_j \in K} \left| \begin{matrix} g_j \\ \rule{1cm}{0.4pt} \\ g_i \end{matrix} \right\rangle. \tag{2.51}$$

Here, $\delta_{g_i^{-1} g_j \in K}$ is defined by

$$\delta_{g_i^{-1} g_j \in K} := \begin{cases} 1 & \text{if } g_i^{-1} g_j \in K, \\ 0 & \text{if } g_i^{-1} g_j \notin K. \end{cases} \tag{2.52}$$

The above model is solvable because the Hamiltonian is a sum of commuting projectors. The ground states on an infinite plane, satisfying $\widehat{\mathsf{h}}_i = 1$ and $\widehat{\mathsf{h}}_{ij} = 1$ for all $i \in P$ and $[ij] \in E$, are given by

$$|\mathrm{GS}; \mu\rangle_{\mathrm{original}} = \sum_{\{k_i \in K\}} \prod_{i \in P} \frac{1}{|K|} |\{g^{(\mu)} k_i\}\rangle, \tag{2.53}$$

where $\mu = 1, 2, \cdots, |G/K|$, and $g^{(\mu)}$ is a representative of the left $K$-coset $g^{(\mu)} K$ in $G$. These ground states spontaneously break the $G$ symmetry down to $K$. See section 7.1.1 for more details.[20] We note that the ground state (2.53) is obtained by applying the projector $\bigotimes_i \widehat{\mathsf{h}}_i$ to the product state $\bigotimes_i |g^{(\mu)}\rangle_i$, and in particular, it is not normalized. Similar comments also apply to the ground states of the other gapped phases that we will discuss later.

---

[19]Equation (2.49) is a special case of (2.6). In particular, $\widehat{\mathsf{h}}_{ij}$ is obtained by choosing $\chi = \delta_{g_i^{-1} g_j \in K}$ in (2.7).

[20]In section 7, we consider the models in which the constraint $\widehat{\mathsf{h}}_{ij} = 1$ is imposed exactly at the level of the state space, rather than energetically. The same comment applies to the other models that we will discuss later in this section.

**Gauged Model.** Now, we gauge the $H$ symmetry in the above model. The state space of the gauged model is spanned by all possible configurations of the following dynamical variables:

- The plaquette degrees of freedom (i.e., matter fields) are labeled by the representatives of right $H$-cosets in $G$.

- The edge degrees of freedom (i.e., gauge fields) are labeled by elements of $H$ and must satisfy the constraint

$$h_{ij}h_{jk} = h_{ik}, \qquad \forall [ijk] \in V, \tag{2.54}$$

where $h_{ij} \in H$ denotes the dynamical variable on $[ij] \in E$.

More concisely, the state space can be written as[21]

$$\mathcal{H}_{\text{gauged}} = \bigotimes_{i \in P} \mathbb{C}^{|H \backslash G|} \otimes \widehat{\pi}_{\text{flat}} \left( \bigotimes_{[ij] \in E} \mathbb{C}^{|H|} \right), \tag{2.55}$$

where $\widehat{\pi}_{\text{flat}}$ is the projection to the subspace in which the gauge fields satisfy (2.54).

The Hamiltonian of the gauged model is then given by

$$H_{\text{gauged}} = -\sum_{i \in P} \widehat{\mathsf{h}}_i - \sum_{[ij] \in E} \widehat{\mathsf{h}}_{ij}, \tag{2.56}$$

where $\widehat{\mathsf{h}}_i$ and $\widehat{\mathsf{h}}_{ij}$ are defined by

$$\widehat{\mathsf{h}}_i \left| \begin{array}{c} h_{48} \\ h_{46} \quad M^{g_4} \quad h_{47} \\ h_{24} \qquad\quad h_{34} \\ h_{14} \end{array} \right\rangle = \frac{1}{|K|} \sum_{g_5 \in S_{H \backslash G}} \sum_{h_{45} \in H} \delta_{g_4^{-1} h_{45} g_5 \in K} \left| \begin{array}{c} h_{58} \\ h_{56} \quad M^{g_5} \quad h_{57} \\ h_{25} \qquad\quad h_{35} \\ h_{15} \end{array} \right\rangle, \tag{2.57}$$

$$\widehat{\mathsf{h}}_{ij} \left| \begin{array}{c} g_j \\ g_i \end{array} h_{ij} \right\rangle = \delta_{g_i^{-1} h_{ij} g_j \in K} \left| \begin{array}{c} g_j \\ g_i \end{array} h_{ij} \right\rangle. \tag{2.58}$$

Here, we defined $h_{i5} := h_{i4}h_{45}$ and $h_{5j} := h_{45}^{-1}h_{4j}$ for $i = 1, 2, 3$ and $j = 6, 7, 8$.

The above model is solvable because the Hamiltonian is a sum of commuting projectors. As we will see in section 7.3.1, the ground states on an infinite plane, again satisfying $\widehat{\mathsf{h}}_i = 1$ and $\widehat{\mathsf{h}}_{ij} = 1$ for all $i \in P$ and $[ij] \in E$, are given by

$$|\text{GS}; \mu\rangle_{\text{gauged}} = \sum_{\{g_i \in S_{H \backslash G}\}} \sum_{\{h_i \in H\}} \sum_{\{k_i \in K\}} \prod_{i \in P} \frac{1}{|K|} \delta_{k_i, (g^{(\mu)})^{-1} h_i g_i} |\{g_i, h_i^{-1} h_j\}\rangle, \tag{2.59}$$

---

[21]In general, there is an anadditional projector $\widehat{\pi}_{\text{LW}}$ as in (2.12), but this is the identity in the present case.

where $\mu = 1, 2, \cdots, |H\backslash G/K|$, and $g^{(\mu)}$ is a representative of the $(H, K)$-double coset $Hg^{(\mu)}K$ in $G$. We note that (2.59) reduces to (2.53) when $H$ is trivial.

We can also construct commuting projector Hamiltonians for the gapped phases with non-trivial $\nu \in Z^3(H, U(1))$ and $\lambda \in Z^3(K, U(1))$. These Hamiltonians are defined on the same state space $\mathcal{H}_{\text{gauged}}$. We refer the reader to section 7.3.1 for the details of these models. The ground states on an infinite plane turn out to be

$$|\text{GS}; \mu\rangle_{\text{gauged}} = \sum_{\{g_i \in S_{H\backslash G}\}} \sum_{\{h_i \in H\}} \sum_{\{k_i \in K\}} \prod_{i \in P} \frac{1}{|K|} \delta_{k_i, (g^{(\mu)})^{-1} h_i g_i} \prod_{[ijk] \in V} \left( \frac{\lambda(k_i, k_{ij}, k_{jk})}{\nu(h_i, h_{ij}, h_{jk})} \right)^{s_{ijk}} |\{g_i, h_{ij}\}\rangle,$$

(2.60)

where $h_{ij} := h_i^{-1} h_j$ and $k_{ij} := k_i^{-1} k_j$. Here, $g^{(\mu)}$ is again a representative of the $(H, K)$-double coset $Hg^{(\mu)}K$, and $s_{ijk}$ is the sign defined by (2.33). See (7.94) for the more general case where $G$ can be anomalous.

**SPT Phases.** The minimal gapped phases with minimal symmetries include SPT phases as a special case. See section 7.3.3 for more details on the specialization to SPT phases. It is natural to expect that all 2+1d non-chiral SPT phases with fusion 2-category symmetries are minimal gapped phases with minimal symmetries, based on the classification of fiber 2-functors of fusion 2-categories [115]. This is opposed to the 1+1d case, where non-group-theoretical fusion 1-categories can also admit SPT phases [26,116]. See, e.g., [95,100,102,103,106,117–119] for more details on 1+1d SPT phases with non-invertible symmetries. See also [79,80,120] for some examples of 2+1d SPT phases with non-invertible symmetries on the lattice.

**Example:** $G = \mathbb{Z}_4$. We will discuss non-trivial examples with genuinely non-invertible symmetries later on in this section. However, to illustrate this general framework, we start with a simple example that is well-known, and connect it to the construction here, namely minimal gapped phases with minimal symmetries arising from $G = \mathbb{Z}_4$. The SymTFT continuum discussion can be found in [1].

When $G = \mathbb{Z}_4 = \langle b | b^4 = 1 \rangle$, the possible choices of $H$ and $K$ are

$$\{1\}, \quad \mathbb{Z}_2 = \{1, b^2\}, \quad \mathbb{Z}_4 = \{1, b, b^2, b^3\}. \tag{2.61}$$

If we pick $H = \{1\}, \mathbb{Z}_2, \mathbb{Z}_4$, the symmetry of the gauged model becomes $2\mathsf{Vec}_{\mathbb{Z}_4}$, $2\mathsf{Vec}^{\omega}_{\mathbb{Z}_2^{(0)} \times \mathbb{Z}_2^{(1)}}$ (i.e. $\mathbb{Z}_2$ 0- and 1-form symmetries with a mixed anomaly), and $2\mathsf{Rep}(\mathbb{Z}_4)$, respectively. Depending on $H$, the different choices for $K$ result in 0-form SSB phase, trivial phase, or topologically ordered phase (when the 1-form symmetry is SSB'ed), or a mixture thereof.

In what follows, we discuss the minimal gapped phases for all choices of $K$, focusing on $H = \mathbb{Z}_2$ so that the symmetry category is the zero-form and 1-form symmetry

$$2\mathsf{Vec}^{\omega}_{\mathbb{Z}_2^{(0)} \times \mathbb{Z}_2^{(1)}} \tag{2.62}$$

with mixed anomaly $\omega$. For simplicity, we choose $\nu$ and $\lambda$ to be trivial. For any choice of $K$, the state space of the gauged model is given by

$$\mathcal{H}_{\text{gauged}} = \bigotimes_{i \in P} \mathbb{C}^2 \otimes \widehat{\pi}_{\text{flat}} \left( \bigotimes_{[ij] \in E} \mathbb{C}^2 \right). \tag{2.63}$$

Namely, we have a qubit on each plaque and each edge. The qubit on each plaquette is labeled by an element of $S_{H \backslash G} = \{1, b\}$,[22] while the qubit on each edge is labeled by an element of $H = \{1, b^2\}$.

- $K = \{1\}$: This choice of $K$ corresponds to the $\mathbb{Z}_2$-gauging of the full $\mathbb{Z}_4$ SSB phase. The gauged model is in the gapped phase where $\mathbb{Z}_2^{(0)}$ symmetry is spontaneously broken, while $\mathbb{Z}_2^{(1)}$ symmetry is preserved. The Hamiltonian of the gauged model is given by $H_{\text{gauged}} = -\sum_{i \in P} \widehat{\mathsf{h}}_i - \sum_{[ij] \in E} \widehat{\mathsf{h}}_{[ij]}$, where

$$\widehat{\mathsf{h}}_i = 1, \qquad \widehat{\mathsf{h}}_{ij} = \frac{1 + Z_i Z_j}{2} \otimes \frac{1 + Z_{ij}}{2}. \tag{2.64}$$

Here, $Z_i$ and $Z_{ij}$ denote the Pauli-$Z$ operators acting on plaquette $i$ and edge $[ij]$ as

$$Z_i |1\rangle_i = |1\rangle_i, \quad Z_i |b\rangle_i = -|b\rangle_i, \quad Z_{ij} |1\rangle_{ij} = |1\rangle_{ij}, \quad Z_{ij} |b^2\rangle_{ij} = -|b^2\rangle_{ij}. \tag{2.65}$$

The ground states are given by the tensor product of the $\mathbb{Z}_2^{(0)}$ SSB states on the plaquettes and the $\mathbb{Z}_2^{(1)}$-symmetric trivial state on the edges. More explicitly, the two ground states on an infinite plane are given by

$$|\text{GS}; 1\rangle = |\{1\}\rangle_{\text{plaquettes}} \otimes |\{1\}\rangle_{\text{edges}}, \tag{2.66}$$

$$|\text{GS}; 2\rangle = |\{b\}\rangle_{\text{plaquettes}} \otimes |\{1\}\rangle_{\text{edges}}, \tag{2.67}$$

where $|\{x\}\rangle_{\text{plaquettes}}$ and $|\{y\}\rangle_{\text{edges}}$ for $x \in \{1, b\}$ and $y \in \{1, b^2\}$ are defined by

$$|\{x\}\rangle_{\text{plaquettes}} := \bigotimes_{i \in P} |x\rangle_i, \qquad |\{y\}\rangle_{\text{edges}} := \bigotimes_{[ij] \in E} |y\rangle_{ij}. \tag{2.68}$$

---

[22]We can equally choose $S_{H \backslash G} = \{1, b^3\}$.

- $K = \mathbb{Z}_2$: This choice of $K$ corresponds to the $\mathbb{Z}_2$-gauging of the $\mathbb{Z}_4/\mathbb{Z}_2$ SSB phase. The gauged model is in the gapped phase where both $\mathbb{Z}_2^{(0)}$ and $\mathbb{Z}_2^{(1)}$ symmetries are spontaneously broken. The Hamiltonian of the gauged model is given by $H_{\text{gauged}} = -\sum_{i \in P} \widehat{h}_i - \sum_{[ij] \in E} \widehat{h}_{[ij]}$, where

$$\widehat{h}_i = \frac{1}{2}\left(1 + \prod_{e \in \partial i} X_e\right), \qquad \widehat{h}_{ij} = \frac{1 + Z_i Z_j}{2}. \tag{2.69}$$

Here, $\partial i$ is the set of edges on the boundary of plaquette $i$, and $X_e$ denotes the Pauli-$X$ operator acting on $e \in E$ as

$$X_e |1\rangle_e = |b^2\rangle_e, \qquad X_e |b^2\rangle_e = |1\rangle_e. \tag{2.70}$$

The ground states are given by the tensor product of the $\mathbb{Z}_2^{(0)}$ SSB states on the plaquettes and the $\mathbb{Z}_2^{(1)}$ SSB state (i.e., the Toric Code ground state [63]) on the edges. More explicitly, the two ground states on an infinite plane are given by

$$|\text{GS}; 1\rangle = |\{1\}\rangle_{\text{plaquettes}} \otimes |\text{TC}\rangle_{\text{edges}}, \tag{2.71}$$

$$|\text{GS}; 2\rangle = |\{b\}\rangle_{\text{plaquettes}} \otimes |\text{TC}\rangle_{\text{edges}}, \tag{2.72}$$

where $|\text{TC}\rangle_{\text{edges}}$ is the Toric Code ground state defined (up to normalization) by

$$|\text{TC}\rangle_{\text{edges}} := \sum_{\{h_i = 1, b^2\}} \bigotimes_{[ij] \in E} |h_i^{-1} h_j\rangle_{ij}. \tag{2.73}$$

- $K = \mathbb{Z}_4$: This choice of $K$ corresponds to the $\mathbb{Z}_2$-gauging of the trivial $\mathbb{Z}_4$-symmetric phase. The gauged model is in the gapped phase where $\mathbb{Z}_2^{(0)}$ symmetry is preserved, while $\mathbb{Z}_2^{(1)}$ symmetry is spontaneously broken. The Hamiltonian of the gauged model is given by $H_{\text{gauged}} = -\sum_{i \in P} \widehat{h}_i - \sum_{[ij] \in E} \widehat{h}_{[ij]}$, where

$$\widehat{h}_i = \frac{1 + X_i}{2} \otimes \frac{1 + \prod_{e \in \partial i} X_e}{2}, \qquad \widehat{h}_{ij} = 1. \tag{2.74}$$

Here, $X_i$ denotes the Pauli-$X$ operator acting on plaquette $i$ as

$$X_i |1\rangle_i = |b\rangle_i, \qquad X_i |b\rangle_i = |1\rangle_i. \tag{2.75}$$

The ground states are given by the tensor product of the $\mathbb{Z}_2^{(0)}$-symmetric trivial state on the plaquettes and the $\mathbb{Z}_2^{(1)}$ SSB state on the edges. More explicitly, the unique ground state on an infinite plane is given by

$$|\text{GS}\rangle = |+\rangle_{\text{plaquettes}} \otimes |\text{TC}\rangle_{\text{edges}}, \tag{2.76}$$

where $|+\rangle_{\text{plaquettes}}$ is the trivial product state on the plaquettes defined by

$$|+\rangle_{\text{plaquettes}} := \bigotimes_{i \in P} |+\rangle_i, \qquad |+\rangle_i := \frac{1}{\sqrt{2}}\left(|1\rangle_i + |b\rangle_i\right). \tag{2.77}$$

### 2.3.2 Minimal Gapped Phases with Non-Minimal Symmetry

Minimal gapped phases with non-minimal symmetries are obtained by the non-minimal gauging of $G$-symmetric gapped phases without topological orders. The corresponding choice of $A$ and $B$ is

$$A: \text{arbitrary}, \qquad B = \mathsf{Vec}_K^\lambda. \tag{2.78}$$

In what follows, we suppose that $A$ is faithfully graded by $H$, which is the gauged subgroup of $G$. For simplicity, we will focus on the case where $\lambda$ is trivial for the moment. The case of non-trivial $\lambda$ will be briefly mentioned at the end of this subsection and will be detailed in section 7.3.1.

**$G$-Symmetric Model.** The $G$-symmetric model before gauging is the same as the one we used in section 2.3.1. Specifically, the state space and the Hamiltonian of the model are given by

$$\mathcal{H}_{\text{original}} = \bigotimes_{i \in P} \mathbb{C}[G], \tag{2.79}$$

$$H_{\text{original}} = -\sum_{i \in P} \widehat{\mathsf{h}}_i - \sum_{[ij] \in E} \widehat{\mathsf{h}}_{ij}, \tag{2.80}$$

where $\widehat{\mathsf{h}}_i$ and $\widehat{\mathsf{h}}_{ij}$ are defined by (2.50) and (2.51), respectively. The ground states on an infinite plane are given by (2.53). The symmetry $G$ is spontaneously broken down to $K$ in these ground states.

**Gauged Model.** As discussed in section 2.2.2, the state space of the gauged model is given by

$$\mathcal{H}_{\text{gauged}} = \widehat{\pi}_{\text{LW}} \mathcal{H}'_{\text{gauged}}. \tag{2.81}$$

Here, $\widehat{\pi}_{\text{LW}}$ is the projector

$$\widehat{\pi}_{\text{LW}} = \prod_{i \in P} \widehat{B}_i, \tag{2.82}$$

where $\widehat{B}_i$ is the Levin-Wen plaquette operator defined by (2.15). The vector space $\mathcal{H}'_{\text{gauged}}$ is spanned by all possible configurations of the following dynamical variables:

- The dynamical variables on the plaquettes are labeled by elements of $S_{H \backslash G}$.

- The dynamical variables on the edges are labeled by simple objects of $A^{\text{rev}}$. These dynamical variables must obey the constraint

$$a_{ik} \in a_{ij} \otimes a_{jk}, \qquad \forall [ijk] \in V, \tag{2.83}$$

where $a_{ij}$ is the dynamical variable on $[ij] \in E$.

Each state of the gauged model is denoted by

$$|\{g_i, a_{ij}\}\rangle = \left| \begin{array}{c} M^{g_l} \\ a_{kl} \\ a_{jl} \\ M^{g_j} \\ a_{ij} \end{array} \begin{array}{c} M^{g_k} \\ a_{jk} \\ a_{ik} \\ M^{g_i} \end{array} \right\rangle, \tag{2.84}$$

where $g_i \in S_{H \backslash G}$ and $a_{ij} \in A^{\mathrm{rev}}$. More concisely, the state space can be written as

$$\mathcal{H}_{\mathrm{gauged}} = \widehat{\pi}_{\mathrm{LW}} \left( \bigotimes_{i \in P} \mathbb{C}^{|H \backslash G|} \otimes \widehat{\pi}_{\mathrm{fusion}} \left( \bigotimes_{[ij] \in E} \mathbb{C}^{\mathrm{rank}(A)} \right) \right), \tag{2.85}$$

where $\mathrm{rank}(A)$ is the number of simple objects of $A$, and $\widehat{\pi}_{\mathrm{fusion}}$ is the projector to the subspace in which the edge degrees of freedom comply with the fusion rules. Here, we assumed that $A$ is multiplicity-free. See section 7.3.1 for the case of more general $A$.

The Hamiltonian of the gauged model is given by

$$H_{\mathrm{gauged}} = -\sum_{i \in P} \widehat{\mathsf{h}}_i - \sum_{[ij] \in P} \widehat{\mathsf{h}}_{ij}, \tag{2.86}$$

where $\widehat{\mathsf{h}}_i$ and $\widehat{\mathsf{h}}_{ij}$ are defined by

$$\widehat{\mathsf{h}}_i \left| \begin{array}{c} M^{g_4} \end{array} \right\rangle = \frac{1}{|K|} \sum_{g_5 \in S_{H \backslash G}} \sum_{h_{45} \in H} \delta_{g_4^{-1} h_{45} g_5 \in K} \sum_{a_{45} \in A_{h_{45}}^{\mathrm{rev}}} \frac{\dim(a_{45})}{\mathcal{D}_{A_{h_{45}}}} \left| \begin{array}{c} M^{g_5} \\ a_{45} \end{array} \right\rangle$$

$$\widehat{\mathsf{h}}_{ij} \left| \begin{array}{c} M^{g_j} \\ \longleftarrow a_{ij} \\ M^{g_i} \end{array} \right\rangle = \delta_{g_i^{-1} h_{ij} g_j \in K} \left| \begin{array}{c} M^{g_j} \\ \longleftarrow a_{ij} \\ M^{g_i} \end{array} \right\rangle, \qquad a_{ij} \in A_{h_{ij}}^{\mathrm{rev}}. \tag{2.87}$$

The right-hand side of (2.87) is evaluated by using the $F$-move of $A^{\mathrm{rev}}$ as in (2.18).

The above model is solvable because the Hamiltonian is a sum of commuting projectors. As we will see in section 7.3.1, the ground states on an infinite plane are given by

$$|\mathrm{GS}; \mu\rangle_{\mathrm{gauged}} = \sum_{\{g_i \in S_{H \backslash G}\}} \sum_{\{k_i \in K\}} \sum_{\{h_i \in H\}} \prod_{i \in P} \frac{1}{|K|} \delta_{k_i, (g^{(\mu)})^{-1} h_i g_i}$$

$$\sum_{\{a_i \in A_{h_i}^{\mathrm{rev}}\}} \sum_{\{a_{ij} \in \overline{a_i} \otimes a_j\}} \prod_{i \in P} \frac{1}{\mathcal{D}_{A_{h_i}}} \prod_{[ijk] \in V} \left( \frac{d_{ij}^a d_{jk}^a}{d_{ik}^a} \right)^{\frac{1}{4}} (\overline{F}_{ijk}^a)^{s_{ijk}} |\{g_i, a_{ij}\}\rangle, \tag{2.88}$$

where $\mu = 1, 2, \cdots, |H \backslash G / K|$, and $g^{(\mu)}$ is a representative of the $(H, K)$-double coset $H g^{(\mu)} K$ in $G$. We recall that $d_{ij}^a$ denotes the quantum dimension of $a_{ij}$, and $(\overline{F}_{ijk}^a)^{s_{ijk}}$ is defined by (2.34). The above equation reduces to (2.59) when $A = \mathsf{Vec}_H$.

More generally, we can also construct commuting projector Hamiltonians for minimal gapped phases with non-trivial $\lambda$ on the same state space $\mathcal{H}_{\text{gauged}}$. The ground states of these models on an infinite plane turn out to be

$$
\begin{aligned}
|\text{GS}; \mu\rangle_{\text{gauged}} = & \sum_{\{g_i \in S_{H \backslash G}\}} \sum_{\{k_i \in K\}} \sum_{\{h_i \in H\}} \prod_{i \in P} \frac{1}{|K|} \delta_{k_i, (g^{(\mu)})^{-1} h_i g_i} \prod_{[ijk] \in V} \lambda(k_i, k_{ij}, k_{jk})^{s_{ijk}} \\
& \sum_{\{a_i \in A_{h_i}^{\text{rev}}\}} \sum_{\{a_{ij} \in \overline{a_i} \otimes a_j\}} \prod_{i \in P} \frac{1}{\mathcal{D}_{A_{h_i}}} \prod_{[ijk] \in V} \left( \frac{d_{ij}^a d_{jk}^a}{d_{ik}^a} \right)^{\frac{1}{4}} (\overline{F}_{ijk}^a)^{s_{ijk}} |\{g_i, a_{ij}\}\rangle,
\end{aligned}
\tag{2.89}
$$

where $k_{ij} := k_i^{-1} k_j$. See (7.87) for more general cases where $G$ can be anomalous and $A$ is not necessarily multiplicity-free.

**Example: $G = \mathbb{Z}_2$ with $A = \text{Ising}$.** Let us discuss some simple examples of the minimal gapped phases with non-minimal symmetries. We consider $G = \mathbb{Z}_2$ and choose $A$ to be the Ising fusion category

$$
\text{Ising} = A_0 \oplus A_1,
\tag{2.90}
$$

which is faithfully graded by $H = \mathbb{Z}_2$. Here, we employ the additive notation $\mathbb{Z}_2 = \{0, 1\}$. The trivial componet $A_0$ has two simple objects $\{\mathbb{1}, \psi\}$, while the non-trivial component $A_1$ has a single simple object $\{\sigma\}$. These objects obey the fusion rules

$$
\psi \otimes \psi \cong \mathbb{1}, \qquad \psi \otimes \sigma \cong \sigma \otimes \psi \cong \sigma, \qquad \sigma \otimes \sigma \cong \mathbb{1} \oplus \psi.
\tag{2.91}
$$

Physically, this choice corresponds to the following generalized gauging: we start from a $\mathbb{Z}_2$-symmetric lattice model, stack the $\mathbb{Z}_2^{\text{em}}$-enriched Toric Code on it, and gauge the diagonal $\mathbb{Z}_2$ subgroup. Here, $\mathbb{Z}_2^{\text{em}}$ denotes the electric-magnetic duality symmetry of the Toric Code, i.e., the $\mathbb{Z}_2$ symmetry that exchanges the $e$-anyon and $m$-anyon. The symmetry of the gauged model is described by

$$
_{\text{Ising}}(2\text{Vec}_{\mathbb{Z}_2})_{\text{Ising}} \cong \Sigma \mathcal{Z}(\text{Ising})_0,
\tag{2.92}
$$

where $\mathcal{Z}(\text{Ising})_0$ is the braided fusion subcategory of $\mathcal{Z}(\text{Ising})$ with five simple objects

$$
\{\mathbb{1}\overline{\mathbb{1}}, \quad \psi\overline{\mathbb{1}}, \quad \mathbb{1}\overline{\psi}, \quad \psi\overline{\psi}, \quad \sigma\overline{\sigma}\}.
\tag{2.93}
$$

Here, $\overline{a} \in \overline{\text{Ising}}$ denotes the time-reversal counterpart of $a \in \text{Ising}$.[23] We recall that $\Sigma$ in (2.92) means the condensation completion [108]. See section 5.3 for the symmetry 2-categories of more general gauged models.

---

[23]We note that $\overline{a}$ here is not the dual of $a$.

Let us consider minimal gapped phases with $B = \mathsf{Vec}_K^\lambda$, where $K$ is either $\{0\}$ or $\mathbb{Z}_2$. When $K = \{0\}$, the choice of $\lambda$ is unique up to coboundary. On the other hand, when $K = \mathbb{Z}_2$, we have two choices of $\lambda$ up to coboundary because $H^3(\mathbb{Z}_2, \mathrm{U}(1)) \cong \mathbb{Z}_2$. Thus, in total, there are three minimal gapped phases with symmetry $\Sigma\mathcal{Z}(\mathsf{Ising})_0$. In what follows, we will discuss the lattice models for all of these gapped phases.

For any choice of $K$ and $\lambda$, the state space of the model is given by

$$\mathcal{H}_{\text{gauged}} = \widehat{\pi}_{\text{fusion}} \left( \bigotimes_{[ij] \in E} \mathbb{C}^3 \right). \tag{2.94}$$

We note that there are no dynamical variables on the plaquettes because $H \backslash G$ is trivial. The edge degrees of freedom are labeled by simple objects of $\mathsf{Ising}^{\text{rev}} \cong \mathsf{Ising}$.[24] The projector $\widehat{\pi}_{\text{fusion}}$ imposes the condition that the edge degrees of freedom obey the fusion rules of $\mathsf{Ising}$ at every vertex.

The Hamiltonian and its ground states for each choice of $(K, \lambda)$ are described as follows:

- $K = \{0\}$, $\lambda = 1$: The Hamiltonian is given by $H_{\text{gauged}} = -\sum_{i \in P} \widehat{\mathsf{h}}_i - \sum_{[ij] \in E} \widehat{\mathsf{h}}_{ij}$, where

$$\widehat{\mathsf{h}}_i = \frac{1}{2} \left( \widehat{L}_i^{(\mathbb{1})} + \widehat{L}_i^{(\psi)} \right), \qquad \widehat{\mathsf{h}}_{ij} |a_{ij}\rangle = \begin{cases} |a_{ij}\rangle & \text{if } a_{ij} = \mathbb{1}, \psi, \\ 0 & \text{if } a_{ij} = \sigma. \end{cases} \tag{2.95}$$

Here, $\widehat{L}_i^{(a)}$ is the loop operator defined by

$$\tag{2.96}$$

which is evaluated by using the $F$-symbols of $\mathsf{Ising}$. In the subspace where $\widehat{\mathsf{h}}_{ij} = 1$ for all $[ij] \in E$, the above model reduces to the Levin-Wen model based on $\mathsf{Vec}_{\mathbb{Z}_2}$ [64], or equivalently, the Toric Code model on a honeycomb lattice [63]. In particular, the ground state subspace of this model agrees with that of the Toric Code model. Thus, the above model realizes the Toric Code topological order, which is described by $\mathcal{Z}(\mathsf{Vec}_{\mathbb{Z}_2})$. The symmetry $\Sigma\mathcal{Z}(\mathsf{Ising})_0$ acts on the Toric Code topological order via the braided monoidal functor

$$\mathcal{Z}(\mathsf{Ising})_0 \to \mathcal{Z}(\mathsf{Vec}_{\mathbb{Z}_2}), \tag{2.97}$$

which maps $\{\mathbb{1}\overline{\mathbb{1}},\ \psi\overline{\mathbb{1}},\ \mathbb{1}\overline{\psi},\ \psi\overline{\psi},\ \sigma\overline{\sigma}\}$ to $\{\mathbb{1},\ f,\ f,\ \mathbb{1},\ e \oplus m\}$, respectively. Here, the simple objects of $\mathcal{Z}(\mathsf{Vec}_{\mathbb{Z}_2})$ are denoted by $\{\mathbb{1}, e, m, f\}$ with $f$ being a fermion.

---

[24]This is a monoidal equivalence, not a braided monoidal equivalence.

- $K = \mathbb{Z}_2$, $\lambda = 1$: The Hamiltonian is given by $H_{\text{gauged}} = -\sum_{i \in P} \widehat{\mathsf{h}}_i - \sum_{[ij] \in E} \widehat{\mathsf{h}}_{ij}$, where

$$\widehat{\mathsf{h}}_i = \frac{1}{4}\left(\widehat{L}_i^{(\mathbb{1})} + \widehat{L}_i^{(\psi)} + \sqrt{2}\widehat{L}_i^{(\sigma)}\right), \qquad \widehat{\mathsf{h}}_{ij} = 1. \tag{2.98}$$

Here, $\widehat{L}_i^{(a)}$ is the loop operator defined by (2.96). This is the Levin-Wen model based on $\mathsf{Ising}$, and hence realizes the doubled Ising topological order $\mathcal{Z}(\mathsf{Ising})$. The symmetry $\Sigma\mathcal{Z}(\mathsf{Ising})_0$ acts on the doubled Ising topological order via the canonical embedding

$$\mathcal{Z}(\mathsf{Ising})_0 \hookrightarrow \mathcal{Z}(\mathsf{Ising}). \tag{2.99}$$

We note that the topological order $\mathcal{Z}(\mathsf{Ising})$ is a minimal non-degenerate extension of the non-invertible 1-form symmetry $\mathcal{Z}(\mathsf{Ising})_0$.

- $K = \mathbb{Z}_2, [\lambda] \neq [1]$: The Hamiltonian is given by $H_{\text{gaugued}} = -\sum_{i \in P} \widehat{\mathsf{h}}_i - \sum_{[ij] \in E} \widehat{\mathsf{h}}_{ij}$, where

$$\widehat{\mathsf{h}}_i = \frac{1}{4}\left(\widehat{R}_i^{(\mathbb{1})} + \widehat{R}_i^{(\psi)} + \sqrt{2}\widehat{R}_i^{(\sigma)}\right), \qquad \widehat{\mathsf{h}}_{ij} = 1. \tag{2.100}$$

The loop operator $\widehat{R}_i^{(a)}$ is defined by the same diagram as (2.96), which is now evaluated by using the $F$-symbols of $\mathsf{Rep}(\mathrm{SU}(2)_2)$ rather than those of $\mathsf{Ising}$.[25] Here, by slight abuse of notation, we denote the simple objects of $\mathsf{Rep}(\mathrm{SU}(2)_2)$ by the same letters as those of $\mathsf{Ising}$. The above model is the Levin-Wen model based on $\mathsf{Rep}(\mathrm{SU}(2)_2)$, and hence realizes the topological order described by $\mathcal{Z}(\mathsf{Rep}(\mathrm{SU}(2)_2))$. The symmerty $\Sigma\mathcal{Z}(\mathsf{Ising})_0$ acts on this topological order via the embedding

$$\mathcal{Z}(\mathsf{Ising})_0 \xrightarrow{\cong} \mathcal{Z}(\mathsf{Rep}(\mathrm{SU}(2)_2))_0 \hookrightarrow \mathcal{Z}(\mathsf{Rep}(\mathrm{SU}(2)_2)), \tag{2.101}$$

where $\mathcal{Z}(\mathsf{Rep}(\mathrm{SU}(2)_2))_0$ is the braided fusion subcategory of $\mathcal{Z}(\mathsf{Rep}(\mathrm{SU}(2)_2))$ with five simple objects (2.93). The braided equivalence $\mathcal{Z}(\mathsf{Ising})_0 \cong \mathcal{Z}(\mathsf{Rep}(\mathrm{SU}(2)_2))_0$ maps the simple objects of $\mathcal{Z}(\mathsf{Ising})_0$ to those of $\mathcal{Z}(\mathsf{Rep}(\mathrm{SU}(2)_2))_0$ labeled by the same letters. We note that the topological order $\mathcal{Z}(\mathsf{Rep}(\mathrm{SU}(2)_2))$ is a minimal non-degenerate extension of the non-invertible 1-form symmetry $\mathcal{Z}(\mathsf{Ising})_0$.

We will discuss another example in section 2.3.5 which originates from $G = S_3$ and has a $2\mathsf{Rep}(\mathbb{G})$ symmetry.

---

[25]The category $\mathsf{Rep}(\mathrm{SU}(2)_2)$ has the same fusion rules as $\mathsf{Ising}$, but have different $F$-symbols. We note that $\mathsf{Rep}(\mathrm{SU}(2)_2)$ and $\mathsf{Ising}$ are the only fusion categories with three simple objects satisfying the fusion rules (2.91). These fusion categories are known as $\mathbb{Z}_2$ Tambara-Yamagami categories [121].

### 2.3.3 Non-Minimal Gapped Phases with Minimal Symmetry

Non-minimal gapped phases with minimal symmetries are obtained by the ordinary (twisted) gauging of $G$-symmetric gapped phases with topological orders. We note that the $G$ symmetry before gauging can be spontaneously broken. The corresponding choice of $A$ and $B$ is

$$A = \mathsf{Vec}_H^\nu, \qquad B : \text{arbitrary}. \tag{2.102}$$

In what follows, we suppose that $B$ is faithfully graded by $K$, which is the unbroken subgroup of $G$. For simplicity, we will focus on the case where $\nu$ is trivial for the moment. The case of non-trivial $\nu$ will be briefly mentioned at the end of this subsection and will be detailed in section 7.3.4.

**$G$-Symmetric Model.** Let us first define the $G$-symmetric lattice model before gauging. The state space of the model is given by

$$\mathcal{H}_{\text{original}} = \bigotimes_{i \in P} \mathbb{C}[G] \otimes \widehat{\pi}_{\text{fusion}} \left( \bigotimes_{[ij] \in E} \mathbb{C}^{\text{rank}(B)} \right). \tag{2.103}$$

More explicitly, the dynamical variables of the model can be described as follows:

- The dynamical variables on the plaquettes are labeled by elements of $G$.

- The dynamical variables on the edges are labeled by simple objects of $B$. These dynamical variables must satisfy the constraint

$$b_{ik} \in b_{ij} \otimes b_{jk}, \qquad \forall [ijk] \in V, \tag{2.104}$$

where $b_{ij} \in B$ is the dynamical variable on $[ij] \in E$.

Each state of this model is denoted by

$$|\{g_i, b_{ij}\}\rangle = \left| \begin{array}{c} \includegraphics \end{array} \right\rangle. \tag{2.105}$$

Here, we assumed that $B$ is multiplicity-free. See section 7.1.3 for the case of more general $B$.[26] We note that the constraint $\widehat{\pi}_{\text{fusion}} = 1$ can also be imposed energetically without explicitly breaking the $G$ symmetry.

---

[26] When $B$ is not multiplicity-free, we have extra degrees of freedom on the vertices. These dynamical variables are labeled by morphisms of $B$.

The Hamiltonian of the model is given by

$$H_{\text{original}} = -\sum_{i \in P} \widehat{\mathsf{h}}_i - \sum_{[ij] \in E} \widehat{\mathsf{h}}_{ij}, \tag{2.106}$$

where $\widehat{\mathsf{h}}_i$ and $\widehat{\mathsf{h}}_{ij}$ are defined by

 $$\tag{2.107}$$

 $$\tag{2.108}$$

Here, $\mathcal{D}_B$ is the total dimension of $B$. The diagram on the right-hand side of (2.107) is evaluated by applying the $F$-move of $B$ succesively. One can write (2.107) more explicitly as

 $$\tag{2.109}$$

where $d_{ij}^b$ is the quantum dimension of $b_{ij}$, and $F_{ijkl}^b$ is the $F$-symbol of $B$, which is defined in the same way as (2.19).

The above model is solvable because the Hamiltonian is a sum of commuting projectors. As we will see in section 7.1.3, the ground states on an infinite plane are given by

$$|\text{GS}; \mu\rangle_{\text{original}} = \sum_{\{k_i \in K\}} \sum_{\{b_i \in B_{k_i}\}} \sum_{\{b_{ij} \in \overline{b_i} \otimes b_j\}} \prod_{i \in P} \frac{1}{\mathcal{D}_B} \prod_{[ijk] \in V} \left( \frac{d_{ij}^b d_{jk}^b}{d_{ik}^b} \right)^{\frac{1}{4}} \prod_{[ijk] \in V} (F_{ijk}^b)^{s_{ijk}} \, |\{g^{(\mu)} k_i, b_{ij}\}\rangle, \tag{2.110}$$

where $\mu = 1, 2, \cdots, |G/K|$, and $g^{(\mu)}$ is a representative of the left $K$-coset $g^{(\mu)} K$ in $G$. The $F$-symbols $(F_{ijk}^b)^{s_{ijk}}$ are defined in the same way as (2.38). We note that the above ground states spontaneously break the $G$ symmetry down to $K$. Each vacuum realizes the same symmetry-enriched topological order as that of the symmetry-enriched string-net model [70, 71].

**Gauged Model.** Now, we gauge the $H$ symmetry in the above model. The state space of the gauged model is given by

$$\mathcal{H}_{\text{gauged}} = \widehat{\pi}_{\text{LW}} \mathcal{H}'_{\text{gauged}}, \tag{2.111}$$

where $\widehat{\pi}_{\text{LW}}$ is trivial in this case, and $\mathcal{H}'_{\text{gauged}}$ is spanned by all possible configurations of the following dynamical variables:

- The dynamical variables on the plaquettes are labeled by elements of $S_{H \backslash G}$.

- The dynamical variables on the edges are labeled by pairs $(h_{ij}, b_{ij})$, where $h_{ij}$ is an element of $H$ and $b_{ij}$ is a simple object of $B$. These dynamical variables must satisfy the constraint

$$h_{ik} = h_{ij} h_{jk}, \qquad b_{ik} \in b_{ij} \otimes b_{jk}, \qquad \forall [ijk] \in V, \tag{2.112}$$

  where $h_{ij}$ and $b_{ij}$ are the dynamical variables on $[ij] \in E$.

Each state of the gauged model is denoted by

$$|\{g_i, (h, b)_{ij}\}\rangle = \left| \begin{array}{c} M^{g_l} \\ (h,b)_{kl} \\ (h,b)_{jl} \quad M^{g_k} \\ (h,b)_{jk} \\ M^{g_j} \quad (h,b)_{ik} \\ (h,b)_{ij} \quad M^{g_i} \end{array} \right\rangle. \tag{2.113}$$

where $g_i \in S_{H \backslash G}$ and $(h, b)_{ij} := (h_{ij}, b_{ij})$ for $h_{ij} \in H$ and $b_{ij} \in B$. More concisely, the state space can be written as

$$\mathcal{H}_{\text{gauged}} = \bigotimes_{i \in P} \mathbb{C}^{|H \backslash G|} \otimes \widehat{\pi}_{\text{flat}} \left( \bigotimes_{[ij] \in E} \mathbb{C}^{|H|} \right) \otimes \widehat{\pi}_{\text{fusion}} \left( \bigotimes_{[ij] \in E} \mathbb{C}^{\text{rank}(B)} \right), \tag{2.114}$$

where $\widehat{\pi}_{\text{flat}}$ and $\widehat{\pi}_{\text{fusion}}$ are the projectors to the subspace in which (2.112) is satisfied.

The Hamiltonian of the gauged model is given by

$$H_{\text{gauged}} = -\sum_{i \in P} \widehat{\mathsf{h}}_i - \sum_{[ij] \in E} \widehat{\mathsf{h}}_{ij}, \tag{2.115}$$

where $\widehat{\mathsf{h}}_i$ and $\widehat{\mathsf{h}}_{ij}$ are defined by

$$\widehat{\mathsf{h}}_i \left| \begin{array}{c} M^{g_4} \end{array} \right\rangle = \sum_{g_5 \in S_{H \backslash G}} \sum_{h_{45} \in H} \delta_{g_4^{-1} h_{45} g_5 \in K} \sum_{b_{45} \in B_{g_4^{-1} h_{45} g_5}} \frac{\dim(b_{45})}{\mathcal{D}_B} \left| \begin{array}{c} M^{g_5} \\ (h,b)_{45} \end{array} \right\rangle, \tag{2.116}$$

$$\widehat{\mathsf{h}}_{ij} \left| \begin{array}{c} M^{g_j} \\ \overleftarrow{\phantom{xx}} \\ M^{g_i} \end{array} (h,b)_{ij} \right\rangle = \delta_{g_i^{-1}h_{ij}g_j, k_{ij}} \left| \begin{array}{c} M^{g_j} \\ \overleftarrow{\phantom{xx}} \\ M^{g_i} \end{array} (h,b)_{ij} \right\rangle, \qquad b_{ij} \in B_{k_{ij}}. \tag{2.117}$$

Here, we recall that $(h,b)_{ij}$ is an abbreviation of $(h_{ij}, b_{ij})$.

The above model is solvable because the Hamiltonian is a sum of commuting projectors. As we will see in section 7.3.4, the ground states on an infinite plane are given by

$$|\text{GS}; \mu\rangle_{\text{gauged}} = \sum_{\{g_i \in S_{H\backslash G}\}} \sum_{\{h_i \in H\}} \sum_{\{k_i \in K\}} \prod_{i \in P} \delta_{k_i, (g^{(\mu)})^{-1} h_i g_i} \prod_{i \in P} \frac{1}{\mathcal{D}_B}$$
$$\sum_{\{b_i \in B_{k_i}\}} \sum_{\{b_{ij} \in \overline{b_i} \otimes b_j\}} \prod_{[ijk] \in V} \left( \frac{d_{ij}^b d_{jk}^b}{d_{ik}^b} \right)^{\frac{1}{4}} \prod_{[ijk] \in V} (F_{ijk}^b)^{s_{ijk}} |\{g_i, (h,b)_{ij}\}\rangle, \tag{2.118}$$

where $h_{ij} := h_i^{-1} h_j$, and $g^{(\mu)}$ is a representative of the $(H, K)$-double coset $Hg^{(\mu)}K$ with $\mu = 1, 2, \cdots, |H\backslash G/K|$.

More generally, we can also construct commuting projector Hamiltonians for non-minimal gapped phases with non-trivial $\nu \in Z^3(H, \text{U}(1))$ on the same state space. The ground states of these models on an infinite plane turn out to be

$$|\text{GS}; \mu\rangle_{\text{gauged}} = \sum_{\{g_i \in S_{H\backslash G}\}} \sum_{\{h_i \in H\}} \sum_{\{k_i \in K\}} \prod_{i \in P} \delta_{k_i, (g^{(\mu)})^{-1} h_i g_i} \prod_{i \in P} \frac{1}{\mathcal{D}_B} \prod_{[ijk] \in V} \nu(h_i, h_{ij}, h_{jk})^{-s_{ijk}}$$
$$\sum_{\{b_i \in B_{k_i}\}} \sum_{\{b_{ij} \in \overline{b_i} \otimes b_j\}} \prod_{[ijk] \in V} \left( \frac{d_{ij}^b d_{jk}^b}{d_{ik}^b} \right)^{\frac{1}{4}} \prod_{[ijk] \in V} (F_{ijk}^b)^{s_{ijk}} |\{g_i, (h,b)_{ij}\}\rangle. \tag{2.119}$$

See (7.119) for more general cases where $G$ can be anomalous and $B$ is non necessarily multiplicity-free.[27]

**Example: $G = \mathbb{Z}_2$ with $B = \text{Ising}$.** Let us discuss simple examples of non-minimal gapped phases with minimal symmetries. We consider $G = \mathbb{Z}_2$ and choose $B$ to be the Ising fusion category

$$\text{Ising} = B_0 \oplus B_1, \tag{2.120}$$

which is faithfully graded by $K = \mathbb{Z}_2$. Here, we denote $\mathbb{Z}_2$ additively, i.e., $\mathbb{Z}_2 = \{0, 1\}$. The simple objects of $B_0$ and $B_1$ are denoted by $\{\mathbb{1}, \psi\}$ and $\{\sigma\}$, which obey the fusion rules (2.91). This choice of $B$ corresponds to the $\mathbb{Z}_2^{\text{em}}$-enriched Toric Code before gauging.

In what follows, we consider the minimal gauging of the above non-minimal gapped phase. The minimal gauging is specified by $A = \text{Vec}_H^\nu$, where $H$ is either $\{0\}$ or $\mathbb{Z}_2$. The choice of $\lambda$

---

[27]Euation (7.119) is the most general case where $A$ is also arbitrary.

is unique when $H = \{0\}$, while there are two choices of $\lambda$ when $H = \mathbb{Z}_2$. Thus, in total, there are three options for the minimal gauging.

- $H = \{0\}, \lambda = 1$: This choice of $H$ corresponds to gauging nothing. Therefore, the corresponding non-minimal gapped phase remains to be the $\mathbb{Z}_2^{\mathrm{em}}$-enriched Toric Code. For completeness, let us provide the lattice model for this phase.

  The state space of the model is given by

  $$\mathcal{H}_{\mathrm{gauged}} = \bigotimes_{i \in P} \mathbb{C}^2 \otimes \widehat{\pi}_{\mathrm{fusion}} \left( \bigotimes_{[ij] \in E} \mathbb{C}^3 \right). \tag{2.121}$$

  Namely, we have a qubit and a qutrit on each plaquette and each edge, respectively. Each state of the qutrit is labeled by a simple object of Ising. The projector $\widehat{\pi}_{\mathrm{fusion}}$ imposes the condition that these qutrits obey the fusion rules of Ising at every vertex.

  The Hamiltonian is given by $H_{\mathrm{gauged}} = -\sum_{i \in P} \widehat{\mathsf{h}}_i - \sum_{[ij] \in E} \widehat{\mathsf{h}}_{ij}$, where

  $$\widehat{\mathsf{h}}_i = \frac{1}{4} \left( \widehat{L}_i^{(\mathbb{1})} + \widehat{L}_i^{(\psi)} + \sqrt{2} \widehat{X}_i \widehat{L}_i^{(\sigma)} \right), \tag{2.122}$$

  $$\widehat{\mathsf{h}}_{ij} |g_i, g_j, b_{ij}\rangle = \begin{cases} |g_i, g_j, b_{ij}\rangle & \text{if } (g_i^{-1} g_j, b_{ij}) = (0, \mathbb{1}),\ (0, \psi),\ (1, \sigma), \\ 0 & \text{otherwise.} \end{cases} \tag{2.123}$$

  Here, $\widehat{X}_i$ is the Pauli-X operator acting on the qubit on plaquette $i$, and $\widehat{L}_i^{(b)}$ is the loop operaror defined by

  $$\widehat{L}_i^{(b)} \left| \text{} \right\rangle = \left| \text{} \right\rangle. \tag{2.124}$$

  The right-hand side is evaluated by using the $F$-symbols of Ising. We note that $\widehat{L}_i^{(b)}$ acts only on the edge degrees of freedom. The above model is the $\mathbb{Z}_2$-enriched string-net model based on Ising [70, 71], which is known to have the $\mathbb{Z}_2^{\mathrm{em}}$-enriched Toric Code topological order.

- $H = \mathbb{Z}_2, \nu = 1$: This choice of $H$ and $\nu$ corresponds to the ordinary gauging of $\mathbb{Z}_2^{\mathrm{em}}$ symmetry without a discrete torsion. The gauged model realizes a non-minimal gapped phase with $2\mathsf{Rep}(\mathbb{Z}_2)$ symmetry. We will describe the lattice model for this phase below.

The state space of the gauged model is

$$\mathcal{H}_{\text{gauged}} = \widehat{\pi}_{\text{flat}} \left( \bigotimes_{[ij] \in E} \mathbb{C}^2 \right) \otimes \widehat{\pi}_{\text{fusion}} \left( \bigotimes_{[ij] \in E} \mathbb{C}^3 \right). \tag{2.125}$$

Namely, we have a pair of a qubit and a qutrit on each edge, which are labeled by an element of $H$ and a simple object of $\mathsf{Ising}$, respectively. These qubits and qutrits obey the flatness condition and the fusion constraint at every vertex, due to the projectors $\widehat{\pi}_{\text{flat}}$ and $\widehat{\pi}_{\text{fusion}}$.

The Hamiltonian of the gauged model is given by $H_{\text{gauged}} = -\sum_{i \in P} \widehat{\mathsf{h}}_i - \sum_{[ij] \in E} \widehat{\mathsf{h}}_{ij}$, where

$$\widehat{\mathsf{h}}_i = \frac{1}{4} \left( \widehat{L}_i^{(\mathbb{1})} + \widehat{L}_i^{(\psi)} + \sqrt{2} \widehat{L}_i^{(\sigma)} \right), \tag{2.126}$$

$$\widehat{\mathsf{h}}_{ij} |(h,b)_{ij}\rangle = \begin{cases} |(h,b)_{ij}\rangle & \text{if } (h,b)_{ij} = (0,\mathbb{1}), \ (0,\psi), \ (1,\sigma), \\ 0 & \text{otherwise.} \end{cases} \tag{2.127}$$

The first term $\widehat{\mathsf{h}}_i$, which acts only on the qutrits $\{b_{ij} \mid [ij] \in E\}$, agrees with the Hamiltonian of the Levin-Wen model based on $\mathsf{Ising}$ [64]. On the other hand, the second term $\widehat{\mathsf{h}}_{ij}$ fixes the configuration of the qubits $\{h_{ij} \mid [ij] \in E\}$ for a given configuration of the qutrits. Therefore, in the subspace where $\widehat{\mathsf{h}}_{ij} = 1$ for all $[ij] \in E$, the model reduces to the Levin-Wen model based on $\mathsf{Ising}$. This implies that the above model realizes the doubled Ising topological order $\mathcal{Z}(\mathsf{Ising})$.

Since we gauged the full $\mathbb{Z}_2$ symmetry, the gauged model has $2\mathsf{Rep}(\mathbb{Z}_2) = \Sigma\mathsf{Vec}_{\mathbb{Z}_2}$ symmetry, where $\mathsf{Vec}_{\mathbb{Z}_2}$ is equipped with the symmetric braiding. This symmetry acts on the doubled Ising topological order via the braided monoidal functor

$$\mathsf{Vec}_{\mathbb{Z}_2} \to \mathcal{Z}(\mathsf{Ising}), \tag{2.128}$$

which maps the non-trivial simple object of $\mathsf{Vec}_{\mathbb{Z}_2}$ to $\psi\overline{\psi} \in \mathcal{Z}(\mathsf{Ising})$.

- $H = \mathbb{Z}_2, [\nu] \neq [1]$: This choice of $H$ and $\nu$ corresponds to the twisted gauging of $\mathbb{Z}_2^{\text{em}}$ symmetry. The gauged model again realizes a non-minimal gapped phase with $2\mathsf{Rep}(\mathbb{Z}_2)$ symmetry. Let us now describe the lattice model for this gapped phase.

The state space of the gauged model is given by (2.125). On the other hand, the Hamiltonian is given by $H_{\text{gauged}} = -\sum_{i \in P} \widehat{\mathsf{h}}_i - \sum_{[ij] \in E} \widehat{\mathsf{h}}_{ij}$, where

$$\widehat{\mathsf{h}}_i = \frac{1}{4} \left( \widehat{L}_i^{(\mathbb{1})} + \widehat{L}_i^{(\psi)} + \sqrt{2} \widehat{L}_i^{(\sigma)} \right) \otimes \left( \widehat{L}_i^{(h=0)} + \widehat{L}_i^{(h=1)} \right), \tag{2.129}$$

$$\widehat{\mathsf{h}}_{ij} |(h,b)_{ij}\rangle = \begin{cases} |(h,b)_{ij}\rangle & \text{if } (h,b)_{ij} = (0,\mathbb{1}), \ (0,\psi), \ (1,\sigma), \\ 0 & \text{otherwise.} \end{cases} \tag{2.130}$$

Here, $\widehat{L}_i^{(b)}$ for $b \in \mathsf{Ising}$ is defined by (2.124), which acts only on the qutrits $\{b_{ij} \mid [ij] \in E\}$. On the other hand, $\widehat{L}_i^{(h)}$ for $h \in \mathbb{Z}_2$ is defined by

$$
\widehat{L}_i \left| \vcenter{\hbox{}} \right\rangle = \left| \vcenter{\hbox{}} \right\rangle , \tag{2.131}
$$

which acts only on the qubits $\{h_{ij} \mid [ij] \in E\}$. The right-hand side of the above equation is evaluated by using the $F$-symbols of $(\mathsf{Vec}_{\mathbb{Z}_2}^{\nu})^{\mathrm{rev}} = \mathsf{Vec}_{\mathbb{Z}_2}^{\nu^{-1}}$. As in the case of trivial $\nu$, the second term $\widehat{\mathsf{h}}_{ij}$ fixes the configuration of the qubits for a given configuration of the qutrits. Furthermore, in the subspace where $\widehat{\mathsf{h}}_{ij} = 1$ for all $[ij] \in E$, the first term $\widehat{\mathsf{h}}_i$ reduces to

$$
\widehat{\mathsf{h}}_i \big|_{\{\widehat{\mathsf{h}}_{ij} = 1 \mid [ij] \in E\}} = \frac{1}{4} \left( \widehat{R}_i^{(\mathbb{1})} + \widehat{R}_i^{(\psi)} + \sqrt{2} \widehat{R}_i^{(\sigma)} \right) , \tag{2.132}
$$

where $\widehat{R}_i^{(b)}$ is defined below (2.100). This agrees with the Hamiltonian of the Levin-Wen model based on $\mathsf{Rep}(\mathrm{SU}(2)_2)$. Therefore, the above model realizes the topological order $\mathcal{Z}(\mathsf{Rep}(\mathrm{SU}(2)_2))$.

The symmetry of the model is again described by $2\mathsf{Rep}(\mathbb{Z}_2) = \Sigma\mathsf{Vec}_{\mathbb{Z}_2}$. This symmetry acts on the topological order $\mathcal{Z}(\mathsf{Rep}(\mathrm{SU}(2)_2))$ via the braided monoidal functor

$$
\mathsf{Vec}_{\mathbb{Z}_2} \to \mathcal{Z}(\mathsf{Rep}(\mathrm{SU}(2)_2)), \tag{2.133}
$$

which maps the non-trivial simple object of $\mathsf{Vec}_{\mathbb{Z}_2}$ to $\psi\overline{\psi} \in \mathcal{Z}(\mathsf{Rep}(\mathrm{SU}(2)_2))$.

### 2.3.4  Non-Minimal Gapped Phases with Non-Minimal Symmetry

Non-minimal gapped phases with non-minimal symmetries are obtained by the non-minimal gauging of $G$-symmetric gapped phases with topological orders. The corresponding choice of $A$ and $B$ is

$$
A : \text{arbitrary}, \qquad B : \text{arbitrary}. \tag{2.134}
$$

In what follows, we suppose that $A$ and $B$ are faithfully graded by $H$ and $K$, respectively. Physically, $H$ is the gauged subgroup of $G$, while $K$ is the unbroken subgroup of $G$.

**$G$-Symmetric Model.**  The $G$-symmetric model before gauging is the same as the one we used in section 2.3.3. Specifically, the state space of the model is given by

$$
\mathcal{H}_{\mathrm{original}} = \bigotimes_{i \in P} \mathbb{C}[G] \otimes \widehat{\pi}_{\mathrm{fusion}} \left( \bigotimes_{[ij] \in E} \mathbb{C}^{\mathrm{rank}(B)} \right) . \tag{2.135}
$$

The Hamiltonian is given by

$$H_{\text{original}} = -\sum_{i \in P} \widehat{\mathsf{h}}_i - \sum_{[ij] \in E} \widehat{\mathsf{h}}_{ij}, \tag{2.136}$$

where $\widehat{\mathsf{h}}_i$ and $\widehat{\mathsf{h}}_{ij}$ are defined by (2.107) and (2.108). The ground states of this model are given by (2.110). We note that the $G$ symmetry is spontaneously broken down to $K$ in these ground states. In addition, each vacuum realizes the SET order of the symmetry-enriched string-net model based on $B$.

**Gauged Model.** Now, we perform the generalized gauging using the $H$-graded fusion category $A$. The state space of the gauged model is given by

$$\mathcal{H}_{\text{gauged}} = \widehat{\pi}_{\text{LW}} \mathcal{H}'_{\text{gauged}}, \tag{2.137}$$

where $\widehat{\pi}_{\text{LW}}$ is the Levin-Wen projector defined in section 2.2.2, and $\mathcal{H}'_{\text{gauged}}$ is spanned by all possible configurations of the following dynamical variables:

- The dynamical variables on the plaquettes are labeled by elements of $S_{H \backslash G}$.

- The dynamical variables on the edges are labeled by the pairs $(a_{ij}, b_{ij})$, where $a_{ij}$ is a simple object of $A^{\text{rev}}$ and $b_{ij}$ is a simple object of $B$.[28] These dynamical variables must satisfy the constraint

$$a_{ik} \in a_{ij} \otimes a_{jk}, \qquad b_{ik} \in b_{ij} \otimes b_{jk}, \qquad \forall [ijk] \in V, \tag{2.138}$$

where $a_{ij}$ and $b_{ij}$ are the dynamical variables on $[ij] \in E$.

Each state of the gauged model is denoted by

$$|\{g_i, (a,b)_{ij}\}\rangle = \left| \begin{array}{c} M^{g_l} \\ (a,b)_{kl} \\ (a,b)_{jl} \quad M^{g_k} \\ (a,b)_{jk} \\ M^{g_j} \quad (a,b)_{ik} \\ (a,b)_{ij} \quad M^{g_i} \end{array} \right\rangle, \tag{2.139}$$

where $g_i \in S_{H \backslash G}$ and $(a,b)_{ij} := (a_{ij}, b_{ij})$ for $a_{ij} \in A^{\text{rev}}$ and $b_{ij} \in B$. More concisely, the state space can be written as

$$\mathcal{H}_{\text{gauged}} = \widehat{\pi}_{\text{LW}} \left( \bigotimes_{i \in P} \mathbb{C}^{|H \backslash G|} \otimes \widehat{\pi}_{\text{fusion}} \left( \bigotimes_{[ij] \in E} \mathbb{C}^{\text{rank}(A)} \right) \otimes \widehat{\pi}_{\text{fusion}} \left( \bigotimes_{[ij] \in E} \mathbb{C}^{\text{rank}(B)} \right) \right). \tag{2.140}$$

---

[28] In other words, the edge degrees of freedom are labeled by simple objects of $A^{\text{rev}} \boxtimes B$.

Here, we assumed that $A$ and $B$ are multiplicity-free. See section 7.3.1 for the case of more general $A$ and $B$. We note that $\widehat{\pi}_{\mathrm{LW}}$ acts non-trivially only on $a_{ij}$'s.

The Hamiltonian of the gauged model is given by

$$H_{\mathrm{gauged}} = -\sum_{i\in P}\widehat{\mathsf{h}}_i - \sum_{[ij]\in E}\widehat{\mathsf{h}}_{ij}, \tag{2.141}$$

where $\widehat{\mathsf{h}}_i$ and $\widehat{\mathsf{h}}_{ij}$ are defined by

$$\widehat{\mathsf{h}}_i \left| \cdots M^{g_4} \cdots \right\rangle = \sum_{g_5\in S_{H\backslash G}}\sum_{h_{45}\in H}\delta_{g_4^{-1}h_{45}g_5\in K}\sum_{a_{45},b_{45}}\frac{\dim(a_{45})}{\mathcal{D}_{A_{h_{45}}}}\frac{\dim(b_{45})}{\mathcal{D}_B}\left| \cdots M^{g_5}\ {}_{(a,b)_{45}} \cdots \right\rangle, \tag{2.142}$$

$$\widehat{\mathsf{h}}_{ij}\left| \begin{matrix} M^{g_j}\\ {}_{(a,b)_{ij}}\\ M^{g_i}\end{matrix} \right\rangle = \delta_{g_i^{-1}h_{ij}g_j,k_{ij}}\left| \begin{matrix} M^{g_j}\\ {}_{(a,b)_{ij}}\\ M^{g_i}\end{matrix} \right\rangle, \qquad a_{ij}\in A^{\mathrm{rev}}_{h_{ij}},\quad b_{ij}\in B_{k_{ij}}. \tag{2.143}$$

Here, we recall that $(a,b)_{ij}$ is an abbreviation of $(a_{ij},b_{ij})$. The last summation on the right-hand side of (2.142) is taken over all simple objects of $A^{\mathrm{rev}}_{h_{45}}$ and $B_{g_4^{-1}h_{45}g_5}$. The diagram on the right-hand of (2.142) is evaluated by using the $F$-move of $A^{\mathrm{rev}}$ and $B$. See (7.115) for a more explicit form of $\widehat{\mathsf{h}}_i$.[29]

The above model is solvable because the Hamiltonian is a sum of commuting projectors. As we will see in section 7.3.4, the ground states on an infinite plane are given by

$$|\mathrm{GS};\mu\rangle_{\mathrm{gauged}} = \sum_{\{g_i\in S_{H\backslash G}\}}\sum_{\{h_i\in H\}}\sum_{\{k_i\in K\}}\prod_{i\in P}\delta_{k_i,(g^{(\mu)})^{-1}h_ig_i}\prod_{i\in P}\frac{1}{\mathcal{D}_{A_{h_i}}}\prod_{i\in P}\frac{1}{\mathcal{D}_B}$$

$$\sum_{\{a_i\in A^{\mathrm{rev}}_{h_i}\}}\sum_{\{a_{ij}\in\overline{a_i}\otimes a_j\}}\sum_{\{b_i\in B_{k_i}\}}\sum_{\{b_{ij}\in\overline{b_i}\otimes b_j\}}\prod_{[ijk]\in V}\left(\frac{d^a_{ij}d^a_{jk}}{d^a_{ik}}\right)^{\frac{1}{4}}\left(\frac{d^b_{ij}d^b_{jk}}{d^b_{ik}}\right)^{\frac{1}{4}} \tag{2.144}$$

$$\prod_{[ijk]\in V}({}^A\overline{F}^a_{ijk})^{s_{ijk}}({}^BF^b_{ijk})^{s_{ijk}}|\{g_i,(a,b)_{ij}\}\rangle,$$

where ${}^AF$ and ${}^BF$ denote the $F$-symbols of $A$ and $B$, and $g^{(\mu)}$ is a representative of the $(H,K)$-double coset $Hg^{(\mu)}K$ with $\mu = 1,2,\cdots,|H\backslash G/K|$. See (7.119) for the most general case where $G$ can be anomalous and $A$ and $B$ are not necessarily multiplicity-free.

**Example:** $G = \mathbb{Z}_2$ with $A = B = \mathsf{Ising}$. We consider an example of a non-minimal gapped phase with non-minimal symmetry by choosing $G = \mathbb{Z}_2$ and $A = B = \mathsf{Ising}$. We view $\mathsf{Ising}$ as

---

[29]Equation (7.115) can also be applied to the case where $G$ is anomalous. When $G$ is non-anomalous, we have $\Omega^{g;h}_{ijkl} = 1$ in (7.115).

a $\mathbb{Z}_2$-graded fusion category so that we have $H = \mathbb{Z}_2$ and $K = \mathbb{Z}_2$. This choice corresponds to the non-minimal gappped phase obtained by stacking the $\mathbb{Z}_2^{\mathrm{em}}$-enriched Toric Code with another copy of the $\mathbb{Z}_2^{\mathrm{em}}$-enriched Toric Code and gauging the diagonal $\mathbb{Z}_2$ symmetry. The resulting topological order is known to be described by $\mathcal{Z}(\mathsf{Rep}(H_8))$ [122], where $H_8$ is the eight-dimensional Kac-Paljutkin algebra [123]. The symmetry of the gauged model is described by $\Sigma\mathcal{Z}(\mathsf{Ising})_0$, which we defined around (2.92). In what follows, we describe the lattice model for this non-minimal gapped phase.

The state space of the gauged model is given by

$$\mathcal{H}_{\mathrm{gauged}} = \widehat{\pi}_{\mathrm{fusion}}\left(\bigotimes_{[ij]\in E} \mathbb{C}^3\right) \otimes \widehat{\pi}_{\mathrm{fusion}}\left(\bigotimes_{[ij]\in E} \mathbb{C}^3\right). \tag{2.145}$$

Namely, we have a pair of qutrits on each edge, both of which are labeled by a simple object of $\mathsf{Ising}$.[30] The projector $\widehat{\pi}_{\mathrm{fusion}}$ imposes the condition that these qutrits obey the fusion rules of $\mathsf{Ising}$ at every vertex.

The Hamiltonian is given by $H_{\mathrm{gauged}} = -\sum_{i\in P} \widehat{\mathsf{h}}_i - \sum_{[ij]\in E} \widehat{\mathsf{h}}_{ij}$, where

$$\widehat{\mathsf{h}}_i = \frac{1}{8}\left(\widehat{L}_i^{(\mathbb{1},\mathbb{1})} + \widehat{L}_i^{(\psi,\mathbb{1})} + \widehat{L}_i^{(\mathbb{1},\psi)} + \widehat{L}_i^{(\psi,\psi)} + 2\widehat{L}_i^{(\sigma,\sigma)}\right), \tag{2.146}$$

$$\widehat{\mathsf{h}}_{ij} |(a,b)_{ij}\rangle = \begin{cases} |(a,b)_{ij}\rangle & \text{if } (a,b)_{ij} = (\mathbb{1},\mathbb{1}),\ (\psi,\mathbb{1}),\ (\mathbb{1},\psi),\ (\psi,\psi),\ (\sigma,\sigma), \\ 0 & \text{otherwise.} \end{cases} \tag{2.147}$$

Here, $\widehat{L}_i^{(a,b)}$ is the loop operator defined by

$$\widehat{L}_i^{(a,b)} := \widehat{L}_i^{(a)} \otimes \widehat{L}_i^{(b)}, \tag{2.148}$$

where $\widehat{L}_i^{(a)}$ and $\widehat{L}_i^{(b)}$ are given by (2.96) and (2.124), respectively. We note that $\widehat{L}^{(a)}$ acts only on $a_{ij}$'s, while $\widehat{L}_i^{(b)}$ acts only on $b_{ij}$'s. In the subspace where $\widehat{\mathsf{h}}_{ij} = 1$ for all $[ij] \in E$, the above model reduces to the Levin-Wen model based on the fusion subcategory of $\mathsf{Ising} \boxtimes \mathsf{Ising}$ with five simple objects

$$\{(\mathbb{1},\mathbb{1}), \quad (\psi,\mathbb{1}), \quad (\mathbb{1},\psi), \quad (\psi,\psi), \quad (\sigma,\sigma)\}. \tag{2.149}$$

Since this subcategory is monoidally equivalent to $\mathsf{Rep}(H_8)$ [26], the above model realizes the topological order described by $\mathcal{Z}(\mathsf{Rep}(H_8))$.

## 2.3.5 Example: $G = S_3$ and $2\mathsf{Rep}(\mathbb{Z}_3^{(1)} \rtimes \mathbb{Z}_2^{(0)})$ SSB Phases

We now turn to the lattice construction of a particularly exotic gapped phase of matter. In the landscape of gapped phases, one typically (almost always) encounters phases whose ground

---

[30]We recall that $\mathsf{Ising}^{\mathrm{rev}}$ is monoidally equivalent to $\mathsf{Ising}$.

states are isomorphic to one another. At the most coarse level, this implies that each ground state has the same entanglement structure. In contrast, non-invertible symmetries[31] can stabilize phases with qualitatively distinct vacua. The simplest such phase was studied in [2]. This phase is the fully spontaneously broken phase of the symmetry category corresponding to 2-representations of a 2-group $\mathbb{G}$ (i.e. $2\mathsf{Rep}(\mathbb{G})$) where

$$\mathbb{G} = \mathbb{Z}_3^{(1)} \rtimes \mathbb{Z}_2^{(0)}. \tag{2.150}$$

This symmetry category is related to the $S_3$ 0-form symmetry (i.e., $2\mathsf{Vec}_{S_3}$) via gauging the non-normal subgroup $\mathbb{Z}_2 < S_3$. We now briefly describe the $S_3$ model, focusing on a specific gapped phase that breaks the $\mathbb{Z}_3$ subgroup of $S_3$ spontaneously. We then perform a generalized gauging of the $S_3$ model to obtain the aforementioned gapped phase. Specifically, we perform both the minimal and non-minimal gauging of the $\mathbb{Z}_2 < S_3$.

**The $S_3$ Lattice Model.** Lattice models with $G$ 0-form symmetries and their associated gapped phases have been studied extensively in the literature. The most convenient construction is via $G$-qudits. This is equivalent to (an example of) the fusion surface model described in later sections. The state space is given by a tensor product of local Hilbert spaces $\mathbb{C}[S_3]$ assigned to each plaquette of the honeycomb lattice. A basis state in this Hilbert space is denoted by $|\{g_i\}\rangle$, where $g_i \in S_3$. We present $S_3$ as

$$S_3 = \langle a, b \mid a^3 = b^2 = 1, bab = a^2 \rangle. \tag{2.151}$$

The $S_3$ symmetry acts as

$$U_g|\{g_i\}\rangle = |\{gg_i\}\rangle. \tag{2.152}$$

Hamiltonians for various gapped phases with $2\mathsf{Vec}_G$ symmetry will be constructed in Sec. 7.1. We focus on the phase that breaks the $\mathbb{Z}_3$ subgroup of $S_3$, which corresponds to the choice

$$B = \mathsf{Vec}_{\mathbb{Z}_2^b}, \tag{2.153}$$

where $\mathbb{Z}_2^b = \{1, b\}$. There are three ground states

$$|\mathrm{GS}; \ell\rangle = \frac{1}{2^{|P|/2}} \prod_i \sum_{g_i \in \mathsf{M}_\ell} |\{g_i\}\rangle, \tag{2.154}$$

where $|P|$ is the number of plaquettes and $\mathsf{M}_0 = \{1, b\}$, $\mathsf{M}_1 = \{a, ab\}$ and $\mathsf{M}_2 = \{a^2, a^2b\}$. It can be readily checked that the $S_3$ symmetry acts on the ground state space as

$$U_a|\mathrm{GS}; \ell\rangle = |\mathrm{GS}; \ell + 1 \bmod 3\rangle, \qquad U_b|\mathrm{GS}; \{0, 1, 2\}\rangle = |\mathrm{GS}; \{0, 2, 1\}\rangle. \tag{2.155}$$

---

[31] Specifically, non-condensation non-invertible 0-form symmetries.

In other words, the three ground states can be arranged on the corners of an equilateral triangle and $S_3$ acts naturally with $U_a$ implementing a $\mathbb{Z}_3$ rotation while $U_b$ implementing a reflection that leaves the vertex 0 fixed.

**Gauging $\mathbb{Z}_2^b$ and the $2\mathrm{Rep}(\mathbb{Z}_3^{(1)} \rtimes \mathbb{Z}_2^{(0)})$ Symmetric Model.** After gauging $\mathbb{Z}_2^b = \{1, b\}$, the degrees of freedom on the plaquettes are labelled as $g_i$ where $g_i \in S_{H\backslash G}$ is the representative of a right $H$-coset in $G$. For the present case, we have $G = S_3$ and $H = \mathbb{Z}_2^b$, and hence there are three right cosets $\mathsf{M}'_0 = \{1, b\}$, $\mathsf{M}'_1 = \{a, ba\}$, and $\mathsf{M}'_2 = \{a^2, ba^2\}$, for which we pick the representatives $S_{\mathbb{Z}_2^b \backslash S_3} = \{1, a, a^2\}$. The degrees of freedom on the plaquettes are unconstrained. On the other hand, as described in section 2.3, the degrees of freedom on the edges are labeled by elements in $\mathbb{Z}_2^b$. The edge degrees of freedom should be interpreted as a $\mathbb{Z}_2$ gauge field which further needs to satisfy a flatness constraint on each vertex. The state space one thus obtains is naturally not tensor decomposable, however it admits a natural embedding into a larger Hilbert space

$$\mathcal{H}_0 = \bigotimes_{i \in P} \mathbb{C}_i^3 \otimes \bigotimes_{[kl] \in E} \mathbb{C}_{[kl]}^2 \, . \tag{2.156}$$

One may define models on this state space that have non-topological 1-form symmetry [79]. In our model, however, we impose the flatness constraint kinematically at the level of the state space via the projector

$$\mathcal{H}_{\mathrm{gauged}} = \widehat{\pi}_{\mathrm{flat}} \mathcal{H}_0 \, . \tag{2.157}$$

As a result, the 1-form symmetry operators become topological.

Now, we write down the gauged Hamiltonian for the minimal gapped phase corresponding to $B = \mathsf{Vec}_{\mathbb{Z}_2^b}$. To this end, we define $\mathbb{Z}_3$ clock and shift operators $Z_i$ and $X_i$ on each plaquette that act on the computational basis as

$$Z|a^p\rangle = \omega^p |a^p\rangle \, , \qquad X|a^p\rangle = |a^{p+1 \bmod 3}\rangle \, , \tag{2.158}$$

It will also be useful to define a "charge conjugation" operator $\Gamma$ that acts as

$$\Gamma|a^p\rangle = |a^{-p}\rangle \, . \tag{2.159}$$

Similarly, on the two-dimensional state space associated to the edges of the lattice, we define the usual Pauli operators, which act as

$$\sigma^z |b^q\rangle = (-1)^q |b^q\rangle \, , \qquad \sigma^x |b^q\rangle = |b^{q+1 \bmod 2}\rangle \, . \tag{2.160}$$

The flatness constraint is then imposed via the projector

$$\widehat{\pi}_{\text{flat}} = \prod_{[ijk]} \frac{1 + \sigma^z_{ij}\sigma^z_{jk}\sigma^z_{ki}}{2} \ . \tag{2.161}$$

The Hamiltonian on $\mathcal{H}_{\text{gauged}}$ takes the form

$$H_{\text{gauged}} = -\sum_{[ij]} P_{ij} - \sum_i \frac{1 + \Gamma_i B_i}{2} \ . \tag{2.162}$$

Here $B_i$ is the operator that inserts a $\mathbb{Z}_2$ flux loop on the boundary of the plaquette labelled by $i$

$$B_i = \prod_j \sigma^x_{ij} \ . \tag{2.163}$$

Meanwhile, $P_{ij}$ is the local projector that implements the constraint that the qubit on $[ij]$ along with the qutrits on the two neighboring plaquettes $i$ and $j$ can only have one of the following six configurations:

$$(g_i\,,h_{ij}\,,g_j) \in \left\{(a^p, 1, a^p)\,,(a^p, b, a^{-p}) \mid p = 0, 1, 2\right\} \ . \tag{2.164}$$

This projector has the concrete form

$$P_{ij} = \sum_{t=\pm} \frac{1 + t\sigma^z_{ij}}{2} \otimes \frac{1 + Z_i Z_j^{-t} + Z_i^2 Z_j^{-2t}}{3} \ . \tag{2.165}$$

The Hamiltonian (2.162) is a commuting projector Hamiltonian and therefore the ground states lie in the projected space. The subspace projected by $P_{ij}$ for all edges $[ij]$ decomposes into the direct sum of two state spaces, which we will denote as $V_1$ and $V_2$. On $V_1$, we have $p = 0$ in (2.164), while on $V_2$, we have $p \neq 0$. Furthermore, since the dynamics generated by the term $\Gamma_i B_i$ do not mix $V_1$ and $V_2$, we may consider the Hamiltonians projected to these two spaces separately. On $V_1$, each plaquette degree of freedom is constrained to be 1 on which $\Gamma_i$ acts trivially, therefore the Hamiltonian simply becomes

$$H\Big|_{V_1} = -\sum_i \frac{1 + B_i}{2} \ . \tag{2.166}$$

This is nothing but the Toric Code Hamiltonian on a triangular lattice (dual to the honeycomb lattice).[32] On the other hand, on $V_2$, the state space on each plaquette is two-dimensional, which is spanned by states labeled as $a$ and $a^2$. On this space, $\Gamma$ acts as the Pauli-$x$ operator

---

[32]Typically the Toric Code model is defined on an unconstrained Hilbert space on which the projector $\widehat{\pi}_{\text{flat}}$ is imposed energetically.

exchanging $a$ and $a^2$. We denote the effective Pauli operators on the constrained space as $\widetilde{\sigma}_i^{\alpha}$. Therefore we obtain

$$H\Big|_{V_2} = -\sum_i \frac{1 + \widetilde{\sigma}_i^x B_i}{2}. \tag{2.167}$$

This can be recognized as the symmetry enriched Levin Wen model which enriches the trivial topological order by a $\mathbb{Z}_2$ symmetry [70, 71].[33] At this point, stating this model as such seems an overkill but it makes the generalization to the non-minimal version of this phase manifest.

**Symmetry Action on the Ground States.** The symmetry of the gauged model is described by $2\mathsf{Rep}(\mathbb{Z}_3^{(1)} \rtimes \mathbb{Z}_2^{(0)})$ [124]. This symmetry 2-category has a unique non-condensation non-invertible object up to condensation. The corresponding 0-form symmetry operator is given by $\mathsf{D}_{\mathbb{Z}_2^b} U_a \overline{\mathsf{D}}_{\mathbb{Z}_2^b}$, where $\mathsf{D}_{\mathbb{Z}_2^b}$ and $\overline{\mathsf{D}}_{\mathbb{Z}_2^b}$ are the gauging and ungauging operators defined by (cf. section 5.2.3)

$$\mathsf{D}_{\mathbb{Z}_2^b} |\{b^{n_i} a^{m_i}\}\rangle = |\{a^{m_i}, b^{-n_i + n_j}\}\rangle, \tag{2.168}$$

$$\overline{\mathsf{D}}_{\mathbb{Z}_2^b} |a^{m_i}, b^{n_{ij}}\rangle = \sum_{\{n_i = 0,1\}} \prod_{[ij] \in E} \delta_{n_{ij}, -n_i + n_j \bmod 2} |\{b^{n_i} a^{m_i}\}\rangle. \tag{2.169}$$

See also [76] for the symmetry operators for $2\mathsf{Rep}(\mathbb{Z}_3^{(1)} \rtimes \mathbb{Z}_2^{(0)})$ symmetry. In what follows, we will compute the action of this symmetry operator on the ground states. In particular, we will see that the trivial and topologically ordered ground states of the gauged model are mixed by the action of the non-invertible symmetry.

To compute the symmetry action, let us first write down the ground states of the gauged model. From (2.59), we find that the ground states on an infinite plane are given, up to normalization, by

$$|\text{GS}; 0\rangle_{\text{gauged}} = \sum_{\{h_i \in \mathbb{Z}_2^b\}} |\{1, h_i^{-1} h_j\}\rangle, \qquad |\text{GS}; 1\rangle_{\text{gauged}} = \sum_{\{h_i \in \mathbb{Z}_2^b\}} |\{g(h_i), h_i^{-1} h_j\}\rangle. \tag{2.170}$$

Here, we defined

$$g(h_i) = \begin{cases} a & \text{if } h_i = 1, \\ a^2 & \text{if } h_i = b. \end{cases} \tag{2.171}$$

We note that $|\text{GS}; 0\rangle_{\text{gauged}}$ is the ground state of the Toric Code Hamiltonian (2.166), while $|\text{GS}; 1\rangle_{\text{gauged}}$ is the ground state of the trivial Hamiltonian (2.167). As such, we denote these ground states as

$$|\text{GS}; 0\rangle_{\text{gauged}} = |\text{TC}\rangle, \qquad |\text{GS}; 1\rangle_{\text{gauged}} = |\text{triv}\rangle. \tag{2.172}$$

---

[33]More specifically, $H\Big|_{V_2}$ realizes the trivial SPT phase with $\mathbb{Z}_2$ symmetry.

On the other hand, the ground states of the original $S_3$-symmetric model are given, up to normalization, by

$$|\text{GS}; \ell\rangle_{\text{original}} = \sum_{\{g_i \in \mathsf{M}_\ell\}} |\{g_i\}\rangle \quad \text{for } \ell = 0, 1, 2, \tag{2.173}$$

where $\mathsf{M}_0 = \{1, b\}$, $\mathsf{M}_1 = \{a, ab\}$, and $\mathsf{M}_2 = \{a^2, a^2 b\}$. The $S_3$ symmetry acts on these ground states as in (2.155). We note that $|\text{GS}; 0\rangle_{\text{original}}$ preserves the $\mathbb{Z}_2^b$ symmetry, while the other two ground states spontaneously break the $\mathbb{Z}_2^b$ symmetry.

Now, we compute the action of the symmetry operator $\mathsf{D}_{\mathbb{Z}_2^b} U_a \overline{\mathsf{D}}_{\mathbb{Z}_2^b}$. To this end, we note that gauging operator $\mathsf{D}_{\mathbb{Z}_2^b}$ acts on the ground states of the $S_3$-symmetric model as

$$\mathsf{D}_{\mathbb{Z}_2^b} |\text{GS}; \ell\rangle_{\text{original}} = \begin{cases} |\text{GS}; 0\rangle_{\text{gauged}} & \text{for } \ell = 0, \\ |\text{GS}; 1\rangle_{\text{gauged}} & \text{for } \ell = 1, 2. \end{cases} \tag{2.174}$$

This means that the trivial $\mathbb{Z}_2^b$-invariant ground state is mapped to the Toric Code ground state, while the $\mathbb{Z}_2^b$-SSB ground states are mapped to the trivial ground state. Similarly, the ungauging operator $\overline{\mathsf{D}}_{\mathbb{Z}_2^b}$ acts on the ground states of the gauged model as

$$\begin{aligned} \overline{\mathsf{D}}_{\mathbb{Z}_2^b} |\text{GS}; 0\rangle_{\text{gauged}} &= (1 + U_b) |\text{GS}; 0\rangle_{\text{original}} = 2 |\text{GS}; 0\rangle_{\text{original}} \\ \overline{\mathsf{D}}_{\mathbb{Z}_2^b} |\text{GS}; 1\rangle_{\text{gauged}} &= (1 + U_b) |\text{GS}; 1\rangle_{\text{original}} = |\text{GS}; 1\rangle_{\text{original}} + |\text{GS}; 2\rangle_{\text{original}} . \end{aligned} \tag{2.175}$$

That is, the Toric Code ground state is mapped to the trivial $\mathbb{Z}_2^b$-invariant ground state, while the trivial ground state is mapped to (the symmetric linear combination of) the $\mathbb{Z}_2^b$-SSB states. Using the above equations, one can immediately compute the action of the non-invertible symmetry operator as

$$\begin{aligned} \mathsf{D}_{\mathbb{Z}_2^b} U_a \overline{\mathsf{D}}_{\mathbb{Z}_2^b} |\text{GS}; 0\rangle_{\text{gauged}} &= 2 |\text{GS}; 1\rangle_{\text{gauged}} , \\ \mathsf{D}_{\mathbb{Z}_2^b} U_a \overline{\mathsf{D}}_{\mathbb{Z}_2^b} |\text{GS}; 1\rangle_{\text{gauged}} &= |\text{GS}; 0\rangle_{\text{gauged}} + |\text{GS}; 1\rangle_{\text{gauged}} . \end{aligned} \tag{2.176}$$

Recalling (2.172), we find that the above non-invertible symmetry operator mixes the trivial ground state and the Toric Code ground state.

**Non-Minimal Generalization.** Consider now a non-minimal gauging of the $S_3$ symmetric model. Specifically, we pick a fusion category $A = A_1 \oplus A_b$ faithfully graded by $\mathbb{Z}_2^b < S_3$. Again, the degrees of freedom on the plaquettes are the representatives of right $\mathbb{Z}_2^b$-cosets in $S_3$, which we choose to be $\{1, a, a^2\}$. The degrees of freedom on the edges, however, are now upgraded from being 1 or $b$, i.e., elements in $\mathbb{Z}_2^b$, to being objects in the graded components $A_1$ and $A_b$ respectively. For simplicity, we assume that the fusion coefficients of $A$ are either zero or one. Then, there are no additional degrees of freedom on the vertices of the honeycomb

lattice. However, there are constraints that impose the correct fusion rules on the vertices. The construction of the model is very analogous to the minimal case discussed above with the replacements

$$B_i \longrightarrow \sum_{x \in A_b^{\text{rev}}} \frac{d_x}{\mathcal{D}_{A_b}} \widehat{L}_i^{(x)} \,. \tag{2.177}$$

Additionally, we need to impose the Levin-Wen constraint

$$\sum_{x \in A_1^{\text{rev}}} \frac{d_x}{\mathcal{D}_{A_1}} \widehat{L}_i^{(x)} = 1 \quad \forall \, i \,. \tag{2.178}$$

The Hamiltonian operators further impose that the low-energy state space only contains states with the following constrained structure: any edge $[ij]$ along with its two neighboring plaquettes $i$ and $j$ can only have one of the following configurations

$$(g_i \,, x_{ij} \,, g_j) \in \{(a^p, x, a^p) \mid x \in A_1\} \, \sqcup \, \{(a^p, x, a^{-p}) \mid x \in A_b\} \,, \qquad p = 0, 1, 2. \tag{2.179}$$

This low-energy state space decomposes into the direct sum of two state spaces, which we will denote as $V_1$ and $V_2$. Again, on $V_1$, $p = 0$, while on $V_2$, $p \neq 0$. The Hamiltonian restricted to $V_1$ has the form

$$H\Big|_{V_1} = -\frac{1}{2} \sum_{h \in \mathbb{Z}_2^b} \sum_{x \in A_h^{\text{rev}}} \frac{d_x}{\mathcal{D}_{A_h}} \widehat{L}_i^{(x)} \,. \tag{2.180}$$

This corresponds to the Levin-Wen model built from the fusion category $A^{\text{rev}}$ as input and describes the $\mathcal{Z}(A^{\text{rev}})$ topological order. Meanwhile, the Hamiltonian restricted to $V_2$ has the form

$$H\Big|_{V_2} = -\frac{1}{2} \sum_{x \in A_1^{\text{rev}}} \frac{d_x}{\mathcal{D}_{A_1}} \widehat{L}_i^{(x)} - \frac{1}{2} \sum_{x \in A_b^{\text{rev}}} \frac{d_x}{\mathcal{D}_{A_b}} \widetilde{\sigma}_i^x \widehat{L}_i^{(x)} \,. \tag{2.181}$$

This describes a $\mathbb{Z}_2$ enrichment of the topological order $\mathcal{Z}(A_1^{\text{rev}})$ [70, 71]. We note that the topological order realized by $H\Big|_{V_1}$ is the $\mathbb{Z}_2$-gauging of the topological order realized by $H\Big|_{V_2}$.

## 2.4   Tensor Network Representation

In this subsection, we provide tensor network representations of the generalized gauging operators and the gapped ground states discussed in the previous subsections. We refer the reader to section 3.4 for a brief review of tensor networks. Throughout the paper, we will employ the following color convention for tensor networks:

- Physical legs are written in black, while virtual bonds are written in different colors:

- A green virtual bond is labeled by an element of $H$.

- A blue virtual bond is labeled by an element of $K$.

- A red virtual bond is labeled by a simple object of $A^{\mathrm{rev}}$.

- An orange virtual bond is labeled by a simple object of $B$.

The labels of the physical legs depend on the case under consideration. However, the following rules are common in all cases:

- The physical leg above each plaquette is labeled by an element of $S_{H\backslash G}$.

- The physical leg below each plaquette is labeled by an element of $G$.

For simplicity, we will focus on the case where $G$ is non-anomalous for the moment. See sections 6.2.4, 7.3.2, and 7.3.5 for the case of anomalous $G$.

### 2.4.1 Generalized Gauging Operators

**Minimal Gauging.** As we discussed in section 2.2.4, the generalized gauging operator for the minimal gauging $A = \mathsf{Vec}_H^\nu$ is given by

$$\mathsf{D}_A \left|\{h_i g_i\}\right\rangle = \prod_{[ijk]\in V} \nu(h_i, h_i^{-1}h_j, h_j^{-1}h_k)^{-s_{ijk}} \left|\{g_i, h_i^{-1}h_j\}\right\rangle, \tag{2.182}$$

where $h_i \in H$ and $g_i \in S_{H\backslash G}$. The above operator can be represented by the following tensor network:

$$\mathsf{D}_A = \quad \text{} \quad . \tag{2.183}$$

Here, the physical leg on each edge is labeled by an element of $H$. The non-zero components of the local tensors are defined by

 $= 1,$ $\qquad$  $= 1,$ $\tag{2.184}$

 $= \nu(h_i, h_i^{-1}h_j, h_j^{-1}h_k)^{-1},$ $\qquad$  $= \nu(h_i, h_i^{-1}h_j, h_j^{-1}h_k),$

$$\tag{2.185}$$

where $h \in H$ and $g \in S_{H \backslash G}$. A tensor network representation of the ungauging operator (2.40) is obtained by flipping the tensor network for $\mathsf{D}_A$ upside down and swapping $\nu$ and $\nu^{-1}$ in (2.185).

**Non-Minimal Gauging.** The generalized gauging operator for the non-minimal gauging is given by

$$\mathsf{D}_A \left| \{h_i g_i\} \right\rangle = \sum_{\{a_i \in A_{h_i}^{\mathrm{rev}}\}} \sum_{\{a_{ij} \in \overline{a_i} \otimes a_j\}} \prod_{i \in P} \frac{1}{\mathcal{D}_{A_{h_i}}} \prod_{[ijk] \in V} \left( \frac{d_{ij}^a d_{jk}^a}{d_{ik}^a} \right)^{\frac{1}{4}} \prod_{[ijk] \in V} (\overline{F}_{ijk}^a)^{s_{ijk}} \left| \{g_i, a_{ij}\} \right\rangle ,$$

(2.186)

where $h_i \in H$ and $g_i \in S_{H \backslash G}$. Here, we assumed that $A$ is multiplicity-free. The above operator can be represented by the following tensor network:

$$\mathsf{D}_A = \quad$$ 

(2.187)

The physical leg on each edge is labeled by a simple object of $A^{\mathrm{rev}}$. The non-zero components of the local tensors are defined by

$$\overset{h \quad g}{\underset{hg \quad h}{\overset{h \diagdown\!\!\!\diagup h}{\underset{h \diagup\!\!\!\diagdown h}{\bullet}}}} = \frac{1}{\mathcal{D}_{A_h}}, \qquad a \overset{a}{\underset{}{\smash{\big\downarrow}}} \bullet\!\!-\!a = 1, \qquad a \overset{h}{\underset{}{\rule{0pt}{1.5em}}}\!\!-\!a = \delta_{a \in A_h},$$

(2.188)

$$\overset{a_{jk} \, a_k}{\underset{a_{ij} \, a_i}{\overset{a_j}{\underset{a_j}{\mathrel{\rotatebox{0}{$\blacktriangleleft$}}}}}}\!\!\!\!\begin{smallmatrix} a_k \\ a_{ik} \\ a_i \end{smallmatrix} = \left( \frac{d_{ij}^a d_{jk}^a}{d_{ik}^a} \right)^{\frac{1}{4}} \overline{F}_{ijk}^a , \qquad \overset{a_k \, a_{jk}}{\underset{a_i \, a_{ij}}{\overset{a_{ik}}{\underset{a_i}{\mathrel{\rotatebox{0}{$\triangleright$}}}}}}\!\!\!\!\begin{smallmatrix} a_j \\ a_j \end{smallmatrix} = \left( \frac{d_{ij}^a d_{jk}^a}{d_{ik}^a} \right)^{\frac{1}{4}} F_{ijk}^a ,$$

(2.189)

where $h \in H$, $g \in S_{H \backslash G}$, and $\delta_{a \in A_h}$ is defined by

$$\delta_{a \in A_h} = \begin{cases} 1 & \text{if } a \in A_h, \\ 0 & \text{if } a \notin A_h. \end{cases}$$

(2.190)

A tensor network representation of the ungauging operator (2.37) is obtained by flipping the tensor network for $\mathsf{D}_A$ upside down, swapping $F$ and $\overline{F}$ in (2.189), and rescaling the non-zero component of the plaquette tensor to 1. See (5.111) for the explicit tensor network representation of $\overline{\mathsf{D}}_A$. See also (6.75) and (6.78) for the generalization to the case of anomalous $G$.

## 2.4.2 Gapped Phases

Let us move on to tensor network representations of the gapped ground states. In what follows, we will focus on the minimal gapped phases with minimal symmetries (i.e., the simplest case) and non-minimal gapped phases with non-minimal symmetries (i..e, the most general case).

**Minimal Gapped Phases with Minimal Symmetries.** We consider the minimal gapped phase with minimal symmetry corresponding to

$$A = \mathsf{Vec}_H^\nu, \qquad B = \mathsf{Vec}_K^\lambda. \tag{2.191}$$

As we discussed in section 2.3.1, the ground states of this gapped phase are given by

$$|\text{GS}; \mu\rangle_{\text{gauged}} = \sum_{\{g_i \in S_{H \backslash G}\}} \sum_{\{h_i \in H\}} \sum_{\{k_i \in K\}} \prod_{i \in P} \frac{1}{|K|} \delta_{k_i, (g^{(\mu)})^{-1} h_i g_i} \prod_{[ijk] \in V} \left( \frac{\lambda(k_i, k_{ij}, k_{jk})}{\nu(h_i, h_{ij}, h_{jk})} \right)^{s_{ijk}} |\{g_i, h_{ij}\}\rangle. \tag{2.192}$$

Here, we recall that $h_{ij} := h_i^{-1} h_j$, $k_{ij} := k_i^{-1} k_j$, and $g^{(\mu)}$ is a representative of the $(H, K)$-double coset $H g^{(\mu)} K$ in $G$. The above ground states can be represented by the following tensor network:

$$|\text{GS}; \mu\rangle_{\text{gauged}} = \quad  \quad . \tag{2.193}$$

The physical leg on each edge is labeled by an element of $H$. The non-zero components of the local tensors are defined by[34]

$$ = \frac{1}{|K|} \delta_{k, (g^{(\mu)})^{-1} hg}, \qquad  = 1, \tag{2.194}$$

$$ = \frac{\lambda(k_i, k_{ij}, k_{jk})}{\nu(h_i, h_{ij}, h_{jk})}, \qquad  = \frac{\nu(h_i, h_{ij}, h_{jk})}{\lambda(k_i, k_{ij}, k_{jk})}. \tag{2.195}$$

When $G$ is anomalous, the tensors in (2.195) are modified as in (7.97) and (7.98). In the case of SPT phases, the tensor network representation (2.193) can be simplified as in (7.106).

---

[34] According to our color convention, the red virtual bonds are labeled by simple objects of $A^{\text{rev}} = \mathsf{Vec}_H^{\nu^{-1}}$. Equivalently, they are labeled by elements of $H$.

**Non-Minimal Gapped Phases with Non-Minimal Symmetries.** We consider the non-minimal gapped phase with non-minimal symmetry corresponding to

$$A : \text{arbitrary}, \qquad B : \text{arbitrary}. \tag{2.196}$$

We suppose that $A$ and $B$ are multiplicity-free. As we discussed in section 2.3.4, the ground states of this gapped phase are given by

$$|\text{GS};\mu\rangle_{\text{gauged}} = \sum_{\{g_i \in S_{H\backslash G}\}} \sum_{\{h_i \in H\}} \sum_{\{k_i \in K\}} \prod_{i \in P} \delta_{k_i,(g^{(\mu)})^{-1}h_i g_i} \prod_{i \in P} \frac{1}{\mathcal{D}_{A_{h_i}}} \prod_{i \in P} \frac{1}{\mathcal{D}_B}$$

$$\sum_{\{a_i \in A_{h_i}^{\text{rev}}\}} \sum_{\{a_{ij} \in \overline{a_i} \otimes a_j\}} \sum_{\{b_i \in B_{k_i}\}} \sum_{\{b_{ij} \in \overline{b_i} \otimes b_j\}} \prod_{[ijk] \in V} \left( \frac{d_{ij}^a d_{jk}^a}{d_{ik}^a} \right)^{\frac{1}{4}} \left( \frac{d_{ij}^b d_{jk}^b}{d_{ik}^b} \right)^{\frac{1}{4}} \tag{2.197}$$

$$\prod_{[ijk] \in V} (^A\overline{F}_{ijk}^a)^{s_{ijk}} (^B F_{ijk}^b)^{s_{ijk}} |\{g_i, (a,b)_{ij}\}\rangle.$$

The above ground states can be represented by the following double-layered tensor network:

$$|\text{GS};\mu\rangle_{\text{gauged}} = \quad \text{} \quad . \tag{2.198}$$

In the top layer, the physical leg on each edge is labeled by a simple object of $A^{\text{rev}}$. On the other hand, in the bottom layer, the physical leg on each edge is labeled by a simple object of $B$. The non-zero components of the local tensors are defined by

$$\underset{h\ \ k\ h\ k}{\overset{h\ g\ k}{\underset{k\qquad\ \ h}{\underset{h\qquad\ \ k}{\mu}}}} = \frac{1}{\mathcal{D}_{A_h}} \frac{1}{\mathcal{D}_B} \delta_{k,(g^{(\mu)})^{-1}hg}, \qquad \underset{a \quad\quad a}{\overset{a}{\bullet}} = \underset{b \quad\quad b}{\overset{b}{\bullet}} = 1, \tag{2.199}$$

$$\underset{a}{\overset{h}{|}}\ a = \delta_{a \in A_h}, \qquad \underset{b}{\overset{k}{|}}\ b = \delta_{b \in B_k}, \tag{2.200}$$

$$\underset{a_{ij}\ a_i}{\overset{a_{jk}\ a_k}{\underset{a_j}{\overset{a_j}{\blacktriangleleft}}}}\ \begin{matrix} a_k \\ a_{ik} \\ a_i \end{matrix} = \left( \frac{d_{ij}^a d_{jk}^a}{d_{ik}^a} \right)^{\frac{1}{4}}\ ^A\overline{F}_{ijk}^a, \qquad \begin{matrix} a_k\ a_{jk} \\ a_k \\ a_{ik} \\ a_i \end{matrix}\overset{\blacktriangleright}{\underset{a_i\ a_{ij}}{}}\begin{matrix} a_j \\ a_j \end{matrix} = \left( \frac{d_{ij}^a d_{jk}^a}{d_{ik}^a} \right)^{\frac{1}{4}}\ ^A F_{ijk}^a, \tag{2.201}$$

$$\underset{b_{ij}\ b_i}{\overset{b_{jk}\ b_k}{\underset{b_j}{\overset{b_j}{\triangleleft}}}}\ \begin{matrix} b_k \\ b_{ik} \\ b_i \end{matrix} = \left( \frac{d_{ij}^b d_{jk}^b}{d_{ik}^b} \right)^{\frac{1}{4}}\ ^B F_{ijk}^b, \qquad \begin{matrix} b_k\ b_{jk} \\ b_k \\ b_{ik} \\ b_i \end{matrix}\overset{\blacktriangleright}{\underset{b_i\ b_{ij}}{}}\begin{matrix} b_j \\ b_j \end{matrix} = \left( \frac{d_{ij}^b d_{jk}^b}{d_{ik}^b} \right)^{\frac{1}{4}}\ ^B\overline{F}_{ijk}^b. \tag{2.202}$$

See section 7.3.5 for the generalization to the case of anomalous $G$.

When $A = \mathsf{Vec}_H^\nu$, the physical legs on the edges of the top layer are uniquely determined by the other physical legs, due to (2.199), (2.200), and (2.201). Accordingly, the diagram in (2.198) reduces to a single-layered tensor network. Similarly, when $B = \mathsf{Vec}_K^\lambda$, the physical legs on the edges of the bottom layer are uniquely determined by the other physical legs, due to (2.199), (2.200), and (2.202). Thus, the diagram again reduces to a single-layered tensor network. In particular, when $A = \mathsf{Vec}_H^\nu$ and $B = \mathsf{Vec}_K^\lambda$, (2.198) reduces to (2.193).

## 2.5   SymTFT Perspective on Lattice Models

$(2+1)$-dimensional lattice models with fusion 2-categorical symmetries, for instance fusion surface models [62] or generalized Ising gauge theories [76], can be naturally situated within the 4d SymTFT. Specifically, the lattice model appears on a codimension-2 interface between the symmetry boundary and the physical boundary of the SymTFT. This is entirely analogous to how the anyon chain model [52, 55, 125–128] (and the corresponding statistical mechanical models [24, 128, 129]) with a fusion category symmetry $\mathcal{C}$ appear within 3d SymTFT based on the Turaev-Viro-Barrett-Westbury state-sum construction with $\mathcal{C}$ being the input fusion category [42, 128].

We first recall that fusion 2-categories fall into two broad families, characterized by the properties of the genuine lines within the fusion 2-category which form a braided fusion 1-category. Specifically, the Müger center of the 1-category of genuine lines corresponds to either $\mathsf{Rep}(G)$ (representations of a group) or $\mathsf{Rep}(G, z)$ (representations of a super-group) [19]. Simply put, the latter corresponds to the case where some of the transparent genuine lines have topological spin $-1$, while all transparent genuine lines are bosonic in the former case. Here we describe the lattice construction for any fusion 2-category of the first kind which are also known as "All Boson type" fusion 2-categories [130, 131].

**The SymTFT and its Topological Boundaries.** The SymTFT for any such symmetry is the 4d Dijkgraaf-Witten theory [132] based on some finite group $G$ and topological action $[\omega] \in H^4(G, U(1))$. The gapped boundary conditions of the SymTFT can be organized in terms of generalized gaugings of a reference Dirichlet boundary condition, which we denote as

$$\mathfrak{B}_{\mathrm{Dir}} . \tag{2.203}$$

More precisely, the boundary $\mathfrak{B}_{\mathrm{Dir}}$ is the Dirichlet boundary for all the SymTFT lines, that are labeled by representations of $G$. The fusion 2-category of defects on this boundary is $2\mathsf{Vec}_G^\omega$. All other topological boundary conditions of the SymTFT are obtainable via generalized

gaugings of $\mathfrak{B}_{\mathrm{Dir}}$, written schematically as

$$\frac{\mathfrak{B}_{\mathrm{Dir}} \boxtimes \mathfrak{T}}{H}, \tag{2.204}$$

where $\mathfrak{T}$ is an $H$-symmetric TFT, $\boxtimes$ denotes the stacking operation on the boundary, and the quotient by $H$ denotes gauging. The set of topological boundaries is further organized in terms of properties of $\mathfrak{T}$ as follows:

- **Minimal Boundaries:** These correspond to the cases, that $\mathfrak{T}$ is an invertible TFT, i.e., an $H$-SPT. Stacking an $H$ SPT, labeled by $[\nu] \in H^3(H, U(1))$ and subsequently gauging $H$ is nothing but gauging with the choice of discrete torsion $\nu$. Clearly, $H$ needs to be a subgroup of $G$ on which the anomaly $\omega$ is trivialized.

- **Non-Minimal Non-Chiral Boundaries:** These correspond to the case where $\mathfrak{T}$ is a non-trivial $H$-symmetric topological order, which admits gapped boundaries. Such a topological order is the Drinfeld center of a fusion category $\mathcal{C}_1$, denoted as $\mathcal{Z}(\mathcal{C}_1)$. The $H$-symmetry enrichment of $\mathcal{Z}(\mathcal{C}_1)$ can be fully captured in terms of an $H$-graded $\omega|_H$-twisted fusion category $\mathcal{C}$, whose identity component is $\mathcal{C}_1$ [71, 115]. Minimal gapped boundaries are a special case of non-minimal boundaries for which $\mathcal{C} = \mathsf{Vec}_H^\nu$, viewed as an $H$-graded fusion category.

- **Non-Minimal Chiral Boundaries:** These correspond to the stacking of $\mathfrak{B}_{\mathrm{Dir}}$ with a non-chiral $H$-symmetric topological order. Algebraically, such a topological order is an $H$-crossed braided fusion category whose underlying modular tensor category is not the center of any fusion category [109].[35] Such a boundary condition does not admit a gapped interface to $\mathfrak{B}_{\mathrm{Dir}}$.

Before moving on, we note that we do not consider the case where $\mathfrak{T}$ is a TFT corresponding to an $H$ symmetry broken phase as these do not furnish any new boundary conditions. Non-chiral boundaries, both minimal and non-minimal, are obtainable via gauging an algebra object $\mathcal{C}$ in $2\mathsf{Vec}_G^\omega$. We will denote non-chiral boundaries as

$$\mathfrak{B}[\mathcal{C}] = \frac{\mathfrak{B}_{\mathrm{Dir}} \boxtimes \mathfrak{T}_\mathcal{C}}{H}, \tag{2.205}$$

where $\mathfrak{T}_\mathcal{C}$ is a single vacua TFT whose underlying category of lines forms the modular tensor category $\mathcal{Z}(\mathcal{C}_1)$ and whose $H$-symmetry is specified by $\mathcal{C}$.

---

[35]More precisely, when $H$ is anomalous, this topological order should be described by an appropriate generalization of an $H$-crossed braided fusion category, which incorporates the data of an anomaly.

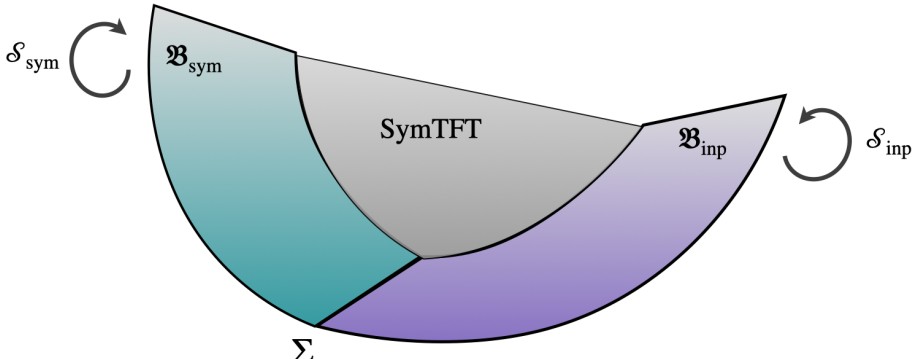

Figure 3: Lattice model with a fusion 2-categorical symmetry $\mathcal{S}_{\text{sym}}$ is naturally situated on the co-dimension-2 interface $\Sigma$ between the physical and input topological boundaries of the SymTFT. One dimension is suppressed in this figure.

**Gapped Phases from the SymTFT.** The SymTFT is a systematic and economical way to fully classify and characterize gapped phases with fusion 2-categorical symmetries [1, 2]. Let us briefly sketch out the procedure to carry this out. To construct a phase with $\mathcal{S}_{\text{sym}}$ symmetry, one first picks a topological symmetry boundary condition $\mathfrak{B}_{\text{sym}}$ of the SymTFT which carries the 2-category of defects given by $\mathcal{S}_{\text{sym}}$. In this paper, we only consider those symmetries that can be obtained by gauging an algebra object $A$ in $2\mathsf{Vec}_G^\omega$, which is a $G$-graded fusion category [133, 134].[36] We denote this symmetry category as $\mathcal{S}_{\text{sym}}[A]$ and the corresponding symmetry boundary is $\mathfrak{B}_{\text{sym}} = \mathfrak{B}[A]$. Then to produce an $\mathcal{S}_{\text{sym}}[A]$ symmetric gapped phase described by an $\mathcal{S}_{\text{sym}}[A]$-equivariant 3d TFT, we pick a topological boundary condition $\mathfrak{B}_{\text{phys}}$ and compactify the interval occupied by the SymTFT. In this work, we only focus on physical boundaries which are obtainable via gauging an algebra object $B$ in $2\mathsf{Vec}_G^\omega$. Therefore, we are interested in lattice models for gapped phases realized by the SymTFT sandwich with $\mathfrak{B}[A]$ and $\mathfrak{B}[B]$ chosen as the symmetry and physical boundaries respectively. In what follows, we outline how the SymTFT construction works for such lattice models.

$2\mathsf{Vec}_G^\omega$ **Symmetric Lattice Model.** First of all, one can construct a (not necessarily gapped) lattice model with a $G$ global symmetry that carries an anomaly $\omega$. To do so, let us consider the SymTFT with the symmetry and input boundaries chosen as $\mathfrak{B}_{\text{sym}} = \mathfrak{B}_{\text{inp}} = \mathfrak{B}_{\text{Dir}}$. We denote the category of defects on these boundaries as $\mathcal{S}_{\text{sym}}$ and $\mathcal{S}_{\text{inp}}$ respectively. There is a co-dimension-2 interface located on $\Sigma$ between these boundaries on which we define a lattice. Mathematically this interface is described by the regular $\mathcal{S}_{\text{sym}} = 2\mathsf{Vec}_G^\omega$ module

---

[36]The $G$-gradinig on $A$ is not necessarily faithful. Physically, the subgroup $H \subset G$ that faithfully grades $A$ corresponds to the gauged subgroup.

2-category. The state space of the lattice model is obtained by endowing $\Sigma$ with a honeycomb lattice (which can be straightforwardly generalized to other lattices) and allowing objects and 1-morphisms in $\mathcal{S}_{\text{inp}}$ to end on $\Sigma$ from the input boundary as illustrated in Fig. 4. Namely, the input boundary $\mathfrak{B}_{\text{inp}}$ is now decorated by a defect network of $\mathcal{S}_{\text{inp}}$. Denote the structure of objects and 1-morphisms ending on $\Sigma$ from $\mathfrak{B}_{\text{inp}}$ as $\Phi$. Given this setup, the different independent configurations of various $p = 2, 1, 0$ strata on $\Sigma$ correspond to basis vectors in the lattice model. The symmetry of the model is $\mathcal{S}_{\text{sym}} = 2\mathsf{Vec}_G^\omega$. Meanwhile, the Hamiltonian operators are given by locally generated endomorphisms of $\Phi$ in $\mathcal{S}_{\text{inp}}$. We note that the lattice

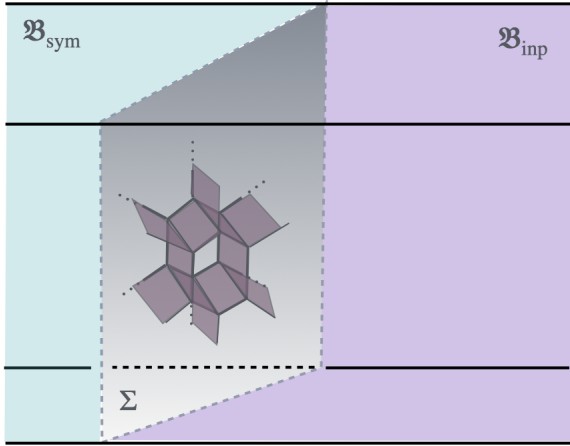

Figure 4: Hamiltonian operators are given by locally generated endomorphisms of $\Phi$ in $\mathcal{S}_{\text{inp}}$. Specifically, picking these endomorphisms to be generated by a 2-algebra in $2\mathsf{Vec}_G^\omega$ furnishes the fixed-point Hamiltonian for a gapped phase.

model obtained in this way is not topological for generic $\Phi$.

**Gauging.** Performing a generalized gauging of this model as described in section 2.2 corresponds to gauging on the symmetry boundary via a choice of algebra $A$. Here, the gauging on the symmetry boundary means that we insert a fine-mesh of topological defects labeled by $A$ on the boundary. This changes the interface on $\Sigma$ from the regular $2\mathsf{Vec}_G^\omega$-module to another module 2-category ${}_A(2\mathsf{Vec}_G^\omega)$, the 2-category of left $A$-modules in $2\mathsf{Vec}_G^\omega$. Accordingly, this modifies the state space and the symmetry operators of the model, while leaving the abstract presentation of the Hamiltonian unchanged.[37] See sections 5 and 6 for a detailed analysis.

---

[37]The abstract presentation of the Hamiltonian refers to the choice of locally generated endomorphisms of $\Phi$. We note that the actual Hamiltonian operators change because the state space changes, even though their abstract presentation is unchanged.

**Fixed-Point Hamiltonians for Gapped Phases.** Meanwhile, the fixed-point Hamiltonian within a gapped phase can be obtained by gauging on the input boundary via the algebra $B$ to obtain a physical boundary $\mathfrak{B}[B]$. Put differently, we choose the defect network $\Phi$ on the input boundary to be a fine-mesh of topological defects labeled by $B$. This gauging is implemented within the lattice model dynamically by imposing that the Hamiltonian is constructed from building blocks defined via the algebra $B$ as described in Sec. 2.3.

Furthermore, the construction of symmetry twisted sectors, generalized charges, symmetry order parameters as well as a constructive treatment of the lattice realizations of certain second order phase transitions can be carried out based on the SymTFT, generalizing the construction of [42] to one dimension higher. We leave these aspects for future work.

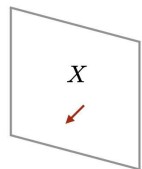 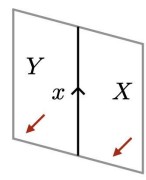 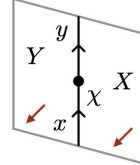

Figure 5: Objects, 1-morphisms, and 2-morphisms of a fusion 2-category are represented by oriented surfaces, lines, and points, respectively. The red arrows specify the coorientations of the surfaces, which will be omitted when the coorientation is clear from the context.

# Part II
# Main Text

## 3 Preliminaries

In this section, we briefly review fusion 2-categories and related mathematical backgrounds. Throughout the paper, we suppose that the base field is $\mathbb{C}$. The tensor product in a fusion 2-category will be denoted by $\square$, while the tensor product in a fusion 1-category will be denoted by $\otimes$.

### 3.1 Fusion 2-Categories

We first recall the basic data of a fusion 2-category. Mathematically, a fusion 2-category is defined as a finite semisimple $\mathbb{C}$-linear rigid monoidal 2-category whose unit object is simple. We refer the reader to [3] for details of the precise definition. The classification of fusion 2-categories was given recently in [19].

A fusion 2-category $\mathcal{C}$ consists of objects, 1-morphisms between objects, and 2-morphisms between 1-morphisms, see figure 5 for the diagrammatic representations of these data. For each pair of objects $X, Y \in \mathcal{C}$, the 1-morphisms between $X$ and $Y$ (and the 2-morphisms between these 1-morphisms) form a finite semisimple 1-category $\mathrm{Hom}_{\mathcal{C}}(X, Y)$. In particular, for each pair of 1-morphisms $x, y \in \mathrm{Hom}_{\mathcal{C}}(X, Y)$, the 2-morphisms between $x$ and $y$ form a finite dimensional vector space $\mathrm{Hom}_{\mathrm{Hom}_{\mathcal{C}}(X,Y)}(x, y)$. The identity 1-morphism of $X$ and the identity 2-morphism of $x$ are denoted by $1_X$ and $\mathrm{id}_x$, respectively. We note that $\mathrm{End}_{\mathcal{C}}(X) := \mathrm{Hom}_{\mathcal{C}}(X, X)$ is a multifusion category, whose monoidal structure is given by the composition of 1-morphisms. The subscripts of Hom and End will be omitted when there is no confusion.

An object $X \in \mathcal{C}$ is said to be simple if and only if the unit object $1_X \in \mathrm{End}(X)$ is simple, i.e., $\mathrm{End}(1_X) \cong \mathbb{C}$. In other words, $X \in \mathcal{C}$ is simple if and only if $\mathrm{End}(X)$ is fusion. Similarly, a 1-morphism $x \in \mathrm{Hom}(X, Y)$ is said to be simple if and only if the endomorphism space of $x$

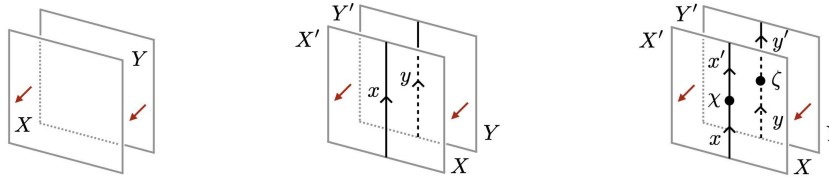

Figure 6: The diagrammatic representation of the tensor product: $X\Box Y \in \mathcal{C}$ (left), $x\Box y \in$ $\text{Hom}(X\Box X', Y\Box Y')$ (middle), and $\chi\Box\zeta \in \text{Hom}(x\Box x', y\Box y')$ (right). Here, the 1-morphism $x\Box y$ is defined by $(x\Box 1_{Y'})\circ(1_X\Box y)$, which is related to $(1_{X'}\Box y)\circ(x\Box 1_Y)$ by the interchanger isomorphism [3].

is one-dimensional, i.e., $\text{End}(x) \cong \mathbb{C}$.[38] The number of (isomorphism classes of) simple objects and simple 1-morphisms are finite. Any objects and 1-morphisms in $\mathcal{C}$ can be decomposed into finite direct sums of simple objects and simple 1-morphisms. Simple objects and simple 1-morphisms cannot be decomposed anymore.

Simple objects $X$ and $Y$ in $\mathcal{C}$ are said to be connected if and only if there exists a non-zero 1-morphism between them. We note that $X$ and $Y$ are not necessarily isomorphic to each other even if they are connected. The set of simple objects connected to each other is called a connected component of $\mathcal{C}$. The number of connected components is finite because the number of simple objects is finite.

A fusion 2-category $\mathcal{C}$ is equipped with a monoidal structure, which allows us to define the tensor product $X\Box Y$ for any pair of objects $X, Y \in \mathcal{C}$. The unit object with respect to this tensor product is denoted by $I$. We note that $I$ is simple.[39] Diagrammatically, the object $X\Box Y$ is represented by a pair of two surfaces stacked on top of each other. The tensor product of 1-morphisms and 2-morphisms are also defined similarly, see figure 6 for their diagrammatic representations.

The basic data of a fusion 2-category $\mathcal{C}$ can be summarized as follows:

- The set of simple objects $\Gamma_{ij}$ of $\mathcal{C}$

- The set of simple 1-morphisms from $\Gamma_{ij}\Box\Gamma_{jk}$ to $\Gamma_{ik}$ for all simple objects $\Gamma_{ij}, \Gamma_{jk}, \Gamma_{ik} \in \mathcal{C}$

- The set of 2-morphisms from $\Gamma_{ikl}\circ(\Gamma_{ijk}\Box 1_{\Gamma_{kl}})$ to $\Gamma_{ijl}\circ(1_{\Gamma_{ij}}\Box\Gamma_{jkl})$

$$\text{(3.1)}$$

---

[38]These definitions of a simple object and a simple 1-morphism can also be applied to more general semisimple 2-categories [3].

[39]When the unit object is not simple, $\mathcal{C}$ is called a multifusion 2-category.

for all simple 1-morphisms $\Gamma_{ijk} \in \mathrm{Hom}_{\mathcal{C}}(\Gamma_{ij}\square\Gamma_{jk}, \Gamma_{ik})$, $\Gamma_{ikl} \in \mathrm{Hom}_{\mathcal{C}}(\Gamma_{ik}\square\Gamma_{kl}, \Gamma_{il})$, $\Gamma_{ijl} \in \mathrm{Hom}_{\mathcal{C}}(\Gamma_{ij}\square\Gamma_{jl}, \Gamma_{il})$, and $\Gamma_{jkl} \in \mathrm{Hom}_{\mathcal{C}}(\Gamma_{jk}\square\Gamma_{kl}, \Gamma_{jl})$

- The complex numbers $z_{\pm}(\Gamma; [01234])$ called 10-j symbols, which are defined by[40]

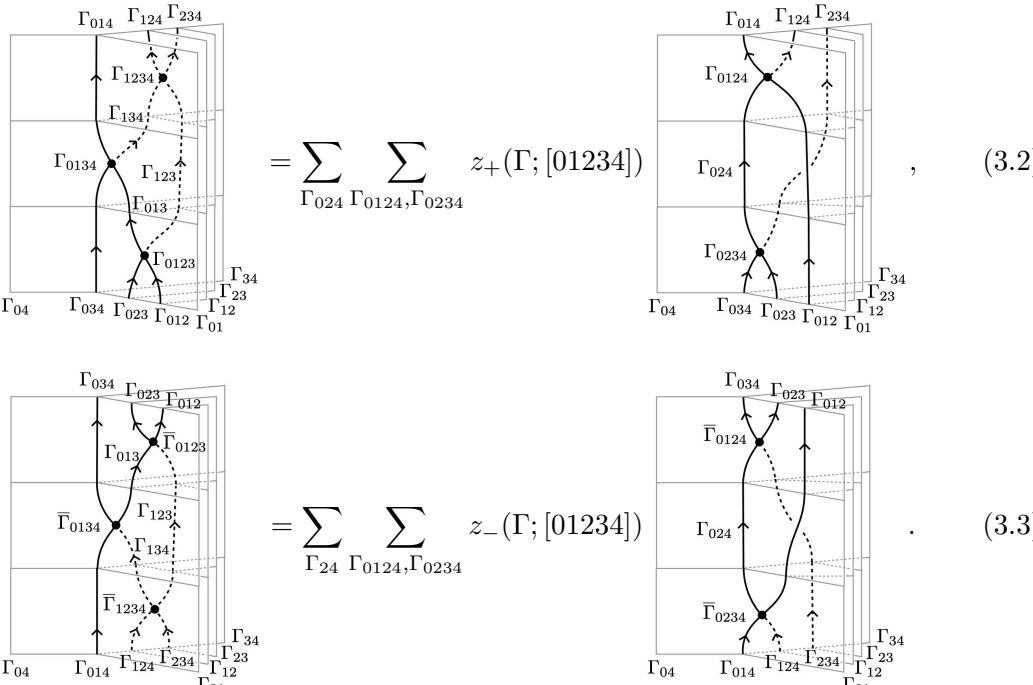

$$= \sum_{\Gamma_{024}} \sum_{\Gamma_{0124}, \Gamma_{0234}} z_+(\Gamma; [01234]) \qquad , \qquad (3.2)$$

$$= \sum_{\Gamma_{24}} \sum_{\Gamma_{0124}, \Gamma_{0234}} z_-(\Gamma; [01234]) \qquad . \qquad (3.3)$$

Here, $\{\Gamma_{ijkl}\}$ is a basis of the vector space $\mathrm{Hom}_{\mathrm{Hom}_{\mathcal{C}}(\Gamma_{ij}\square\Gamma_{jk}\square\Gamma_{kl}, \Gamma_{il})}(\Gamma_{ikl}\circ(\Gamma_{ijk}\square 1_{\Gamma_{kl}}), \Gamma_{ijl}\circ(1_{\Gamma_{ij}}\square\Gamma_{jkl}))$ and $\{\overline{\Gamma}_{ijkl}\}$ is a basis of the dual vector space. The summation on the right-hand side is taken over all simple 1-morphisms $\Gamma_{024}$ and basis 2-morphisms $\Gamma_{0124}$ and $\Gamma_{0234}$. We note that the 10-j symbols depend on the choice of bases $\{\Gamma_{ijkl}\}$ and $\{\overline{\Gamma}_{ijkl}\}$. The coherence condition on the 10-j symbols is given by the commutativity of the Stasheff polytope $K_5$ [135–137], which is the higher-dimensional anologue of the pentagon equation [138].

A fusion 2-category $\mathcal{C}$ is further equipped with the data regarding the duality of objects and 1-morphisms. These data allow us to define the quantum dimensions of objects and 1-morphisms. If every object has the same quantum dimension as its dual, $\mathcal{C}$ is said to be spherical. Unless otherwise stated, we suppose that a fusion 2-category is spherical in this paper.

---

[40]We note that $z_{\pm}$ in (3.2) and (3.3) is slightly different from the original definition of the 10-j symbols in [3]. Namely, the 10-j symbols in [3] are defined as $z_{\pm}(\Gamma; [01234])$ divided by the quantum dimension of the 1-morphism $\Gamma_{024}$, supposing that $\Gamma_{ijkl}$ and $\overline{\Gamma}_{ijkl}$ are chosen to be the dual bases with respect to the non-degenerate trace pairing of 2-morphisms. For more details, see [62, Section 2.1], in which the original 10-j symbols were denoted by $z_{\pm}$.

**Example.** A simple example of a fusion 2-category is $2\mathsf{Vec}_G^\omega$, where $G$ is a finite group and $\omega \in Z^4(G, \mathrm{U}(1))$. Mathematically, $2\mathsf{Vec}_G^\omega$ is defined as the 2-category of $G$-graded finite semisimple 1-categories with the 10-j symbols twisted by the 4-cocycle $\omega$. More explicitly, the basic data of $2\mathsf{Vec}_G^\omega$ are given as follows:

- Simple objects are labeled by elements of $G$ and are denoted by $\{D_2^g \mid g \in G\}$.[41]

- For each triple $g, h, k \in G$, there is only one simple 1-morphism from $D_2^g \Box D_2^h$ to $D_2^k$ when $gh = k$, while otherwise there is no non-zero 1-morphism between them.

- The 2-morphism represented by the following diagram is unique up to a scalar:

$$
\tag{3.4}
$$

Here, $1_{g \cdot h}$ denotes the non-zero simple 1-morphism from $D_2^g \Box D_2^h$ to $D_2^{gh}$.

- For a given choice of the basis 2-morphisms (3.4), the 10-j symbols are given by

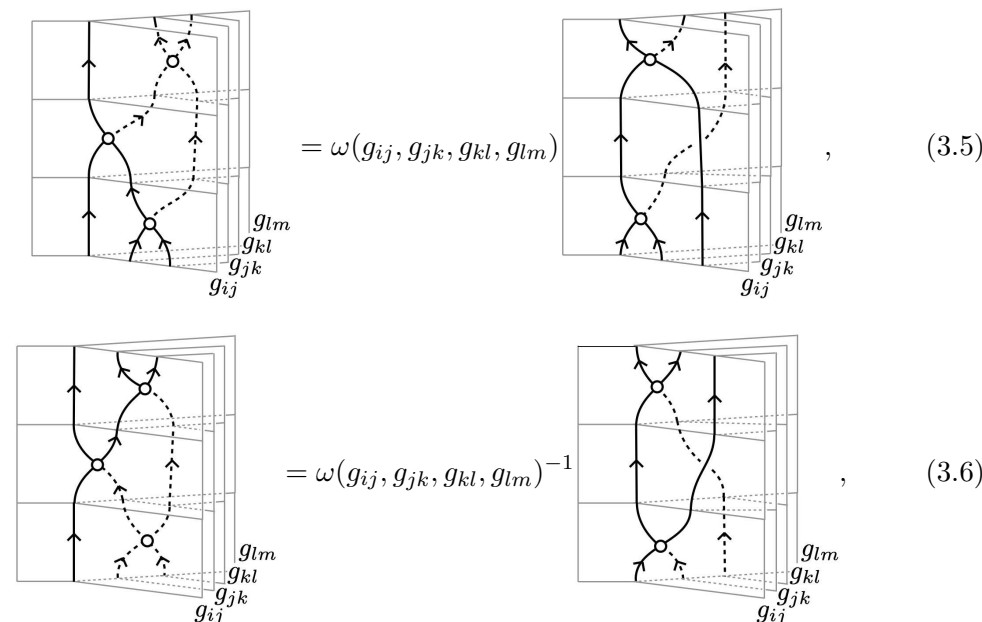

$$
= \omega(g_{ij}, g_{jk}, g_{kl}, g_{lm}) \qquad , \tag{3.5}
$$

$$
= \omega(g_{ij}, g_{jk}, g_{kl}, g_{lm})^{-1} \qquad , \tag{3.6}
$$

where we omitted the labels of various surfaces and lines because they are uniquely determined by $g_{ij}, g_{jk}, g_{kl}, g_{lm} \in G$. The consistency condition on the 10-j symbols reduces to the cocycle condition on $\omega$.

---

[41] As an object of $2\mathsf{Vec}_G^\omega$, $D_2^g$ is regarded as a finite semisimple 1-category $\mathsf{Vec}$ with the grading $g \in G$.

## 3.2 Separable Algebras

In this subsection, we recall the definition of a separable algebra in a fusion 2-category [134, 139]. The role of a separable algebra in this paper is twofold. On the one hand, it allows us to define the generalized gauging of fusion 2-category symmetries. On the other hand, it allows us to construct lattice models of gapped phases with fusion 2-category symmetries.

An **algebra** $A$ in a fusion 2-category $\mathcal{C}$ consists of the following data:

- The underlying object of $A$, which we also denote by $A$

- The multiplication 1-morphism $m : A\square A \to A$

- The associativity 2-isomorphism $\mu$, which is represented diagrammatically as

$$\tag{3.7}$$

The 2-isomorphism $\mu$ has to satisfy the following consistency condition:

$$\tag{3.8}$$

More precisely, an algebra $A \in \mathcal{C}$ is also equipped with the unit 1-morphism $i : I \to A$, where $I$ denotes the unit object of $\mathcal{C}$. The unit 1-morphism $i$ is associated with structure 2-isomorphisms that satisfy appropriate consistency conditions [134]. We do not write them down here because we will not use them explicitly in what follows.

An algebra $A$ is said to be **rigid** if its multiplication 1-morphism $m : A\square A \to A$ admits a right adjoint $m^* : A \to A\square A$. The unit and counit of the adjoint pair $(m, m^*)$ are denoted by $\eta : 1_{A\square A} \to m^* \circ m$ and $\epsilon : m \circ m^* \to 1_A$, respectively. These 2-morphisms must satisfy the following condition called the cusp equations:

$$= \mathrm{id}_m, \qquad = \mathrm{id}_{m^*}. \tag{3.9}$$

Here and in what follows, we omit to draw the surfaces labeled by $A$ when no confusion can arise.[42] By using $\eta$ and $\epsilon$, we can define the duals of $\mu$ and $\mu^{-1}$, which are denoted by $\psi_r$ and $\psi_l$ respectively:

$$(3.10)$$

$$(3.11)$$

These 2-morphisms must satisfy consistency conditions analogous to the higher associativity condition (3.8). Specifically, $\psi_r$ and $\psi_l$ must satisfy the following three conditions:

$$(3.12)$$

$$(3.13)$$

$$(3.14)$$

---

[42]One can always restore the surfaces from the lines labeled by $m$ and $m^*$.

For later use, we also define another 2-morphism $\mu^*$ by

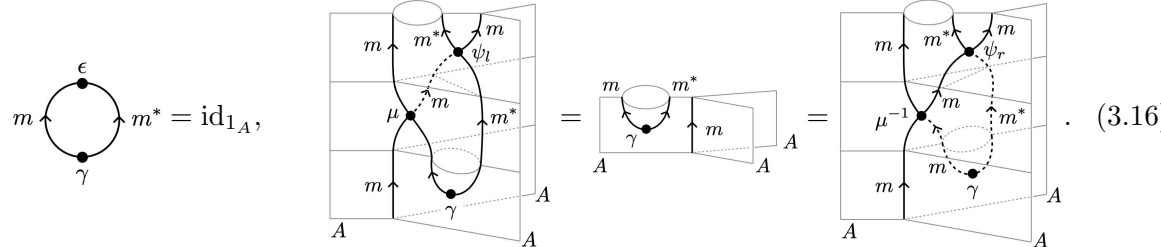

$$(3.15)$$

A rigid algebra $A$ is said to be **separable** if it is further equipped with a 2-morphism $\gamma : 1_A \Rightarrow m \circ m^*$ that satisfies

$$(3.16)$$

Every rigid algebra in a fusion 2-category over $\mathbb{C}$ is known to be separable [107, 139]. Namely, for any rigid algebra $A \in \mathcal{C}$, there always exists a 2-morphism $\gamma$ that satisfies (3.16). See [134] for more details on separable algebras, and see also [140, 141] for a closely related notion called orbifold datum.

**Example.** A separable algebra $A$ in $2\mathsf{Vec}_G$ is a $G$-graded multifusion category [133, 134]. The multiplication 1-morphism $m : A \square A \to A$ is given by the tensor product of $A$, while the associativity 2-isomorphism $\mu$ is given by the associator of $A$. See section 5.1.1 for more details on the separable algebra structure on $A \in 2\mathsf{Vec}_G$. More generally, a separable algebra in $2\mathsf{Vec}_G^\omega$ is a $G$-graded $\omega$-twisted multifusion category [115], see section 6.1.1 for more details.

## 3.3   Module 2-Categories

In this subsection, we recall the basic data of a module 2-category over a fusion 2-category $\mathcal{C}$. We refer the reader to [133] for more details on module 2-categories.

A right $\mathcal{C}$-module 2-category $\mathcal{M}$ is a (finite semisimple) 2-category on which $\mathcal{C}$ acts from the right. In what follows, a $\mathcal{C}$-module 2-category always means a right $\mathcal{C}$-module 2-category unless otherwise stated. We denote the action of $\mathcal{C}$ on $\mathcal{M}$ as $\triangleleft: \mathcal{M} \times \mathcal{C} \to \mathcal{M}$. The basic data of a $\mathcal{C}$-module 2-category $\mathcal{M}$ can be summarized as follows:

- The set of simple objects of $\mathcal{M}$

- The set of simple 1-morphisms from $M_{ij} \triangleleft \Gamma_{jk}$ to $M_{ik}$ for all simple objects $M_{ij}, M_{ik} \in \mathcal{M}$ and $\Gamma_{jk} \in \mathcal{C}$

- The set of 2-morphisms from $M_{ikl} \circ (M_{ijk} \square 1_{\Gamma_{jkl}})$ to $M_{ijl} \circ (1_{M_{ij}} \square M_{jkl})$

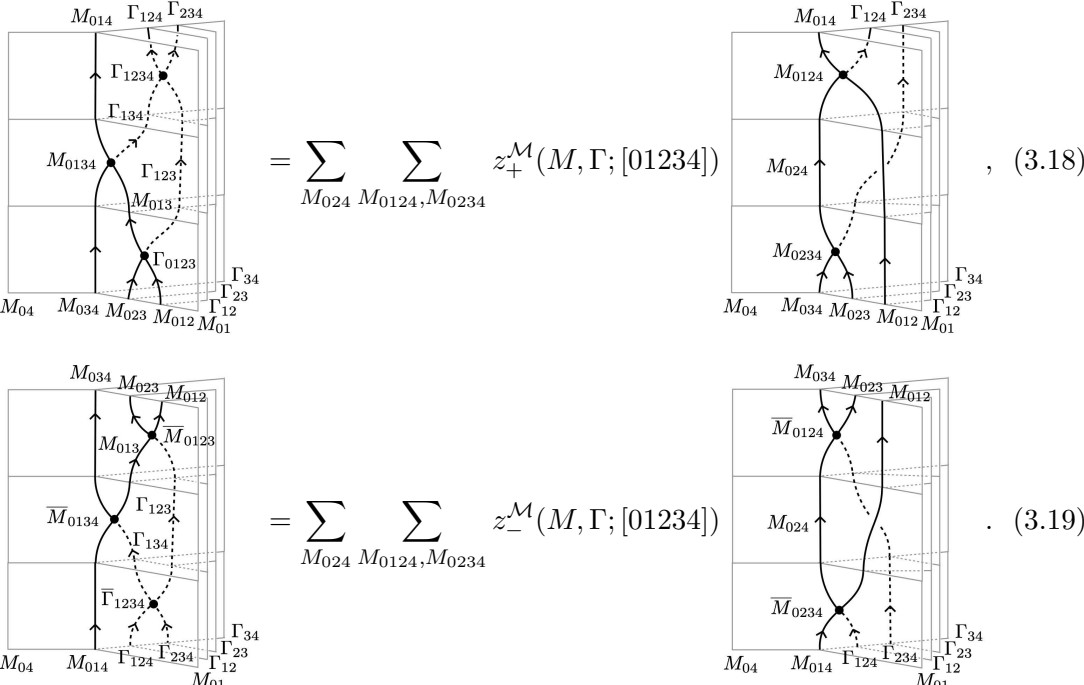

$$(3.17)$$

for all simple 1-morphisms $M_{ijk} \in \mathrm{Hom}_{\mathcal{M}}(M_{ij} \triangleleft \Gamma_{jk}, M_{ik})$, $M_{ikl} \in \mathrm{Hom}_{\mathcal{M}}(M_{ik} \triangleleft \Gamma_{kl}, M_{il})$, $M_{ijl} \in \mathrm{Hom}_{\mathcal{M}}(M_{ij} \triangleleft \Gamma_{jl}, M_{il})$, and $M_{jkl} \in \mathrm{Hom}_{\mathcal{M}}(M_{jk} \triangleleft \Gamma_{kl}, M_{jl})$

- The complex numbers $z_{\pm}^{\mathcal{M}}(\Gamma, M; [01234])$ defined by

$$= \sum_{M_{024}} \sum_{M_{0124}, M_{0234}} z_+^{\mathcal{M}}(M, \Gamma; [01234]) \qquad , \quad (3.18)$$

$$= \sum_{M_{024}} \sum_{M_{0124}, M_{0234}} z_-^{\mathcal{M}}(M, \Gamma; [01234]) \qquad . \quad (3.19)$$

We call $z_{\pm}^{\mathcal{M}}$ module 10-j symbols. Here, $\{M_{ijkl}\}$ is a basis of the vector space $\mathrm{Hom}(M_{ikl} \circ (M_{ijk} \square 1_{\Gamma_{kl}}), M_{ijl} \circ (1_{M_{ij}} \square M_{jkl}))$ and $\{\overline{M}_{ijkl}\}$ is a basis of the dual vector space. The summation on the right-hand side is taken over all simple 1-morphisms $M_{024}$ and basis 2-morphisms $M_{0124}$ and $M_{0234}$. The module 10-j symbols of $\mathcal{M}$ are required to satisfy coherence conditions similar to those for the 10-j symbols of $\mathcal{C}$.

For a $\mathcal{C}$-module 2-category $\mathcal{M}$, the 2-category $\mathsf{Fun}_{\mathcal{C}}(\mathcal{M}, \mathcal{M})$ of $\mathcal{C}$-module endofunctors of $\mathcal{M}$ is called the dual of $\mathcal{C}$ with respect to $\mathcal{M}$ and is denoted by

$$\mathcal{C}_{\mathcal{M}}^* := \mathsf{Fun}_{\mathcal{C}}(\mathcal{M}, \mathcal{M}). \qquad (3.20)$$

We note that $\mathcal{C}_{\mathcal{M}}^*$ is a multifusion 2-category in general. In particular, $\mathcal{C}_{\mathcal{M}}^*$ is a fusion 2-category when $\mathcal{M}$ is indecomposable [142].

Any module 2-category over $\mathcal{C}$ is equivalent to the 2-category $_A\mathcal{C}$ of left $A$-modules in $\mathcal{C}$, where $A$ is a separable algebra in $\mathcal{C}$ [133]. The right $\mathcal{C}$-action on $_A\mathcal{C}$ is defined by using the tensor product of $\mathcal{C}$:

$$M \lhd \Gamma := M \square \Gamma, \qquad \forall M \in {}_A\mathcal{C}, \quad \forall \Gamma \in \mathcal{C}. \tag{3.21}$$

When $\mathcal{M}$ is equivalent to $_A\mathcal{C}$ as a $\mathcal{C}$-module 2-category, the dual 2-category $\mathcal{C}_{\mathcal{M}}^*$ is monoidally equivalent to $_A\mathcal{C}_A$, the 2-category of $(A, A)$-bimodules in $\mathcal{C}$ [142].

**Example.** Any module 2-category over $2\mathsf{Vec}_G$ is equivalent to $_A(2\mathsf{Vec}_G)$ for some $G$-graded multifusion category $A$. The concrete data of $_A(2\mathsf{Vec}_G)$ are summarized in section 5.1.2 assuming that $A$ is fusion, i.e., the unit object of $A$ is simple. More generally, any module 2-category over $2\mathsf{Vec}_G^\omega$ is equivalent to $_A(2\mathsf{Vec}_G^\omega)$ for some $G$-graded $\omega$-twisted multifusion category. The concrete data of $_A(2\mathsf{Vec}_G^\omega)$ are summarized in section 6.1.2 assuming that $A$ is fusion. When $A$ is fusion, the module 2-category $_A(2\mathsf{Vec}_G^\omega)$ is indecomposable, and thus the dual 2-category $_A(2\mathsf{Vec}_G^\omega)_A$ becomes a fusion 2-category. The basic data of the dual 2-category $_A(2\mathsf{Vec}_G^\omega)_A$ are partially described in section 5.3 when $\omega$ is trivial and in section 6.3 when $\omega$ is non-trivial.

## 3.4 Tensor Networks

In this subsection, we recall the definition of tensor network operators, including tensor network states as a special case. For a recent review of the tensor networks, see, e.g., [84]. In later sections, we will use the tensor networks to represent the generalized gauging operators for (possibly anomalous) finite group symmetries. The tensor networks will also be used to represent the ground states of gapped phases with fusion 2-category symmetries.

As a warm-up, we start from the tensor network operators on a one-dimensional lattice. For simplicity, we will assume translation invariance and consider operators whose source and target vector spaces are given by

$$\mathcal{H}^s := \bigotimes_{\text{all sites}} \mathcal{H}^s_{\text{phys}}, \qquad \mathcal{H}^t := \bigotimes_{\text{all sites}} \mathcal{H}^t_{\text{phys}}. \tag{3.22}$$

Here, $\mathcal{H}^s_{\text{phys}}$ and $\mathcal{H}^t_{\text{phys}}$ are finite-dimensional vector spaces on each site. The tensor products are taken over all sites of the lattice. We note that the source and target vector spaces can be different in general.[43]

---

[43]In particualr, the generalized gauging operators that we will discuss in later sections typically have different source and target vector spaces.

A tensor network operator in 1+1d, known as a matrix product operator, is represented by the following diagram:

$$\widehat{\mathcal{O}} = \quad \text{(diagram)} \quad .$$ (3.23)

The vertical and horizontal edges are called physical legs and virtual bonds, respectively. The physical leg at the bottom takes values in the source vector space $\mathcal{H}^s_{\text{phys}}$, while the physical leg at the top takes values in the target vector space $\mathcal{H}^t_{\text{phys}}$. On the other hand, the virtual bond takes values in another finite-dimensional vector space $\mathcal{H}_{\text{virt}}$. We note that $\mathcal{H}_{\text{virt}}$ has nothing to do with the physical state spaces. Each vertex of the diagram represents a four-leg tensor, which takes values in the complex numbers:

$$y_{i-1} \quad \begin{matrix} x_i \\ \bullet \\ x'_i \end{matrix} \quad y_i \quad = W^{x_i, x'_i}_{y_{i-1}, y_i} \in \mathbb{C}.$$ (3.24)

Here, the labels $x_i$, $x'_i$, and $y_i$ are basis elements of $\mathcal{H}^s_{\text{phys}}$, $\mathcal{H}^t_{\text{phys}}$, and $\mathcal{H}_{\text{virt}}$. The matrix element of the operator $\widehat{\mathcal{O}}$ is then given by

$$\langle \{x_i\} | \widehat{\mathcal{O}} | \{x'_i\} \rangle = \sum_{\{y_i\}} \prod_i W^{x_i, x'_i}_{y_{i-1}, y_i},$$ (3.25)

where the summation is taken over all basis elements and the product is taken over all sites.

When the source vector space is trivial, i.e., when $\mathcal{H}^s_{\text{phys}} = \mathbb{C}$, the operator $\widehat{\mathcal{O}}$ can be canonically identified with an element of the target vector space $\bigotimes \mathcal{H}^t_{\text{phys}}$. This vector is represented by a tensor network of the form

$$|\mathcal{O}\rangle = \quad \text{(diagram)} \quad ,$$ (3.26)

where each three-leg tensor is defined by

$$y_{i-1} \quad \begin{matrix} x_i \\ \bullet \end{matrix} \quad y_i \quad = W^{x_i}_{y_{i-1}, y_i} \in \mathbb{C}.$$ (3.27)

The wavefunction of the tensor network state (3.26) is then given by

$$\langle \{x_i\} | \mathcal{O} \rangle = \sum_{\{y_i\}} \prod_i W^{x_i}_{y_{i-1}, y_i}.$$ (3.28)

This tensor network state is known as a matrix product state.

The above definition of tensor network operators in 1+1d can be generalized to any dimension. As in the 1+1d case, a tensor network operator in general dimensions consists of local

tensors that connect physical legs via virtual bonds. For example, a tensor network operator in 2+1d typically looks like

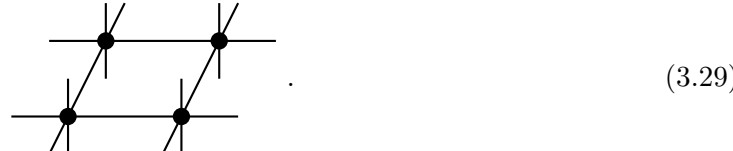

$$(3.29)$$

Given a tensor network representation of an operator $\widehat{\mathcal{O}}$, the matrix element $\langle\{x_i\}|\widehat{\mathcal{O}}|\{x_i'\}\rangle$ is defined as follows.

- We first assign the basis elements $x_i' \in \mathcal{H}_{\text{phys}}^s$ and $x_i \in \mathcal{H}_{\text{phys}}^t$ to the physical legs corresponding to the source and target vector spaces.

- We also assign the basis elements $y_i \in \mathcal{H}_{\text{virt}}$ to the virtual bonds.

- We then take the product of the complex numbers associated with the local tensors for the given configuration of the basis elements. We denote the resulting complex number by $W(\{x_i\}, \{x_i'\}; \{y_i\})$.

- The matrix element $\langle\{x_i\}|\widehat{\mathcal{O}}|\{x_i'\}\rangle$ is then given by the sum

$$\langle\{x_i\}|\widehat{\mathcal{O}}|\{x_i'\}\rangle = \sum_{\{y_i\}} W(\{x_i\}, \{x_i'\}; \{y_i\}), \qquad (3.30)$$

where the summation is taken over all configurations of the basis elements on the virtual bonds.

The above definition can also be applied to the case where $\mathcal{H}_{\text{phys}}^s$, $\mathcal{H}_{\text{phys}}^t$, and $\mathcal{H}_{\text{virt}}$ depend on the positions on the lattice.

These tensor networks provide a concise way to express (generalized) gauging operators [52, 55, 98–100, 143], non-invertible symmetry operators [52, 55, 94–96, 98–100], and gapped ground states of lattice models [144–148]. In particular, they are useful for extracting the universal properties of gapped phases with and without symmetries [85–93, 95, 102, 103].

# 4 Generalized Gauging in Fusion Surface Model

In this section, we describe the generalized gauging of fusion 2-category symmetry in the fusion surface model [62]. We will also discuss the systematic construction of gapped phases in the gauged fusion surface model.

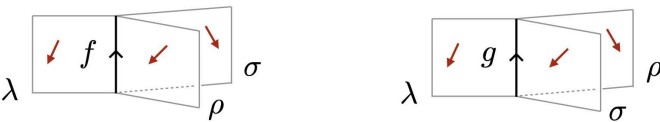

Figure 7: The input data of the fusion surface model. The surfaces represent objects $\rho$, $\sigma$, and $\lambda$, while the lines represent 1-morphisms $f$ and $g$.

## 4.1 Fusion Surface Model

Let us first recall the definition of the fusion surface model [62], which is a 2+1d analogue of the 1+1d anyon chain model [125, 126]. For simplicity, we consider the model on a honeycomb lattice. The generalization to other lattices is straightforward.

To define the model, we first fix the following input data:

- a spherical fusion 2-category $\mathcal{C}$

- objects $\rho, \sigma, \lambda \in \mathcal{C}$

- 1-morphisms $f \in \mathrm{Hom}_{\mathcal{C}}(\rho \square \sigma, \lambda)$ and $g \in \mathrm{Hom}_{\mathcal{C}}(\sigma \square \rho, \lambda)$

See figure 7 for a diagrammatic representation of the above data. We note that objects $\rho$, $\sigma$, and $\lambda$ are not necessarily simple, meaning that they can be isomorphic to finite direct sums of simple objects. Similarly, 1-morphisms $f$ and $g$ are not necessarily simple.

The state space $\mathcal{H}$ of the model is the vector space of 2-morphisms that are represented by the fusion diagrams of the following form:

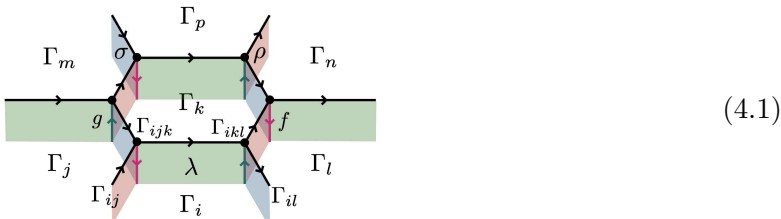

(4.1)

The dynamical variables of the model are living on the plaquettes, edges, and vertices of the honeycomb lattice. These dynamical variables are labeled by objects $\{\Gamma_i \mid i \in P\}$, 1-morphisms $\{\Gamma_{ij} \mid [ij] \in E\}$, and 2-morphisms $\{\Gamma_{ijk} \mid [ijk] \in V\}$ of the input fusion 2-category $\mathcal{C}$. Here, $P$, $E$, and $V$ denote the set of all plaquettes, the set of all edges, and the set of all vertices. The configuration of the dynamical variables is constrained by the input data $(\rho, \sigma, \lambda; f, g)$ so that the fusion diagram in (4.1) makes sense. For example, $\Gamma_{ij}$ in (4.1) is a 1-morphism from $\Gamma_i \square \rho$ to $\Gamma_j$.[44] We note that different configurations of dynamical variables $\{\Gamma_i, \Gamma_{ij}, \Gamma_{ijk}\}$ may lead to the same 2-morphism, i.e., the same state in $\mathcal{H}$.

---

[44] All the colored surfaces in (4.1) are oriented from back to front, while all the white plaquettes are oriented from bottom to top.

The state space $\mathcal{H}$ can be described more explicitly as follows. First, we choose a representative of each connected component of simple objects of $\mathcal{C}$. In addition, we also choose a representative of each isomorphism class of simple 1-morphisms of $\mathcal{C}$. We then consider a vector space $\mathcal{H}'$ spanned by all possible configurations of the dynamical variables on the honeycomb lattice. Here, the dynamical variables on the plaquettes and edges take values in the set of representatives that we chose, while those on the vertices take values in the set of basis 2-morphisms that fit into the fusion diagram. To obtain the state space $\mathcal{H}$, we apply a projector $\widehat{\pi}_{\mathrm{LW}}$ to $\mathcal{H}'$ so that the states in $\mathcal{H}'$ are identified with each other in $\mathcal{H}$ if they give rise to the same 2-morphism in $\mathcal{C}$. That is, we have

$$\mathcal{H} \cong \widehat{\pi}_{\mathrm{LW}} \mathcal{H}'. \tag{4.2}$$

The projector $\widehat{\pi}_{\mathrm{LW}}$ is given by the product of local projectors on all plaquettes, i.e.,

$$\widehat{\pi}_{\mathrm{LW}} = \prod_{i \in P} \widehat{B}_i, \tag{4.3}$$

where $\widehat{B}_i$ is the Levin-Wen plaquette operator defined by [64]

$$\tag{4.4}$$

Here, $\dim(a)$ denotes the quantum dimension of an object $a$ of the fusion 1-category $\mathrm{End}(\Gamma_i)$, and $\mathcal{D}_{\mathrm{End}(\Gamma_i)} := \sum_{a \in \mathrm{End}(\Gamma_i)} \dim(a)^2$ is the total dimension of $\mathrm{End}(\Gamma_i)$. The summation on the right-hand side is taken over all (isomorphism classes of) simple objects of $\mathrm{End}(\Gamma_i)$. The Levin-Wen operator satisfies $\widehat{B}_i^2 = 1$ and $\widehat{B}_i \widehat{B}_j = \widehat{B}_j \widehat{B}_i$ for all $i, j \in P$ [64].

We note that the state space $\widehat{\pi}_{\mathrm{LW}} \mathcal{H}'$ does not depend on the choice of representatives up to isomorphism. An isomorphism between the state spaces for two different choices of representatives is given by $\prod_i \widehat{C}_i$, where $\widehat{C}_i$ is a Levin-Wen-like operator defined by

$$\tag{4.5}$$

Here, $\Gamma_i$ and $\Gamma'_i$ are different representatives of the same connected component. The quantities $\dim_{\mathcal{C}}(a)$ and $\dim_{\mathcal{C}}(\Gamma_i)$ are the quantum dimensions of a 1-morphism $a$ and an object $\Gamma_i$ in $\mathcal{C}$.[45]

---

[45]The ratio $\dim_{\mathcal{C}}(a)/\dim_{\mathcal{C}}(\Gamma_i)$ is the scalar that appears if we shrink the loop of $a$ into a point. In particular, when $\Gamma_i = \Gamma'_i$, we have $\dim(a) = \dim_{\mathcal{C}}(a)/\dim_{\mathcal{C}}(\Gamma_i)$.

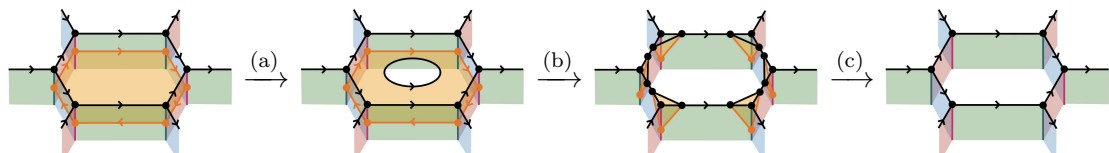

Figure 8: The Hamiltonian is evaluated in the following steps: (a) the partial fusion of surfaces, (b) the partial fusion of lines, (c) the bubble removal using the 10-j move defined by (3.2) and (3.3).

The operator $\widehat{C}_i$ is invertible on $\widehat{\pi}_{\mathrm{LW}}\mathcal{H}'$, with the inverse given by the same operator with $\Gamma_i$ and $\Gamma_i'$ exchanged.

The Hamiltonian of the model is defined as a 2-morphism in $\mathcal{C}$, whose action on the state space $\mathcal{H}$ is given by the composition of 2-morphisms. For simplicity, in this paper, we consider a Hamiltonian of the form

$$H = -\sum_{i\in P}\widehat{\mathsf{h}}_i, \tag{4.6}$$

where $\widehat{\mathsf{h}}_i$ acts on the dynamical variables only around a single plaquette $i$. Each local term $\widehat{\mathsf{h}}_i$ in the Hamiltonian can be expressed diagrammatically as

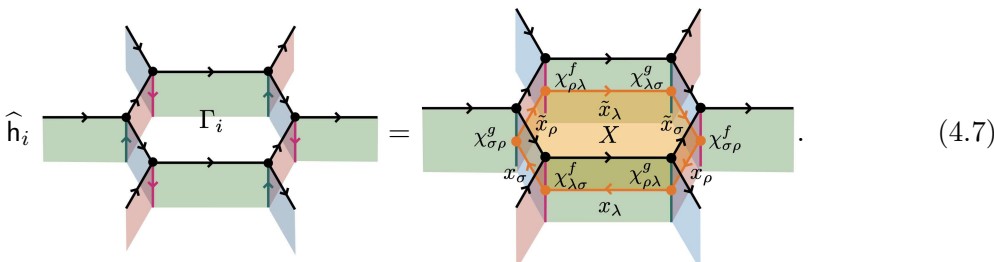
$$\tag{4.7}$$

Here, object $X$, 1-morphisms $\{x_\rho, x_\sigma, x_\lambda, \tilde{x}_\rho, \tilde{x}_\sigma, \tilde{x}_\lambda\}$, and 2-morphisms $\{\chi^f_{\lambda\sigma}, \chi^g_{\sigma\rho}, \chi^f_{\rho\lambda}, \chi^g_{\lambda\sigma}, \chi^f_{\sigma\rho}, \chi^g_{\rho\lambda}\}$ can be chosen arbitrarily as long as the above fusion diagram makes sense. This Hamiltonian is evaluated by fusing the middle orange surface into the honeycomb lattice from below, see figure 8. One can explicitly write down the matrix elements of $\widehat{\mathsf{h}}_i$ in terms of 10-j symbols of $\mathcal{C}$. We refer the reader to [62] for a more detailed expression of the Hamiltonian.

**Symmetry Category.** By construction, the fusion surface model obtained from $\mathcal{C}$ has a fusion 2-category symmetry $\mathcal{C}$. The symmetry operators are represented by surfaces and lines labeled by objects and 1-morphisms of $\mathcal{C}$. They act on states by the fusion from above, see figure 9. These symmetry operators automatically commute with the Hamiltonian due to the coherence condition on the 10-j symbols. Intuitively, this commutativity follows from the fact that the symmetry operators act from above while the Hamiltonian acts from below.

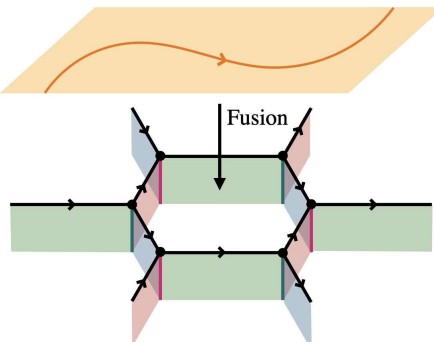

Figure 9: The symmetry action is defined by the fusion of topological surface/line operators from above. A topological surface can generally have topological lines on it.

**Symmetry TFT Interpretation.** The fusion surface model can be obtained from the symmetry TFT construction [62]. The symmetry TFT for this model is the 4d Douglas-Reutter state sum TFT based on a spherical fusion 2-category $\mathcal{C}$ [3]. We believe that the braided fusion 2-category of topological defects associated with this state sum TFT is described by the Drinfeld center of $\mathcal{C}$ and thus denote it by $\mathcal{Z}(\mathcal{C})$.[46] To obtain the fusion surface model, we put the Douglas-Reutter TFT $\mathcal{Z}(\mathcal{C})$ on a four-dimensional slab $I \times M_3$, where $I = [0, 1]$ is a finite interval and $M_3$ is a three-dimensional oriented manifold. On the left boundary $\{0\} \times M_3$, we impose the Dirichlet boundary condition of $\mathcal{Z}(\mathcal{C})$, which is a canonical topological boundary condition labeled by a regular $\mathcal{C}$-module 2-category. On this boundary, the topological surfaces, topological lines, and topological junctions between them form the fusion 2-category $\mathcal{C}$, which describes the symmetry of the 2+1d model. On the other hand, on the right boundary $\{1\} \times M_3$, we impose another boundary condition obtained by decorating the Dirichlet boundary with a defect network shown in figure 10. This boundary condition is not necessarily topological. Now, since the 4d bulk is topological, we can squash the interval $I$ into a point. This produces a 3d classical statistical mechanical model called the 3d height model [62], which is a higher dimensional generalization of the 2d height model [128]. The anisotropic limit of the 3d height model gives us the corresponding 2+1d quantum lattice model, which is the fusion surface model [62].

## 4.2 Gauged Fusion Surface Model

We can generalize the original fusion surface model by using a module 2-category $\mathcal{M}$ over the input fusion 2-category $\mathcal{C}$. The new model can be regarded as a gauged version of the original model. Hence, we call it the gauged fusion surface model. This is a 2+1d analogue of

---

[46]This is a natural generalization of the fact that the 3d Turaev-Viro-Barrett-Westbury TFT based on a spherical fusion 1-category [149, 150] is described by its Drinfeld center [151–153].

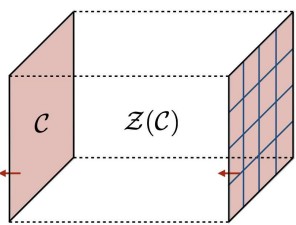 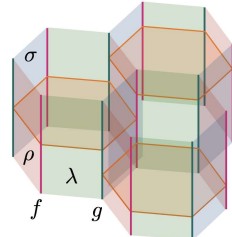

Figure 10: The symmetry TFT construction of the fusion surface model. The bulk TFT is the 4d Douglas-Reutter TFT based on $\mathcal{C}$. The left boundary is the Dirichlet boundary, i.e., a topological boundary labeled by the regular $\mathcal{C}$-module 2-category. The right boundary is the Dirichlet boundary decorated with the defect network shown on the right.

the gauging of fusion 1-category symmetries in 1+1d anyon chain models [42, 52, 55].[47] The gauged model reduces to the original fusion surface model when $\mathcal{M}$ is the regular $\mathcal{C}$-module 2-category. In later sections, we will discuss the gauged fusion surface models for $\mathcal{C} = 2\mathsf{Vec}_G^\omega$ in detail. The gauged fusion surface models for another class of fusion 2-category $\mathcal{C} = \Sigma\mathcal{B}$, the codensation completion of a braided fusion 1-category $\mathcal{B}$ [3, 108], were studied in [81].

The dynamical variables of the gauged model are labeled by objects, 1-morphisms, and 2-morphisms of $\mathcal{M}$. The state space $\mathcal{H}$ of the new model is given by the vector space of 2-morphisms that are represented by the fusion diagrams of the following form:

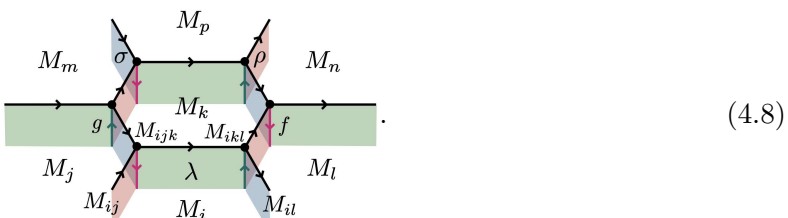 (4.8)

Here, $M_i$, $M_{ij}$, and $M_{ijk}$ are objects, 1-morphisms, and 2-morphisms of $\mathcal{M}$. We note that the input data $(\rho, \sigma, \lambda; f, g)$ remain the same as those of the original fusion surface model. Namely, $\rho$, $\sigma$, and $\lambda$ are objects of $\mathcal{C}$, and $f$ and $g$ are 1-morphisms of $\mathcal{C}$ as shown in figure 7.

As in the original fusion surface model, we can realize the state space $\mathcal{H}$ as the image of a projector $\widehat{\pi}_{\mathrm{LW}}$ acting on a larger vector space $\mathcal{H}'$ spanned by all possible configurations of dynamical variables. Here, the dynamical variables on (1) plaquettes, (2) edges, and (3) vertices take values in (1) the set of representatives of connected components of simple objects of $\mathcal{M}$, (2) the set of representatives of isomorphism classes of simple 1-morphisms of $\mathcal{M}$, and (3) the set of basis 2-morphisms of $\mathcal{M}$, respectively. The projector $\widehat{\pi}_{\mathrm{LW}}$ acting on $\mathcal{H}'$ is given by the product of Levin-Wen plaquette operators $\widehat{B}_i$, which are defined by the same equation

---

[47]In continuum QFTs, the gauging of fusion 1-category symmetries in 1+1d was introduced in [154–156].

as (4.4) except that the dynamical variables are now labeled by objects and morphisms of $\mathcal{M}$ rather than those of $\mathcal{C}$.

The Hamiltonian of the new model is specified by the same data $\{X, x_\rho, \cdots, \chi^f_{\lambda\sigma}, \cdots\}$ as those in the original model. More specifically, the Hamiltonian of the gauged model is given by

$$H = -\sum_i \widehat{\mathsf{h}}_i, \tag{4.9}$$

where the action of $\widehat{\mathsf{h}}_i$ on a state (4.8) is defined by the same diagram as that in (4.7):

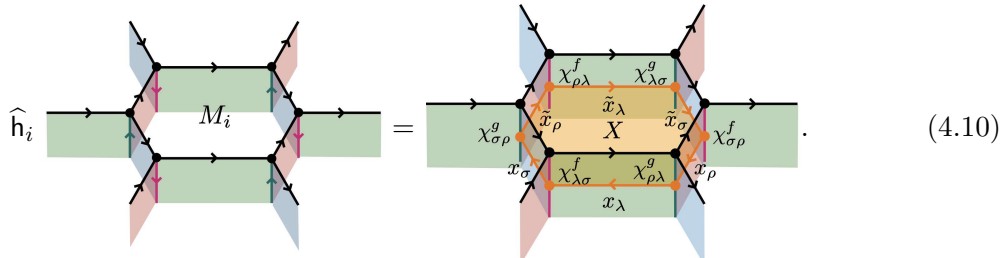

This Hamiltonian can be evaluated explicitly using the module 10-j symbols defined by (3.18) and (3.19).

**Symmetry Category.** To identify the symmetry of the above model, we notice that a right $\mathcal{C}$-module 2-category $\mathcal{M}$ is automatically a left module 2-category over the dual 2-category $\mathcal{C}^*_{\mathcal{M}}$. The left $\mathcal{C}^*_{\mathcal{M}}$-module structure on $\mathcal{M}$ is given by the identity functor id : $\mathcal{C}^*_{\mathcal{M}} \to \mathrm{End}(\mathcal{M})$.[48] This left $\mathcal{C}^*_{\mathcal{M}}$-action is compatible with the right $\mathcal{C}$-action on $\mathcal{M}$, meaning that $\mathcal{M}$ is a $(\mathcal{C}^*_{\mathcal{M}}, \mathcal{C})$-bimodule 2-category [142]. This implies that $\mathcal{C}^*_{\mathcal{M}}$ naturally acts on the state space of our model, and this action commutes with the Hamiltonian (4.10). Therefore, the new model constructed from a $\mathcal{C}$-module 2-category $\mathcal{M}$ has a fusion 2-category symmetry $\mathcal{C}^*_{\mathcal{M}}$.

**Symmetry TFT Interpretation.** From the SymTFT perspective, changing the module 2-category corresponds to changing the topological boundary condition on the symmetry boundary of the 4d bulk TFT. Physically, this operation is interpreted as stacking a 3d $\mathcal{C}^{\mathrm{op}}$-symmetric non-chiral TFT on the Dirichlet boundary and gauging the canonical algebra

$$A_{\mathrm{can}} := \underline{\mathrm{End}}_\mathcal{C}(I) \in \mathcal{C} \boxtimes \mathcal{C}^{\mathrm{op}} \tag{4.11}$$

on the boundary. Here, $\mathcal{C}^{\mathrm{op}}$ denotes the 2-category $\mathcal{C}$ equipped with the opposite tensor product, and $\underline{\mathrm{End}}_\mathcal{C}(I)$ is the internal End of the unit object $I \in \mathcal{C}$, where $\mathcal{C}$ is viewed as a

---

[48]For example, the action of $F \in \mathcal{C}^*_{\mathcal{M}}$ on $M \in \mathcal{M}$ is given by $F \triangleright M = F(M)$.

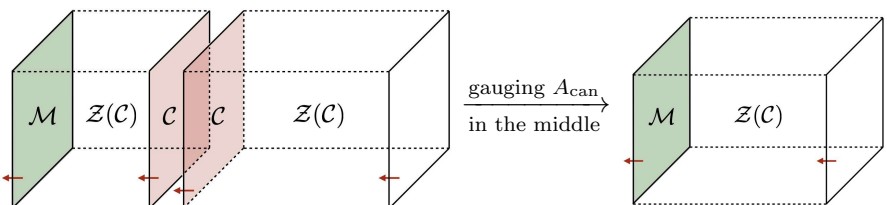

Figure 11: The topological boundary labeled by a $\mathcal{C}$-module 2-category $\mathcal{M}$ is obtained by stacking a 3d $\mathcal{C}^{\mathrm{op}}$-symmetric TFT on the Dirichlet boundary and gauging the canonical algebra $A_{\mathrm{can}} = \underline{\mathrm{End}}_{\mathcal{C}}(I) \in \mathcal{C} \boxtimes \mathcal{C}^{\mathrm{op}}$. The red interface after the gauging is labeled by a $\mathcal{C} \boxtimes \mathcal{C}^{\mathrm{op}}$-module 2-category $(C \boxtimes \mathcal{C}^{\mathrm{op}})_{A_{\mathrm{can}}}$, which is equivalent to $\mathcal{C}$ due to [133, Theorem 5.3.4]. This interface corresponds to the identity surface of the 4d TFT $\mathcal{Z}(\mathcal{C})$, and thus nothing remains at the interface after the gauging.

left $\mathcal{C} \boxtimes \mathcal{C}^{\mathrm{op}}$-module 2-category.[49] See figure 11 for a schematic picture of this operation. The lower dimensional analogue of this operation is discussed in [157, Appendix B].

When $\mathcal{C} = 2\mathsf{Vec}_G^\omega$, the canonical algebra $A_{\mathrm{can}}$ is given by

$$A_{\mathrm{can}} = \bigoplus_{g \in G} D_2^g \boxtimes \overline{D_2^g} \ \in \ 2\mathsf{Vec}_G^\omega \boxtimes (2\mathsf{Vec}_G^\omega)^{\mathrm{op}}, \tag{4.12}$$

where $D_2^g$ is a simple object of $2\mathsf{Vec}_G^\omega$ and $\overline{D_2^g}$ is its dual. Using the monoidal equivalence

$$(2\mathsf{Vec}_G^\omega)^{\mathrm{op}} \cong 2\mathsf{Vec}_G^{\omega^{-1}} : D_2^g \mapsto \overline{D_2^g}, \tag{4.13}$$

we can write the canonical algebra as

$$A_{\mathrm{can}} = \bigoplus_{g \in G} D_2^g \boxtimes D_2^g \ \in \ 2\mathsf{Vec}_G^\omega \boxtimes 2\mathsf{Vec}_G^{\omega^{-1}}. \tag{4.14}$$

This implies that the gauging of $A_{\mathrm{can}}$ reduces to the gauging of the non-anomalous diagonal $G$ subgroup when $\mathcal{C} = 2\mathsf{Vec}_G^\omega$. Therefore, changing a module 2-category is equivalent to the generalized gauging, i.e, stacking a $(2\mathsf{Vec}_G^\omega)^{\mathrm{op}} \cong 2\mathsf{Vec}_G^{\omega^{-1}}$ symmetric TFT and gauging the diagonal $G$ subgroup. See [20, Section 3] and appendix D for more details on the relation between the generalized gauging and the change of a module 2-category.

## 4.3 Gapped Phases

The fusion surface model can realize various gapped phases with fusion 2-category symmetries. In this subsection, following [62, Section 5.3], we construct commuting projector Hamiltonians for gapped phases in the gauged fusion surface model. We expect that these Hamiltonians exhaust all non-chiral gapped phases with symmetry $\mathcal{C}_{\mathcal{M}}^*$.[50]

---

[49]More concretely, $\underline{\mathrm{End}}_{\mathcal{C}}(I)$ is an object of $\mathcal{C} \boxtimes \mathcal{C}^{\mathrm{op}}$ such that $\mathrm{Hom}_{\mathcal{C}}(X \square Y, I) \cong \mathrm{Hom}_{\mathcal{C} \boxtimes \mathcal{C}^{\mathrm{op}}}(X \boxtimes Y, \underline{\mathrm{End}}_{\mathcal{C}}(I))$ for all $X, Y \in \mathcal{C}$ [133].

[50]The fusion surface model can also realize chiral topological phases, see [78, 158] for some examples.

The commuting projector model is defined by using a separable algebra $B \in \mathcal{C}$.[51] To define the state space, we choose the input data $(\rho, \sigma, \lambda; f, g)$ to be

$$\rho = \sigma = \lambda = B, \quad f = g = m. \tag{4.15}$$

The Hamiltonian is given by the sum $H = -\sum_{i \in P} \widehat{\mathsf{h}}_i$, where each term $\widehat{\mathsf{h}}_i$ is defined by[52]

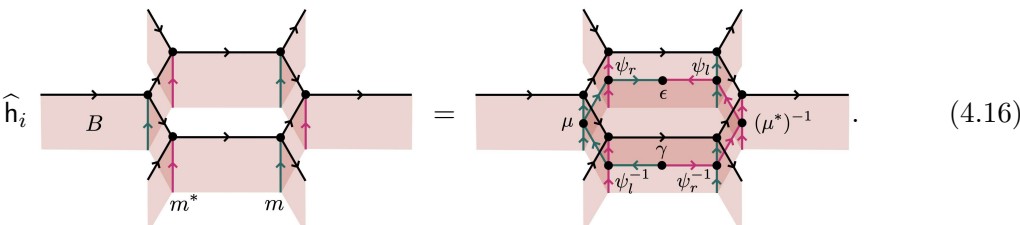

$$\tag{4.16}$$

We note that all 2-morphisms involved in the Hamiltonian are given by $\mu$, $\mu^{-1}$, or their duals. The choice of the dual 2-morphisms is uniquely determined by the configuration of surfaces labeled by $B$ and lines labeled by $m$ or $m^*$. It follows from the separable algebra structure on $B$ that the local terms $\widehat{\mathsf{h}}_i$ obey [62]

$$\widehat{\mathsf{h}}_i^2 = \widehat{\mathsf{h}}_i, \quad [\widehat{\mathsf{h}}_i, \widehat{\mathsf{h}}_j] = 0, \qquad \forall i, j \in P. \tag{4.17}$$

Thus, the Hamiltonian $H$ is a sum of commuting projectors. This implies that the above model realizes a gapped phase that admits a gapped boundary. Examples of such a gapped phase include non-chiral topologically ordered phases, symmetry protected topological phases, spontaneous symmetry-breaking phases, and mixtures thereof. In section 7, we will write down the gapped Hamiltonian (4.16) and their ground states more explicitly for $\mathcal{C} = 2\mathsf{Vec}_G^\omega$ and a general $\mathcal{M}$.

**Symmetry TFT Interpretation.** From the symmetry TFT perspective, a gapped phase is obtained by imposing topological boundary conditions on both the left and right boundaries of the 4d TFT $\mathcal{Z}(\mathcal{C})$. The topological boundary on the left is labeled by a right $\mathcal{C}$-module 2-category $\mathcal{M} = {}_A\mathcal{C}$, which determines the gauging of $\mathcal{C}$ symmetry. On the other hand, the topological boundary on the right is labeled by a left $\mathcal{C}$-module 2-category $\mathcal{N}$, which determines the gapped phase. The left $\mathcal{C}$-module $\mathcal{N}$ corresponding to the gapped Hamiltonian (4.16) is the 2-category $\mathcal{C}_B$ of right $B$-modules in $\mathcal{C}$. The topological boundary labeled by $\mathcal{C}_B$ is obtained by decorating the Dirichlet boundary of $\mathcal{Z}(\mathcal{C})$ with a fine mesh of a topological defect network whose plaquettes, edges, and vertices are labeled by $B$, $m$, and $\mu$ (or their duals). In particular,

---

[51]In this manuscript, a separable algebra that specifies a gapped phase is always denoted by $B$. On the other hand, a separable algebra that specifies generalized gauging is always denoted by $A$.

[52]The duality data such as $\gamma$ and $\epsilon$ in (4.16) were implicit in [62].

if the defect network forms a honeycomb lattice as shown in figure 10, the symmetry TFT construction produces the 2+1d gapped lattice model that we defined above. This symmetry TFT interpretation suggests that the gapped phase realized by the Hamiltonian (4.16) is characterized by the left $_A\mathcal{C}_A$-module 2-category $_A\mathcal{C}_B$.[53] The same construction of gapped phases in one lower dimension was discussed in [26, 36, 42].

# 5 Generalized Gauging of $2\mathsf{Vec}_G$ Symmetry

In this section, we discuss the generalized gauging of a non-anomalous finite group symmetry $G$ in the fusion surface model. The input fusion 2-category $\mathcal{C}$ is the 2-category $2\mathsf{Vec}_G$ of $G$-graded finite semisimple 1-categories. We denote the simple objects of $2\mathsf{Vec}_G$ as $\{D_2^g \mid g \in G\}$.[54] For any elements $g, h, k \in G$, there is an equivalence of 1-categories

$$\mathrm{Hom}_{2\mathsf{Vec}_G}(D_2^g \square D_2^h, D_2^k) \cong \mathrm{Hom}_{2\mathsf{Vec}_G}(D_2^k, D_2^g \square D_2^h) \cong \delta_{gh,k}\mathsf{Vec}. \tag{5.1}$$

Namely, there is only one isomorphism class of non-zero simple 1-morphisms between $D_2^g \square D_2^h$ and $D_2^k$ when $gh = k$, while there are no non-zero 1-morphisms between them when $gh \neq k$. The unique (non-zero) simple 1-morphism from $D_2^g \square D_2^h$ to $D_2^{gh}$ is denoted by $1_{g \cdot h}$. Similarly, the unique (non-zero) simple 1-morphism from $D_2^{gh}$ to $D_2^g \square D_2^h$ is denoted by $\overline{1}_{g \cdot h}$. Without loss of generality, we can choose the basis 2-morphisms of $2\mathsf{Vec}_G$ so that the associated 10-j symbols $z_\pm$ become trivial.[55] In what follows, the basis 2-morphisms will be represented by white dots in the fusion diagrams.

## 5.1 Preliminaries

### 5.1.1 Separable Algebras in $2\mathsf{Vec}_G$

We first review the separable algebra in $2\mathsf{Vec}_G$, which is the input data for a gapped phase with and generalized gauging of $2\mathsf{Vec}_G$ symmetry. By definition, the underlying object of a separable algebra $A \in 2\mathsf{Vec}_G$ is a $G$-graded finite semisimple 1-category:

$$A = \bigoplus_{g \in G} A_g. \tag{5.2}$$

The multiplication 1-morphism $m : A\square A \to A$ endows $A$ with a monoidal structure, whose associator is given by the associativity 2-isomorphism $\mu$. Thus, an algebra $A \in 2\mathsf{Vec}_G$ has the

---

[53]For instance, the number of ground states on a sphere should be equal to the number of connected components of $_A\mathcal{C}_B$.

[54]More precisely, $D_2^g$ is a representative of an isomorphism class of simple objects. We choose the representative $D_2^g$ so that $\dim D_2^g = 1$. In general, a simple object isomorphic to $D_2^g$ is of the form $D_2^g \square I_\lambda$, where $I_\lambda$ is a 2d invertible TFT with Euler term $\lambda \in \mathbb{R}$. The quantum dimension of $D_2^g \square I_\lambda$ is given by $\lambda^2$, which is the partition function of $I_\lambda$ on a 2-sphere. We will not use the objects $I_\lambda$ in what follows.

[55]For a generic choice of the basis 2-morphisms, the 10-j symbol of $2\mathsf{Vec}_G$ is given by a 3-coboundary on $G$.

structure of a $G$-graded finite semisimple monoidal category. The algebra $A$ is separable if and only if it is a $G$-graded multifusion category [133,134]. Furthermore, $A$ is indecomposable if and only if it is indecomposable as a $G$-graded multifusion category. For instance, $G$-graded fusion categories are indecomposable separable algebras in $2\mathsf{Vec}_G$. In this paper, $A$ is always chosen to be a $G$-graded unitary fusion category. The $G$-grading on $A$ is not necessarily faithful.

In what follows, we describe the structure of a separable algebra $A \in 2\mathsf{Vec}_G$ in more detail.

**Algebra Structure** $(A, m, \mu)$. As an object of $2\mathsf{Vec}_G$, the algebra $A$ can be written as

$$A = \bigoplus_{g \in G} \bigoplus_{a_g \in A_g} D_2^g = \bigoplus_{g \in G} (D_2^g)^{\oplus \mathrm{rank}(A_g)}. \tag{5.3}$$

Here, $a_g$ is a simple object of $A_g$,[56] and $\mathrm{rank}(A_g)$ is the number of isomorphism classes of simple objects of $A_g$. The above equation implies that each simple object $a_g \in A_g$ is associated with $D_2^g \in 2\mathsf{Vec}_G$. The object $D_2^g$ associated with $a_g \in A_g$ is denoted by $D_2^g[a_g]$.

The multiplication 1-morphism $m : A \square A \to A$ is given by the tensor product in $A$. More specifically, $m$ is defined component-wise as follows:

$$D_2^{g_{ik}}[a_{ik}] \quad \boxed{\begin{array}{c} m \uparrow \\[2mm] \end{array}} \quad \begin{array}{c} D_2^{g_{jk}}[a_{jk}] \\ D_2^{g_{ij}}[a_{ij}] \end{array} = 1_{g_{ij} \cdot g_{jk}}^{\oplus n_{ijk}} \tag{5.4}$$

Here, $n_{ijk} := \dim(\mathrm{Hom}_A(a_{ij} \otimes a_{jk}, a_{ik}))$ denotes the fusion coefficient of $A$. The above equation implies that each basis 2-morphism $\alpha_{ijk} \in \mathrm{Hom}_A(a_{ij} \otimes a_{jk}, a_{ik})$ is associated with a simple 1-morphism $1_{g_{ij} \cdot g_{jk}}[\alpha_{ijk}] := 1_{g_{ij} \cdot g_{jk}} \in \mathrm{Hom}_{2\mathsf{Vec}_G}(D_2^{g_{ij}}[a_{ij}] \square D_2^{g_{jk}}[a_{jk}], D_2^{g_{ik}}[a_{ik}])$.

The associativity 2-isomorphism $\mu$ is given by the $F$-symbol of $A$. Namely, $\mu$ is defined component-wise as

$$\begin{array}{c} a_{jl} \\ \alpha_{ijl} \begin{array}{c} \mu \\ \nearrow \end{array} \alpha_{jkl} \\ \alpha_{ikl} \begin{array}{c} \\ \end{array} \alpha_{ijk} \quad a_{kl} \\ a_{il} \quad a_{ik} \quad \begin{array}{c} a_{jk} \\ a_{ij} \end{array} \end{array} = (F_{a_{il}}^{a_{ij} a_{jk} a_{kl}})_{(a_{ik}; \alpha_{ijk}, \alpha_{ikl}), (a_{jl}; \alpha_{ijl}, \alpha_{jkl})} \quad \begin{array}{c} 1_{g_{ij} \cdot g_{jl}} \quad 1_{g_{jk} \cdot g_{kl}} \\ \\ 1_{g_{ik} \cdot g_{kl}} \quad 1_{g_{ij} \cdot g_{jk}} \end{array} , \tag{5.5}$$

where the white dot on the right-hand side represents the basis 2-morphism. On the left-hand side, an object $D_2^{g_{ij}}[a_{ij}]$ and a 1-morphism $1_{g_{ij} \cdot g_{jk}}[\alpha_{ijk}]$ are simply written as $a_{ij}$ and $\alpha_{ijk}$.

---

[56]More precisely, $a_g$ is a representative of the isomorphism class of a simple object of $A_g$.

Similarly, $\mu^{-1}$ is given by the inverse of the $F$-symbol, i.e.,

$$[\text{diagram}] = \left(\overline{F}^{a_{ij}a_{jk}a_{kl}}_{a_{il}}\right)_{(a_{ik};\alpha_{ijk},\alpha_{ikl}),(a_{jl};\alpha_{ijl},\alpha_{jkl})}\ [\text{diagram}]. \tag{5.6}$$

Here, $F$ and $\overline{F}$ are defined by

$$[\text{diagram}] = \sum_{a_{jl}}\sum_{\alpha_{ijl},\alpha_{jkl}} \left(F^{a_{ij}a_{jk}a_{kl}}_{a_{il}}\right)_{(a_{ik};\alpha_{ijk},\alpha_{ikl}),(a_{jl};\alpha_{ijl},\alpha_{jkl})}\ [\text{diagram}],$$

$$[\text{diagram}] = \sum_{a_{ik}}\sum_{\alpha_{ijk},\alpha_{ikl}} \left(\overline{F}^{a_{ij}a_{jk}a_{kl}}_{a_{il}}\right)_{(a_{ik};\alpha_{ijk},\alpha_{ikl}),(a_{jl};\alpha_{ijl},\alpha_{jkl})}\ [\text{diagram}], \tag{5.7}$$

or equivalently,

$$[\text{diagram}] = \sum_{a_{jl}}\sum_{\alpha_{ijl},\alpha_{jkl}} \left(\overline{F}^{a_{ij}a_{jk}a_{kl}}_{a_{il}}\right)_{(a_{ik};\alpha_{ijk},\alpha_{ikl}),(a_{jl};\alpha_{ijl},\alpha_{jkl})}\ [\text{diagram}],$$

$$[\text{diagram}] = \sum_{a_{ik}}\sum_{\alpha_{ijk},\alpha_{ikl}} \left(F^{a_{ij}a_{jk}a_{kl}}_{a_{il}}\right)_{(a_{ik};\alpha_{ijk},\alpha_{ikl}),(a_{jl};\alpha_{ijl},\alpha_{jkl})}\ [\text{diagram}], \tag{5.8}$$

which satisfy the pentagon equation [138]. We suppose that the basis morphisms $\alpha_{ijk}$ and $\overline{\alpha}_{ijk}$ are normalized so that

$$[\text{diagram}] = \sum_{a_{ik}}\sum_{\alpha_{ijk}} \sqrt{\frac{\dim(a_{ik})}{\dim(a_{ij})\dim(a_{jk})}}\ [\text{diagram}],$$

$$[\text{diagram}] = \delta_{a_{ik},a'_{ik}}\,\delta_{\alpha_{ijk},\alpha'_{ijk}} \sqrt{\frac{\dim(a_{ij})\dim(a_{jk})}{\dim(a_{ik})}}\ [\text{diagram}], \tag{5.9}$$

where $\dim(a_{ij})$ is the quantum dimension of $a_{ij}$. The coherence condition (3.8) on $\mu$ follows from the pentagon equation for the $F$-symbols. For simplicity of notation, we often abbreviate the $F$-symbol and the quantum dimension as follows:

$$F^{a;\alpha}_{ijkl} := \left(F^{a_{ij}a_{jk}a_{kl}}_{a_{il}}\right)_{(a_{ik};\alpha_{ijk},\alpha_{ikl}),(a_{jl};\alpha_{ijl},\alpha_{jkl})}, \qquad d^a_{ij} := \dim(a_{ij}). \tag{5.10}$$

We also write $F$ and $\overline{F}$ as $F^+$ and $F^-$, respectively. We note that $F$ is unitary because $A$ is assumed to be unitary:

$$\overline{F}^{a;\alpha}_{ijkl} = (F^{a;\alpha}_{ijkl})^*. \tag{5.11}$$

Furthermore, since $A$ is unitary, $d^a_{ij}$ agrees with the Frobenius-Perron dimension of $a_{ij}$, which is a positive real number greater than or equal to one [159]:

$$d^a_{ij} \geq 1. \tag{5.12}$$

For later use, we recall that the $F$-symbols have the following tetrahedral symmetry [160]:

$$\frac{1}{\sqrt{d^a_{ik}d^a_{jl}}}F^{a;\alpha}_{ijkl} = \frac{1}{\sqrt{d^a_{\sigma(i)\sigma(k)}d^a_{\sigma(j)\sigma(l)}}}(F^{a;\alpha}_{\sigma(i)\sigma(j)\sigma(k)\sigma(l)})^{\mathrm{sgn}(\sigma)}. \tag{5.13}$$

Here, $\sigma \in S_4$ is an arbitrary permutation of four letters $\{i, j, k, l\}$, and $\mathrm{sgn}(\sigma) = \pm 1$ is the sign of $\sigma$. For instance, when $\sigma$ is the permutation of $k$ and $l$, the above equation reduces to

$$\frac{1}{\sqrt{d^a_{ik}d^a_{jl}}}F^{a;\alpha}_{ijkl} = \frac{1}{\sqrt{d^a_{il}d^a_{jk}}}(F^{a;\alpha}_{ijlk})^- = \frac{1}{\sqrt{d^a_{il}d^a_{jk}}}(\overline{F}^{a_{ij}a_{jl}a_{lk}}_{a_{il}})_{(a_{il};\alpha_{ijl},\alpha_{ilk}),(a_{jk};\alpha_{ijk},\alpha_{jlk})}, \tag{5.14}$$

where $a_{lk} := \overline{a_{kl}}$ is the dual of $a_{kl}$, and $\alpha_{xlk}$ for $x = i, j$ is defined by

$$\tag{5.15}$$

**Rigid Algebra Structure** $(m^*, \eta, \epsilon)$. Since $A$ is a rigid algebra in $2\mathsf{Vec}_G$, the multiplication 1-morphism $m : A\square A \to A$ has its dual. The dual 1-morphism $m^* : A \to A\square A$ is defined component-wise as

$$\tag{5.16}$$

where $n_{ijk} = \dim(\mathrm{Hom}_A(a_{ik}, a_{ij} \otimes a_{jk}))$ is the fusion coefficient of $A$. We note that the simple components of $m^*$ are in one-to-one correspondence with basis 2-morphisms of $\mathrm{Hom}_A(a_{ik}, a_{ij} \otimes a_{jk})$. The 1-morphism corresponding to $\overline{\alpha}_{ijk} \in \mathrm{Hom}_A(a_{ik}, a_{ij} \otimes a_{jk})$ is denoted by $\overline{1}_{g_{ij} \cdot g_{jk}}[\overline{\alpha}_{ijk}]$.

The unit $\eta$ and the counit $\epsilon$ for the dual pair $(m, m^*)$ are given component-wise as

$$\tag{5.17}$$

$$\text{} = \delta_{\alpha_{ijk},\alpha'_{ijk}} \sqrt{\frac{d^a_{ij} d^a_{jk}}{d^a_{ik}}} \; \text{}. \tag{5.18}$$

On the left-hand side, $a_{ij}$ and $\alpha_{ijk}$ represent $D^{g_{ij}}_2[a_{ij}]$ and $1_{g_{ij}\cdot g_{jk}}[\alpha_{ijk}]$, respectively. The unit and counit defined by (5.17) and (5.18) satisfy the cusp equations (3.9). By using the above $\eta$ and $\epsilon$, we can compute $\psi_r$ in (3.10) as follows:

$$\text{} = F^{a;\alpha}_{ijkl} \sqrt{\frac{d^a_{il} d^a_{jk}}{d^a_{ik} d^a_{jl}}} \; \text{} = \overline{F}^{a;\alpha}_{ijlk} \; \text{}. \tag{5.19}$$

Here, we used the tetrahedral symmetry (5.14). Similarly, $\psi_l$ in (3.11) and $\mu^*$ in (3.15) can be computed as

$$\text{} = F^{a;\alpha}_{ijlk} \; \text{}, \tag{5.20}$$

$$\text{} = F^{a;\alpha}_{ijkl} \; \text{}. \tag{5.21}$$

The 2-morphisms $\psi_r$ and $\psi_l$ satisfy the consistency conditions (3.12)–(3.14) due to the pentagon equation for the $F$-symbols.

**Separable Algebra Structure $\gamma$.** Finally, the 2-morphism $\gamma$ that satisfies the separability condition (3.16) is given component-wise as

$$\text{} = \delta_{\alpha_{ijk},\alpha'_{ijk}} \frac{1}{\mathcal{D}_A} \sqrt{\frac{d^a_{ij} d^a_{jk}}{d^a_{ik}}} \; \text{}, \tag{5.22}$$

where $\mathcal{D}_A = \sum_{a \in A} \dim(a)^2$ is the total dimension of $A$. A direct computation shows that $\epsilon$ and $\gamma$ defined by (5.18) and (5.22) satisfy the separability condition (3.16). For example, the first equality of (3.16) follows from the identity

$$\frac{1}{\mathcal{D}_A} \sum_{a_{ij}, a_{jk} \in A} n_{ijk} \frac{d^a_{ij} d^a_{jk}}{d^a_{ik}} = \frac{1}{\mathcal{D}_A} \sum_{a_{ij} \in A} (d^a_{ij})^2 = 1, \qquad \forall a_{ik} \in A. \tag{5.23}$$

Here, we used the equality $\sum_{a_{jk} \in A} n_{ijk} d^a_{jk} = d^a_{ij} d^a_{ik}$. One can also check the other two equalities of (3.16) using the tetrahedral symmetry (5.13).

### 5.1.2 Module 2-Categories over $2\mathsf{Vec}_G$

Let us describe the basic structures of the $2\mathsf{Vec}_G$-module 2-category $_A(2\mathsf{Vec}_G)$, where $A$ is a $G$-graded fusion category. We refer the reader to [115, Section 2.3] for a detailed definition of this module 2-category. We suppose that $A$ is faithfully graded by a subgroup $H \subset G$, meaning that $A_g$ is empty if and only if $g$ is not in $H$.[57]

**Objects.** An object in $_A(2\mathsf{Vec}_G)$ is a left $A$-module 1-category equipped with a $G$-grading compatible with the left $A$-action. More specifically, an object $M \in {}_A(2\mathsf{Vec}_G)$ is a $G$-graded finite semisimple 1-category that is equipped with a grading-preserving left $A$-action $\rhd \colon A \times M \to M$ together with a natural isomorphism[58]

$$l_{a,b,m} : (a \otimes b) \rhd m \to a \rhd (b \rhd m), \quad \forall a, b \in A, \quad \forall m \in M. \tag{5.24}$$

The isomorphism $l_{a,b,m}$ satisfies the pentagon equation, i.e., the following diagram commutes:

$$\begin{array}{ccc}
((a \otimes b) \otimes c) \rhd m \xrightarrow{l_{a \otimes b,c,m}} (a \otimes b) \rhd (c \rhd m) \xrightarrow{l_{a,b,c \rhd m}} a \rhd (b \rhd (c \rhd m)) \\
{\scriptstyle \alpha_{a,b,c} \rhd \mathrm{id}_m} \downarrow \qquad\qquad\qquad\qquad\qquad\qquad\qquad \uparrow {\scriptstyle \mathrm{id}_a \rhd l_{b,c,m}} \\
(a \otimes (b \otimes c)) \rhd m \xrightarrow[\quad l_{a,b \otimes c,m} \quad]{} a \rhd ((b \otimes c) \rhd m)
\end{array} \tag{5.25}$$

Here, $\alpha_{a,b,c} : (a \otimes b) \otimes c \to a \otimes (b \otimes c)$ is the associator of $A$.

The simplest example of a left $A$-module in $2\mathsf{Vec}_G$ is the regular $A$-module, i.e., $A$ itself. More generally, for any $g \in G$, an object of the form $A\square D_2^g$ is a left $A$-module in $2\mathsf{Vec}_G$. The left $A$-action on $A\square D_2^g$ is given by the multiplication of $A$. We note that $A\square D_2^g$ is equivalent to the regular module $A$ as a left $A$-module category. However, $A\square D_2^g$ and $A$ are inequivalent as left $A$-modules in $2\mathsf{Vec}_G$ because they have different $G$-gradings. Specifically, an object $a \in A_{g'}$ viewed as an object of $A\square D_2^g$ is graded by $g'g$ rather than $g'$. Since the regular module over a fusion category is indecomposable, $A\square D_2^g$ is also indecomposable as a left $A$-module category. In particular, $A\square D_2^g$ is an indecomposable left $A$-module in $2\mathsf{Vec}_G$.

Any left $A$-module in $2\mathsf{Vec}_G$ is connected to $A\square D_2^g$ for some $g \in G$. This follows from the fact that for any monoidal 2-category $\mathcal{C}$, there is an equivalence of 1-categories (see, e.g., [115, Lemma 3.2.2])

$$\mathrm{Hom}_{{}_A\mathcal{C}_{A'}}(A\square X\square A', M) \cong \mathrm{Hom}_{\mathcal{C}}(X, M), \tag{5.26}$$

---

[57]In this manuscript, an algebra object specifying the generalized gauging is denoted by $A$ and is faithfully graded by $H \subset G$. On the other hand, an algebra object specifying a gapped phase is denoted by $B$ and is faithfully graded by $K \subset G$.

[58]A left $A$-module $M \in {}_A(2\mathsf{Vec}_G)$ is also equipped with an isomorphism $i_M : \mathbb{1}_A \rhd M \to M$, where $\mathbb{1}_A$ is the unit object of $A$. The isomorphism $i_M$ must satisfy an appropriate coherence condition [115]. We will supress $i_M$ in what follows.

where $A$ and $A'$ are algebras in $\mathcal{C}$, $X$ is an arbitrary object of $\mathcal{C}$, and $M$ is an arbitrary object of the 2-category $_A\mathcal{C}_{A'}$ of $(A, A')$-bimodules in $\mathcal{C}$. When $\mathcal{C} = 2\mathsf{Vec}_G$, $A' = D_2^e$, and $X = D_2^g$, the above equation leads to

$$\mathrm{Hom}_{A(2\mathsf{Vec}_G)}(A\square D_2^g, M) \cong \mathrm{Hom}_{2\mathsf{Vec}_G}(D_2^g, M). \tag{5.27}$$

Since $M$ is non-zero, the right-hand side of the above equation is non-zero for some $g \in G$. This implies that there exists $g \in G$ such that $M$ is connected to $A\square D_2^g$.

The connected components of $_A(2\mathsf{Vec}_G)$ are in one-to-one correspondence with right $H$-cosets in $G$. This also follows from the equivalence (5.27). Specifically, if we choose $M$ to be $A\square D_2^{g'}$, (5.27) reduces to

$$\mathrm{Hom}_{A(2\mathsf{Vec}_G)}(A\square D_2^g, A\square D_2^{g'}) \cong \mathrm{Hom}_{2\mathsf{Vec}_G}(D_2^g, A\square D_2^{g'}) \cong A_{g(g')^{-1}}. \tag{5.28}$$

Since $A_{g(g')^{-1}} \neq 0$ if and only if $g(g')^{-1} \in H$, the objects $A\square D_2^g$ and $A\square D_2^{g'}$ are connected to each other if and only if $g$ and $g'$ are in the same right $H$-coset, i.e., $Hg = Hg'$. Thus, the connected components of $_A(2\mathsf{Vec}_G)$ are labeled by right $H$-cosets.

For later convenience, we fix a representative of each connected component. To this end, we first choose a representative of each right $H$-coset. The set of representatives of the cosets is denoted by $S_{H\backslash G}$. For each element $g \in S_{H\backslash G}$, we choose $A\square D_2^g$ as a representative of the connected component labeled by $Hg$. We denote this representative as $M^g := A\square D_2^g$.

The right $2\mathsf{Vec}_G$-module action on $_A(2\mathsf{Vec}_G)$ is given by the tensor product of $2\mathsf{Vec}_G$. In particular, the action of $D_2^{g'} \in 2\mathsf{Vec}_G$ on $M^g \in {}_A(2\mathsf{Vec}_G)$ is given by

$$M^g \lhd D_2^{g'} = A\square D_2^g\square D_2^{g'} = A\square D_2^{gg'}. \tag{5.29}$$

We note that $M_g \lhd D_2^{g'}$ is not necessarily a representative.

**1- and 2-Morphisms.** A 1-morphism in $_A(2\mathsf{Vec}_G)$ is a grading-preserving left $A$-module functor. Namely, a 1-morphism $F : M \to N$ between left $A$-modules $M, N \in {}_A(2\mathsf{Vec}_G)$ is a grading-preserving functor equipped with a natural isomorphism

$$\xi_{a,m} : F(a \rhd m) \to a \rhd F(m), \quad \forall a \in A,\ \forall m \in M, \tag{5.30}$$

which makes the following diagram commute:

$$
\begin{array}{ccc}
F((a \otimes b) \rhd m) \xrightarrow{\xi_{a\otimes b,m}} (a \otimes b) \rhd F(m) \xrightarrow{l^N_{a,b,F(m)}} a \rhd (b \rhd F(m)) \\
{\scriptstyle F(l^M_{a,b,m})}\Big\downarrow \qquad\qquad\qquad\qquad\qquad\qquad\qquad \Big\uparrow{\scriptstyle \mathrm{id}_a\rhd\xi_{b,m}} \\
F(a \rhd (b \rhd m)) \xrightarrow{\qquad\qquad \xi_{a,b\rhd m} \qquad\qquad} a \rhd F(b \rhd m)
\end{array}
\tag{5.31}
$$

Here, $l^M_{a,b,m}$ and $l^N_{a,b,F(m)}$ denote the natural isomorphisms associated with $M$ and $N$, respectively. Similarly, a 2-morphism in $_A(2\mathsf{Vec}_G)$ is a natural transformation of $A$-module functors. That is, a 2-morphism $\eta : F \Rightarrow G$ between two 1-morphisms $F, G : M \to N$ is a natural transformation that makes the following diagram commute:

$$
\begin{array}{ccc}
F(a \triangleright m) & \xrightarrow{\xi^F_{a,m}} & a \triangleright F(m) \\
\eta_{a \triangleright m} \downarrow & & \downarrow \mathrm{id}_a \triangleright \eta_m \\
G(a \triangleright m) & \xrightarrow[\xi^G_{a,m}]{} & a \triangleright G(m)
\end{array}
\tag{5.32}
$$

Here, $\xi^F_{a,m}$ and $\xi^G_{a,m}$ denote the natural isomorphisms associated with $F$ and $G$, respectively. By definition, the 1-category $\mathrm{Hom}_{A(2\mathsf{Vec}_G)}(M, N)$ consisting of 1- and 2-morphisms of $_A(2\mathsf{Vec}_G)$ is a subcategory of the 1-category $\mathrm{Hom}_{A(2\mathsf{Vec})}(M, N)$ consisting of left $A$-module functors and natural transformations between them. In what follows, we will describe the structure of the 1-category $\mathrm{Hom}_{A(2\mathsf{Vec}_G)}(M, N)$ in more detail for some $M$ and $N$ of our interest.

**Reverse and Opposite Category.** We first consider the case where $M = N = A \square D^g_2$. In this case, the 1-category

$$
\mathrm{Hom}_{A(2\mathsf{Vec}_G)}(M, N) = \mathrm{End}_{A(2\mathsf{Vec}_G)}(A \square D^g_2)
\tag{5.33}
$$

has the structure of a monoidal category, whose tensor product is given by the composition of 1-morphisms. As we show in appendix B.1, there is an equivalence of monoidal categories

$$
\mathrm{End}_{A(2\mathsf{Vec}_G)}(A \square D^g_2) \cong A^{\mathrm{op}}_e \cong A^{\mathrm{rev}}_e.
\tag{5.34}
$$

Here, $A^{\mathrm{op}}$ is the $G^{\mathrm{op}}$-graded fusion category whose objects are given by those of $A$ and the tensor product $\otimes^{\mathrm{op}} : A^{\mathrm{op}} \times A^{\mathrm{op}} \to A^{\mathrm{op}}$ is defined by

$$
a \otimes^{\mathrm{op}} b := b \otimes a, \quad \forall a, b \in A.
\tag{5.35}
$$

On the other hand, $A^{\mathrm{rev}}$ is the $G$-graded fusion category obtained by reversing all morphisms of $A$. In particular, objects of $A^{\mathrm{rev}}$ are those of $A$, and the tensor product of $A^{\mathrm{rev}}$ is given by the tensor product of $A$, while the associator of $A^{\mathrm{rev}}$ is given the inverse of the associator of $A$. The categories $A^{\mathrm{op}}$ and $A^{\mathrm{rev}}$ are called the opposite category and the reverse category, respectively.[59] We note that $A^{\mathrm{op}}$ and $A^{\mathrm{rev}}$ are monoidally equivalent to each other [159]. The monoidal equivalence between them is given by the functor that takes the dual

$$
A^{\mathrm{op}} \xrightarrow{\cong} A^{\mathrm{rev}} : \quad a \mapsto \bar{a}.
\tag{5.36}
$$

---

[59]In some literature, e.g., in nLab, the opposite category in this manuscript is called the reverse category https://ncatlab.org/nlab/show/reverse+monoidal+category, while the reverse category in this manuscript is called the opposite category https://ncatlab.org/nlab/show/opposite+category.

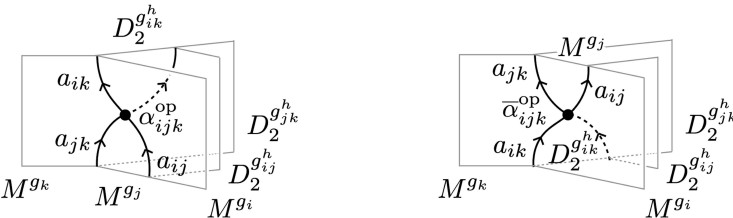

Figure 12: A 1-morphism from $M^{g_i} \lhd D_2^{g_{ij}^h}$ to $M^{g_j}$ is labeled by an object $a_{ij} \in A_{h^{ij}}^{\mathrm{op}}$. A 2-morphism sitting at the junction of four 1-morphisms $a_{ij}$, $a_{jk}$, $a_{ik}$, and $1_{g_{ij}^h \cdot g_{jk}^h}$ is labeled by a morphism $\alpha_{ijk}^{\mathrm{op}} \in \mathrm{Hom}_{A^{\mathrm{op}}}(a_{jk} \otimes^{\mathrm{op}} a_{ij}, a_{ik}) = \mathrm{Hom}_A(a_{ij} \otimes a_{jk}, a_{ik})$ or $\overline{\alpha}_{ijk}^{\mathrm{op}} \in \mathrm{Hom}_{A^{\mathrm{op}}}(a_{ik}, a_{jk} \otimes^{\mathrm{op}} a_{ij}) = \mathrm{Hom}_A(a_{ik}, a_{ij} \otimes a_{jk})$. The dotted lines in the diagrams are labeled by the identity 1-morphism $1_{g_{ij}^h \cdot g_{jk}^h}$.

When $M \neq N$, the category $\mathrm{Hom}_{A(2\mathsf{Vec}_G)}(M, N)$ does not have the structure of a monoidal category. Nevertheless, it has the structure of an $(\mathrm{End}_{A(2\mathsf{Vec}_G)}(N), \mathrm{End}_{A(2\mathsf{Vec}_G)}(M))$-bimodule category. In particular, when $M = A \square D_2^g$ and $N = A \square D_2^{g'}$, the category $\mathrm{Hom}_{A(2\mathsf{Vec}_G)}(M, N)$ has the structure of an $(A_e^{\mathrm{op}}, A_e^{\mathrm{op}})$-bimodule category due to the monoidal equivalence (5.34). As we show in appendix B.1, there is an equivalence of $(A_e^{\mathrm{op}}, A_e^{\mathrm{op}})$-bimodule category

$$\mathrm{Hom}_{A(2\mathsf{Vec}_G)}(A \square D_2^g, A \square D_2^{g'}) \cong A_{g(g')^{-1}}^{\mathrm{op}}. \tag{5.37}$$

Here, $A_{g(g')^{-1}}^{\mathrm{op}}$ is the $g(g')^{-1}$-graded component of $A^{\mathrm{op}}$. The $(A_e^{\mathrm{op}}, A_e^{\mathrm{op}})$-bimodule structure on $A_{g(g')^{-1}}^{\mathrm{op}}$ is given by the tensor product of $A^{\mathrm{op}}$. We emphasize that (5.37) is an equivalence of $(A_e^{\mathrm{op}}, A_e^{\mathrm{op}})$-bimodule categories, while (5.28) is an equivalence of (semisimple) categories.

If we choose $g$ and $g'$ appropriately, the equivalence (5.37) reduces to

$$\mathrm{Hom}_{A(2\mathsf{Vec}_G)}(M^{g_i} \lhd D_2^{g_{ij}^h}, M^{g_j}) \cong A_{h_{ij}}^{\mathrm{op}}, \tag{5.38}$$

$$\mathrm{Hom}_{A(2\mathsf{Vec}_G)}(M^{g_j}, M^{g_i} \lhd D_2^{g_{ij}^h}) \cong A_{h_{ij}^{-1}}^{\mathrm{op}}. \tag{5.39}$$

Here, $g_{ij}^h$ is defined by $g_i^{-1} h_{ij} g_j$ for $g_i, g_j \in S_{H \backslash G}$ and $h_{ij} \in H$. The equivalences (5.38) and (5.39) imply that 1-morphisms and 2-morphisms in $_A(2\mathsf{Vec}_G)$ can be labeled by objects and morphisms of $A^{\mathrm{op}}$, see figure 12.

Since $A^{\mathrm{op}}$ is monoidally equivalent to $A^{\mathrm{rev}}$, (5.38) and (5.39) can also be written as the equivalences of $(A_e^{\mathrm{rev}}, A_e^{\mathrm{rev}})$-bimodule categories. Recalling that the monoidal equivalence (5.36) changes the grading from $h$ to $h^{-1}$, we find

$$\mathrm{Hom}_{A(2\mathsf{Vec}_G)}(M^{g_i} \lhd D_2^{g_{ij}^h}, M^{g_j}) \cong A_{h_{ij}^{-1}}^{\mathrm{rev}}, \tag{5.40}$$

$$\mathrm{Hom}_{A(2\mathsf{Vec}_G)}(M^{g_j}, M^{g_i} \lhd D_2^{g_{ij}^h}) \cong A_{h_{ij}}^{\mathrm{rev}}. \tag{5.41}$$

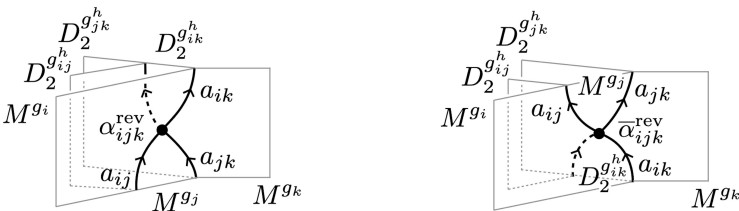

Figure 13: A 1-morphism from $M^{g_j}$ to $M^{g_i} \lhd D_2^{g_{ij}^h}$ is labeled by an object $a_{ij} \in A_{h^{ij}}^{\mathrm{rev}}$. A 2-morphism sitting at the junction of four 1-morphisms $a_{ij}$, $a_{jk}$, $a_{ik}$, and $\overline{1}_{g_{ij}^h \cdot g_{jk}^h}$ is labeled by a morphism $\alpha_{ijk}^{\mathrm{rev}} \in \mathrm{Hom}_{A^{\mathrm{rev}}}(a_{ij} \otimes a_{jk}, a_{ik}) = \mathrm{Hom}_A(a_{ik}, a_{ij} \otimes a_{jk})$ or $\overline{\alpha}_{ijk}^{\mathrm{rev}} \in \mathrm{Hom}_{A^{\mathrm{rev}}}(a_{ik}, a_{ij} \otimes a_{jk}) = \mathrm{Hom}_A(a_{ij} \otimes a_{jk}, a_{ik})$. The dotted lines in the diagrams are labeled by the identity 1-morphism $\overline{1}_{g_{ij}^h \cdot g_{jk}^h}$.

Here, $A_{h_{ij}}^{\mathrm{rev}}$ is the $h_{ij}$-graded component of the reverse category $A^{\mathrm{rev}}$, and the $(A_e^{\mathrm{rev}}, A_e^{\mathrm{rev}})$-bimodule structure on it is given by the tensor product of $A^{\mathrm{rev}}$. The equivalences (5.40) and (5.41) imply that 1-morphisms and 2-morphisms in $_A(2\mathsf{Vec}_G)$ can be labeled by objects and morphisms of $A^{\mathrm{rev}}$, see figure 13.

**Module 10-j Symbols.** The module 10-j symbols of the $2\mathsf{Vec}_G$-module 2-category $_A(2\mathsf{Vec}_G)$ are given by the $F$-symbols of $A^{\mathrm{op}}$, i.e.,

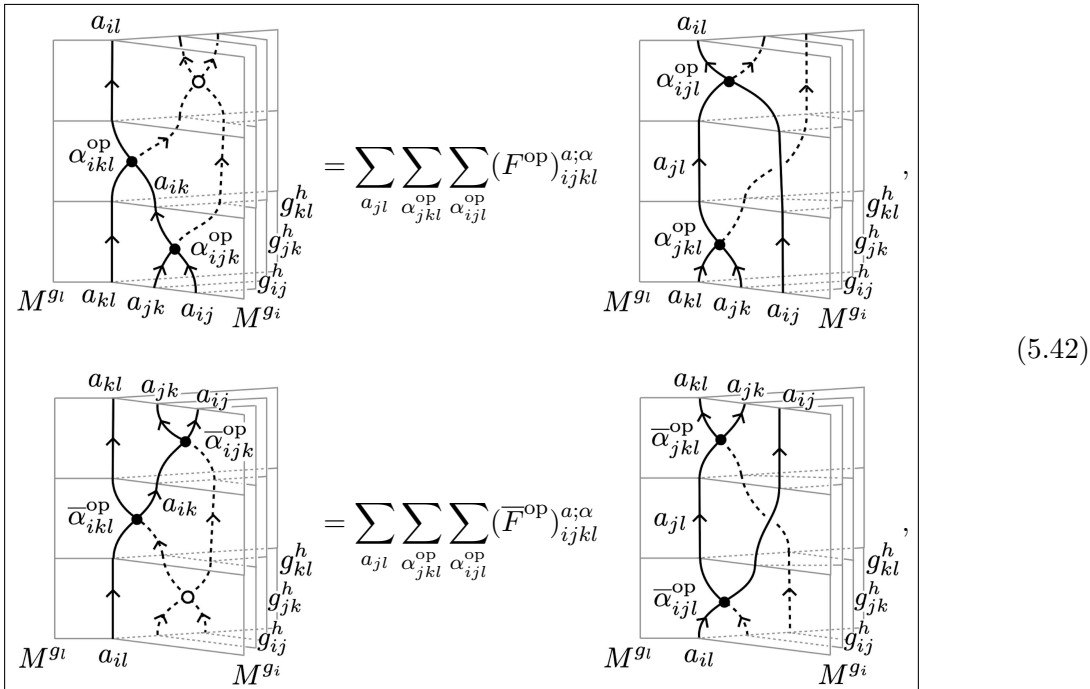

$$(5.42)$$

where $a_{ij} \in A^{\mathrm{op}}$, $\alpha_{ijk}^{\mathrm{op}} \in \mathrm{Hom}_{A^{\mathrm{op}}}(a_{jk} \otimes^{\mathrm{op}} a_{ij}, a_{ik})$, etc. The $F$-symbol $F^{\mathrm{op}}$ of the opposite category $A^{\mathrm{op}}$ is related to the $F$-symbol of $A$ as follows:

$$(F^{\mathrm{op}})_{ijkl}^{a;\alpha} := ((F^{\mathrm{op}})_{a_{il}}^{a_{kl}a_{jk}a_{ij}})_{(a_{jl};\alpha_{jkl}^{\mathrm{op}},\alpha_{ijl}^{\mathrm{op}}),(a_{il};\alpha_{jkl}^{\mathrm{op}},\alpha_{ijk}^{\mathrm{op}})} = \overline{F}_{ijkl}^{a;\alpha}. \tag{5.43}$$

Here, $\alpha_{ijk}^{\mathrm{op}}$ is equal to $\alpha_{ijk}$ as an element of $\mathrm{Hom}_{A^{\mathrm{op}}}(a_{jk} \otimes^{\mathrm{op}} a_{ij}, a_{ik}) = \mathrm{Hom}_A(a_{ij} \otimes a_{jk}, a_{ik})$.

Similarly, we also have

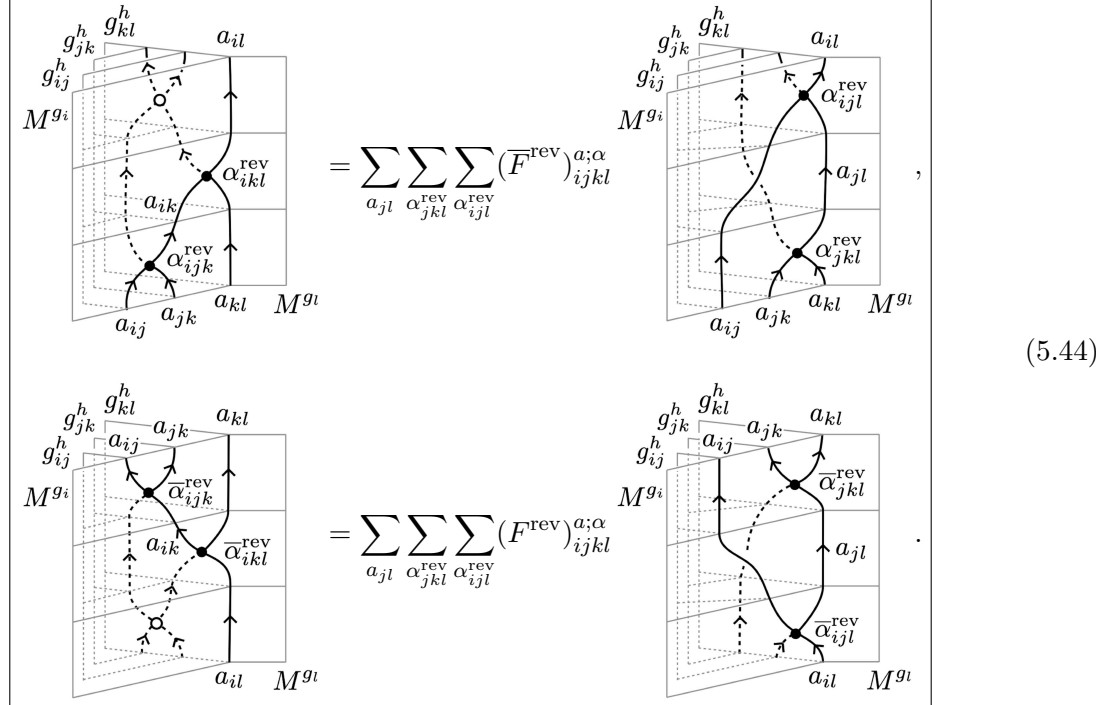

$$\tag{5.44}$$

where $a_{ij} \in A^{\mathrm{rev}}$, $\alpha_{ijk}^{\mathrm{rev}} \in \mathrm{Hom}_{A^{\mathrm{rev}}}(a_{ij} \otimes a_{jk}, a_{ik})$, etc. The $F$-symbols $F^{\mathrm{rev}}$ of the reverse category $A^{\mathrm{rev}}$ is related to the $F$-symbol of $A$ as follows:

$$(F^{\mathrm{rev}})_{ijkl}^{a;\alpha} := ((F^{\mathrm{rev}})_{a_{il}}^{a_{ij}a_{jk}a_{kl}})_{(a_{ik};\alpha_{ijk}^{\mathrm{rev}},\alpha_{ikl}^{\mathrm{rev}}),(a_{jl};\alpha_{ijl}^{\mathrm{rev}},\alpha_{jkl}^{\mathrm{rev}})} = \overline{F}_{ijkl}^{a;\alpha}. \tag{5.45}$$

Here, $\alpha_{ijk}^{\mathrm{rev}}$ is equal to $\overline{\alpha}_{ijk}$ as an element of $\mathrm{Hom}_{A^{\mathrm{rev}}}(a_{ij} \otimes a_{jk}, a_{ik}) = \overline{\mathrm{Hom}}_A(a_{ik}, a_{ij} \otimes a_{jk})$.

**Underlying Objects and Morphisms.** Since $_A(2\mathsf{Vec}_G)$ is a sub-2-category of $2\mathsf{Vec}_G$, objects and morphisms of $_A(2\mathsf{Vec}_G)$ have the underlying objects and morphisms in $2\mathsf{Vec}_G$. The underlying object of $M^g \in {}_A(2\mathsf{Vec}_G)$ is given by

$$M^g = A \square D_2^g = \bigoplus_{h \in H} \bigoplus_{a_h \in A_h} D_2^h[a_h] \square D_2^g = \bigoplus_{h \in H} \bigoplus_{a_h \in A_h} D_2^{hg}[a_h]. \tag{5.46}$$

Here, we defined $D_2^{hg}[a_h] := D_2^h[a_h] \square D_2^g$, where $D_2^h[a_h]$ is the direct sum component of $A \in 2\mathsf{Vec}_G$ labeled by a simple object $a_h \in A_h$ (cf. section 5.1.1).

The underlying 1-morphism of $a_{ij} \in \mathrm{Hom}_{A(2\mathsf{Vec}_G)}(M^{g_i} \lhd D_2^{g_{ij}^h}, M^{g_j}) \cong A_{h_{ij}}^{\mathrm{op}}$ is given by

$$D_2^{h_j g_j}[a_j] \boxed{\begin{array}{c} a_{ij}\uparrow \end{array}} D_2^{g_{ij}^h} \atop D_2^{h_i g_i}[a_i] = 1_{h_i g_i \cdot g_{ij}^h}^{\oplus n_{ij}}, \tag{5.47}$$

where $n_{ij} := \dim(\mathrm{Hom}_{A^{\mathrm{op}}}(a_j \otimes^{\mathrm{op}} \overline{a_i}, a_{ij}))$, $a_i \in A_{h_i}$, and $a_j \in A_{h_j}$. We note that $n_{ij} = 0$ unless $h_{ij} = h_i^{-1} h_j$. The above equation implies that each component of the underlying 1-morphism is associated with a basis morphism $\alpha_{ij}^{\mathrm{op}} \in \mathrm{Hom}_{A^{\mathrm{op}}}(a_j \otimes^{\mathrm{op}} \overline{a_i}, a_{ij})$. Similarly, the underlying 1-morphism of $a_{ij} \in \mathrm{Hom}_{A(2\mathsf{Vec}_G)}(M^{g_j}, M^{g_i} \lhd D_2^{g_{ij}^h}) \cong A_{h_{ij}}^{\mathrm{rev}}$ is given by

$$D_2^{g_{ij}^h} \atop D_2^{h_i g_i}[a_i] \boxed{\begin{array}{c} \uparrow a_{ij} \end{array}} D_2^{h_j g_j}[a_j] = \overline{1}_{h_i g_i \cdot g_{ij}^h}^{\oplus m_{ij}}, \tag{5.48}$$

where $m_{ij} := \dim(\mathrm{Hom}_{A^{\mathrm{rev}}}(\overline{a_i} \otimes a_j, a_{ij}))$. The above equation implies that each component of the underlying 1-morphism is associated with a basis morphism $\alpha_{ij}^{\mathrm{rev}} \in \mathrm{Hom}_{A^{\mathrm{rev}}}(\overline{a_i} \otimes a_j, a_{ij})$.

The underlying 2-morphisms of $\alpha_{ijk}^{\mathrm{op}}$ and $\overline{\alpha}_{ijk}^{\mathrm{op}}$ in figure 12 are given component-wise as

$$\tag{5.49}$$

where $F_{ijk}^{a;\alpha}$ is defined by

$$F_{ijk}^{a;\alpha} := (F_{a_k}^{a_i a_{ij} a_{jk}})_{(a_j; \alpha_{ij}, \alpha_{jk}),(a_{ik}; \alpha_{ik}, \alpha_{ijk})}. \tag{5.50}$$

The 1-morphism labeled by $\alpha_{ij}^{\mathrm{op}}$ in (5.49) represents a component of the 1-morphism $a_{ij}$ in

(5.47). Similarly, the underlying 2-morphisms of $\alpha_{ijk}^{\text{rev}}$ and $\overline{\alpha}_{ijk}^{\text{rev}}$ in figure 13 are given by

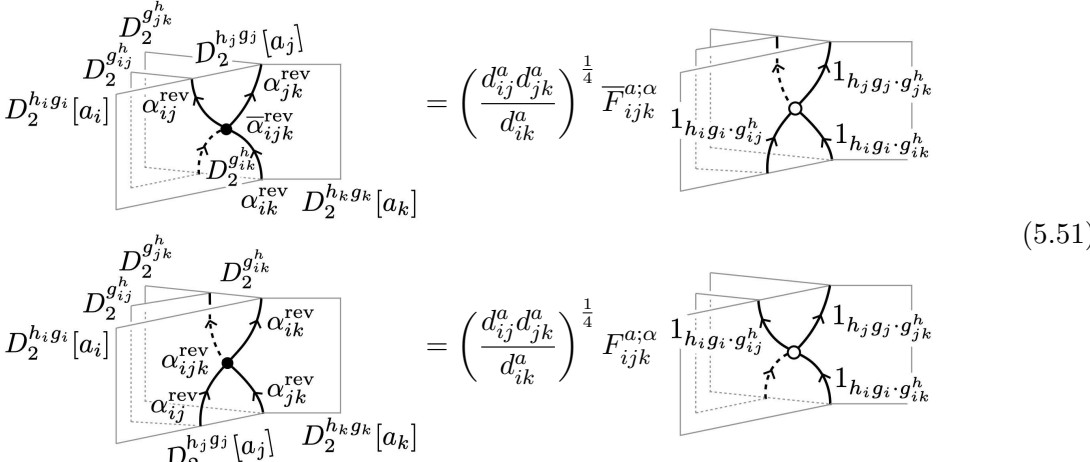

$$(5.51)$$

In the above equations, the factor of $(d_{ij}^a d_{jk}^a / d_{ik}^a)^{\frac{1}{4}}$ is necessary so that (5.9) holds. One can check that the above 2-morphisms are indeed 2-morphisms in $_A(2\mathsf{Vec}_G)$, i.e., they are compatible with the $A$-module structures, see appendix B.2. Equations (5.49) and (5.51), together with the pentagon equation, guarantee that the module 10-j symbols of $_A(2\mathsf{Vec}_G)$ are given by the $F$-symbols of $A$ as in (5.42) and (5.44).

## 5.2 Generalized Gauging on the Lattice

In this subsection, we discuss the generalized gauging of $2\mathsf{Vec}_G$ symmetry in the fusion surface model. As discussed in section 4.2, the generalized gauging of symmetry $\mathcal{C}$ is specified by a right $\mathcal{C}$-module 2-category $\mathcal{M} \cong {}_A\mathcal{C}$, where $A$ is a separable algebra in $\mathcal{C}$. When $\mathcal{C} = 2\mathsf{Vec}_G$, a separable algebra $A$ is a $G$-graded multifusion category, see section 5.1.1. In what follows, we will consider the generalized gauging of $2\mathsf{Vec}_G$ symmetry with $A$ being a $G$-graded unitary fusion category.

### 5.2.1 $2\mathsf{Vec}_G$-Symmetric Model

We begin with the fusion surface model with $2\mathsf{Vec}_G$ symmetry. The input fusion 2-category $\mathcal{C}$ and the module 2-category $\mathcal{M}$ are

$$\mathcal{C} = \mathcal{M} = 2\mathsf{Vec}_G. \tag{5.52}$$

**State Space.** To define the state space, we fix the input data $(\rho, \sigma, \lambda; f, g)$, cf. section 4.1. For simplicity, we suppose

$$\rho = \sigma = \lambda, \quad f = g. \tag{5.53}$$

Furthermore, for concreteness, we choose $\rho$ to be the direct sum of all simple objects with multiplicity one:

$$\rho = \bigoplus_{g \in G} D_2^g. \tag{5.54}$$

Similarly, we choose $f$ to be the direct sum of the identity 1-morphisms for all possible fusion channels, i.e., $f$ is given component-wise as

$$D_2^{g_3} \boxed{\quad f \quad} D_2^{g_2} = \delta_{g_1 g_2, g_3} 1_{g_1 \cdot g_2}. \tag{5.55}$$

For the above choice of the input data, the dynamical variables of the model can be described as follows:

- The dynamical variables on the plaquettes are labeled by elements of $G$. The configurations of these dynamical variables can be arbitrary because $\rho$ contains all simple objects.

- There are no dynamical variables on the edges and vertices.

The Levin-Wen plaquette operator (4.4) is the identity because $\mathrm{End}_{2\mathsf{Vec}_G}(D_2^g) \cong \mathsf{Vec}$ for all $g \in G$. Therefore, the state space of the model is given by the space of all possible configurations of the dynamical variables, which we denote by

$$\mathcal{H}_{\text{original}} = \bigotimes_{i \in P} \mathbb{C}^{|G|}. \tag{5.56}$$

Each state in $\mathcal{H}_{\text{original}}$ will be written as

$$|\{g_i\}\rangle = \left| \begin{array}{c} g_l \\ g_j \\ g_i \end{array} \ g_k \right\rangle. \tag{5.57}$$

The generalization to other choices of $\rho$ and $f$ is straightforward.[60]

**Hamiltonian.** The Hamiltonian of the original model is given by

$$H_{\text{original}} = -\sum_{i \in P} \widehat{h}_i, \tag{5.58}$$

---

[60]Some cases with a more general choice of $\rho$ and $f$ will be studied in section 7.

where the local term $\widehat{\mathsf{h}}_i$ is defined by the following fusion diagram:

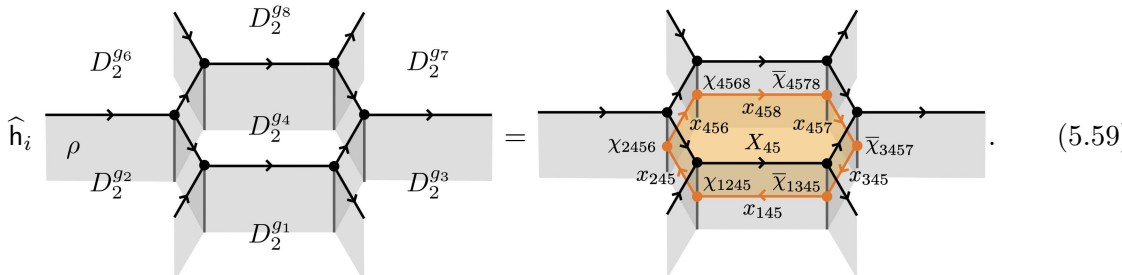

$$\tag{5.59}$$

Here, $X_{45}$ is an object of $2\mathsf{Vec}_G$, $\{x_{145}, x_{245}, x_{345}, x_{456}, x_{457}, x_{458}\}$ are 1-morphisms of $2\mathsf{Vec}_G$, and $\{\chi_{1245}, \chi_{2456}, \chi_{4568}, \overline{\chi}_{1345}, \overline{\chi}_{3457}, \overline{\chi}_{4578}\}$ are 2-morphisms of $2\mathsf{Vec}_G$. For simplicity, we choose $X_{45}$ to be the direct sum of all simple objects with multiplicity one, i.e.,

$$X_{45} = \bigoplus_{g \in G} D_2^g. \tag{5.60}$$

Furthermore, we choose the 1-morphisms $x_{i45} : \rho \square X_{45} \to \rho$ and $x_{45j} : X_{45} \square \rho \to \rho$ to be the direct sum of the identity 1-morphisms, which is given component-wise as

$$\begin{array}{c} x_{i45} \\ \hline D_2^{g_{i5}} \subset \rho \quad D_2^{g_{i4}} \subset \rho \end{array} D_2^{g_{45}} \subset X_{45} = \delta_{g_{i4} g_{45}, g_{i5}} 1_{g_{i4} \cdot g_{45}}, \tag{5.61}$$

$$\begin{array}{c} x_{45j} \\ \hline D_2^{g_{4j}} \subset \rho \quad D_2^{g_{45}} \subset X_{45} \end{array} D_2^{g_{5j}} \subset \rho = \delta_{g_{45} g_{5j}, g_{4j}} 1_{g_{45} \cdot g_{5j}}. \tag{5.62}$$

Finally, the 2-morphisms $\chi_{ijkl}$ and $\overline{\chi}_{ijkl}$ are defined component-wise as

$$\chi_{ijkl} = \chi_{ijkl}(g_{ij}, g_{jk}, g_{kl}) \tag{5.63}$$

$$\overline{\chi}_{ijkl} = \overline{\chi}_{ijkl}(g_{ij}, g_{jk}, g_{kl}) \tag{5.64}$$

where $\chi_{ijkl}(g_{ij}, g_{jk}, g_{kl})$ and $\overline{\chi}_{ijkl}(g_{ij}, g_{jk}, g_{kl})$ are arbitrary complex numbers.[61] These complex numbers will be abbreviated as

$$\chi_{ijkl}^g := \chi_{ijkl}(g_{ij}, g_{jk}, g_{kl}), \quad \overline{\chi}_{ijkl}^g = \overline{\chi}_{ijkl}(g_{ij}, g_{jk}, g_{kl}). \tag{5.65}$$

---

[61] We note that $\chi_{ijkl}$ and $\overline{\chi}_{ijkl}$ are independent functions.

For the above choice of $(X, x, \chi)$, the action of $\widehat{\mathsf{h}}_i$ can be computed as

$$\widehat{\mathsf{h}}_i \left| \begin{array}{c} g_8 \\ g_6 \qquad g_7 \\ g_4 \\ g_2 \qquad g_3 \\ g_1 \end{array} \right\rangle = \sum_{g_{45} \in G} \chi^g_{1245} \chi^g_{2456} \chi^g_{4568} \overline{\chi}^g_{1345} \overline{\chi}^g_{3457} \overline{\chi}^g_{4578} \left| \begin{array}{c} g_8 \\ g_6 \qquad g_7 \\ g_5 \\ g_2 \qquad g_3 \\ g_1 \end{array} \right\rangle, \qquad (5.66)$$

where $g_{ij}$ in $\chi_{ijkl}$ and $\overline{\chi}_{ijkl}$ is defined by $g_{ij} := g_i^{-1} g_j$. The generalization to a more general choice of $(X, x, \chi)$ is straightforward.

**Symmetry.** The above model has $2\mathsf{Vec}_G$ symmetry by construction. The symmetry operator for $D_2^g \in 2\mathsf{Vec}_G$ is given by the left multiplication of $g \in G$:

$$U_g \left|\{g_i\}\right\rangle = \left|\{gg_i\}\right\rangle. \qquad (5.67)$$

It immediately follows from (5.66) that $U_g$ commutes with the Hamiltonian $H_{\text{original}}$.

### 5.2.2 Gauged Model

We now describe the gauged fusion surface model. The input fusion 2-category $\mathcal{C}$ and the module 2-category $\mathcal{M}$ are

$$\mathcal{C} = 2\mathsf{Vec}_G, \qquad \mathcal{M} = {}_A(2\mathsf{Vec}_G), \qquad (5.68)$$

where $A$ is a $G$-graded unitary fusion category faithfully graded by $H \subset G$. The gauging procedure is called minimal gauging when $A = \mathsf{Vec}_H^\nu$, and non-minimal gauging otherwise.

**State Space.** We choose the input data $(\rho, \sigma, \lambda; f, g)$ to be the same as those in the original model, see (5.53)–(5.55). In this case, possible configurations of dynamical variables of the gauged model can be described as follows:

- The dynamical variables on the plaquettes are labeled by the representatives of connected components of ${}_A(2\mathsf{Vec}_G)$. As discussed in section 5.1.2, the representative of each connected component is given by $M^{g_i} = A \square D_2^{g_i}$, where $g_i \in S_{H\backslash G}$ is the representative of a right $H$-coset in $G$. A configuration of these dynamical variables is denoted by $\{g_i \mid i \in P\}$.[62] There is no constraint on this configuration because $\rho$ contains all simple objects.

---

[62]We recall that $P$ denotes the set of all plaquettes of the honeycomb lattice. Similarly, the set of all edges is denoted by $E$, while the set of all vertices is denoted by $V$.

- For a given configuration $\{g_i \mid i \in P\}$, the vertical surface below edge $[ij]$ must be labeled by a group element of the form $g_{ij}^h = g_i^{-1} h_{ij} g_j$, where $h_{ij}$ is an arbitrary element of $H$. When the vertical surface is labeled by $g_{ij}^h$, the dynamical variable on edge $[ij]$ takes values in the set of simple objects of

$$\mathrm{Hom}_{2\mathsf{Vec}_G}(M^{g_i} \lhd D_2^{g_{ij}^h}, M^{g_j}) \cong A_{h_{ij}}^{\mathrm{op}}. \tag{5.69}$$

Since $h_{ij} \in H$ is arbitrary, this dynamical variable can actually take values in the set of all simple objects of $A^{\mathrm{op}} = \bigoplus_{h \in H} A_h^{\mathrm{op}}$ for any configuration $\{g_i \mid i \in P\}$. A configuration of the dynamical variables on the plaquettes and edges is denoted by $\{g_i, a_{ij} \mid i \in P, \ [ij] \in E\}$, where $a_{ij} \in A^{\mathrm{op}}$.

- For a given configuration $\{g_i, a_{ij} \mid i \in P, \ [ij] \in E\}$, the dynamical variable on vertex $[ijk]$ takes values in the vector space $\mathrm{Hom}_{A^{\mathrm{op}}}(a_{jk} \otimes^{\mathrm{op}} a_{ij}, a_{ik})$ or $\mathrm{Hom}_{A^{\mathrm{op}}}(a_{ik}, a_{jk} \otimes^{\mathrm{op}} a_{ij})$ depending on whether $[ijk]$ is in the $A$-sublattice or $B$-sublattice. A configuration of the dynamical variables on the plaquettes, edges, and vertices is denoted by

$$\{g_i, a_{ij}, \alpha_{ijk} \mid i \in P, \ [ij] \in E, \ [ijk] \in V\}. \tag{5.70}$$

Pictorially, the state corresponding to the above configuration can be written as

$$|\{g_i, a_{ij}, \alpha_{ijk}\}\rangle := \left| \begin{array}{c} \includegraphics \end{array} \right\rangle, \tag{5.71}$$

where the vertices are labeled by $\alpha_{ijk}^{\mathrm{op}}$, $\overline{\alpha}_{jkl}^{\mathrm{op}}$, etc. The labels on the vertices will often be omitted to avoid cluttering the diagram.

The state space of the gauged model is given by

$$\mathcal{H}_{\mathrm{gauged}} = \widehat{\pi}_{\mathrm{LW}} \mathcal{H}'_{\mathrm{gauged}}, \tag{5.72}$$

where $\mathcal{H}'_{\mathrm{gauged}}$ is the space of all possible configurations (5.70) and $\widehat{\pi}_{\mathrm{LW}} = \prod_{i \in P} \widehat{B}_i$ is the product of the Levin-Wen plaquette operators defined by

$$\widehat{B}_i \left| \begin{array}{c} \includegraphics \end{array} \right\rangle = \sum_{a \in A_e^{\mathrm{op}}} \frac{\dim(a)}{\mathcal{D}_{A_e}} \left| \begin{array}{c} \includegraphics \end{array} \right\rangle. \tag{5.73}$$

The inside and outside of the loop on the right-hand side are both labeled by $M^{g_i}$.

**Hamiltonian.** The Hamiltonian is given by the sum of local operators

$$H_{\text{gauged}} = -\sum_{i \in P} \widehat{\mathsf{h}}_i, \tag{5.74}$$

where $\widehat{\mathsf{h}}_i$ is represented by the following fusion diagram:

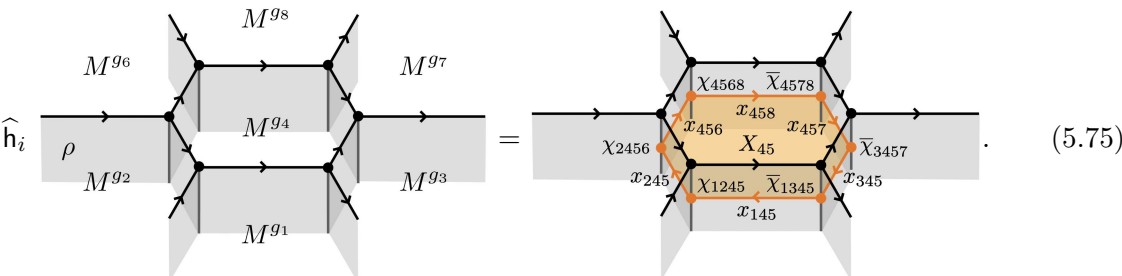

$$\tag{5.75}$$

The input data $(X, x, \chi)$ are chosen to be the same as those in the original model, see (5.60)–(5.64).

The action of the local Hamiltonian $\widehat{\mathsf{h}}_i$ can be computed as

$$\widehat{\mathsf{h}}_i \left| \begin{array}{c} \cdots \end{array} \right\rangle = \sum_{g_5 \in S_{H \backslash G}} \sum_{h_{45} \in H} \chi^{g;h}_{1245} \chi^{g;h}_{2456} \chi^{g;h}_{4568} \overline{\chi}^{g;h}_{1345} \overline{\chi}^{g;h}_{3457} \overline{\chi}^{g;h}_{4578} \quad \cdots$$

$$= \sum_{g_5 \in S_{H \backslash G}} \sum_{h_{45} \in H} \chi^{g;h}_{1245} \chi^{g;h}_{2456} \chi^{g;h}_{4568} \overline{\chi}^{g;h}_{1345} \overline{\chi}^{g;h}_{3457} \overline{\chi}^{g;h}_{4578} \sum_{a_{45} \in A^{\text{op}}_{h_{45}}} \frac{d^a_{45}}{\mathcal{D}_{A_{h_{45}}}} \quad \cdots \quad ,$$

$$\tag{5.76}$$

where $\chi^{g;h}_{ijkl}$ and $\overline{\chi}^{g;h}_{ijkl}$ are defined by

$$\chi^{g;h}_{ijkl} := \chi_{ijkl}(g^h_{ij}, g^h_{jk}, g^h_{kl}), \quad \overline{\chi}^{g;h}_{ijkl} := \overline{\chi}_{ijkl}(g^h_{ij}, g^h_{jk}, g^h_{kl}). \tag{5.77}$$

On the first line of (5.76), we used the identity $\sum_{g \in G} \mathcal{O}(g) = \sum_{g_5 \in S_{H \backslash G}} \sum_{h_{45} \in H} \mathcal{O}(g_4^{-1} h_{45} g_5)$ for any quantity $\mathcal{O}$. The second line of (5.76) goes back to the first line if we shrink the loop of $a_{45}$ into a point. A straightforward computation shows that the diagram on the second line

of (5.76) can be evaluated explicitly as follows:

$$
\begin{aligned}
&\left|\; M^{g_5},\, a_{45} \;\right\rangle
= \left|\;
\begin{array}{c}
a_{68}\ \ a_{78}\\
a_{48}\\
a_{46}\ M^{g_4}\ a_{47}\ \ a_{37}\\
a_{26}\ \ a_{24}\ \ \underset{a_{14}}{a_{45}\ M^{g_5}\ } a_{34}\\
a_{12}\ \ a_{13}
\end{array}
\;\right\rangle
\end{aligned}
$$

$$
= \sum_{a_{15},\cdots,a_{58}}\ \sum_{\alpha_{145},\cdots,\alpha_{458}}\ \prod_{i=1,2,3}\sqrt{\frac{d^a_{i5}d^a_{45}}{d^a_{i4}}}\ \prod_{j=6,7,8}\sqrt{\frac{d^a_{5j}d^a_{45}}{d^a_{4j}}}\ \left|\;
\begin{array}{c}
a_{68}\ \ a_{78}\\
a_{58}\\
a_{56}\ a_{45}\ \ a_{45}\ a_{57}\ \ a_{37}\\
a_{26}\ \ a_{45}\ M^{g_5}\ a_{45}\\
a_{25}\ a_{45}\ \ a_{45}\ a_{35}\\
a_{15}\\
a_{12}\ \ a_{13}
\end{array}
\;\right\rangle \qquad (5.78)
$$

$$
= \sum_{a,\alpha}\ \overline{F}^{a;\alpha}_{1245}\,\overline{F}^{a;\alpha}_{2456}\,\overline{F}^{a;\alpha}_{4568}\,F^{a;\alpha}_{1345}\,F^{a;\alpha}_{3457}\,F^{a;\alpha}_{4578}\ \sqrt{\frac{d^a_{15}d^a_{56}d^a_{57}}{d^a_{14}d^a_{46}d^a_{47}}}\ \sqrt{\frac{d^a_{24}d^a_{34}d^a_{48}}{d^a_{25}d^a_{35}d^a_{58}}}\ \left|\;
\begin{array}{c}
a_{68}\ \ a_{78}\\
a_{58}\\
a_{56}\ \ a_{57}\ a_{37}\\
a_{26}\ \ M^{g_5}\\
a_{25}\ \ a_{35}\\
a_{15}\\
a_{12}\ \ a_{13}
\end{array}
\;\right\rangle .
$$

Here, we omitted to draw the surfaces below the honeycomb lattice. The summation on the last line is taken over $\{a_{15}, a_{25}, a_{35}, a_{56}, a_{57}, a_{58}\}$, $\{\alpha_{145}, \alpha_{245}, \alpha_{345}, \alpha_{456}, \alpha_{457}, \alpha_{458}\}$, and $\{\alpha_{125}, \alpha_{256}, \alpha_{568}, \alpha_{135}, \alpha_{357}, \alpha_{578}\}$. Combining (5.76) and (5.78) leads us to

$$
\widehat{\mathsf{h}}_i\left|\;
\begin{array}{c}
a_{68}\ \ a_{78}\\
a_{48}\\
a_{46}\ M^{g_4}\ a_{47}\ a_{37}\\
a_{26}\ a_{24}\ \ a_{34}\\
a_{14}\\
a_{12}\ \ a_{13}
\end{array}
\;\right\rangle
= \sum_{g_5\in S_{H\backslash G}}\ \sum_{h_{45}\in H}\ \chi^{g;h}_{1245}\chi^{g;h}_{2456}\chi^{g;h}_{4568}\overline{\chi}^{g;h}_{1345}\overline{\chi}^{g;h}_{3457}\overline{\chi}^{g;h}_{4578}\ \sum_{a_{45}\in A^{\mathrm{op}}_{h_{45}}}\frac{d^a_{45}}{\mathcal{D}_{A_{h_{45}}}}
$$

$$
\sum_{a_{15},\cdots,a_{58}}\ \sum_{\alpha_{145},\cdots,\alpha_{458}}\ \sum_{\alpha_{125},\cdots,\alpha_{578}}\ \overline{F}^{a;\alpha}_{1245}\,\overline{F}^{a;\alpha}_{2456}\,\overline{F}^{a;\alpha}_{4568}\,F^{a;\alpha}_{1345}\,F^{a;\alpha}_{3457}\,F^{a;\alpha}_{4578}
$$

$$
\sqrt{\frac{d^a_{15}d^a_{56}d^a_{57}}{d^a_{14}d^a_{46}d^a_{47}}}\ \sqrt{\frac{d^a_{24}d^a_{34}d^a_{48}}{d^a_{25}d^a_{35}d^a_{58}}}\ \left|\;
\begin{array}{c}
a_{68}\ \ a_{78}\\
a_{58}\\
a_{56}\ \ a_{57}\ a_{37}\\
a_{26}\ \ M^{g_5}\\
a_{25}\ \ a_{35}\\
a_{15}\\
a_{12}\ \ a_{13}
\end{array}
\;\right\rangle .
$$

$$(5.79)$$

The above Hamiltonian is well-defined as an operator acting on the state space $\mathcal{H}_{\mathrm{gauged}}$ because it satisfies

$$
\widehat{\mathsf{h}}_i\widehat{B}_i = \widehat{B}_i\widehat{\mathsf{h}}_i = \widehat{\mathsf{h}}_i, \qquad (5.80)
$$

where $\widehat{B}_i$ is the Levin-Wen plaquette operator (5.73).

**Gauged Model Written in terms of $A^{\mathrm{rev}}$.** The monoidal equivalence between $A^{\mathrm{op}}$ and $A^{\mathrm{rev}}$ implies that the gauged model can also be written in terms of $A^{\mathrm{rev}}$ instead of $A^{\mathrm{op}}$. In the model written in terms of $A^{\mathrm{rev}}$, the dynamical variables on the plaquettes, edges, and vertices are labeled by elements of $S_{H\backslash G}$, simple objects of $A^{\mathrm{rev}}$, and basis morphisms of $A^{\mathrm{rev}}$, respectively. The state corresponding to a possible configuration of these dynamical variables can be written as

$$
|\{g_i, a_{ij}, \alpha_{ijk}\}\rangle = \left| \begin{array}{c} \includegraphics \end{array} \right\rangle , \tag{5.81}
$$

where the vertices are labeled by $\overline{\alpha}^{\mathrm{rev}}_{ijk}$, $\alpha^{\mathrm{rev}}_{jkl}$, etc. We note that the orientations of the edges in (5.81) are opposite to those in (5.71). The Hamiltonian is again given by

$$
H_{\mathrm{gauged}} = -\sum_{i \in P} \widehat{\mathsf{h}}_i , \tag{5.82}
$$

where $\widehat{\mathsf{h}}_i$ is defined by the same diagram as that in (5.75) except that the edges of the honeycomb lattice (written in black) are now oriented in the opposite direction. The action of $\widehat{\mathsf{h}}_i$ on the state (5.81) can be computed as

$$
\widehat{\mathsf{h}}_i \left| \begin{array}{c} \includegraphics \end{array} \right\rangle = \sum_{g_5 \in S_{H\backslash G}} \sum_{h_{45} \in H} \chi^{g;h}_{1245} \chi^{g;h}_{2456} \chi^{g;h}_{4568} \overline{\chi}^{g;h}_{1345} \overline{\chi}^{g;h}_{3457} \overline{\chi}^{g;h}_{4578} \sum_{a_{45} \in A^{\mathrm{rev}}_{h_{45}}} \frac{d^a_{45}}{\mathcal{D}_{A_{h_{45}}}}
$$

$$
\sum_{a_{15},\cdots,a_{58}} \sum_{\alpha_{145},\cdots,\alpha_{458}} \sum_{\alpha_{125},\cdots,\alpha_{578}} \overline{F}^{a;\alpha}_{1245} \overline{F}^{a;\alpha}_{2456} \overline{F}^{a;\alpha}_{4568} F^{a;\alpha}_{1345} F^{a;\alpha}_{3457} F^{a;\alpha}_{4578}
$$

$$
\sqrt{\frac{d^a_{15} d^a_{56} d^a_{57}}{d^a_{14} d^a_{46} d^a_{47}}} \sqrt{\frac{d^a_{24} d^a_{34} d^a_{48}}{d^a_{25} d^a_{35} d^a_{58}}} \left| \begin{array}{c} \includegraphics \end{array} \right\rangle .
$$

$$
\tag{5.83}
$$

The above Hamiltonian agrees with (5.79) upon identifying the states defined by (5.71) and (5.81).

Since the two models written in terms of $A^{\mathrm{op}}$ and $A^{\mathrm{rev}}$ are equivalent, we can use whichever description we want. In what follows, we will employ the description in terms of $A^{\mathrm{rev}}$. Namely, the state and the Hamiltonian of the gauged model are given by (5.81) and (5.83).

**Ordinary Gauging as a Special Case.** The generalized gauging described above includes ordinary (twisted) gauging of a finite group symmetry. The ordinary gauging of non-anomalous finite group symmetries in 2+1d lattice models was studied in detail in [76].

To recover the ordinary gauging, we choose the $G$-graded fusion category $A$ to be

$$A = \mathsf{Vec}_H^\nu, \tag{5.84}$$

where $H \subset G$ is a subgroup of $G$ and $\nu \in Z^3(H, \mathrm{U}(1))$ is a 3-cocycle on $H$. For this choice of $A$, the dynamical variables of the gauged model are given as follows:

- The dynamical variables on the plaquettes take values in $S_{H\backslash G}$.

- The dynamical variables on the edges take values in $H$.[63] These dynamical variables are subject to the constraint $h_{ik} = h_{ij}h_{jk}$ around each vertex $[ijk]$. Here, $h_{ij}$ denotes the dynamical variable on edge $[ij]$.

- There are no dynamical variables on the vertices.

The Levin-Wen plaquette operator is trivial because $A_e$ is equivalent to $\mathsf{Vec}$. Therefore, the state space $\mathcal{H}_{\mathrm{gauged}}$ is given by the space of all possible configurations of the dynamical variables on the lattice. The Hamiltonian (5.83) reduces to

$$\widehat{\mathsf{h}}_i \left| \begin{array}{c} h_{68} \quad h_{78} \\ h_{48} \\ h_{46} \quad h_{47} \, h_{37} \\ h_{26} \, h_{24} \quad M^{g_4} \quad h_{34} \\ h_{14} \\ h_{12} \quad h_{13} \end{array} \right\rangle = \sum_{g_5 \in S_{H\backslash G}} \sum_{h_{45} \in H} \chi_{1245}^{g;h} \chi_{2456}^{g;h} \chi_{4568}^{g;h} \overline{\chi}_{1345}^{g;h} \overline{\chi}_{3457}^{g;h} \overline{\chi}_{4578}^{g;h}$$

$$\frac{\nu(h_{13}, h_{34}, h_{45})\nu(h_{34}, h_{45}, h_{57})\nu(h_{45}, h_{57}, h_{78})}{\nu(h_{12}, h_{24}, h_{45})\nu(h_{24}, h_{45}, h_{56})\nu(h_{45}, h_{56}, h_{68})} \left| \begin{array}{c} h_{68} \quad h_{78} \\ h_{58} \\ h_{56} \quad h_{57} \, h_{37} \\ h_{26} \, h_{25} \quad M^{g_5} \quad h_{35} \\ h_{15} \\ h_{12} \quad h_{13} \end{array} \right\rangle. \tag{5.85}$$

Physically, $h_{ij}$ represents an $H$-gauge field. The constraint $h_{ik} = h_{ij}h_{jk}$ implies that this gauge field is flat. The Gauss law constraint is imposed exactly (not energetically) on the state space $\mathcal{H}_{\mathrm{gauged}}$. We note that the Gauss law constraint is already solved explicitly, which is why the dynamical variables on the plaquettes are labeled by $H$-cosets in $G$ rather than elements of $G$. The minimal coupling is implemented through $\chi_{ijkl}^{g;h}$ and $\overline{\chi}_{ijkl}^{g;h}$.

---

[63] As discussed in section 5.2.2, the dynamical variables on the edges are generally labeled by simple objects of $A^{\mathrm{rev}}$. When $A = \mathsf{Vec}_H^\nu$, (the isomorphism classes of) simple objects of $A^{\mathrm{rev}}$ are in one-to-one correspondence with elements of $H$. Hence, the dynamical variables on the edges are labeled by elements of $H$.

### 5.2.3 Generalized Gauging Operators

In this subsection, we write down the generalized gauging operators that implement the generalized gauging of $2\mathsf{Vec}_G$ symmetry. We will also discuss the generalized gauging operators that map the gauged model back to the original model. The gauging operators for the minimal gauging were studied in [76].[64]

**From the Original Model to the Gauged Model.**  Let us first consider the generalized gauging operator that maps the original model to the gauged model:

$$\mathsf{D}_A : \mathcal{H}_{\text{original}} \to \mathcal{H}_{\text{gauged}}. \tag{5.86}$$

We recall that in the original model, the dynamical degrees of freedom are labeled by objects, 1-morphisms, and 2-morphisms of $2\mathsf{Vec}_G$. On the other hand, in the gauged model, the dynamical degrees of freedom are labeled by objects, 1-morphisms, and 2-morphisms of $_A(2\mathsf{Vec}_G)$. Thus, the generalized gauging operator maps objects, 1-morphisms, and 2-morphisms of $2\mathsf{Vec}_G$ to those of $_A(2\mathsf{Vec}_G)$. Mathematically, this map is given by the free module functor

$$F_A : 2\mathsf{Vec}_G \to {}_A(2\mathsf{Vec}_G). \tag{5.87}$$

The free module functor $F_A$ is a functor that maps an object $X \in 2\mathsf{Vec}_G$ to $A\square X \in {}_A(2\mathsf{Vec}_G)$, a 1-morphism $x \in \text{Hom}_{2\mathsf{Vec}_G}(X, X')$ to $1_A\square x \in \text{Hom}_{A(2\mathsf{Vec}_G)}(A\square X, A\square X')$, and a 2-morphism $\chi \in \text{Hom}_{2\mathsf{Vec}_G}(x, x')$ to $\text{id}_{1_A}\square \chi \in \text{Hom}_{A(2\mathsf{Vec}_G)}(1_A\square x, 1_A\square x')$, where $1_A$ denotes the identity 1-morphism of $A \in {}_A(2\mathsf{Vec}_G)$. Pictorially, the free module functor $F_A$ is represented by the stacking of the surface labeled by $A$:

$$\tag{5.88}$$

Accordingly, the generalized gauging operator $\mathsf{D}_A$ is given by the fusion with the surface labeled $A$.

To write down the generalized gauging operator $\mathsf{D}_A$ on the lattice, we consider the action of $\mathsf{D}_A$ on a state $|\{h_i g_i\}\rangle \in \mathcal{H}_{\text{original}}$, where $h_i \in H$ and $g_i \in S_{H\backslash G}$.[65] Based on the diagrammatic

---

[64]In 1+1d, the gauging operators for general fusion category symmetries are studied in [52, 55]. In 2+1d, some of the gauging operators for $2\mathsf{Rep}(G)$ symmetry are also studied in [99] in the context of abelian lattice gauge theories. In 3+1d, the gauging operator for $\mathbb{Z}_2$ 1-form symmetry is studied in [98]. See also [54] for sequential quantum circuit representations of some gauging operators.

[65]The decomposition of an element of $G$ into the product of elements of $H$ and $S_{H\backslash G}$ is unique.

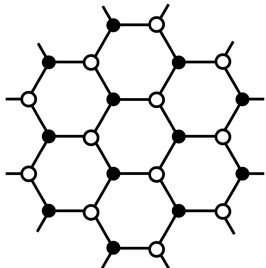

Figure 14: The black dots constitute the $A$-sublattice, while the white dots constitute the $B$-sublattice.

representation (5.88), one can compute the action of $\mathsf{D}_A$ as follows:

$$
\begin{aligned}
\mathsf{D}_A \left|\{h_i g_i\}\right\rangle &= \sum_{\{a_i \in A_{h_i}^{\mathrm{rev}}\}} \prod_{i \in P} \frac{d_i^a}{\mathcal{D}_{A_{h_i}}} \\
&= \sum_{\{a_i \in A_{h_i}^{\mathrm{rev}}\}} \prod_{i \in P} \frac{d_i^a}{\mathcal{D}_{A_{h_i}}} \sum_{\{a_{ij}, \alpha_{ij}\}} \prod_{[ij] \in E} \sqrt{\frac{d_{ij}^a}{d_i^a d_j^a}} \\
&= \sum_{\{a_i \in A_{h_i}^{\mathrm{rev}}\}} \prod_{i \in P} \frac{1}{\mathcal{D}_{A_{h_i}}} \sum_{\{a_{ij}, \alpha_{ij}, \alpha_{ijk}\}} \prod_{[ijk] \in V} \left(\frac{d_{ij}^a d_{jk}^a}{d_{ik}^a}\right)^{\frac{1}{4}} \prod_{[ijk] \in V} (\overline{F}_{ijk}^{a;\alpha})^{s_{ijk}} \left|\{g_i, a_{ij}, \alpha_{ijk}\}\right\rangle .
\end{aligned}
$$
(5.89)

Here, $a_{ij} \in \overline{a_i} \otimes a_j$ is a simple object of $A_{h_i^{-1} h_j}^{\mathrm{rev}}$, $\alpha_{ij}$ and $\alpha_{ijk}$ are the basis morphisms of $\mathrm{Hom}_A(\overline{a_i} \otimes a_j, a_{ij})$ and $\mathrm{Hom}_A(a_{ij} \otimes a_{jk}, a_{ik})$, and $s_{ijk}$ is defined by

$$
s_{ijk} = \begin{cases} +1 & \text{if } [ijk] \text{ is in the } A\text{-sublattice,} \\ -1 & \text{if } [ijk] \text{ is in the } B\text{-sublattice.} \end{cases}
$$
(5.90)

See figure 14 for the definitions of the $A$- and $B$-sublattices. We note that the above generalized gauging operator satisfies $\widehat{B}_i \mathsf{D}_A = \mathsf{D}_A$ for all plaquettes $i$, meaning that the image of $\mathsf{D}_A$ is contained in the state space of the gauged model. Thus, $\mathsf{D}_A$ is well-defined as a linear map from $\mathcal{H}_{\mathrm{original}}$ to $\mathcal{H}_{\mathrm{gauged}}$. By construction, the generalized gauging operator $\mathsf{D}_A$ intertwines

the Hamiltonians of the original and gauged models, i.e.,

$$\mathsf{D}_A H_{\text{original}} = H_{\text{gauged}} \mathsf{D}_A. \tag{5.91}$$

**From the Gauged Model to the Original Model.** We can also consider a linear map that implements the ungauging from the gauged model to the original model:

$$\overline{\mathsf{D}}_A : \mathcal{H}_{\text{gauged}} \to \mathcal{H}_{\text{original}}. \tag{5.92}$$

This operator maps the dynamical degrees of freedom of the gauged model to those of the original model. Namely, it maps objects, 1-morphisms, and 2-morphisms of $_A(2\mathsf{Vec}_G)$ to those of $2\mathsf{Vec}_G$. Mathematically, this map is described by the forgetful functor[66]

$$\text{Forg} : {}_A(2\mathsf{Vec}_G) \to 2\mathsf{Vec}_G, \tag{5.93}$$

which maps objects, 1-morphisms, 2-morphisms of $_A(2\mathsf{Vec}_G)$ to their underlying objects (5.46), underlying 1-morphisms (5.47) (5.48), and underlying 2-morphisms (5.49) (5.51), respectively. The corresponding generalized gauging operator $\overline{\mathsf{D}}_A$ thus acts on a state as

$$\boxed{\overline{\mathsf{D}}_A \left| \{g_i, a_{ij}, \alpha_{ijk}\} \right\rangle = \sum_{\{h_i \in H\}} \sum_{\{a_i \in A_{h_i}^{\text{rev}}\}} \sum_{\{\alpha_{ij}\}} \prod_{[ijk]} \left( \frac{d_{ij}^a d_{jk}^a}{d_{ik}^a} \right)^{\frac{1}{4}} \prod_{[ijk]} (F_{ijk}^{a;\alpha})^{s_{ijk}} \left| \{h_i g_i\} \right\rangle.} \tag{5.94}$$

We note that $\overline{\mathsf{D}}_A$ satisfies $\overline{\mathsf{D}}_A \widehat{B}_i = \overline{\mathsf{D}}_A$ for all plaquettes $i$. Hence, $\overline{\mathsf{D}}_A$ is well-defined as a linear map from the state space of the gauged model to that of the original model.

The action of $\overline{\mathsf{D}}_A$ on a state $\left| \{g_i, a_{ij}, \alpha_{ijk}\} \right\rangle$ vanishes if the gauge field $\{h_{ij} \mid [ij] \in E\}$, defined by the grading of $a_{ij} \in A_{h_{ij}}$, has a non-trivial holonomy around a closed loop. Indeed, when the gauge field has a non-trivial holonomy, there is no configuration $\{h_i \mid i \in P\}$ that satisfies $h_{ij} = h_i^{-1} h_j$, meaning that the summand on the right-hand side of (5.94) is empty. Thus, for any oriented loop $\gamma$ on the dual of the honeycomb lattice, we have

$$\prod_{[ij] \in E_\gamma} h_{ij}^{s_{ij}} \neq e \implies \overline{\mathsf{D}}_A \left| \{g_i, a_{ij}, \alpha_{ijk}\} \right\rangle = 0. \tag{5.95}$$

Here, $\prod_{[ij] \in E_\gamma}$ is the path-ordered product over all edges intersecting $\gamma$. The sign $s_{ij}$ in (5.95) is defined by

$$s_{ij} = \begin{cases} +1 & \text{if } \gamma \text{ intersects } [ij] \text{ from the right of } [ij], \\ -1 & \text{if } \gamma \text{ intersects } [ij] \text{ from the left of } [ij]. \end{cases} \tag{5.96}$$

The left and right are defined with respect to the orientation of edge $[ij]$.

---

[66]We note that the forgetful functor Forg is the right adjoint to the free module functor $F_A$.

As a sanity check, let us consider the product $\overline{\mathsf{D}}_A \mathsf{D}_A$ of the generalized gauging operators. Since $\mathsf{D}_A$ and $\overline{\mathsf{D}}_A$ implement the free module functor (5.88) and the forgetful functor (5.93), the product $\overline{\mathsf{D}}_A \mathsf{D}_A$ implements the composite functor $\mathrm{Forg} \circ F_A : 2\mathsf{Vec}_G \to 2\mathsf{Vec}_G$, which maps $X \in 2\mathsf{Vec}_G$ to $A \square X \in 2\mathsf{Vec}_G$. Recalling that the underlying object of $A$ is given by (5.3), we find

$$\overline{\mathsf{D}}_A \mathsf{D}_A = \sum_{h \in H} \mathrm{rank}(A_h) U_h. \tag{5.97}$$

One can check the above equality by a direct computation using (5.89) and (5.94), see appendix C for details.

**Relation to Hermitian Conjugate.** The generalized gauging operators $\mathsf{D}_A$ and $\overline{\mathsf{D}}_A$ are related to each other by the Hermitian conjugate (up to normalization). To see this, we first define the Hermitian inner products on the state space of the original model and that of the gauged model as follows:[67]

$$\langle \{h_i' g_i'\} | \{h_i g_i\} \rangle = \prod_i \delta_{h_i, h_i'} \delta_{g_i, g_i'},$$
$$\langle \{g_i', a_{ij}', \alpha_{ijk}'\} | \{g_i, a_{ij}, \alpha_{ijk}\} \rangle = \prod_i \delta_{g_i, g_i'} \prod_{[ij]} \delta_{a_{ij}, a_{ij}'} \prod_{[ijk]} \delta_{\alpha_{ijk}, \alpha_{ijk}'}. \tag{5.98}$$

The Hermitian conjugate of $\mathsf{D}_A$ with respect to the above inner product is given by

$$\langle \{h_i g_i'\} | \mathsf{D}_A^\dagger | \{g_i, a_{ij}, \alpha_{ijk}\} \rangle = \prod_i \frac{\delta_{g_i, g_i'}}{\mathcal{D}_{A_{h_i}}} \sum_{\{a_i\}} \sum_{\{\alpha_{ij}\}} \prod_{[ijk]} \left( \frac{d_{ij}^a d_{jk}^a}{d_{ik}^a} \right)^{\frac{1}{4}} \prod_{[ijk]} (F_{ijk}^{a;\alpha})^{s_{ijk}}, \tag{5.99}$$

where $a_i$ is a simple object of $A_{h_i}^{\mathrm{rev}}$, $\alpha_{ij}$ is a basis morphism of $\mathrm{Hom}_A(\overline{a_i} \otimes a_j, a_{ij})$, and we used the unitarity of the $F$-symbol, i.e., $(F_d^{abc})^*_{(e;\mu,\nu),(f;\rho,\sigma)} = (\overline{F}_d^{abc})_{(e;\mu,\nu),(f;\rho,\sigma)}$. On the other hand, the matrix element of $\overline{\mathsf{D}}_A$ is given by

$$\langle \{h_i g_i'\} | \overline{\mathsf{D}}_A | \{g_i, a_{ij}, \alpha_{ijk}\} \rangle = \prod_i \delta_{g_i, g_i'} \sum_{\{a_i\}} \sum_{\{\alpha_{ij}\}} \prod_{[ijk]} \left( \frac{d_{ij}^a d_{jk}^a}{d_{ik}^a} \right)^{\frac{1}{4}} \prod_{[ijk]} (F_{ijk}^{a;\alpha})^{s_{ijk}}. \tag{5.100}$$

By comparing (5.99) and (5.100), we find

$$\overline{\mathsf{D}}_A = \mathsf{D}_A^\dagger \prod_i \mathcal{D}_{A_e}. \tag{5.101}$$

Here, we used the identity $\mathcal{D}_{A_e} = \mathcal{D}_{A_h}$ for all $h \in H$ [159]. Equation (5.101) shows that $\overline{\mathsf{D}}_A$ agrees with $\mathsf{D}_A^\dagger$ up to normalization. In the case of the minimal gauging $A = \mathsf{Vec}_H^\nu$, we have $\overline{\mathsf{D}}_A = \mathsf{D}_A^\dagger$.

---

[67]The second line of (5.98) defines the Hermitian inner product on the vector space $\mathcal{H}_{\mathrm{gauged}}'$ spanned by all possible configurations of dynamical variables of the gauged model. This induces the Hermitian inner product on the state space $\mathcal{H}_{\mathrm{gauged}} \subset \mathcal{H}_{\mathrm{gauged}}'$.

**Generalized Gauging Operators for Twisted Sectors.** One can also define the generalized gauging operators that map the $H$-twisted sectors of the original model to the untwisted sector of the gauged model. We will use such generalized gauging operators in section 5.3.4 to construct topological surface operators decorated by topological line operators.

We first define the twisted sectors of the original model. Each state in the $g$-twisted sector for $g \in G$ is represented by the following fusion diagram:

$$|\{h_i g_i\}\rangle = \quad \quad = \quad \quad . \tag{5.102}$$

The line on which the defect $D_2^g$ ends is denoted by $\gamma$, which we suppose to be straight for simplicity.[68] For later use, we define $\tilde{\gamma}$ as the set of edges located immediately to the left of $\gamma$. The state space of the twisted sector is spanned by all the fusion diagrams of the above form with fixed $\gamma$. The Hamiltonian acting on the state (5.102) is defined by the same diagram as in the Hamiltonian of the untwisted sector.

A generalized gauging operator that maps the $h$-twisted sector for $h \in H$ to the untwisted sector of the gauged model is defined by the following fusion diagram:[69]

$$\mathsf{D}_A^a |\{h_i g_i\}\rangle = \quad \quad . \tag{5.103}$$

Here, the surface operator on the right-hand side is labeled by a left $A$-module $A \in {}_A(2\mathsf{Vec}_G)$. At the junction of surfaces $A$ and $D_2^h$, we have a topological line $a$, which is a 1-morphism from $A$ to $A \lhd D_2^h$ in ${}_A(2\mathsf{Vec}_G)$:

$$a \in \mathrm{Hom}_{A(2\mathsf{Vec}_G)}(A, A \lhd D_2^h) \cong A_h^{\mathrm{rev}}. \tag{5.104}$$

We note that a different choice of $a \in A_h^{\mathrm{rev}}$ gives rise to a different generalized gauging operator for the $h$-twisted sector. The operator $\mathsf{D}_A^a$ defined by (5.103) can be evaluated explicitly as

---

[68]It is straightforward to generalize the definition of the twisted sector to the case where $\gamma$ is more complicated. More generally, one can also define the sector twisted by a general network of topological defects.

[69]The states in the $g$-twisted sector for $g \notin H$ are mapped to the twisted sector states of the gauged model.

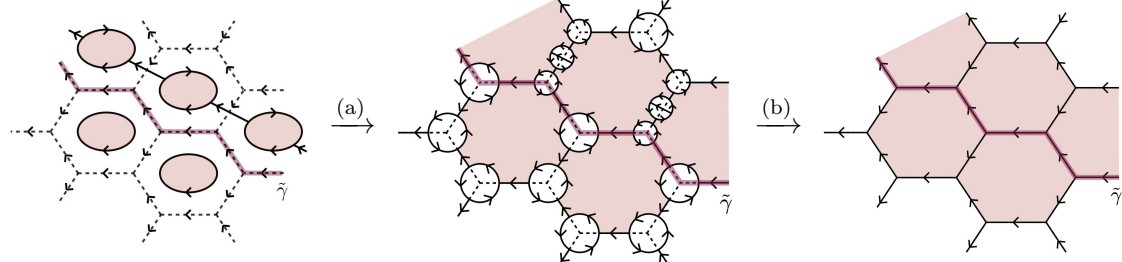

Figure 15: (a) The partial fusion of topological lines around all edges of the honeycomb lattice. (b) The $F$-move around all vertices of the honeycomb lattice and around the edges that intersect $\gamma$.

follows:

$$\mathsf{D}_A^a \left|\{h_i g_i\}\right\rangle = \sum_{\{a_i \in A_{h_i}^{\mathrm{rev}}\}} \prod_i \frac{d_i^a}{\mathcal{D}_{A_{h_i}}} \quad \cdots$$

$$= \sum_{\{a_i \in A_{h_i}^{\mathrm{rev}}\}} \prod_i \frac{d_i^a}{\mathcal{D}_{A_{h_i}}} \sum_{\{a_i' \in a \otimes a_i | i \in P_\gamma\}} \prod_{i \in P_\gamma} \sqrt{\frac{d_i^{a'}}{d^a d_i^a}} \quad \cdots \quad .$$

$$\tag{5.105}$$

Here, $P_\gamma$ is the set of plaquettes that intersect $\gamma$. The diagram on the right-hand side can be evaluated by doing the partial fusion and the $F$-move as shown in figure 15. As a result, the matrix element of $\mathsf{D}_A^a$ can be expressed in terms of the $F$-symbols and the quantum dimensions. We note that $\mathsf{D}_A^a$ reduces to $\mathsf{D}_A$ when $h$ is the unit element of $H$ and $a$ is the unit object of $A_e^{\mathrm{rev}}$.

Let us write down the matrix element of $\mathsf{D}_A^a$ more explicitly in the case of the minimal gauging.[70] For the minimal gauging $A = \mathsf{Vec}_H^\nu$, the simple topological line $a \in A_h^{\mathrm{rev}}$ is unique because $A_h^{\mathrm{rev}}$ has only one simple object. Thus, the gauging operator for the $h$-twisted sector

---

[70]Even in the case of the non-minimal gauging, it is not difficult to compute the matrix element of $\mathsf{D}_A^a$. However, the explicit expression becomes highly complicated in general.

is unique and is denoted by $\mathsf{D}_A^h$. The action of $\mathsf{D}_A^h$ on a state can be computed as

$$\mathsf{D}_A^h \,|\{h_i g_i\}\rangle = \prod_{[ijk]} \nu(h_i, \tilde{h}_{ij}, \tilde{h}_{jk})^{-s_{ijk}} \prod_{[ij]\in E_\gamma} \nu(h, h_i, \tilde{h}_{ij})^{-1} \,|\{g_i, \tilde{h}_{ij}\}\rangle , \qquad (5.106)$$

where $\gamma$ is chosen as in (5.102), $E_\gamma$ is the set of edges that intersect $\gamma$, and $\tilde{h}_{ij}$ is defined by

$$\tilde{h}_{ij} := \begin{cases} h_i^{-1} h_j & \text{for } [ij] \notin \tilde{\gamma}, \\ h_i^{-1} h h_j & \text{for } [ij] \in \tilde{\gamma}. \end{cases} \qquad (5.107)$$

### 5.2.4 Tensor Network Representation

Let us write down the tensor network representations of the generalized gauging operators $\mathsf{D}_A$ and $\overline{\mathsf{D}}_A$. For simplicity, we suppose that $A$ is multiplicity-free, i.e., the fusion coefficient of $A$ is either zero or one. See, e.g., [84] for a review of the tensor networks. A minimal background on the tensor networks is provided in section 3.4.

The generalized gauging operator $\mathsf{D}_A$ defined by (5.89) can be represented by the following tensor network:

$$\mathsf{D}_A = \qquad\qquad\qquad\qquad . \qquad (5.108)$$

Here, the physical legs are written in black, while the virtual bonds are written in red and green. The physical leg below the plaquette takes values in $G$, the physical leg above the plaquette takes values in $S_{H\backslash G}$, and the physical leg on each edge takes values in the set of simple objects of $A$. On the other hand, the virtual bonds written in green take values in $H$, and those written in red take values in the set of simple objects of $A$. The non-zero components of the local tensors in (5.108) are defined as follows:

$$\begin{array}{ccc} & & \\ \underset{hg \ \ h}{\overset{h \ \ g}{\times}} = \dfrac{1}{\mathcal{D}_{A_h}}, & a \underset{a}{\overset{a}{\bullet}} a = 1, & a \underset{a}{\overset{h}{\rule{0pt}{0pt}}} a = \delta_{a\in A_h}, \end{array} \qquad (5.109)$$

$$\underset{a_{ij} \ a_i}{\overset{a_{jk} \ a_k}{\blacktriangleleft}} a_{ik} = \left(\frac{d_{ij}^a d_{jk}^a}{d_{ik}^a}\right)^{\frac{1}{4}} \overline{F}_{ijk}^a, \qquad a_{ik} \underset{a_i \ a_{ij}}{\overset{a_k \ a_{jk}}{\blacktriangleright}} = \left(\frac{d_{ij}^a d_{jk}^a}{d_{ik}^a}\right)^{\frac{1}{4}} F_{ijk}^a. \qquad (5.110)$$

Here, $h \in H$, $g \in S_{H\backslash G}$, $a \in A$, and $\delta_{a\in A_h}$ is one if $a \in A_h$ and zero otherwise. We omitted the superscript $\alpha$ of the $F$-symbols because $A$ is multiplicity-free.

Similarly, $\overline{\mathsf{D}}_A$ defined by (5.94) can be represented by the following tensor network:

$$\overline{\mathsf{D}}_A = \quad \text{} \quad . \tag{5.111}$$

We note that the physical legs are reversed from those in (5.108). In particular, the physical leg above the plaquette takes values in $G$, while the physical leg below the plaquette takes values in $S_{H\backslash G}$. The non-zero components of the local tensors in (5.111) are defined as

$$\text{} = 1, \qquad \text{} = 1, \qquad \text{} = \delta_{a \in A_h}, \tag{5.112}$$

$$\text{} = \left( \frac{d_{ij}^a d_{jk}^a}{d_{ik}^a} \right)^{\frac{1}{4}} F_{ijk}^a, \qquad \text{} = \left( \frac{d_{ij}^a d_{jk}^a}{d_{ik}^a} \right)^{\frac{1}{4}} \overline{F}_{ijk}^a, \tag{5.113}$$

where $h \in H$, $g \in S_{H\backslash G}$, and $a \in A$.

## 5.3  Symmetry of the Gauged Model

As argued in section 4.2, if we gauge a fusion 2-category symmetry $\mathcal{C}$ by using a right $\mathcal{C}$-module 2-category $\mathcal{M}$, the symmetry category of the gauged model becomes the dual 2-category $\mathcal{C}_{\mathcal{M}}^*$, the 2-category of $\mathcal{C}$-module 2-endofunctors of $\mathcal{M}$. When $\mathcal{M}$ is the 2-category $_A\mathcal{C}$ of left $A$-modules in $\mathcal{C}$, the dual 2-category $\mathcal{C}_{\mathcal{M}}^*$ is equivalent to the 2-category $_A\mathcal{C}_A$ of $(A, A)$-bimodules in $\mathcal{C}$ [142]. In particular, when $\mathcal{C} = 2\mathsf{Vec}_G$, the symmetry operators of the gauged model form a fusion 2-category $_A(2\mathsf{Vec}_G)_A$, where $A \in 2\mathsf{Vec}_G$ is a $G$-graded fusion category. In what follows, we will describe the structure of this symmetry category and write down some of the symmetry operators explicitly on the lattice. For the minimal gauging $A = \mathsf{Vec}_H^\nu$, the symmetry operators of the gauged model were studied in detail in [76].

### 5.3.1  Dual 2-Category $_A(2\mathsf{Vec}_G)_A$

Let us first recall the basic data of the dual 2-category $_A(2\mathsf{Vec}_G)_A$. See [115, Section 2.3] for more details.

An object of $_A(2\mathsf{Vec}_G)_A$ is an $(A, A)$-bimodule 1-category equipped with a $G$-grading compatible with the left and right $A$-actions. More specifically, an object $M \in {}_A(2\mathsf{Vec}_G)_A$ is a

$G$-graded finite semisimple 1-category that is equipped with grading-preserving left and right $A$-actions $\triangleright: A \times M \to M$ and $\triangleleft: M \times A \to M$ together with natural isomorphisms

$$l_{a,b,m} : (a \otimes b) \triangleright m \to a \triangleright (b \triangleright m),$$
$$r_{m,a,b} : (m \triangleleft a) \triangleleft b \to m \triangleleft (a \otimes b), \qquad (5.114)$$
$$\beta_{a,m,b} : (a \triangleright m) \triangleleft b \to a \triangleright (m \triangleleft b),$$

for all $a, b \in A$ and $m \in M$. The above natural isomorphisms are required to satisfy the usual coherence conditions of a bimodule category.

A 1-morphism of $_A(2\mathsf{Vec}_G)_A$ is an $(A, A)$-bimodule functor that preserves the $G$-grading. More specifically, a 1-morphism $F \in \mathrm{Hom}_{_A(2\mathsf{Vec}_G)_A}(M, M')$ is a grading-preserving functor $F : M \to M'$ that is equipped with natural isomorphisms

$$s_{a,m} : F(a \triangleright m) \to a \triangleright F(m),$$
$$t_{m,a} : F(m \triangleleft a) \to F(m) \triangleleft a, \qquad (5.115)$$

for all $a \in A$ and $m \in M$. The above natural isomorphisms are required to satisfy the usual coherence conditions of a bimodule functor.

A 2-morphism $\eta : F \Rightarrow F'$ for $F, F' \in \mathrm{Hom}_{_A(2\mathsf{Vec}_G)_A}(M, M')$ is an $(A, A)$-bimodule natural transformation from $F$ to $F'$. Namely, $\eta$ is a natural transformation that is compatible with the $(A, A)$-bimodule structures in the usual sense.

### 5.3.2 0-Form Symmetry

The 0-form symmetry operators of the gauged model are labeled by objects of $_A(2\mathsf{Vec}_G)_A$, i.e., $(A, A)$-bimodules in $2\mathsf{Vec}_G$. A simple example of an $(A, A)$-bimodule in $2\mathsf{Vec}_G$ is $A\square D_2^g\square A$ for any $g \in G$. The $(A, A)$-bimodule structure on $A\square D_2^g\square A$ is given by the left and right multiplication of $A$. As we will see below, $A\square D_2^g\square A$ is a simple object of $_A(2\mathsf{Vec}_G)_A$.

Any $(A, A)$-bimodule in $2\mathsf{Vec}_G$ is connected to $A\square D_2^g\square A$ for some $g \in G$. This follows from the equivalence of 1-categories

$$\mathrm{Hom}_{_A(2\mathsf{Vec}_G)_A}(A\square D_2^g\square A, M) \cong \mathrm{Hom}_{2\mathsf{Vec}_G}(D_2^g, M), \qquad (5.116)$$

where $M \in {}_A(2\mathsf{Vec}_G)_A$ is an arbitrary non-zero $(A, A)$-bimodule in $2\mathsf{Vec}_G$. Since $M$ is non-zero, there exists some $g \in G$ such that the right-hand side of (5.116) is not empty, meaning that $M$ is connected to $A\square D_2^g\square A$. In particular, two objects $A\square D_2^g\square A$ and $A\square D_2^{g'}\square A$ are connected to each other if and only if $g$ and $g'$ are in the same double $H$-coset in $G$:

$$HgH = Hg'H \iff \mathrm{Hom}_{_A(2\mathsf{Vec}_G)_A}(A\square D_2^g\square A, A\square D_2^{g'}\square A) \neq 0. \qquad (5.117)$$

Therefore, the connected components of simple objects of $_A(2\mathsf{Vec}_G)_A$ are in one-to-one correspondence with double $H$-cosets in $G$ [115].

As mentioned above, the object $A\square D_2^g\square A$ is simple in $_A(2\mathsf{Vec}_G)_A$. Equivalently, the unit object of the endomorphism 1-category $\mathrm{End}_{_A(2\mathsf{Vec}_G)_A}(A\square D_2^g\square A)$ is simple. To see this, we again use the equivalence of 1-categories

$$\mathrm{End}_{_A(2\mathsf{Vec}_G)_A}(A\square D_2^g\square A) \cong \mathrm{Hom}_{2\mathsf{Vec}_G}(D_2^g, A\square D_2^g\square A). \tag{5.118}$$

A 1-morphism $f \in \mathrm{End}_{_A(2\mathsf{Vec}_G)_A}(A\square D_2^g\square A)$ is mapped to $f\circ(i\square 1_{D_2^g}\square i) \in \mathrm{Hom}_{2\mathsf{Vec}_G}(D_2^g, A\square D_2^g\square A)$ by the above equivalence. Here, $i : D_2^e \to A$ is the unit 1-morphism of the algebra $A \in 2\mathsf{Vec}_G$, which is given by the inclusion 1-morphism

$$i : D_2^e \xrightarrow{\iota_{D_2^e[\mathbb{1}_A]}} A = \bigoplus_{g\in G}\bigoplus_{a_g\in A_g} D_2^g[a_g]. \tag{5.119}$$

We note that the unit 1-morphism $i$ is simple in $2\mathsf{Vec}_G$.[71] Under the equivalence (5.118), the unit object of $\mathrm{End}_{_A(2\mathsf{Vec}_G)_A}(A\square D_2^g\square A)$ is mapped to the 1-morphism $i\square 1_{D_2^g}\square i$, which is also simple in $2\mathsf{Vec}_G$. This in turn implies that the unit object of $\mathrm{End}_{_A(2\mathsf{Vec}_G)_A}(A\square D_2^g\square A)$ is simple. Therefore, $A\square D_2^g\square A$ is a simple object of $_A(2\mathsf{Vec}_G)_A$.

**Symmetry Operators on the Lattice.** The symmetry operator corresponding to the object $A\square D_2^g\square A$ is given by $\mathsf{D}_A U_g \overline{\mathsf{D}}_A$, where $\mathsf{D}_A$ and $\overline{\mathsf{D}}_A$ are the generalized gauging operators given by (5.89) and (5.94). The action of this operator can be computed as

$$
\begin{aligned}
\mathsf{D}_A U_g \overline{\mathsf{D}}_A \left|\{g_i, a_{ij}, \alpha_{ijk}\}\right\rangle = \sum_{\{h_i, a_i, \alpha_{ij}\}} \sum_{\{a_i', a_{ij}', \alpha_{ij}', \alpha_{ijk}'\}} \prod_{[ijk]} \left(\frac{d_{ij}^a d_{jk}^a}{d_{ik}^a}\right)^{\frac{1}{4}} \prod_{[ijk]} (F_{ijk}^{a;\alpha})^{s_{ijk}} \\
\prod_i \frac{1}{\mathcal{D}_{A_{h_i'}}} \prod_{[ijk]} \left(\frac{d_{ij}^{a'} d_{jk}^{a'}}{d_{ik}^{a'}}\right)^{\frac{1}{4}} \prod_{[ijk]} (\overline{F}_{ijk}^{a';\alpha'})^{s_{ijk}} \left|\{g_i', a_{ij}', \alpha_{ijk}'\}\right\rangle,
\end{aligned} \tag{5.120}
$$

where $h_i' \in H$ and $g_i' \in S_{H\backslash G}$ are uniquely determined by $h_i' g_i' = g h_i g_i$. The summations on the right-hand side are taken over $h_i \in H$, $a_i \in A_{h_i}^{\mathrm{rev}}$, $\alpha_{ij} \in \mathrm{Hom}_A(\overline{a_i}\otimes a_j, a_{ij})$, $a_i' \in A_{h_i'}^{\mathrm{rev}}$, $a_{ij}' \in \overline{a_i'}\otimes a_j'$, $\alpha_{ij}' \in \mathrm{Hom}_A(\overline{a_i'}\otimes a_j', a_{ij}')$, and $\alpha_{ijk}' \in \mathrm{Hom}_A(a_{ij}'\otimes a_{jk}', a_{ik}')$. The symmetry operators corresponding to the other simple objects of $_A(2\mathsf{Vec}_G)_A$ should be related by condensation to the above symmetry operators, because any object of $_A(2\mathsf{Vec}_G)_A$ is connected to $A\square U_g\square A$ for some $g \in G$.

---

[71]When $A$ is a $G$-graded multifusion category whose unit object $\mathbb{1}_A = \bigoplus_i \mathbb{1}_A^{(i)}$ is non-simple, the unit 1-morphism $i : D_2^e \to A$ is given by the direct sum of the inclusion 1-morphisms $\iota_{D_2^e[\mathbb{1}_A^{(i)}]}$ for all simple components of $\mathbb{1}_A$. In particular, $i$ is not simple when $\mathbb{1}_A$ is not simple, i.e., when $A$ is not a fusion category.

In the case of the minimal gauging $A = \mathsf{Vec}_H^\nu$, the symmetry action (5.120) reduces to

$$
\boxed{\mathsf{D}_A U_g \overline{\mathsf{D}}_A \,|\{g_i, h_{ij}\}\rangle = \sum_{\{h_i\}} \prod_{[ij]} \delta_{h_{ij}, h_i^{-1} h_j} \prod_{[ijk]} \frac{\nu(h_i, h_{ij}, h_{jk})}{\nu(h_i', h_{ij}', h_{jk}')} \,|\{g_i', h_{ij}'\}\rangle \,,}
\tag{5.121}
$$

where $h_{ij}' = (h_i')^{-1} h_j'$. When $g = e$, the symmetry action on a closed surface becomes

$$
\mathsf{D}_A \overline{\mathsf{D}}_A \,|\{g_i, h_{ij}\}\rangle = \delta_{\mathrm{hol}(\{h_{ij}\})} |H| \,|\{g_i, h_{ij}\}\rangle \,,
\tag{5.122}
$$

where $\delta_{\mathrm{hol}(\{h_{ij}\})}$ is one if the holonomy of the gauge field $\{h_{ij}\}$ is zero for any loop, and it is zero otherwise. We note that the scalar factor $\delta_{\mathrm{hol}(\{h_{ij}\})} |H|$ is the partition function of the 2d TFT with a spontaneously broken $H$ symmetry. The above operator (5.122) is an example of what is known as a condensation defect [161] or a theta defect [8, 9].

### 5.3.3   1-Form Symmetry

The 1-form symmetry operators of the gauged model are labeled by 1-endomorphisms of the unit object $A \in {}_A(2\mathsf{Vec}_G)_A$. These symmetry operators, together with topological point operators between them, form a braided fusion 1-category $\mathrm{End}_{A(2\mathsf{Vec}_G)_A}(A)$. Objects of $\mathrm{End}_{A(2\mathsf{Vec}_G)_A}(A)$ are $(A, A)$-bimodule endofunctors of $A$ that preserves the $G$-grading, and morphisms of $\mathrm{End}_{A(2\mathsf{Vec}_G)_A}(A)$ are $(A, A)$-bimodule natural transformations.

As we will see below, there is an equivalence of braided fusion 1-categories

$$
\mathrm{End}_{A(2\mathsf{Vec}_G)_A}(A) \cong \mathcal{Z}(A)_0,
\tag{5.123}
$$

where $\mathcal{Z}(A)_0$ is the full subcategory of the Drinfeld center $\mathcal{Z}(A)$ consisting of objects $(a, \Phi) \in \mathcal{Z}(A)$ with $a \in A_e$ having the trivial grading. Here, $\Phi$ denotes the half-braiding, which is a natural family of isomorphisms $\{\Phi_b : b \otimes a \to a \otimes b \mid b \in A\}$ that satisfies the usual coherence condition. The above equivalence (5.123) implies that the 1-form symmetry operators of the gauged model can be labeled by objects of $\mathcal{Z}(A)_0$. We will explicitly write down these symmetry operators on the lattice later in this subsection.

To show the equivalence (5.123), we first recall that the category $\mathrm{End}_{A(2\mathsf{Vec})_A}(A)$ of $(A, A)$-bimodule endofunctors of $A$ is equivalent to the Drinfeld center $\mathcal{Z}(A)$ of $A$ [162]. Namely, there is an equivalence of braided fusion 1-categories[72]

$$
\mathrm{End}_{A(2\mathsf{Vec})_A}(A) \cong \mathcal{Z}(A).
\tag{5.124}
$$

Under this equivalence, an $(A, A)$-bimodule functor $F \in \mathrm{End}_{A(2\mathsf{Vec})_A}(A)$ is mapped to an object $(F(\mathbb{1}_A), \Phi) \in \mathcal{Z}(A)$, where the half-braiding $\Phi$ is given by

$$
\Phi_a : a \otimes F(\mathbb{1}_A) \xrightarrow{s_{a,\mathbb{1}_A}^{-1}} F(a \otimes \mathbb{1}_A) \cong F(\mathbb{1}_A \otimes a) \xrightarrow{t_{\mathbb{1}_A, a}} F(\mathbb{1}_A) \otimes a.
\tag{5.125}
$$

---

[72]We note that $\mathrm{End}_{A(2\mathsf{Vec})_A}(A)$ is not the same as $\mathrm{End}_{A(2\mathsf{Vec}_G)_A}(A)$: the latter is a subcategory of the former.

Here, $s_{a,b} : F(a \otimes b) \to a \otimes F(b)$ and $t_{a,b} : F(a \otimes b) \to F(a) \otimes b$ are the natural isomorphisms associated to $F$. The (weak) inverse of this equivalence maps an object $(a, \Psi) \in \mathcal{Z}(A)$ to an $(A, A)$-bimodule endofunctor $F$ such that $F(b) = b \otimes a$ for all $b \in A$. The natural isomorphisms associated to this functor are given by

$$s_{b,b'} = \alpha_{b,b',a}, \qquad t_{b,b'} = \alpha^{-1}_{b,a,b'} \circ (\mathrm{id}_b \otimes \Psi_{b'}) \circ \alpha_{b,b',a}. \tag{5.126}$$

Now, when $F$ is an object of $\mathrm{End}_{A(2\mathsf{Vec}_G)_A}(A) \subset \mathrm{End}_{A(2\mathsf{Vec})_A}(A)$, the underlying object $F(\mathbb{1}_A)$ of $(F(\mathbb{1}_A), \Phi) \in \mathcal{Z}(A)$ has the trivial grading because $F$ preserves the $G$-grading. That is, $(F(\mathbb{1}_A), \Phi)$ is an object of the subcategory $\mathcal{Z}(A)_0$. Conversely, if $(a, \Psi)$ is an object of $\mathcal{Z}(A)_0$, its image under the equivalence (5.124) is an $(A, A)$-bimodule endofunctor $F$ such that $F(\mathbb{1}_A) \cong a \in A_e$. In particular, $F$ preserves the $G$-grading because $F(a_g) \cong a_g \otimes F(\mathbb{1}_A) \in A_g$ for any $a_g \in A_g$. In other words, $F$ is an object of $\mathrm{End}_{A(2\mathsf{Vec}_G)_A}(A)$. Thus, (5.124) gives an equivalece between the subcategories $\mathrm{End}_{A(2\mathsf{Vec}_G)_A}(A)$ and $\mathcal{Z}(A)_0$. This shows (5.123).

**Another Description.** We can also show that $\mathcal{Z}(A)_0$ is equivalent to the Müger centralizer of $\mathsf{Rep}(H)$ in $\mathcal{Z}(A)$, i.e., there is a braided equivalence

$$\mathcal{Z}(A)_0 \cong \mathsf{Rep}(H)'_{\mathcal{Z}(A)}. \tag{5.127}$$

Here, the Müger centralizer $\mathsf{Rep}(H)'_{\mathcal{Z}(A)}$ is the full subcategory of $\mathcal{Z}(A)$ consisting of objects $x \in \mathcal{Z}(A)$ satisfying $c_{y,x} \circ c_{x,y} = \mathrm{id}_{x \otimes y}$ for all objects $y \in \mathsf{Rep}(H) \subset \mathcal{Z}(A)$, where $c_{x,y} : x \otimes y \to y \otimes x$ denotes the braiding isomorphism of $\mathcal{Z}(A)$. Combining the two equivalences (5.123) and (5.127), we find

$$\mathrm{End}_{A(2\mathsf{Vec}_G)_A}(A) \cong \mathsf{Rep}(H)'_{\mathcal{Z}(A)}. \tag{5.128}$$

This shows that the 1-form symmetry of the gauged model is equivalently described by a braided fusion category $\mathsf{Rep}(H)'_{\mathcal{Z}(A)}$.

To show the equivalence (5.127), let us recall an explicit description of the subcategory $\mathsf{Rep}(H) \subset \mathcal{Z}(A)$ following [113, Section 3.2]. As a subcategory of $\mathcal{Z}(A)$, an object of $\mathsf{Rep}(H)$ is given by a pair $(V_\rho \otimes \mathbb{1}_A, \Phi^\rho)$ where $\rho$ is a representation of $H$, $V_\rho$ is the representation space of $\rho$, and $\Phi^\rho$ is the half-braiding defined by

$$\Phi^\rho_{a_h} = \mathrm{id}_{a_h} \otimes \rho(h) : a_h \otimes (V_\rho \otimes \mathbb{1}_A) \to a_h \otimes (V_\rho \otimes \mathbb{1}_A) \cong (V_\rho \otimes \mathbb{1}_A) \otimes a_h, \quad \forall a_h \in A_h. \tag{5.129}$$

The last isomorphism in the above equation is given by the left and right unitors $\mathbb{1}_A \otimes a_h \cong a_h \cong a_h \otimes \mathbb{1}_A$, which we can choose to be trivial without loss of generality. By definition, an object $(a, \Psi) \in \mathcal{Z}(A)$ is in the Müger centralizer of $\mathsf{Rep}(H)$ if and only if the double braiding

$$c_{(V_\rho \otimes \mathbb{1}_A, \Phi^\rho),(a,\Psi)} \circ c_{(a,\Psi),(V_\rho \otimes \mathbb{1}_A, \Phi^\rho)} = \Psi_{V_\rho \otimes \mathbb{1}_A} \circ \Phi^\rho_a \tag{5.130}$$

is trivial. We note that $\Psi_{V_\rho \otimes \mathbb{1}_A}$ is trivial for any $(a, \Psi) \in \mathcal{Z}(A)$ because $V_\rho \otimes \mathbb{1}_A$ consists only of (finitely many copies of) the unit object. Therefore, the double braiding (5.130) becomes trivial if and only if $\Phi_a^\rho$ is trivial, which is the case if and only if $a$ has the trivial grading. This shows the equivalence (5.127).

**Relation to Stacking and Gauging.** The 1-form symmetry of the gauged model can also be understood from the point of view of stacking and gauging. To see this, let us consider the gauged model obtained by stacking a 3d $H$-symmetric TFT $\mathfrak{T}_H$ on the original $G$-symmetric model and gauging the diagonal subgroup $H^{\text{diag}}$. As discussed in [1], the 1-form symmetry of this model is described by the Müger centralizer of $\mathsf{Rep}(H)$ in the $H$-equivariantization of the $H$-crossed braided fusion category $\mathcal{M}_H^\times$ that describes the anyons of $\mathfrak{T}_H$. In our context, $\mathcal{M}_H^\times$ is the relative center $\mathcal{Z}_{A_e}(A)$ of the $H$-graded fusion category $A$ and its $H$-equivariantization is given by $\mathcal{Z}(A)$ [113]. Therefore, the 1-form symmetry of the gauged model is described by the Müger centralizer of $\mathsf{Rep}(H)$ in $\mathcal{Z}(A)$, which agrees with (5.128).

**Symmetry Operators on the Lattice.** Let us write down the 1-form symmetry operators on the lattice. The symmetry operator labeled by $(a, \Phi) \in \mathcal{Z}(A)_0$ is denoted by $\widehat{D}_{(a,\Phi)}^\gamma$, where $\gamma$ is the path on which the symmetry operator is supported. The action of $\widehat{D}_{(a,\Phi)}^\gamma$ on a state is given by the following diagrammatic equation:

$$
\widehat{D}_{(a,\Phi)}^\gamma \left| \begin{array}{c} M^{g_l} \\ a_{il} \\ M^{g_j}_{a_{ij}} \quad \quad M^{g_k} \\ M^{g_i} \quad a_{ik} \end{array} \right\rangle = \left| \begin{array}{c} M^{g_j} \quad \overset{M^{g_l}}{\underset{a_{il}}{}} \quad M^{g_k} \\ a \quad a_{ij} \quad \quad a_{ik} \quad a \\ M^{g_j} \, a_{ij} \, \Phi_{a_{ij}}^{-1} \quad \Phi_{a_{ik}} \, a_{ik} \, M^{g_k} \\ M^{g_i} \\ a \\ M^{g_i} \end{array} \right\rangle . \tag{5.131}
$$

We use the half-braiding isomorphism $\Phi$ wherever the symmetry operator intersects the edges of the honeycomb lattice. The right-hand side of (5.131) is evaluated by fusing the symmetry line into the edges. Roughly speaking, the above symmetry operator changes the dynamical variable $a_{ij}$ into $a \otimes a_{ij}$ on every edge where it acts. In particular, this symmetry action does not change the grading of the dynamical variables on the edges because $a$ has the trivial grading. This is consistent with our description of the state space: that is, the grading of $a_{ij}$ is determined by the vertical surface below edge $[ij]$, and this surface does not change under the action of the 1-form symmetry operators because they act on states from above.

In the case of the minimal gauging $A = \mathsf{Vec}_H^\nu$, the braided fusion category $\mathcal{Z}(A)_0$ is

equivalent to $\mathsf{Rep}(H)$, i.e., there is a braided equivalence

$$\mathcal{Z}(A)_0 \cong \mathsf{Rep}(H). \tag{5.132}$$

The symmetry operator labeled by $\rho \in \mathsf{Rep}(H)$ is simply written as $\widehat{D}_\rho^\gamma := \widehat{D}_{(V_\rho \otimes \mathbb{1}_A, \Phi^\rho)}^\gamma$. Using the half-braiding (5.129), we can write down the action of $\widehat{D}_\rho^\gamma$ for a closed loop $\gamma$ as

$$\widehat{D}_\rho^\gamma \, |\{g_i, h_{ij}\}\rangle = \mathrm{tr}_{V_\rho} \left( \rho \left( \prod_{[ij] \in E_\gamma} h_{ij}^{s_{ij}} \right) \right) |g_i, h_{ij}\rangle, \tag{5.133}$$

where $s_{ij} = \pm 1$ is the sign defined by (5.96). Namely, $\widehat{D}_\rho^\gamma$ is the Wilson line operator.

### 5.3.4 Line Operators on Surface Operators

In this subsection, we consider line operators on the surface operator labeled by $A \square D_2^g \square A \in {}_A(2\mathsf{Vec}_G)_A$. In particular, we describe the general construction of such line operators and compute their fusion rules and the $F$-symbols. We will also briefly discuss the corresponding symmetry operators on the lattice. Given that every simple object of ${}_A(2\mathsf{Vec}_G)_A$ is connected to a simple object of the form $A \square D_2^g \square A$, one can obtain any topological surface by gauging some algebra object consisting of line operators on the surface labeled by $A \square D_2^g \square A$.

In general, topological lines on a topological surface labeled by $X \in {}_A(2\mathsf{Vec}_G)_A$ are labeled by 1-endomorphisms of $X$. Such line operators form a multifusion category $\mathrm{End}_{{}_A(2\mathsf{Vec}_G)_A}(X)$, which becomes a fusion category when $X$ is simple. In particular, $\mathrm{End}_{{}_A(2\mathsf{Vec}_G)_A}(A \square D_2^g \square A)$ is fusion because $A \square D_2^g \square A$ is a simple object of ${}_A(2\mathsf{Vec}_G)_A$.

To describe the fusion category structure on $\mathrm{End}_{{}_A(2\mathsf{Vec}_G)_A}(A \square D_2^g \square A)$, let us first recall that there is an equivalence of 1-categories (cf. (5.26))

$$\mathrm{End}_{{}_A(2\mathsf{Vec}_G)_A}(A \square D_2^g \square A) \cong \mathrm{Hom}_{2\mathsf{Vec}_G}(D_2^g, A \square D_2^g \square A). \tag{5.134}$$

Using the direct sum decomposition (5.3) of $A$ into simples of $2\mathsf{Vec}_G$, we find that the above equivalence reduces to[73]

$$\mathrm{End}_{{}_A(2\mathsf{Vec}_G)_A}(A \square D_2^g \square A) \cong \bigoplus_{h \in H \cap gHg^{-1}} \mathsf{Vec}^{\oplus \mathrm{rank}(A_h)\mathrm{rank}(A_{g^{-1}hg})}. \tag{5.135}$$

The above equivalence implies that simple objects of $\mathrm{End}_{{}_A(2\mathsf{Vec}_G)_A}(A \square D_2^g \square A)$ are in one-to-one correspondence with pairs of simple objects of $A_h$ and $A_{g^{-1}hg}$ for any $h \in H \cap gHg^{-1}$. The simple object corresponding to the pair of $a \in A_h$ and $a' \in A_{g^{-1}hg}$ is denoted by $(a, a')$.

Pictorially, the simple object $(a, a') \in \mathrm{End}_{{}_A(2\mathsf{Vec}_G)_A}(A \square D_2^g \square A)$ can be represented as shown in figure 16. The top surface in figure 16 is labeled by a left $A$-module $A \in {}_A(2\mathsf{Vec}_G)$,

---

[73]To obtain (5.135), we used the fact that the rank of $A_h$ is equal to the rank of $A_{h^{-1}}$.

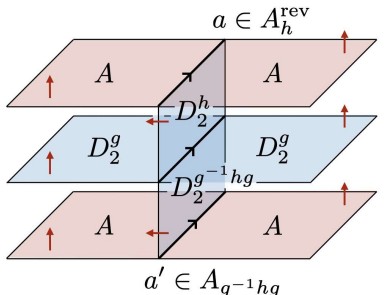

Figure 16: A topological line on a surface $A\Box D_2^g\Box A$ is obtained by inserting a surface $D_2^h$ between the top and bottom surfaces labeled by $A$ so that it intersects the middle surface $D_2^g$ perpendicularly. The topological line at the intersection is the identity 1-morphism.

while the bottom surface is labeled by a right $A$-module $A \in (2\mathsf{Vec}_G)_A$. Accordingly, the topological lines on the top and bottom surfaces are labeled by a 1-morphism of $_A(2\mathsf{Vec}_G)$ and a 1-morphism of $(2\mathsf{Vec}_G)_A$, respectively. Equivalently, these line operators are labeled by objects of $A_h^{\mathrm{rev}}$ and $A_{g^{-1}hg}$ due to the equivalences

$$\mathrm{Hom}_{A(2\mathsf{Vec}_G)}(A, A \triangleleft D_2^h) \cong A_h^{\mathrm{rev}},$$
$$\mathrm{Hom}_{(2\mathsf{Vec}_G)_A}(D_2^{g^{-1}hg} \triangleright A, A) \cong A_{g^{-1}hg}. \tag{5.136}$$

Here, the first line is an equivalence of $(A_e^{\mathrm{rev}}, A_e^{\mathrm{rev}})$-bimodule categories, while the second line is an equivalence of $(A_e, A_e)$-bimodule categories. Since objects of $A_h^{\mathrm{rev}}$ are objects of $A_h$, the line operator shown in figure 16 is indeed labeled by a pair of objects of $A_h$ and $A_{g^{-1}hg}$.

The tensor product of objects in $\mathrm{End}_{A(2\mathsf{Vec}_G)_A}(A\Box D_2^g\Box A)$ is given by the fusion of the line operators defined by figure 16. Specifically, the fusion rule of $\mathrm{End}_{A(2\mathsf{Vec}_G)_A}(A\Box D_2^g\Box A)$ can be computed as

$$(a_{ij}, a'_{ij}) \otimes (a_{jk}, a'_{jk}) \cong (a_{ij} \otimes a_{jk}, a'_{ij} \otimes a'_{jk}) \cong \bigoplus_{a_{ik}, a'_{ik}} N^{a_{ik}}_{a_{ij}, a_{jk}} N^{a'_{ik}}_{a'_{ij}, a'_{jk}} (a_{ik}, a'_{ik}), \tag{5.137}$$

where $N^{a_{ik}}_{a_{ij}, a_{jk}}$ and $N^{a'_{ik}}_{a'_{ij}, a'_{jk}}$ are the fusion coefficients of $A$. Similarly, the $F$-symbols of $\mathrm{End}_{A(2\mathsf{Vec}_G)_A}(A\Box D_2^g\Box A)$ can also be computed as

$$\vcenter{\hbox{\includegraphics{lhs}}} = \sum_{a_{jl}, a'_{jl}} \sum_{\alpha_{ijl}, \alpha'_{ijl}} \sum_{\alpha_{jkl}, \alpha'_{jkl}} \overline{F}^{a;\alpha}_{ijkl} F^{a';\alpha'}_{ijkl} \vcenter{\hbox{\includegraphics{rhs}}}, \tag{5.138}$$

where $(a, a')_{ij} := (a_{ij}, a'_{ij})$ and $(\overline{\alpha}^{\mathrm{rev}}, \overline{\alpha}')_{ijk} := (\overline{\alpha}^{\mathrm{rev}}_{ijk}, \overline{\alpha}'_{ijk})$.

In the case of the minimal gauging $A = \mathsf{Vec}_H^\nu$, simple objects of $\mathrm{End}_{A(2\mathsf{Vec}_G)_A}(A\Box D_2^g\Box A)$ are labeled by elements of $H \cap gHg^{-1}$ because $a \in A_h^{\mathrm{rev}}$ and $a' \in A_{g^{-1}hg}$ are unique in this case.

The simple object labeled by $h \in H \cap gHg^{-1}$ is denoted by $h$ by a slight abuse of notation. The fusion rule of these line operators is given by the multiplication of $H \cap gHg^{-1}$, i.e.,

$$h_{ij} \otimes h_{jk} \cong h_{ij}h_{jk}, \quad \forall h_{ij}, h_{jk} \in H \cap gHg^{-1}. \tag{5.139}$$

Furthermore, the $F$-symbol (5.138) reduces to

$$= \frac{\nu(g^{-1}h_{ij}g, g^{-1}h_{jk}g, g^{-1}h_{kl}g)}{\nu(h_{ij}, h_{jk}, h_{kl})} \tag{5.140}$$

where $h_{ik} := h_{ij}h_{jk}$. Thus, we find a monoidal equivalence

$$\mathrm{End}_{A(2\mathsf{Vec}_G)_A}(A\square D_2^g \square A) \cong \mathsf{Vec}_{H \cap gHg^{-1}}^{\nu_g}, \tag{5.141}$$

where the 3-cocycle $\nu_g \in Z^3(H \cap gHg^{-1}, \mathrm{U}(1))$ is defined by

$$\nu_g(h_{ij}, h_{jk}, h_{kl}) := \frac{\nu(g^{-1}h_{ij}g, g^{-1}h_{jk}g, g^{-1}h_{kl}g)}{\nu(h_{ij}, h_{jk}, h_{kl})} \tag{5.142}$$

for $h_{ij}, h_{jk}, h_{kl} \in H \cap gHg^{-1}$. The monoidal equivalence (5.141) implies that the 2-category of topological surfaces in the connected component of $A\square D_2^g \square A$ is equivalent to $\mathsf{Mod}(\mathsf{Vec}_{H \cap gHg^{-1}}^{\nu_g})$. Hence, we obtain an equivalence of semisimple 2-categories

$$_{\mathsf{Vec}_H^\nu}(2\mathsf{Vec}_G)_{\mathsf{Vec}_H^\nu} \cong \bigoplus_{g \in S_{H\backslash G/H}} \mathsf{Mod}(\mathsf{Vec}_{H \cap gHg^{-1}}^{\nu_g}), \tag{5.143}$$

where $S_{H\backslash G/H}$ denotes the set of representatives of double $H$-cosets in $G$. This agrees with the result in [13, 115].[74]

**Symmetry Operators on the Lattice.** Let us see how the line operators on a surface operator $A\square D_2^g \square A$ act on states in the lattice model. For simplicity, we focus on the case of the minimal gauging $A = \mathsf{Vec}_H^\nu$. In this case, the line operators on $A\square D_2^g \square A$ are labeled by elements of $H \cap gHg^{-1}$. figure 16 suggests that the line operator labeled by $h \in H \cap gHg^{-1}$ is given by the composition $\mathsf{D}_A^h U_g (\mathsf{D}_A^{g^{-1}hg})^\dagger$, where $\mathsf{D}_A^h$ is the gauging operator for the $h$-twisted sector (cf. section 5.2.3) and $U_g$ is the symmetry operator of the original model. Using the explicit form (5.106) of $\mathsf{D}_A^h$, we can compute the action of $\mathsf{D}_A^h U_g (\mathsf{D}_A^{g^{-1}hg})^\dagger$ as follows:

$$\mathsf{D}_A^h U_g (\mathsf{D}_A^{g^{-1}hg})^\dagger |\{g_i, h_{ij}\}\rangle = \sum_{\{h_i \in H\}} \prod_{[ij] \in E\backslash\tilde{\gamma}} \delta_{h_{ij}, h_i^{-1}h_j} \prod_{[ij] \in \tilde{\gamma}} \delta_{h_{ij}, h_i^{-1}g^{-1}hgh_j}$$
$$\prod_{[ijk] \in V} \left( \frac{\nu(h_i, h_{ij}, h_{jk})}{\nu(h_i', \tilde{h}_{ij}', \tilde{h}_{jk}')} \right)^{s_{ijk}} \prod_{[ij] \in E_\gamma} \frac{\nu(g^{-1}hg, h_i, h_{ij})}{\nu(h, h_i', \tilde{h}_{ij}')} |\{g_i', \tilde{h}_{ij}'\}\rangle. \tag{5.144}$$

---

[74]Refs. [13,115] studied the 2-categories of the form $_{\mathsf{Vec}_H^\nu}(2\mathsf{Vec}_G^\omega)_{\mathsf{Vec}_K^\lambda}$. When $K = H$, $\lambda = \nu$, and $\omega$ is trivial, their results reduce to (5.143).

Here, $\gamma$ and $\tilde{\gamma}$ are defined as in (5.102), $h_i'$ and $g_i'$ are the unique elements of $H$ and $S_{H \backslash G}$ that satisfy $gh_ig_i = h_i'g_i'$, and $\tilde{h}_{ij}'$ is defined by

$$\tilde{h}_{ij}' = \begin{cases} (h_i')^{-1}h_j' & \text{for } [ij] \notin \tilde{\gamma}, \\ (h_i')^{-1}hh_j' & \text{for } [ij] \in \tilde{\gamma}. \end{cases} \tag{5.145}$$

We note that (5.144) reduces to (5.121) when $h = e$.

# 6   Generalized Gauging of $2\mathsf{Vec}_G^\omega$ Symmetry

In this section, we discuss the generalized gauging of an anomalous finite group symmetry $G$ in the fusion surface model. The content of this section will be parallel to that of section 5, and hence, our exposition will be brief.

The input fusion 2-category $\mathcal{C}$ of the model is $2\mathsf{Vec}_G^\omega$, which is the 2-category of finite semisimple $G$-graded 1-categories with the 10-j symbols twisted by a 4-cocycle $\omega \in Z^4(G, \mathrm{U}(1))$. Simple objects of $2\mathsf{Vec}_G^\omega$ are labeled by elements of $G$ and are denoted by $\{D_2^g \mid g \in G\}$. The Hom categories are given by

$$\mathrm{Hom}_{2\mathsf{Vec}_G^\omega}(D_2^g \square D_2^h, D_2^k) \cong \mathrm{Hom}_{2\mathsf{Vec}_G^\omega}(D_2^k, D_2^g \square D_2^h) \cong \delta_{gh,k}\mathsf{Vec}. \tag{6.1}$$

The unique simple 1-morphism from $D_2^g \square D_2^h$ to $D_2^{gh}$ is denoted by $1_{g \cdot h}$, while the unique simple 1-morphism from $D_2^{gh}$ to $D_2^g \square D_2^h$ is denoted by $\overline{1}_{g \cdot h}$. The 10-j symbols of $2\mathsf{Vec}_G^\omega$ are given by

$$\cdots = \omega(g_{ij}, g_{jk}, g_{kl}, g_{lm}) \cdots , \tag{6.2}$$

$$\cdots = \omega(g_{ij}, g_{jk}, g_{kl}, g_{lm})^{-1} \cdots , \tag{6.3}$$

where the white dots denote the basis 2-morphisms. Without loss of generality, we suppose that the 4-cocycle $\omega$ is normalized, i.e., $\omega(g, h, k, l) = 1$ if either of $g$, $h$, $k$, and $l$ is the unit element.

## 6.1 Preliminaries

Let us first spell out the details of separable algebras in $2\mathsf{Vec}_G^\omega$ and the corresponding module 2-categories over $2\mathsf{Vec}_G^\omega$. We will use these data later in this section to describe the generalized gauging of $2\mathsf{Vec}_G^\omega$ symmetry on the lattice. The same data will also be used in section 7 to obtain the lattice models of gapped phases.

### 6.1.1 Separable Algebras in $2\mathsf{Vec}_G^\omega$

A separable algebra $A \in 2\mathsf{Vec}_G^\omega$ is a $G$-graded $\omega$-twisted multifusion category, which is a $G$-graded category

$$A = \bigoplus_{g \in G} A_g \tag{6.4}$$

equipped with the tensor product $\otimes : A \times A \to A$ whose associator $\alpha_{a,b,c} : (a \otimes b) \otimes c \to a \otimes (b \otimes c)$ satisfies the twisted pentagon equation [115]

$$
\begin{array}{ccc}
((a \otimes b) \otimes c) \otimes d & \xrightarrow{\alpha_{a \otimes b, c, d}} (a \otimes b) \otimes (c \otimes d) \xrightarrow{\alpha_{a,b,c \otimes d}} a \otimes (b \otimes (c \otimes d)) \\
\omega(g,h,k,l) \Big\downarrow & \Big\uparrow \mathrm{id}_a \otimes \alpha_{b,c,d} \\
((a \otimes b) \otimes c) \otimes d & \xrightarrow[\alpha_{a,b,c} \otimes \mathrm{id}_d]{} (a \otimes (b \otimes c)) \otimes d \xrightarrow[\alpha_{a, b \otimes c, d}]{} a \otimes ((b \otimes c) \otimes d)
\end{array} \tag{6.5}
$$

where $a \in A_g$, $b \in A_h$, $c \in A_k$, and $d \in A_l$. The multiplication 1-morphism $m$ is given by the tensor product $\otimes$, while the associativity 2-isomorphism $\mu$ is given by the associator $\alpha$. The consistency condition (3.8) on $\mu$ follows from the twisted pentagon equation (6.5). We note that the trivially graded component $A_e$ is an ordinary multifusion category because the twisted pentagon equation for $A_e$ reduces to the ordinary pentagon equation due to the normalization of $\omega$.

Throughout this section, we only consider a separable algebra $A \in 2\mathsf{Vec}_G^\omega$ whose trivially graded component $A_e$ is fusion. Furthermore, we assume that the associator $\alpha_{a,b,c}$ is unitary, meaning that the corresponding $F$-symbols become unitary matrices. We call such $A$ a $G$-graded $\omega$-twisted unitary fusion category.

In what follows, we will describe the separable algebra structure on $A$ in more detail. It turns out that the data of a separable algebra in $2\mathsf{Vec}_G^\omega$ are almost the same as those in $2\mathsf{Vec}_G$, which we have already discussed in section 5.1.1.

**Algebra Structure** $(A, m, \mu)$**.** The underlying object of an algebra $A \in 2\mathsf{Vec}_G^\omega$ is decomposed as

$$A = \bigoplus_{g \in G} \bigoplus_{a_g \in A_g} D_2^g[a_g], \tag{6.6}$$

where $D_2^g[a_g]$ denotes the simple object $D_2^g \in 2\mathsf{Vec}_G^\omega$ associated to a simple object $a_g \in A_g$. The summation on the right-hand side is taken over all simple objects of $A$.

The underlying 1-morphism of the multiplication 1-morphism $m : A \square A \to A$ is given component-wise as

$$D_2^{g_{ik}}[a_{ik}]\ \begin{array}{c} m \uparrow \\ \end{array}\ \begin{array}{c} D_2^{g_{jk}}[a_{jk}] \\ D_2^{g_{ij}}[a_{ij}] \end{array} = \bigoplus_{\alpha_{ijk} \in \mathrm{Hom}_A(a_{ij} \otimes a_{jk}, a_{ik})} 1_{g_{ij} \cdot g_{jk}}[\alpha_{ijk}], \tag{6.7}$$

where $1_{g_{ij} \cdot g_{jk}}[\alpha_{ijk}]$ denotes the simple 1-morphism $1_{g_{ij} \cdot g_{jk}}$ associated to a basis morphism $\alpha_{ijk} \in \mathrm{Hom}_A(a_{ij} \otimes a_{jk}, a_{ik})$. The summation on the right-hand side is taken over all basis morphisms. The basis morphisms are normalized so that the following equation holds:

$$\begin{aligned} \left| \begin{array}{cc} \uparrow & \uparrow \\ a_{ij} & a_{jk} \end{array} \right| &= \sum_{a_{ik}} \sum_{\alpha_{ijk}} \sqrt{\frac{d_{ik}^a}{d_{ij}^a d_{jk}^a}}\ \begin{array}{c} a_{ij} \searrow \quad \nearrow a_{jk} \\ a_{ik} \uparrow \quad \overline{\alpha}_{ijk} \\ \quad \alpha_{ijk} \\ a_{ij} \quad a_{jk} \end{array}, \\[2em] a_{ij} \begin{array}{c} a_{ik}' \\ \uparrow \alpha_{ijk} \\ \bullet \\ \uparrow \overline{\alpha}_{ijk} \\ a_{ik} \end{array} a_{jk} &= \delta_{a_{ik}, a_{ik}'} \delta_{\alpha_{ijk}, \alpha_{ijk}'} \sqrt{\frac{d_{ij}^a d_{jk}^a}{d_{ik}^a}}\ \begin{array}{c} \uparrow \\ a_{ik} \end{array}. \end{aligned} \tag{6.8}$$

Here, $d_{ij}^a$ denotes the Frobenius-Perron dimension of $a_{ij}$.

The underlying 2-morphisms of the associativity 2-isomorphism $\mu$ and its inverse $\mu^{-1}$ are given component-wise as

$$\begin{array}{c} a_{jl} \\ \alpha_{ijl} \diagup \mu \quad \alpha_{jkl} \\ \alpha_{ikl} \quad \alpha_{ijk} \quad a_{kl} \\ a_{il} \quad a_{ik} \quad a_{jk} \\ \quad a_{ij} \end{array} = F_{ijkl}^{a;\alpha} \begin{array}{c} 1_{g_{ij} \cdot g_{jl}} \quad 1_{g_{jk} \cdot g_{kl}} \\ \\ 1_{g_{ik} \cdot g_{kl}} \quad 1_{g_{ij} \cdot g_{jk}} \end{array}, \tag{6.9}$$

$$\begin{array}{c} a_{jl} \\ \alpha_{ikl} \quad \alpha_{ijk} \\ \mu^{-1} \bullet \\ \alpha_{ijl} \quad a_{ik} \quad \alpha_{jkl} \quad a_{kl} \\ a_{il} \quad a_{jk} \\ \quad a_{ij} \end{array} = \overline{F}_{ijkl}^{a;\alpha} \begin{array}{c} 1_{g_{ik} \cdot g_{kl}} \quad 1_{g_{ij} \cdot g_{jk}} \\ \\ 1_{g_{ij} \cdot g_{jl}} \quad 1_{g_{jk} \cdot g_{kl}} \end{array}, \tag{6.10}$$

where $F^{a;\alpha}_{ijkl}$ and $\overline{F}^{a;\alpha}_{ijkl}$ denote the $F$-symbols of $A$ defined by

$$
\begin{aligned}
&\text{(diagram)} = \sum_{a_{jl}} \sum_{\alpha_{ijl},\alpha_{jkl}} F^{a;\alpha}_{ijkl}\ \text{(diagram)},\\
&\text{(diagram)} = \sum_{a_{ik}} \sum_{\alpha_{ijk},\alpha_{ikl}} \overline{F}^{a;\alpha}_{ijkl}\ \text{(diagram)}.
\end{aligned}
\tag{6.11}
$$

We note that the $F$-symbols are unitary

$$
\overline{F}^{a;\alpha}_{ijkl} = (F^{a;\alpha}_{ijkl})^*
\tag{6.12}
$$

because we assumed that the associator of $A$ is unitary. The twisted pentagon equation (6.5) implies

$$
\sum_{\alpha_{ikm}} F^{a;\alpha}_{ijkm} F^{a;\alpha}_{iklm} = \omega^g_{ijklm} \sum_{a_{jl}} \sum_{\alpha_{ijl}} \sum_{\alpha_{jkl}} \sum_{\alpha_{jlm}} F^{a;\alpha}_{ijkl} F^{a;\alpha}_{ijlm} F^{a;\alpha}_{jklm},
\tag{6.13}
$$

where $\omega^g_{ijklm} := \omega(g_{ij}, g_{jk}, g_{kl}, g_{lm})$.[75]

**Rigid Algebra Structure** $(m^*, \eta, \epsilon)$. The underlying 1-morphism of the dual 1-morphism $m^* : A \to A\square A$ is given component-wise as

$$
\text{(diagram)} = \bigoplus_{\overline{\alpha}_{ijk}\in\mathrm{Hom}_A(a_{ik},a_{ij}\otimes a_{jk})} \overline{1}_{g_{ij}\cdot g_{jk}}[\overline{\alpha}_{ijk}],
\tag{6.14}
$$

where $\overline{1}_{g_{ij}\cdot g_{jk}}[\overline{\alpha}_{ijk}]$ is the simple 1-morphism $\overline{1}_{g_{ij}\cdot g_{jk}}$ assoiciated to a basis morphism $\overline{\alpha}_{ijk} \in \mathrm{Hom}_A(a_{ik}, a_{ij}\otimes a_{jk})$. The summation on the right-hand side is taken over all basis morphisms.

The underlying 2-morphisms of the unit $\eta$ and the counit $\epsilon$ for the dual pair $(m, m^*)$ are given component-wise as

$$
\text{(diagram)} = \delta_{\alpha_{ijk},\alpha'_{ijk}} \sqrt{\frac{d^a_{ik}}{d^a_{ij} d^a_{jk}}}\ \text{(diagram)},
\tag{6.15}
$$

$$
\text{(diagram)} = \delta_{\alpha_{ijk},\alpha'_{ijk}} \sqrt{\frac{d^a_{ij} d^a_{jk}}{d^a_{ik}}}\ \text{(diagram)}.
\tag{6.16}
$$

---

[75]The twisted pentagon equation (6.13) appeared in [71] in the context of anomalous SET phases.

By using the above $\eta$ and $\epsilon$, one can define $\psi_r$, $\psi_l$, and $\mu^*$ as in (3.10), (3.11), and (3.15). Assuming that the $F$-symbols have the tetrahedral symmetry (5.13), we find

$$= \overline{F}^{a;\alpha}_{ijlk}. \tag{6.17}$$

$$= F^{a;\alpha}_{ijlk}, \tag{6.18}$$

$$= F^{a;\alpha}_{ijkl}. \tag{6.19}$$

**Separable Algebra Structure $\gamma$.** The underlying 2-morphism of $\gamma$ satisfying the separability condition (3.16) is given component-wise as

$$= \delta_{\alpha_{ijk},\alpha'_{ijk}} \frac{1}{\mathcal{D}_A} \sqrt{\frac{d^a_{ij} d^a_{jk}}{d^a_{ik}}} \tag{6.20}$$

where $\mathcal{D}_A = \sum_{a \in A} \dim(a)^2$ is the total dimension of $A$.

### 6.1.2 Module 2-Categories over $2\mathsf{Vec}^{\omega}_G$

Let us describe the basic data of a $2\mathsf{Vec}^{\omega}_G$-module 2-category $_A(2\mathsf{Vec}^{\omega}_G)$, which generalizes the $2\mathsf{Vec}_G$-module 2-category discussed in section 5.1.2. We will see that several data described in section 5.1.2 will be modified by appropriate phase factors when the twist $\omega$ is non-trivial. In what follows, we suppose that $A$ is faithfully graded by $H \subset G$.

**Objects.** An object $M \in {}_A(2\mathsf{Vec}^{\omega}_G)$ is a finite semisimple $G$-graded category

$$M = \bigoplus_{g \in G} M_g \tag{6.21}$$

equipped with a left $A$-action $\triangleright : A \times M \to M$ together with a natural isomorphism

$$l_{a,b,m} : (a \otimes b) \triangleright m \to a \triangleright (b \triangleright m), \qquad \forall a, b \in A, \quad \forall m \in M, \tag{6.22}$$

which makes the following diagram commute [115]:

$$
\begin{array}{ccc}
((a \otimes b) \otimes c) \rhd m \xrightarrow{l_{a \otimes b, c, m}} (a \otimes b) \rhd (c \rhd m) \xrightarrow{l_{a,b,c \rhd m}} a \rhd (b \rhd (c \rhd m)) \\
\omega(g,h,k,l) \downarrow \qquad\qquad\qquad\qquad\qquad\qquad\qquad\qquad\qquad \uparrow \mathrm{id}_a \rhd l_{b,c,m} \\
((a \otimes b) \otimes c) \rhd m \xrightarrow[\alpha_{a,b,c} \rhd \mathrm{id}_m]{} (a \otimes (b \otimes c)) \rhd m \xrightarrow[l_{a,b \otimes c, m}]{} a \rhd ((b \otimes c) \rhd m)
\end{array}
\tag{6.23}
$$

where $a \in A_g$, $b \in A_h$, $c \in A_k$, and $m \in M_l$. The connected components of $_A(2\mathsf{Vec}_G^\omega)$ are in one-to-one correspondence with right $H$-cosets in $G$. The representative of each connected component can be chosen to be

$$
M^g := A \square D_2^g,
\tag{6.24}
$$

where $g \in S_{H \backslash G}$ is the representative of the right $H$-coset $Hg$. We recall that $S_{H \backslash G}$ denotes the set of representatives of right $H$-cosets in $G$.

**1- and 2-Morphisms.** A 1-morphism $F : M \to N$ between $M, N \in {}_A(2\mathsf{Vec}_G^\omega)$ is a grading preserving functor equipped with a natural isomorphism

$$
\xi_{a,m} : F(a \rhd m) \to a \rhd F(m), \qquad \forall a \in A, \quad \forall m \in M,
\tag{6.25}
$$

which satisfies the ordinary coherence condition (5.31) [115]. Similarly, a 2-morphism $\eta : F \Rightarrow G$ between $F, G \in \mathrm{Hom}_{A(2\mathsf{Vec}_G^\omega)}(M, N)$ is a natural transformation that satisfies the ordinary coherence condition (5.32) [115].

The endomorphism 1-category $\mathrm{End}_{A(2\mathsf{Vec}_G^\omega)}(M^g)$ is monoidally equivalent to

$$
\mathrm{End}_{A(2\mathsf{Vec}_G^\omega)}(M^g) \cong A_e^{\mathrm{op}} \cong A_e^{\mathrm{rev}}.
\tag{6.26}
$$

More generally, we have the equivalences of $(A_e^{\mathrm{op}}, A_e^{\mathrm{op}})$-bimodule categories

$$
\mathrm{Hom}_{A(2\mathsf{Vec}_G)}(M^{g_i} \lhd D_2^{g_{ij}^h}, M^{g_j}) \cong A_{h_{ij}}^{\mathrm{op}},
\tag{6.27}
$$

$$
\mathrm{Hom}_{A(2\mathsf{Vec}_G)}(M^{g_j}, M^{g_i} \lhd D_2^{g_{ij}^h}) \cong A_{h_{ij}^{-1}}^{\mathrm{op}},
\tag{6.28}
$$

or equivalently, we have the equivalences of $(A_e^{\mathrm{rev}}, A_e^{\mathrm{rev}})$-bimodule categories

$$
\mathrm{Hom}_{A(2\mathsf{Vec}_G)}(M^{g_i} \lhd D_2^{g_{ij}^h}, M^{g_j}) \cong A_{h_{ij}^{-1}}^{\mathrm{rev}},
\tag{6.29}
$$

$$
\mathrm{Hom}_{A(2\mathsf{Vec}_G)}(M^{g_j}, M^{g_i} \lhd D_2^{g_{ij}^h}) \cong A_{h_{ij}}^{\mathrm{rev}},
\tag{6.30}
$$

where $g_{ij}^h := g_i^{-1} h_{ij} g_j$ for $g_i, g_j \in S_{H \backslash G}$ and $h_{ij} \in H$. The above equivalences imply that 1-morphisms and 2-morphisms of $_A(2\mathsf{Vec}_G^\omega)$ are labeled by objects and morphisms of $A^{\mathrm{op}}$ or $A^{\mathrm{rev}}$ as shown in figures 12 and 13.

**Module 10-j Symbols.** The module 10-j symbols of $_A(2\mathsf{Vec}_G^\omega)$ are given by

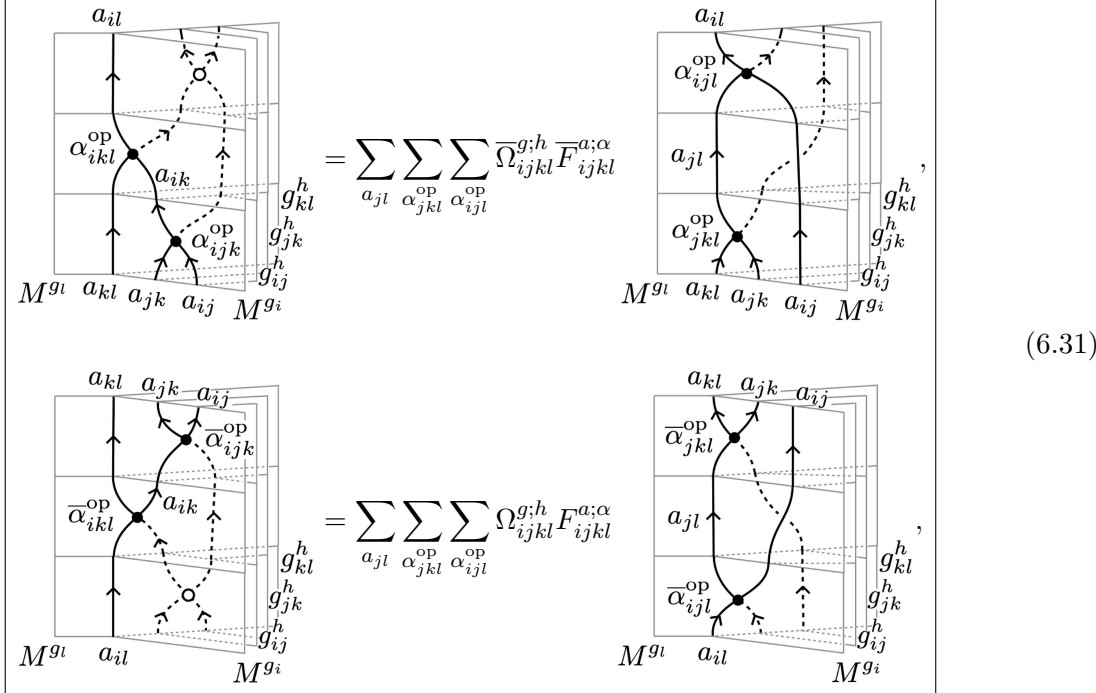

$$\tag{6.31}$$

where $\Omega_{ijkl}^{g;h}$ is defined by[76]

$$\boxed{\Omega_{ijkl}^{g;h} := \frac{\omega(h_{ij}, h_{jk}, h_{kl}, g_l)\omega(h_{ij}, g_j, g_{jk}^h, g_{kl}^h)}{\omega(h_{ij}, h_{jk}, g_k, g_{kl}^h)\omega(g_i, g_{ij}^h, g_{jk}^h, g_{kl}^h)}.} \tag{6.32}$$

The cocycle condition on $\omega$ implies that $\Omega_{ijkl}^{g;h}$ satisfies

$$\frac{\Omega_{jklm}^{g;h}\Omega_{ijlm}^{g;h}\Omega_{ijkl}^{g;h}}{\Omega_{iklm}^{g;h}\Omega_{ijkm}^{g;h}} = \frac{\omega(h_{ij}, h_{jk}, h_{kl}, h_{lm})}{\omega(g_{ij}^h, g_{jk}^h, g_{kl}^h, g_{lm}^h)}. \tag{6.33}$$

This, in turn, implies that the module 10-j symbols in (5.42) obey the correct consistency condition (i.e., the commutativity of the Stasheff polytope) due to the twisted pentagon equation

---

[76]One can easily guess the phase factor (6.32) by generalizing the module $F$-symbols of a $\mathsf{Vec}_G$-module category $_A(\mathsf{Vec}_G)$, where $A$ is a $G$-graded vector space faithfully graded by $H$. One can then show by direct computation that $\Omega_{ijkl}^{g;h}F_{ijkl}^{a;\alpha}$ satisfies the correct consistency condition of the module 10-j symbols.

for the $F$-symbols. Similarly, we also have

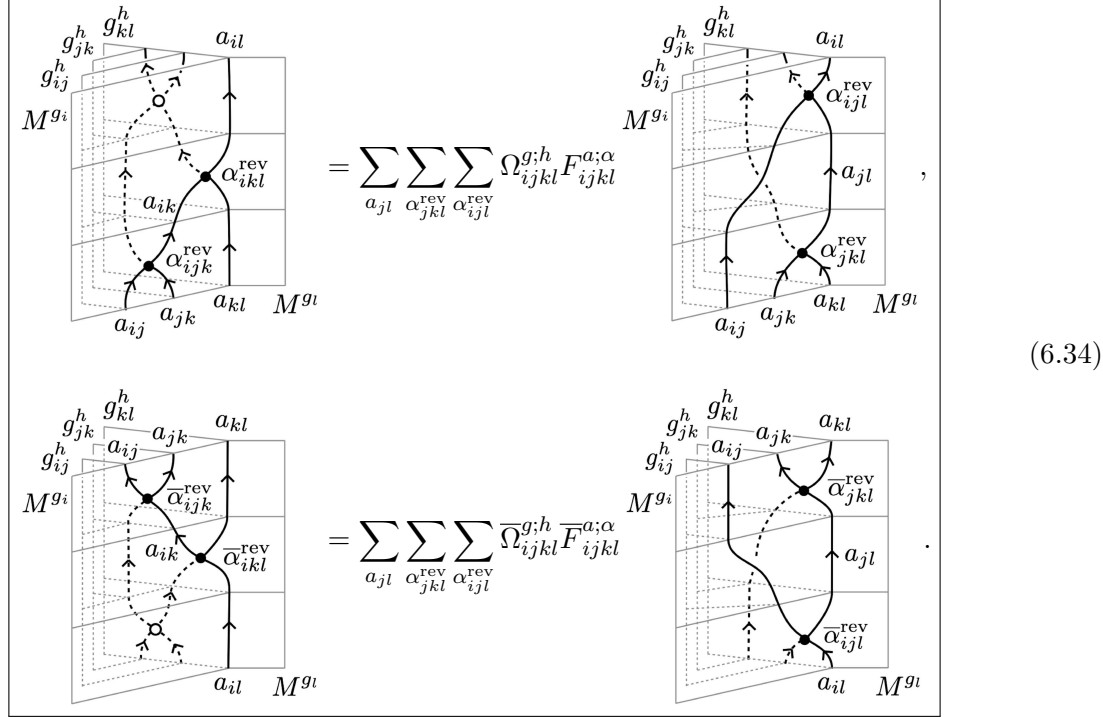

$$(6.34)$$

**Underlying Objects and Morphisms.** The underlying object of $M^g \in {}_A(2\mathsf{Vec}_G^\omega)$ is given by

$$M^g = \bigoplus_{h \in H} (D_2^{hg})^{\oplus \mathrm{rank}(A_h)} = \bigoplus_{h \in H} \bigoplus_{a_h \in A_h} D_2^{hg}[a_h], \tag{6.35}$$

where $D_2^{hg}[a_h] := D_2^h[a_h] \Box D_2^g$. The summation on the right-hand side is taken over all simple objects of $A$.

The underlying 1-morphism of $a_{ij} \in \mathrm{Hom}_{A(2\mathsf{Vec}_G^\omega)}(M^{g_i} \lhd D_2^{g_{ij}^h}, M^{g_j}) \cong A_{h_{ij}}^{\mathrm{op}}$ is given component-wise as

$$D_2^{h_j g_j}[a_j] \raisebox{-1em}{$\begin{array}{c}a_{ij}\uparrow\\ \rule{0pt}{1.2em}\end{array}$}\,\begin{array}{c}D_2^{g_{ij}^h}\\ D_2^{h_i g_i}[a_i]\end{array} = \bigoplus_{\alpha_{ij}^{\mathrm{op}} \in \mathrm{Hom}_{A^{\mathrm{op}}}(a_j \otimes^{\mathrm{op}} \overline{a_i}, a_{ij})} 1_{h_i g_i \cdot g_{ij}^h}[\alpha_{ij}^{\mathrm{op}}], \tag{6.36}$$

where $1_{h_i g_i \cdot g_{ij}^h}[\alpha_{ij}^{\mathrm{op}}]$ denotes the simple 1-morphism $1_{h_i g_i \cdot g_{ij}^h}$ associated to a basis morphism $\alpha_{ij}^{\mathrm{op}} \in \mathrm{Hom}_{A^{\mathrm{op}}}(a_j \otimes^{\mathrm{op}} \overline{a_i}, a_{ij})$. The summation on the right-hand side is taken over all basis morphisms. Similarly, the underlying 1-morphism of $a_{ij} \in \mathrm{Hom}_{A(2\mathsf{Vec}_G^\omega)}(M^{g_j}, M^{g_i} \lhd D_2^{g_{ij}^h}) \cong A_{h_{ij}}^{\mathrm{rev}}$ is given component-wise as

$$\begin{array}{c}D_2^{g_{ij}^h}\\ D_2^{h_i g_i}[a_i]\end{array}\raisebox{-1em}{$\begin{array}{c}\uparrow a_{ij}\\ \rule{0pt}{1.2em}\end{array}$}\,D_2^{h_j g_j}[a_j] = \bigoplus_{\alpha_{ij}^{\mathrm{rev}} \in \mathrm{Hom}_{A^{\mathrm{rev}}}(\overline{a_i} \otimes a_j, a_{ij})} 1_{h_i g_i \cdot g_{ij}^h}[\alpha_{ij}^{\mathrm{rev}}], \tag{6.37}$$

where $\overline{1}_{h_i g_i \cdot g^h_{ij}}[\alpha^{\mathrm{rev}}_{ij}]$ denotes the simple 1-morphism $\overline{1}_{h_i g_i \cdot g^h_{ij}}$ associated to a basis morphism $\alpha^{\mathrm{rev}}_{ij} \in \mathrm{Hom}_{A^{\mathrm{rev}}}(\overline{a_i} \otimes a_j, a_{ij})$.

The underlying 2-morphisms of $\alpha^{\mathrm{op}}_{ijk}$ and $\overline{\alpha}^{\mathrm{op}}_{ijk}$ in figure 12 are given component-wise as

$$\tag{6.38}$$

where $F^{a;\alpha}_{ijk}$ and $\Omega^{g;h}_{ijk}$ are defined by

$$\overline{F}^{a;\alpha}_{ijk} = (F^{a;\alpha}_{ijk})^*, \tag{6.39}$$

$$\boxed{\Omega^{g;h}_{ijk} := \frac{\omega(h_i, h_{ij}, h_{jk}, g_k)\omega(h_i, g_i, g^h_{ij}, g^h_{jk})}{\omega(h_i, h_{ij}, g_j, g^h_{jk})}, \qquad \overline{\Omega}^{g;h}_{ijk} := (\Omega^{g;h}_{ijk})^{-1}.} \tag{6.40}$$

We note that $F^{a;\alpha}_{ijk}$ and $\Omega^{g;h}_{ijk}$ are special cases of $F^{g;h}_{ijkl}$ and $\Omega^{g;h}_{ijkl}$. Specifically, we have

$$F^{a;\alpha}_{ijk} = F^{a;\alpha}_{0ijk}, \qquad \Omega^{g;h}_{ijk} = \Omega^{g;h}_{0ijk}, \tag{6.41}$$

where $a_{0x} := a_x$, $\alpha_{0xy} := \alpha_{xy}$, $g_0 := e$, and $h_{0x} := h_x$ for $x, y = i, j, k$. Similarly, the underlying 2-morphisms of $\alpha^{\mathrm{rev}}_{ijk}$ and $\overline{\alpha}^{\mathrm{rev}}_{ijk}$ in figure 13 are given component-wise as

$$\tag{6.42}$$

The cocycle condition on $\omega$ implies

$$\frac{\Omega^{g;h}_{jkl}\Omega^{g;h}_{ijl}}{\Omega^{g;h}_{ikl}\Omega^{g;h}_{ijk}} = \Omega^{g;h}_{ijkl}\frac{\omega(h_i g_i, g^h_{ij}, g^h_{jk}, g^h_{kl})}{\omega(h_i, h_{ij}, h_{jk}, h_{kl})}. \tag{6.43}$$

Using the above equation and the twisted pentagon equation, one can show that (6.38) and (6.42) lead to the module 10-j symbols given by (6.31) and (6.34).

## 6.2 Generalized Gauging on the Lattice

In this subsection, we discuss the generalized gauging of $2\mathsf{Vec}^\omega_G$ symmetry in the fusion surface model.

### 6.2.1 $2\mathsf{Vec}^\omega_G$-Symmetric Model

We first define the fusion surface model with $2\mathsf{Vec}^\omega_G$ symmetry. To this end, we choose the input fusion 2-category $\mathcal{C}$ and the module 2-category $\mathcal{M}$ to be

$$\mathcal{C} = \mathcal{M} = 2\mathsf{Vec}^\omega_G. \tag{6.44}$$

**State Space.** As discussed in section 4.1, the state space of the model is determined by the input data $(\rho, \sigma, \lambda; f, g)$, where $\rho$, $\sigma$, and $\lambda$ are objects of $\mathcal{C}$, and $f$ and $g$ are 1-morphisms of $\mathcal{C}$. For simplicity, we suppose

$$\rho = \sigma = \lambda, \qquad f = g. \tag{6.45}$$

Furthermore, as in section 5.2, we choose

$$\rho = \bigoplus_{g \in G} D^g_2, \qquad \begin{array}{c} D^{g_3}_2 \\ \end{array} \begin{array}{c} f \\ \end{array} \begin{array}{c} D^{g_2}_2 \\ D^{g_1}_2 \end{array} = \delta_{g_1 g_2, g_3} 1_{g_1 \cdot g_2}. \tag{6.46}$$

In this case, the state space of the model is given by

$$\mathcal{H}_{\text{original}} = \bigotimes_{i \in P} \mathbb{C}^{|G|}, \tag{6.47}$$

where $P$ denotes the set of all plaquettes of the honeycomb lattice. The generalization to other $\rho$ and $f$ is straightforward.

**Hamiltonian.** The Hamiltonian of the model is given by

$$H_{\text{original}} = -\sum_{i \in P} \widehat{\mathsf{h}}_i, \tag{6.48}$$

where the local term $\widehat{\mathsf{h}}_i$ is defined by the fusion diagram

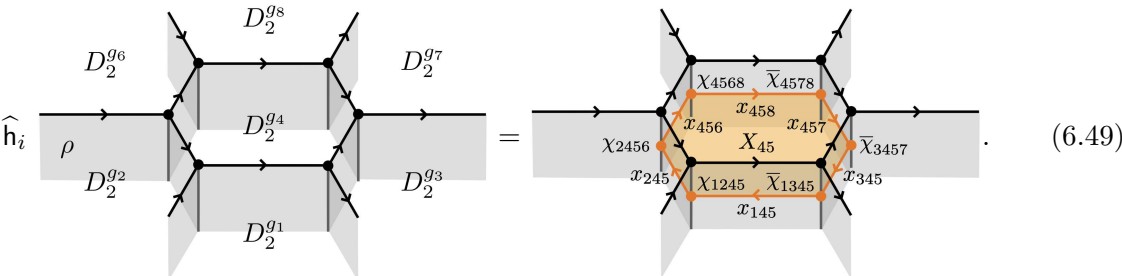

$$\widehat{\mathsf{h}}_i \qquad = \qquad . \qquad (6.49)$$

For simplicity, we choose the object $X_{45} \in 2\mathsf{Vec}_G^\omega$ to be

$$X_{45} = \bigoplus_{g \in G} D_2^g. \qquad (6.50)$$

Similarly, we choose the 1-morphisms $x_{i45}$ and $x_{45j}$ to be

$$x_{i45} \uparrow \quad D_2^{g_{45}} \subset X_{45} \quad = \delta_{g_{i4}g_{45}, g_{i5}} 1_{g_{i4} \cdot g_{45}}, \qquad (6.51)$$
$$D_2^{g_{i5}} \subset \rho \qquad D_2^{g_{i4}} \subset \rho$$

$$x_{45j} \uparrow \quad D_2^{g_{5j}} \subset \rho \quad = \delta_{g_{45}g_{5j}, g_{4j}} 1_{g_{45} \cdot g_{5j}}, \qquad (6.52)$$
$$D_2^{g_{4j}} \subset \rho \qquad D_2^{g_{45}} \subset X_{45}$$

for $i = 1, 2, 3$ and $j = 6, 7, 8$. The 2-morphisms $\chi_{ijkl}$ and $\overline{\chi}_{ijkl}$ are generally given by

$$\chi_{ijkl} \quad D_2^{g_{jl}} \quad D_2^{g_{kl}} = \chi_{ijkl}(g_{ij}, g_{jk}, g_{kl}) \quad D_2^{g_{jl}} \quad D_2^{g_{kl}}, \qquad (6.53)$$
$$D_2^{g_{il}} \quad D_2^{g_{ik}} \quad D_2^{g_{jk}} \qquad D_2^{g_{il}} \quad D_2^{g_{ik}} \quad D_2^{g_{jk}}$$
$$D_2^{g_{ij}} \qquad D_2^{g_{ij}}$$

$$\overline{\chi}_{ijkl} \quad D_2^{g_{ik}} \quad D_2^{g_{kl}} = \overline{\chi}_{ijkl}(g_{ij}, g_{jk}, g_{kl}) \quad D_2^{g_{ik}} \quad D_2^{g_{kl}}, \qquad (6.54)$$
$$D_2^{g_{il}} \quad D_2^{g_{jl}} \quad D_2^{g_{jk}} \qquad D_2^{g_{il}} \quad D_2^{g_{jl}} \quad D_2^{g_{jk}}$$
$$D_2^{g_{ij}} \qquad D_2^{g_{ij}}$$

where $\chi_{ijkl}(g_{ij}, g_{jk}, g_{kl})$ and $\overline{\chi}_{ijkl}(g_{ij}, g_{jk}, g_{kl})$ are arbitrary complex numbers, which are denoted by

$$\chi_{ijkl}^g := \chi_{ijkl}(g_{ij}, g_{jk}, g_{kl}), \qquad \overline{\chi}_{ijkl}^g := \overline{\chi}_{ijkl}(g_{ij}, g_{jk}, g_{kl}). \qquad (6.55)$$

We note that $\overline{\chi}_{ijkl}^g$ is not necessarily the complex conjugate of $\chi_{ijkl}^g$. For the above choice of $(X, x, \chi)$, the Hamiltonian term (6.49) can be evaluated as

$$\widehat{\mathsf{h}}_i \left| \begin{array}{c} g_6 \quad g_8 \quad g_7 \\ g_4 \\ g_2 \quad g_3 \\ g_1 \end{array} \right\rangle = \sum_{g_{45} \in G} \chi_{1245}^g \chi_{2456}^g \chi_{4568}^g \overline{\chi}_{1345}^g \overline{\chi}_{3457}^g \overline{\chi}_{4578}^g \frac{\omega_{1245}^g \omega_{2456}^g \omega_{4568}^g}{\omega_{1345}^g \omega_{3457}^g \omega_{4578}^g} \left| \begin{array}{c} g_6 \quad g_8 \quad g_7 \\ g_5 \\ g_2 \quad g_3 \\ g_1 \end{array} \right\rangle, \qquad (6.56)$$

where $g_{ij} := g_i^{-1} g_j$ and $\omega_{ijkl}^g$ is defined by

$$\omega_{ijkl}^g := \omega(g_i, g_{ij}, g_{jk}, g_{kl}). \tag{6.57}$$

The generalization to other choices of $(X, x, \chi)$ is straightforward.

**Symmetry.** The above model has an anomalous finite group symmetry described by $2\mathsf{Vec}_G^\omega$. The action of the symmetry operator $U_g$ for $g \in G$ is given by [62]

$$U_g \left| \{g_i\} \right\rangle = \prod_{[ijk] \in V} \omega(g, g_i, g_i^{-1} g_j, g_j^{-1} g_k)^{s_{ijk}} \left| \{g g_i\} \right\rangle, \tag{6.58}$$

where $V$ is the set of all vertices of the honeycomb lattice and $s_{ijk}$ is the sign defined by (5.90). The commutativity of $U_g$ and $\widehat{\mathsf{h}}_i$ follows from the cocycle condition on $\omega$.

### 6.2.2 Gauged Model

Let us now describe the gauged model. The input fusion 2-category $\mathcal{C}$ and the module 2-category $\mathcal{M}$ are

$$\mathcal{C} = 2\mathsf{Vec}_G^\omega, \qquad \mathcal{M} = {}_A(2\mathsf{Vec}_G^\omega), \tag{6.59}$$

where $A$ is a $G$-graded $\omega$-twisted unitary fusion category faithfully graded by $H \subset G$.

**State Space.** The input data $(\rho, \sigma, \lambda; f, g)$ are chosen to be the same as those of the original model. In this case, the dynamical variables on the plaquettes, edges, and vertices in the gauged model are labeled by the representatives of the connected components of ${}_A(2\mathsf{Vec}_G^\omega)$, simple objects of $A^{\mathrm{op}}$, and basis morphisms of $A^{\mathrm{op}}$, respectively. Each state with a fixed configuration of the dynamical variables will be written as

$$\left| \{g_i, a_{ij}, \alpha_{ijk}\} \right\rangle := \left| \begin{array}{c} \\ M^{g_l} \\ a_{kl} \\ a_{jl} \longrightarrow M^{g_k} \\ a_{jk} \alpha_{ijk}^{\mathrm{op}} \\ M^{g_j} \\ a_{ik} \\ a_{ij} \quad M^{g_i} \end{array} \right\rangle. \tag{6.60}$$

To avoid cluttering the diagrams, we will omit the labels on the vertices in what follows. The state space of the gauged model is given by

$$\mathcal{H}_{\mathrm{gauged}} := \widehat{\pi}_{\mathrm{LW}} \mathcal{H}'_{\mathrm{gauged}}, \tag{6.61}$$

where $\mathcal{H}'_{\mathrm{gauged}}$ is the vector space spanned by all possible configurations of the dynamical variables and $\widehat{\pi}_{\mathrm{LW}} = \prod_{i \in P} \widehat{B}_i$ is the product of the Levin-Wen plaquette operators

$$\widehat{B}_i \left| \begin{array}{c} \text{[hexagon diagram with } M^{g_i}\text{]} \end{array} \right\rangle = \sum_{a \in A_e^{\mathrm{op}}} \frac{\dim(a)}{\mathcal{D}_{A_e}} \left| \begin{array}{c} \text{[hexagon diagram with } M^{g_i} \text{ and } a\text{]} \end{array} \right\rangle . \tag{6.62}$$

**Hamiltonian.** The Hamiltonian of the gauged model is given by

$$H_{\mathrm{gauged}} = -\sum_{i \in P} \widehat{\mathsf{h}}_i, \tag{6.63}$$

where the local term $\widehat{\mathsf{h}}_i$ is defined by the fusion diagram

$$\tag{6.64}$$

The input data $(X, x, \chi)$ are chosen to be the same as those in the original model. In this case, the action of the Hamiltonian term $\widehat{\mathsf{h}}_i$ can be computed as

$$= \sum_{g_5 \in S_{H \backslash G}} \sum_{h_{45} \in H} \chi_{1245}^{g;h} \chi_{2456}^{g;h} \chi_{4568}^{g;h} \overline{\chi}_{1345}^{g;h} \overline{\chi}_{3457}^{g;h} \overline{\chi}_{4578}^{g;h} \ \overline{\Omega}_{1245}^{g;h} \overline{\Omega}_{2456}^{g;h} \overline{\Omega}_{4568}^{g;h} \Omega_{1345}^{g;h} \Omega_{3457}^{g;h} \Omega_{4578}^{g;h}$$

$$\sum_{a_{45} \in A_{h_{45}}^{\mathrm{op}}} \frac{d_{45}^a}{\mathcal{D}_{A_{h_{45}}}} \sum_{a_{15}, \cdots, a_{58}} \sum_{\alpha_{145}, \cdots, \alpha_{458}} \sum_{\alpha_{125}, \cdots, \alpha_{578}} \overline{F}_{1245}^{a;\alpha} \overline{F}_{2456}^{a;\alpha} \overline{F}_{4568}^{a;\alpha} F_{1345}^{a;\alpha} F_{3457}^{a;\alpha} F_{4578}^{a;\alpha}$$

$$\tag{6.65}$$

$$\sqrt{\frac{d_{15}^a d_{56}^a d_{57}^a}{d_{14}^a d_{46}^a d_{47}^a}} \sqrt{\frac{d_{24}^a d_{34}^a d_{48}^a}{d_{25}^a d_{35}^a d_{58}^a}} \left| \begin{array}{c} \text{[hexagon diagram with } M^{g_5}\text{]} \end{array} \right\rangle .$$

The above equation can be obtained by replacing $F_{ijkl}^{a;\alpha}$ in (5.79) with $\Omega_{ijkl}^{g;h} F_{ijkl}^{a;\alpha}$. When $A = \mathsf{Vec}$, the above Hamiltonian reduces to the original Hamiltonian (6.56) due to (6.32).

**Gauged Model Written in terms of $A^{\text{rev}}$.** The gauged model can also be described in terms of $A^{\text{rev}}$ rather than $A^{\text{op}}$. In the model written in terms of $A^{\text{rev}}$, the dynamical variables on the edges and vertices are labeled by objects and morphisms of $A^{\text{rev}}$, while the dynamical variables on the plaquettes remain unchanged. Each state with a fixed configuration of the dynamical variables is written as

$$|\{g_i, a_{ij}, \alpha_{ijk}\}\rangle := \left| \begin{array}{c} \vcenter{\hbox{(diagram)}} \end{array} \right\rangle . \tag{6.66}$$

We note that the edges are reversed from those in (6.60). The action of the Hamiltonian (6.64) on the above state is again given by (6.65) except that the edges of the honeycomb lattice are now oriented in the opposite direction.

**Ordinary Gauging as a Special Case.** The generalized gauging by $A \in 2\mathsf{Vec}_G^\omega$ reduces to the ordinary twisted gauging of a non-anomalous subgroup $H \subset G$ if we choose

$$A = \mathsf{Vec}_H^\nu. \tag{6.67}$$

Here, $\nu$ is a 3-cochain on $H$ such that $d\nu = \omega|_H^{-1}$, where $\omega|_H$ is the restriction of $\omega$ to $H$. The condition $d\nu = \omega|_H^{-1}$ is required by the twisted pentagon equation (6.5). For the above choice of $A$, the dynamical variables of the gauged model can be described as follows:

- The dynamical variables on the plaquettes are labeled by elements of $S_{H\backslash G}$.

- The dynamical variables on the edges are labeled by elements of $H$ and must satisfy $h_{ik} = h_{ij} h_{jk}$ for every vertex $[ijk]$. Here, $h_{ij} \in H$ denotes the dynamical variable on the edge $[ij]$.

- There are no dynamical variables on the vertices.

Since the Levin-Wen operator in this case is trivial, the state space of the gauged model is given by the vector space spanned by all possible configurations of the dynamical variables.

The Hamiltonian (6.65) reduces to

$$
\widehat{\mathsf{h}}_i \left| \begin{array}{c} \text{(hexagon graph with } M^{g_4}\text{)} \end{array} \right\rangle = \sum_{g_5 \in S_{H\backslash G}} \sum_{h_{45} \in H} \chi^{g;h}_{1245} \chi^{g;h}_{2456} \chi^{g;h}_{4568} \overline{\chi}^{g;h}_{1345} \overline{\chi}^{g;h}_{3457} \overline{\chi}^{g;h}_{4578}
$$

$$
\frac{\Omega^{g;h}_{1345} \Omega^{g;h}_{3457} \Omega^{g;h}_{4578}}{\Omega^{g;h}_{1245} \Omega^{g;h}_{2456} \Omega^{g;h}_{4568}} \frac{\nu^{h}_{1345} \nu^{h}_{3457} \nu^{h}_{4578}}{\nu^{h}_{1245} \nu^{h}_{2456} \nu^{h}_{4568}} \left| \begin{array}{c} \text{(hexagon graph with } M^{g_5}\text{)} \end{array} \right\rangle ,
$$

(6.68)

where $\nu^{h}_{ijkl} := \nu(h_{ij}, h_{jk}, h_{kl})$.

### 6.2.3 Generalized Gauging Operators

Let us consider the generalized gauging operator that implements the generalized gauging of $2\mathsf{Vec}^{\omega}_G$ symmetry. We will also discuss the generalized gauging operator that maps the gauged model back to the original model.

**From the Original Model to the Gauged Model.** The generalized gauging from the original model to the gauged model is described by the free module functor

$$
F_A : 2\mathsf{Vec}^{\omega}_G \to {}_A(2\mathsf{Vec}^{\omega}_G), \tag{6.69}
$$

which maps an object $X \in 2\mathsf{Vec}^{\omega}_G$ to $A\square X \in {}_A(2\mathsf{Vec}^{\omega}_G)$. The action of the corresponding generalized gauging operator $\mathsf{D}_A$ can be computed as

$$
\mathsf{D}_A \left|\{h_i g_i\}\right\rangle = \sum_{\{a_i \in A^{\mathrm{rev}}_{h_i}\}} \prod_{i \in P} \frac{1}{\mathcal{D}_{A_{h_i}}} \sum_{\{a_{ij}, \alpha_{ij}, \alpha_{ijk}\}} \prod_{[ijk]} \left( \frac{d^a_{ij} d^a_{jk}}{d^a_{ik}} \right)^{\frac{1}{4}} (\overline{\Omega}^{g;h}_{ijk})^{s_{ijk}} (\overline{F}^{a;\alpha}_{ijk})^{s_{ijk}} \left|\{g_i, a_{ij}, \alpha_{ijk}\}\right\rangle ,
$$

(6.70)

where $a_{ij} \in \overline{a_i} \otimes a_j$ is a simple object of $A^{\mathrm{rev}}_{h_i^{-1} h_j}$, $\alpha_{ij}$ is a basis morphism of $\mathrm{Hom}_A(\overline{a_i} \otimes a_j, a_{ij})$, $\alpha_{ijk}$ is a basis morphism of $\mathrm{Hom}_A(a_{ij} \otimes a_{jk}, a_{ik})$, $s_{ijk}$ is the sign defined by (5.90), and $h_{ij}$ used in $\Omega^{g;h}_{ijk}$ (cf. (6.40)) is defined by $h_i^{-1} h_j$. On the left-hand side of (6.70), we used the unique decomposition of an element of $G$ into the product of $h_i \in H$ and $g_i \in S_{H\backslash G}$. We note that the above equation is obtained by replacing $F^{a;\alpha}_{ijk}$ in (5.89) with $\Omega^{g;h}_{ijk} F^{a;\alpha}_{ijk}$. The generalized gauging operator $\mathsf{D}_A$ intertwines the Hamiltonian of the original model and that of the gauged model:

$$
\mathsf{D}_A H_{\mathrm{original}} = H_{\mathrm{gauged}} \mathsf{D}_A. \tag{6.71}
$$

**From the Gauged Model to the Original Model.** The ungauging process from the gauged model to the original model is described by the forgetful functor

$$\text{Forg} : {}_A(2\mathsf{Vec}_G^\omega) \to 2\mathsf{Vec}_G^\omega, \tag{6.72}$$

which maps an object $M \in {}_A(2\mathsf{Vec}_G^\omega)$ to its underlying object in $2\mathsf{Vec}_G^\omega$. The corresponding generalized gauging operator $\overline{\mathsf{D}}_A$ is given by

$$\overline{\mathsf{D}}_A \,|\{g_i, a_{ij}, \alpha_{ijk}\}\rangle = \sum_{\{h_i \in H\}} \sum_{\{a_i \in A^{\text{rev}}_{h_i}\}} \sum_{\{\alpha_{ij}\}} \prod_{[ijk]} \left(\frac{d^a_{ij} d^a_{jk}}{d^a_{ik}}\right)^{\frac{1}{4}} (\Omega^{g;h}_{ijk})^{s_{ijk}} (F^{a;\alpha}_{ijk})^{s_{ijk}} \,|\{h_i g_i\}\rangle, \tag{6.73}$$

where $\alpha_{ij}$ is a basis morphism of $\text{Hom}_A(\overline{a_i} \otimes a_j, a_{ij})$. The above equation is obtained by replacing $F^{a;\alpha}_{ijk}$ in (5.94) with $\Omega^{g;h}_{ijk} F^{a;\alpha}_{ijk}$. The right-hand side of (6.73) vanishes if the gauge field $\{h_{ij} \mid [ij] \in E\}$ defined by the grading of $a_{ij} \in A^{\text{rev}}_{h_{ij}}$ has a non-trivial holonomy around a closed loop. That is, for any closed loop $\gamma$ on the dual of the honeycomb lattice, we have

$$\prod_{[ij] \in E_\gamma} h^{s_{ij}}_{ij} \neq e \;\Rightarrow\; \overline{\mathsf{D}}_A \,|\{g_i, a_{ij}, \alpha_{ijk}\}\rangle = 0, \tag{6.74}$$

where $\prod_{[ij] \in E_\gamma}$ is the path-ordered product over all edges that intersect $\gamma$ and $s_{ij}$ is the sign defined by (5.96).

### 6.2.4 Tensor Network Representation

When $A$ is multiplicity-free, the generalized gauging operator $\mathsf{D}_A$ can be represented by the following tensor network:

$$\mathsf{D}_A = \vcenter{\hbox{}}. \tag{6.75}$$

The physical leg above the plaquette takes values in $S_{H\backslash G}$, while the physical leg below the plaquette takes values in $G$. The physical leg on each plaquette takes values in the set of simple objects of $A$. On the other hand, the green bond takes values in $H$, the yellow bond takes values in $S_{H\backslash G}$, and the red bond takes values in the set of simple objects of $A$. The non-zero components of the local tensors in (6.75) are given as follows:

$$\vcenter{\hbox{}} = \frac{1}{\mathcal{D}_{A_h}}, \qquad \vcenter{\hbox{}} = 1, \qquad \vcenter{\hbox{}} = \delta_{a \in A_h}, \tag{6.76}$$

$$\left(\frac{d_{ij}^a d_{jk}^a}{d_{ik}^a}\right)^{\frac{1}{4}} \overline{\Omega}_{ijk}^{g;h} \overline{F}_{ijk}^a, \qquad \left(\frac{d_{ij}^a d_{jk}^a}{d_{ik}^a}\right)^{\frac{1}{4}} \Omega_{ijk}^{g;h} F_{ijk}^a, \qquad (6.77)$$

where $h \in H$, $g \in S_{H \backslash G}$, and $a \in A$.

Similarly, when $A$ is multiplicity-free, the generalized gauging operator $\overline{\mathsf{D}}_A$ can be represented by the following tensor network:

$$\overline{\mathsf{D}}_A = \qquad\qquad\qquad\qquad\qquad\qquad\qquad\qquad . \qquad (6.78)$$

The physical legs are flipped from those of (6.75), while the virtual bonds remain the same. The non-zero components of the local tensors in (6.78) are given as follows:

$$= 1, \qquad a \underset{a}{\overset{}{\bullet}} a = 1, \qquad a \underset{a}{\overset{h}{\rule{0pt}{0pt}}} a = \delta_{a \in A_h}, \qquad (6.79)$$

$$\left(\frac{d_{ij}^a d_{jk}^a}{d_{ik}^a}\right)^{\frac{1}{4}} \Omega_{ijk}^{g;h} F_{ijk}^a, \qquad \left(\frac{d_{ij}^a d_{jk}^a}{d_{ik}^a}\right)^{\frac{1}{4}} \overline{\Omega}_{ijk}^{g;h} \overline{F}_{ijk}^a, \qquad (6.80)$$

where $h \in H$, $g \in S_{H \backslash G}$, and $a \in A$.

## 6.3 Symmetries of the Gauged Model

The symmetry of the gauged model is described by the dual 2-category

$$\mathcal{C}_{\mathcal{M}}^* \cong {}_A(2\mathsf{Vec}_G^\omega)_A, \qquad (6.81)$$

where $\mathcal{C} = 2\mathsf{Vec}_G^\omega$ and $\mathcal{M} = {}_A(2\mathsf{Vec}_G^\omega)$. The objects, 1-morphisms, and 2-morphisms of ${}_A(2\mathsf{Vec}_G^\omega)_A$ are given by $(A, A)$-bimodules, $(A, A)$-bimodule 1-morphisms, and $(A, A)$-bimodule 2-morphisms in $2\mathsf{Vec}_G^\omega$. We refer the reader to [115, Section 2.3] for more details of this 2-category. In this subsection, we will write down some of the symmetry operators labeled by objects and morphisms of ${}_A(2\mathsf{Vec}_G^\omega)_A$. As in the previous subsection, we suppose that $A$ is a $G$-graded $\omega$-twisted unitary fusion category faithfully graded by $H \subset G$.

**0-Form Symmetry.** The 0-form symmetry operators are labeled by objects of $_A(2\mathsf{Vec}_G^\omega)_A$. The connected components of $_A(2\mathsf{Vec}_G^\omega)_A$ are in one-to-one correspondence with double $H$-cosets in $G$. Hence, the symmetry operators up to condensation are in one-to-one correspondence with double $H$-cosets in $G$. We choose $A\square D_2^g\square A$ as a representative of each connected component,[77] where $g \in G$ is a representative of the double $H$-coset $HgH$.

The symmetry operator labeled by $A\square D_2^g\square A$ is given by $\mathsf{D}_A U_g \overline{\mathsf{D}}_A$, where $\mathsf{D}_A$ and $\overline{\mathsf{D}}_A$ are the generalized gauging operators and $U_g$ is the symmetry operator of the original model. By using (6.58), (6.70), and (6.73), we can compute the action of $\mathsf{D}_A U_g \overline{\mathsf{D}}_A$ as follows:

$$
\mathsf{D}_A U_g \overline{\mathsf{D}}_A \left| \{g_i, a_{ij}, \alpha_{ijk}\} \right\rangle = \sum_{\{h_i, a_i, \alpha_{ij}\}} \sum_{\{a_i', a_{ij}', \alpha_{ij}', \alpha_{ijk}'\}} \prod_{i \in P} \frac{1}{\mathcal{D}_{A_{h_i'}}} \prod_{[ijk]} \left( \frac{d_{ij}^a d_{jk}^a}{d_{ik}^a} \right)^{\frac{1}{4}} (\Omega_{ijk}^{g;h})^{s_{ijk}} (F_{ijk}^{a;\alpha})^{s_{ijk}}
$$
$$
\omega(g, h_i g_i, g_{ij}^h, g_{jk}^h)^{s_{ijk}} \left( \frac{d_{ij}^{a'} d_{jk}^{a'}}{d_{ik}^{a'}} \right)^{\frac{1}{4}} (\overline{\Omega}_{ijk}^{g';h'})^{s_{ijk}} (\overline{F}_{ijk}^{a';\alpha'})^{s_{ijk}} \left| \{g_i', a_{ij}', \alpha_{ijk}'\} \right\rangle.
$$
(6.82)

Here, $h_i' \in H$ and $g_i' \in S_{H\backslash G}$ are uniquely determined by $h_i' g_i' = g h_i g_i$, and we defined $h_{ij} := h_i^{-1} h_j$ and $h_{ij}' := (h_i')^{-1} h_j'$. The summations on the right-hand side are taken over $h_i \in H$, $a_i \in A_{h_i}^{\mathrm{rev}}$, $\alpha_{ij} \in \mathrm{Hom}_A(\overline{a_i} \otimes a_j, a_{ij})$, $a_i' \in A_{h_i'}^{\mathrm{rev}}$, $a_{ij}' \in \overline{a_i'} \otimes a_j'$, $\alpha_{ij}' \in \mathrm{Hom}_A(\overline{a_i'} \otimes a_j', a_{ij}')$, and $\alpha_{ijk}' \in \mathrm{Hom}_A(a_{ij}' \otimes a_{jk}', a_{ik}')$.

**1-Form Symmetry.** The 1-form symmetry operators are labeled by 1-endomorphisms of $A \in {}_A(2\mathsf{Vec}_G^\omega)_A$. These 1-morphisms form a braided fusion 1-category $\mathrm{End}_{_A(2\mathsf{Vec}_G^\omega)_A}(A)$. This category is braided equivalent to

$$
\mathrm{End}_{_A(2\mathsf{Vec}_G^\omega)_A}(A) \cong \mathcal{Z}(A)_0,
$$
(6.83)

where, by slight abuse of notation, $\mathcal{Z}(A)_0$ denotes the braided fusion category defined as follows:

- An object of $\mathcal{Z}(A)_0$ is a pair $(a, \Phi)$, where $a$ is an object of $A_e$ and $\Phi$ is a natural family of isomorphisms

$$
\Phi_b : b \otimes a \to a \otimes b, \qquad \forall b \in A,
$$
(6.84)

  that satisfies the usual coherence condition of the half-braiding.

- A morphism $f : (a, \Phi) \to (a', \Phi')$ in $\mathcal{Z}(A)_0$ is a morphism $f : a \to a'$ in $A_e$ that is compatible with the half-braiding, i.e.,

$$
\Phi_b' \circ (\mathrm{id}_b \otimes f) = (f \otimes \mathrm{id}_b) \circ \Phi_b, \qquad \forall b \in A.
$$
(6.85)

---

[77]We note that $A\square D_2^g\square A$ is a simple object of $_A(2\mathsf{Vec}_G^\omega)_A$ becasue the unit object of $A$ is supposed to be simple, cf. section 5.3.2.

- The braided monoidal structure on $\mathcal{Z}(A)_0$ is defined as in the usual definition of the Drinfeld center.[78]

Under the equivalence (6.83), a functor $F \in \mathrm{End}_{A(2\mathsf{Vec}_G^\omega)_A}(A)$ is mapped to an object $(F(\mathbb{1}_A), \Phi) \in \mathcal{Z}(A)_0$, where the half-braiding $\Phi$ is defined by (5.125). On the other hand, an object $(a, \Psi) \in \mathcal{Z}(A)_0$ is mapped to a functor $F \in \mathrm{End}_{A(2\mathsf{Vec}_G^\omega)_A}(A)$ such that $F(b) = b \otimes a$ for all $b \in A$ and the associated natural isomorphisms are given by (5.126).

The 1-form symmetry operator $\widehat{D}^\gamma_{(a,\Phi)}$ labeled by $(a, \Phi) \in \mathcal{Z}(A)_0$ acts on a state as

$$\widehat{D}^\gamma_{(a,\Phi)} \left| \begin{array}{c} \end{array} \right\rangle = \left| \begin{array}{c} \end{array} \right\rangle . \tag{6.86}$$

The blue dots on the right-hand side represent the half-braiding. The superscript $\gamma$ of $\widehat{D}^\gamma_{(a,\Phi)}$ denotes the path on which the symmetry operator is supported.

# 7  Gapped Phases

In this section, we construct commuting projector Hamiltonians for gapped phases with all-boson fusion 2-category symmetries. These models are obtained by the generalized gauging of gapped phases with $2\mathsf{Vec}_G^\omega$ symmetry. We will also write down the ground states of these models using tensor networks.

## 7.1  Gapped Phases with $2\mathsf{Vec}_G$ Symmetry

We begin with gapped phases with $2\mathsf{Vec}_G$ symmetry. We choose

$$\mathcal{C} = 2\mathsf{Vec}_G, \quad \mathcal{M} = 2\mathsf{Vec}_G, \tag{7.1}$$

and consider a gapped phase specified by a separable algebra $B \in 2\mathsf{Vec}_G$.

### 7.1.1  Minimal Gapped Phases

We first focus on minimal gapped phases, which are $G$-symmetric gapped phases without topological orders. These gapped phases are labeled by the pairs $(K, [\lambda])$, where $K \subset G$ is

---

[78]When the twist $\omega$ is trivial, $\mathcal{Z}(A)_0$ reduces to the full subcategory of the Drinfeld center $\mathcal{Z}(A)$ consisting of objects whose underlying objects are in $A_e$.

a subgroup of $G$ and $[\lambda] \in H^3(K, \mathrm{U}(1))$ is an element of the third group cohomology of $K$. Physically, $K$ and $[\lambda]$ represent the unbroken symmetry and its SPT class, respectively.

The minimal gapped phase labeled by $(K, [\lambda])$ is realized in the fusion surface model by choosing the algebra object $B \in 2\mathsf{Vec}_G$ as

$$B = \mathsf{Vec}_K^\lambda. \tag{7.2}$$

Here, $\mathsf{Vec}_K^\lambda$ is the 1-category of finite dimensional $K$-graded vector spaces with the associator twisted by $\lambda \in Z^3(K, \mathrm{U}(1))$. The algebra structure on $\mathsf{Vec}_K^\lambda$ is given as follows:

- The underlying object of $\mathsf{Vec}_K^\lambda \in 2\mathsf{Vec}_G$ is the direct sum $\bigoplus_{k \in K} D_2^k$.

- The multiplication 1-morphism $m : \mathsf{Vec}_K^\lambda \square \mathsf{Vec}_K^\lambda \to \mathsf{Vec}_K^\lambda$ is defined component-wise as

$$\tag{7.3}$$

- The associativity 2-isomorphism $\mu$ is defined component-wise as

$$\tag{7.4}$$

where the white dot on the right-hand side represents a basis 2-morphism. The coherence condition on $\mu$ follows from the cocycle condition on $\lambda$.

For this choice of the algebra $B$, possible configurations of the dynamical variables on the lattice can be described as follows.

- The dynamical variables on the plaquettes are labeled by elements of $G$. These dynamical variables obey the constraint[79]

$$g_i^{-1} g_j \in K \tag{7.5}$$

for any pair of adjacent plaquettes $i$ and $j$.

- There are no dynamical variables on the edges and vertices.

---

[79]This constraint was imposed energetically in section 2.3. Similar comments will also apply to the other lattice models that we will discuss later on.

The Levin-Wen plaquette operator is the identity operator because the endomorphism 1-category $\mathrm{End}_{2\mathsf{Vec}_G}(D_2^g)$ is monoidally equivalent to $\mathsf{Vec}$ for any $g \in G$. Therefore, the state space of the model is given by the vector space spanned by all possible configurations of the dynamical variables. A configuration of the dynamical variables is collectively denoted by

$$\{g_i \mid i \in P\}, \tag{7.6}$$

where $P$ is the set of all plaquettes on the honeycomb lattice. The state corresponding to the above configuration is written as $|\{g_i\}\rangle$. The symmetry operator $U_g$ acts on this state as

$$U_g \left|\{g_i\}\right\rangle = \left|\{gg_i\}\right\rangle. \tag{7.7}$$

The Hamiltonian of the model is given by the sum

$$H = -\sum_{i \in P} \widehat{\mathsf{h}}_i, \tag{7.8}$$

where $\widehat{\mathsf{h}}_i$ is defined in terms of the algebra $\mathsf{Vec}_K^\lambda$ as in (4.16). Unpacking (4.16) leads to

$$\widehat{\mathsf{h}}_i \left| \begin{array}{c} g_8 \\ g_6 \quad \diagup \quad g_7 \\ g_4 \\ g_2 \quad g_3 \\ g_1 \end{array} \right\rangle = \frac{1}{|K|} \sum_{g_5 \in G} \delta_{g_{45} \in K} \frac{\lambda_{1245}^g \lambda_{2456}^g \lambda_{4568}^g}{\lambda_{1345}^g \lambda_{3457}^g \lambda_{4578}^g} \left| \begin{array}{c} g_8 \\ g_6 \quad \diagup \quad g_7 \\ g_5 \\ g_2 \quad g_3 \\ g_1 \end{array} \right\rangle, \tag{7.9}$$

where $\lambda_{ijkl}^g := \lambda(g_{ij}, g_{jk}, g_{kl})$, $g_{ij} := g_i^{-1} g_j$, and $\delta_{g \in K}$ is defined by[80]

$$\delta_{g \in K} = \begin{cases} 1 & \text{if } g \in K, \\ 0 & \text{if } g \notin K. \end{cases} \tag{7.10}$$

Since the above Hamiltonian is the sum of commuting projectors, the ground states are given by the simultaneous eigenstates of $\widehat{\mathsf{h}}_i$'s with eigenvalue 1. Such states can be obtained by applying $\prod_i \widehat{\mathsf{h}}_i$ to generic states.

Let us write down the ground states explicitly. To this end, we first notice that the state space $\mathcal{H}$ can be decomposed into the direct sum

$$\mathcal{H} = \bigoplus_{\mu = 1, 2, \cdots, |G/K|} \mathcal{H}^{(\mu)}, \tag{7.11}$$

where $\mathcal{H}^{(\mu)}$ is the vector space spanned by the states such that all the plaquettes are labeled by elements of the same left $K$-coset $g^{(\mu)}K$, that is, we have[81]

$$\mathcal{H}^{(\mu)} := \mathrm{Span}\{|\{g_i\}\rangle \mid g_i \in g^{(\mu)}K\}. \tag{7.12}$$

---

[80]We note that $\lambda(g_{ij}, g_{jk}, g_{kl})$ makes sense because $g_{ij}, g_{jk}, g_{kl} \in K$ due to the constraint (7.5).

[81]When the dynamical variable on some plaquette is labeled by an element of $g^{(\mu)}K$, the dynamical variables on all the other plaquettes must also be labeled by elements of the same coset due to the constraint (7.5).

Here, $g^{(\mu)} \in G$ denotes a representative of the coset $g^{(\mu)}K$. Each sector has its own ground states because the action of the Hamiltonian does not mix these sectors. Furthermore, the ground state in each sector is unique on an infinite plane.[82] Thus, there are as many ground states on an infinite plane as the number of left $K$-cosets, which is equal to $|G|/|K|$. The ground state $|\mathrm{GS}; \mu\rangle$ within each sector $\mathcal{H}^{(\mu)}$ is given by

$$|\mathrm{GS}; \mu\rangle = \prod_{i \in P} \widehat{\mathsf{h}}_i \,|\{g_i = g^{(\mu)}\}\rangle, \tag{7.13}$$

where $|\{g_i = g^{(\mu)}\}\rangle$ denotes the state where all the plaquettes are labeled by $g^{(\mu)} \in G$. A direct computation shows that the above ground state can be written more explicitly as follows:

$$\boxed{|\mathrm{GS}; \mu\rangle = \sum_{\{k_i \in K\}} \prod_{i \in P} \frac{1}{|K|} \prod_{[ijk] \in V} \lambda(k_i, k_{ij}, k_{jk})^{s_{ijk}} \,|\{g^{(\mu)}k_i\}\rangle.} \tag{7.14}$$

Here, $V$ denotes the set of all vertices on the honeycomb lattice, $k_{ij}$ is defined by $k_i^{-1}k_j$, and $s_{ijk}$ is defined by (5.90). See (7.37) for a detailed computation of the ground states of more general gapped phases. The symmetry operators act on the above ground states as

$$U_g \,|\mathrm{GS}; \mu\rangle = |\mathrm{GS}; \nu\rangle, \tag{7.15}$$

where $g \in g^{(\nu)}K(g^{(\mu)})^{-1} := \{g^{(\nu)}k(g^{(\mu)})^{-1} \mid k \in K\}$. The explicit form of the ground states (7.14) implies that the above gapped phase spontaneously breaks the $G$ symmetry down to $K$ and realizes an SPT order labeled by $[\lambda] \in H^3(K, \mathrm{U}(1))$. In particular, when $K = G$, our model agrees with the group cohomology model of SPT phases studied in [163].

**Embedding into Tensor Product State Space.** The above model can be embedded into a tensor product state space without breaking the symmetry by imposing the constraint (7.5) energetically. Specifically, we define the state space of the new model as

$$\widetilde{\mathcal{H}} = \bigotimes_{i \in P} \mathbb{C}^{|G|}. \tag{7.16}$$

The Hamiltonian acting on this state space is given by

$$H = -\sum_{i \in P} \widehat{\mathsf{h}}_i - \sum_{[ij] \in E} \widehat{\mathsf{h}}_{ij}, \tag{7.17}$$

---

[82]When restricted to the sector $\mathcal{H}^{(1)}$ labeled by the trivial left $K$-coset $K$, the model reduces to the group cohomology model of SPT phases with $K$ symmetry [163]. Hence, it has a unique ground state in $\mathcal{H}^{(1)}$. The other sectors have the same energy spectrum as $\mathcal{H}^{(1)}$ because any states in the other sectors are obtained by acting with the symmetry operators on the states in $\mathcal{H}^{(1)}$. Thus, the model has a unique ground state in every sector.

where $E$ is the set of all edges, and $\widehat{\mathsf{h}}_i$ and $\widehat{\mathsf{h}}_{ij}$ are defined as follows:

$$\widehat{\mathsf{h}}_i \left|\begin{array}{c} \vcenter{\hbox{hexagon with } g_8, g_6, g_7, g_4, g_2, g_3, g_1} \end{array}\right\rangle = \frac{1}{|K|} \sum_{g_5 \in G} \left( \prod_{j=1,2,\cdots,8} \delta_{g_{4j} \in K} \right) \frac{\lambda^g_{1245} \lambda^g_{2456} \lambda^g_{4568}}{\lambda^g_{1345} \lambda^g_{3457} \lambda^g_{4578}} \left|\begin{array}{c} \vcenter{\hbox{hexagon with } g_8, g_6, g_7, g_5, g_2, g_3, g_1} \end{array}\right\rangle , \qquad (7.18)$$

$$\widehat{\mathsf{h}}_{ij} \left|\begin{array}{c} g_j \\ \rule{1.5cm}{0.4pt} \\ g_i \end{array}\right\rangle = \delta_{g_{ij} \in K} \left|\begin{array}{c} g_j \\ \rule{1.5cm}{0.4pt} \\ g_i \end{array}\right\rangle . \qquad (7.19)$$

Here, we recall $g_{ij} := g_i^{-1} g_j$. The first and second terms of the Hamiltonian (7.17) commute with each other. Therefore, the ground states are the simultaneous eigenstates of both terms with the lowest eigenvalues. The ground state subspace of the second term agrees with the state space (7.11) of the original model. The first term acts on this subspace in the same way as the original Hamiltonian (7.9). Thus, the ground states of the new model agree with those of the original model. We emphasize that the new model still has $2\mathsf{Vec}_G$ symmetry whose action is given by (7.7). This model is also an example of the fusion surface model with a specific choice of the input data $(X, x, \chi)$.

### 7.1.2 Tensor Network Representation for Minimal Gapped Phases

The ground state (7.14) can be written in terms of a tensor network as follows:

$$|\mathrm{GS}; \mu\rangle = \vcenter{\hbox{tensor network diagram with central node } \mu} . \qquad (7.20)$$

Here, the physical leg on each plaquette takes values in $G$. On the other hand, the virtual bond written in blue takes values in $K$. The non-zero components of the local tensors in (7.20) are given by

$$\vcenter{\hbox{tensor with legs } k, g, k, k, \mu, k, k} = \frac{1}{|K|} \delta_{k, (g^{(\mu)})^{-1} g} , \qquad (7.21)$$

$$\vcenter{\hbox{triangle with } k_j, k_k, k_i} = \lambda(k_i, k_{ij}, k_{jk}), \qquad \vcenter{\hbox{shaded triangle with } k_k, k_j, k_i} = \lambda(k_i, k_{ij}, k_{jk})^{-1} . \qquad (7.22)$$

### 7.1.3 Non-minimal Gapped Phases

A $G$-symmetric gapped phase is said to be non-minimal if it has a non-trivial topological order. There are two types of such non-minimal gapped phases: one is chiral and the other is non-chiral. In what follows, we will restrict our attention to non-chiral phases, i.e., gapped phases that admit gapped boundaries.

In the fusion surface model, a non-chiral gapped phase with $2\mathsf{Vec}_G$ symmetry is realized by choosing $B \in 2\mathsf{Vec}_G$ to be a general $G$-graded unitary fusion category:

$$B = \bigoplus_{g \in G} B_g. \tag{7.23}$$

We suppose that $B$ is faithfully graded by a subgroup $K \subset G$, that is,

$$g \notin K \ \Leftrightarrow\ B_g = 0. \tag{7.24}$$

The minimal gapped phase labeled by $(K, [\lambda])$ is included as a special case where $B = \mathsf{Vec}_K^\lambda$.

For the above choice of the algebra $B \in 2\mathsf{Vec}_G$, possible configurations of the dynamical variables of the model can be described as follows:

- The dynamical variables on the plaquettes are labeled by elements of $G$. These dynamical variables must satisfy the constraint

  $$g_i^{-1} g_j \in K \tag{7.25}$$

  for any pair of adjacent plaquettes $i$ and $j$. A configuration of the dynamical variables on the plaquettes is denoted by $\{g_i \mid i \in P\}$.

- For a given configuration $\{g_i \mid i \in P\}$, the dynamical variable on edge $[ij]$ takes values in the set of simple objects of $B_{g_{ij}}$, where $g_{ij} := g_i^{-1} g_j$. Each simple object $b_{ij} \in B_{g_{ij}}$ labels the vertical surface that ends on edge $[ij]$.[83] A configuration of the dynamical variables on the plaquettes and edges is denoted by $\{g_i, b_{ij} \mid i \in P,\ [ij] \in E\}$.

- For a given configuration $\{g_i, b_{ij} \mid i \in P,\ [ij] \in E\}$, the dynamical variable on vertex $[ijk]$ takes values in $\mathrm{Hom}_B(b_{ik}, b_{ij} \otimes b_{jk})$ or $\mathrm{Hom}_B(b_{ij} \otimes b_{jk}, b_{ik})$ depending on whether $[ijk]$ is in the $A$-sublattice or $B$-sublattice. A configuration of the dynamical variables on the plaquettes, edges, and vertices is denoted by

  $$\{g_i, b_{ij}, \beta_{ijk} \mid i \in P,\ [ij] \in E,\ [ijk] \in V\}. \tag{7.26}$$

---

[83]More precisely, the surface that ends on edge $[ij]$ is labeled by $D_2^{g_{ij}}[b_{ij}] \in 2\mathsf{Vec}_G$.

The Levin-Wen plaquette operator is trivial because $\mathrm{End}_{2\mathsf{Vec}_G}(D_2^g) \cong \mathsf{Vec}$ for all $g \in G$. Thus, the state space is given by the vector space spanned by all possible configurations of the dynamical variables. The state corresponding to the configuration (7.26) is denoted by $|\{g_i, b_{ij}, \beta_{ijk}\}\rangle$. Diagrammatically, this state can be written as

$$
|\{g_i, b_{ij}, \beta_{ijk}\}\rangle = \left| \vphantom{\begin{array}{c}A\\B\\C\end{array}} \right. \left. \begin{array}{c} \text{(diagram)} \end{array} \right\rangle . \tag{7.27}
$$

The labels on the vertices will often be omitted in order to avoid cluttering the diagram. The symmetry operator $U_g$ acts on the above state as

$$
U_g |\{g_i, b_{ij}, \beta_{ijk}\}\rangle = |\{gg_i, b_{ij}, \beta_{ijk}\}\rangle . \tag{7.28}
$$

As in the minimal gapped phases, the state space consists of multiple sectors labeled by left $K$-cosets in $G$:

$$
\mathcal{H} = \bigoplus_{\mu=1,2,\cdots,|G/K|} \mathcal{H}^{(\mu)} . \tag{7.29}
$$

Here, each sector $\mathcal{H}^{(\mu)}$ is defined by

$$
\mathcal{H}^{(\mu)} := \mathrm{Span}\{|g_i, b_{ij}, \beta_{ijk}\rangle \mid g_i \in g^{(\mu)}K, \ b_{ij} \in B_{g_{ij}}, \ \beta_{ijk} \in \mathrm{Hom}_B(b_{ij} \otimes b_{jk}, b_{ik})\} . \tag{7.30}
$$

We note that the states in different sectors are related to each other by the symmetry action.

The Hamiltonian of the model is again given by the sum

$$
H = -\sum_{i \in P} \widehat{\mathsf{h}}_i , \tag{7.31}
$$

where $\widehat{\mathsf{h}}_i$ is defined by (4.16). Using the defining data of the separable algebra $B \in 2\mathsf{Vec}_G$ (cf. section 5.1.1), we can write down the action of $\widehat{\mathsf{h}}_i$ explicitly as follows:

$$
\widehat{\mathsf{h}}_i \left| \begin{array}{c} \text{(diagram)} \end{array} \right\rangle = \sum_{g_5 \in G} \delta_{g_{45} \in K} \sum_{b_{45} \in B_{g_{45}}} \frac{d_{45}^b}{\mathcal{D}_B} \sum_{b_{15},\cdots,b_{58}} \sum_{\beta_{145},\cdots,\beta_{458}} \sum_{\beta_{125},\cdots,\beta_{578}} \sqrt{\frac{d_{15}^b d_{56}^b d_{57}^b}{d_{14}^b d_{46}^b d_{47}^b}}
$$

$$
\sqrt{\frac{d_{24}^b d_{34}^b d_{48}^b}{d_{25}^b d_{35}^b d_{58}^b}} \, F_{1245}^{b;\beta} F_{2456}^{b;\beta} F_{4568}^{b;\beta} \overline{F}_{1345}^{b;\beta} \overline{F}_{3457}^{b;\beta} \overline{F}_{4578}^{b;\beta} \left| \begin{array}{c} \text{(diagram)} \end{array} \right\rangle . \tag{7.32}
$$

The summations on the right-hand side are taken over all fusion channels $b_{i5} \in b_{i4} \otimes b_{45}$, $b_{5j} \in \overline{b_{45}} \otimes b_{4j}$, and all basis morphisms $\beta_{ijk} \in \mathrm{Hom}_B(b_{ij} \otimes b_{jk}, b_{ik})$. More concisely, the above Hamiltonian can also be written as

$$\widehat{\mathsf{h}}_i = \sum_{k \in K} \sum_{b \in B_k} \frac{\dim(b)}{\mathcal{D}_B} \widehat{L}_i^{(b)}, \tag{7.33}$$

where the loop operator $\widehat{L}_i^{(b)}$ for $b \in B_k$ is defined by

$$\widehat{L}_i^{(b)} \left| \begin{array}{c} \includegraphics \end{array} \right\rangle = \left| \begin{array}{c} \includegraphics \end{array} \right\rangle . \tag{7.34}$$

The loop on the right-hand side acts on the dynamical variables on the nearby edges by the fusion. As in the minimal gapped phases, the above model can be embedded into a tensor product state space without breaking the symmetry.

The ground states of the Hamiltonian (7.31) are obtained by applying the local commuting projectors (7.32) to generic states. As in the case of the minimal gapped phases, we expect that each sector $\mathcal{H}^{(\mu)}$ has a unique ground state on an infinite plane. Therefore, the model on an infinite plane has as many ground states as the number of left $K$-cosets in $G$. The ground state in each sector $\mathcal{H}^{(\mu)}$ is given by

$$|\mathrm{GS}; \mu\rangle = \prod_{i \in P} \widehat{\mathsf{h}}_i \left| \{ g_i = g^{(\mu)}, b_{ij} = \mathbb{1}, \beta_{ijk} = \mathrm{id} \} \right\rangle, \tag{7.35}$$

where $\left| \{ g_i = g^{(\mu)}, b_{ij} = \mathbb{1}, \beta_{ijk} = \mathrm{id} \} \right\rangle$ denotes the state where all plaquettes are labeled by the representative $g^{(\mu)} \in g^{(\mu)}K$, all edges are labeled by the unit object $\mathbb{1} \in B$, and all vertices are labeled by the identity morphism $\mathrm{id} \in \mathrm{Hom}_B(\mathbb{1}, \mathbb{1}) \cong \mathrm{Hom}_B(\mathbb{1} \otimes \mathbb{1}, \mathbb{1})$. The above ground state can be computed diagrammatically as

$$|\mathrm{GS}; \mu\rangle = \sum_{\{k_i \in K\}} \sum_{\{b_i \in B_{k_i}\}} \prod_{i \in P} \frac{d_i^b}{\mathcal{D}_B} \quad \includegraphics$$

$$= \sum_{\{k_i \in K\}} \sum_{\{b_i \in B_{k_i}\}} \sum_{\{b_{ij} \in \overline{b_i} \otimes b_j\}} \sum_{\{\beta_{ij}\}} \prod_{i \in P} \frac{d_i^b}{\mathcal{D}_B} \prod_{[ij] \in E} \sqrt{\frac{d_{ij}^b}{d_i^b d_j^b}} \quad \includegraphics , \tag{7.36}$$

where the dotted edges on the first line are labeled by $\mathbb{1} \in B$, and $\beta_{ij} \in \mathrm{Hom}_B(\overline{b_i} \otimes b_j, b_{ij})$ on the second line are the morphisms at the vertices of the last diagram. Removing the bubbles around the vertices via the $F$-move leads us to

$$|\mathrm{GS};\mu\rangle = \sum_{\{k_i \in K\}} \sum_{\{b_i \in B_{k_i}\}} \sum_{\{b_{ij},\beta_{ij},\beta_{ijk}\}} \prod_{i \in P} \frac{1}{\mathcal{D}_B} \prod_{[ijk]} \left( \frac{d_{ij}^b d_{jk}^b}{d_{ik}^b} \right)^{\frac{1}{4}} \prod_{[ijk]} (F_{ijk}^{b;\beta})^{s_{ijk}} \, |\{g^{(\mu)}k_i, b_{ij}, \beta_{ijk}\}\rangle \, .$$

(7.37)

Here, the sign $s_{ijk}$ is defined by (5.90), and $F_{ijk}^{b;\beta}$ is defined by

$$F_{ijk}^{b;\beta} := (F_{b_k}^{b_i b_{ij} b_{jk}})_{(b_j;\beta_{ij},\beta_{jk}),(b_{ik};\beta_{ik},\beta_{ijk})}.$$

(7.38)

We note that (7.37) reduces to the ground states (7.14) of the minimal gapped phase when $B = \mathsf{Vec}_K^\lambda$. The symmetry operators act on the above ground states as

$$U_g \, |\mathrm{GS};\mu\rangle = |\mathrm{GS};\nu\rangle \, ,$$

(7.39)

where $g \in g^{(\nu)} K (g^{(\mu)})^{-1}$.

When $B$ is faithfully graded by $G$ and is multiplicity-free, the above model agrees with the symmetry-enriched string-net model studied in [70, 71]. In particular, when $G$ is the trivial group, the above model reduces to the Levin-Wen model [64].

### 7.1.4 Tensor Network Representation for Non-minimal Gapped Phases

The ground state (7.14) can be written in terms of a tensor network. In particular, when $B$ is multiplicity-free, the tensor network representation of the ground state (7.37) is given by[84]

$$|\mathrm{GS};\mu\rangle = \quad \raisebox{-2em}{} \quad .$$

(7.40)

Here, the physical leg on each plaquette takes values in $G$, while the physical leg on each edge takes values in the set of simple objects of $B$. On the other hand, the virtual bond written in blue takes values in $K$, while the virtual bond written in orange takes values in the set of simple objects of $B$. The non-zero components of the local tensors in (7.40) are given as follows:

   (7.41)

---

[84]When $B$ is not multiplicity-free, the model has the dynamical variables also on the vertices, which makes the tensor network representation slightly more complicated.

$$
\begin{array}{cc}
\begin{array}{c}
b_{jk}\ b_k \\
b_j \diagdown\diagdown\ b_k \\
\diagdown\!\!\triangleright\ b_{ik} \\
b_j \diagup\diagup\ b_i \\
b_{ij}\ b_i
\end{array}
= \left( \frac{d^b_{ij} d^b_{jk}}{d^b_{ik}} \right)^{\frac{1}{4}} F^b_{ijk},
&
\begin{array}{c}
b_k\ b_{jk} \\
b_k \diagdown\diagdown\ b_j \\
b_{ik}\ \blacktriangleright\!\!\diagdown\ b_j \\
b_i \diagup\diagup\ b_j \\
b_i\ b_{ij}
\end{array}
= \left( \frac{d^b_{ij} d^b_{jk}}{d^b_{ik}} \right)^{\frac{1}{4}} \overline{F}^b_{ijk},
\end{array}
\qquad (7.42)
$$

where $\delta_{b \in B_k}$ in (7.41) is one if $b \in B_k$ and zero otherwise. When $B = \mathsf{Vec}^\lambda_K$, (7.40) reduces to the tensor network representation (7.20) of the ground states of the minimal gapped phase. When $G$ is trivial, (7.40) reduces to the tensor network representation of the ground state of the Levin-Wen model based on $B$ [147, 148]. When $K = G$, (7.40) reduces to the tensor network representation of the ground state of the $G$-enriched string-net model [93].[85]

## 7.2   Gapped Phases with $2\mathsf{Vec}^\omega_G$ Symmetry

Let us generalize the previous subsection to incorporate an anomaly $[\omega] \in H^4(G, \mathrm{U}(1))$. We choose the input fusion 2-category $\mathcal{C}$ and the module 2-category $\mathcal{M}$ to be

$$
\mathcal{C} = \mathcal{M} = 2\mathsf{Vec}^\omega_G. \qquad (7.43)
$$

We consider the gapped phase specified by a $G$-graded $\omega$-twisted unitary fusion category $B \in 2\mathsf{Vec}^\omega_G$, which is faithfully graded by $K \subset G$.

### 7.2.1   Minimal Gapped Phases

A minimal gapped phase with $2\mathsf{Vec}^\omega_G$ symmetry is obtained by choosing separable algebra $B \in 2\mathsf{Vec}^\omega_G$ to be

$$
B = \mathsf{Vec}^\lambda_K, \qquad (7.44)
$$

where $K$ is a subgroup of $G$ and $\lambda$ is a 3-cochain on $K$ that satisfies

$$
d\lambda = \omega|^{-1}_K. \qquad (7.45)
$$

The right-hand side is the restriction of $\omega$ to $K$. The underlying object of $\mathsf{Vec}^\lambda_K \in 2\mathsf{Vec}^\omega_G$ is the direct sum $\bigoplus_{k \in K} D^k_2$ and the algebra structure on it is given by (7.3) and (7.4). The consistency condition on the associativity 2-isomorphism $\mu$ follows from (7.45).

When $B = \mathsf{Vec}^\lambda_K$, possible configurations of the dynamical variables on the lattice can be described as follows:

---

[85]Our tensors are slightly different from those in some references, e.g., [91,93,164], due to different conventions for the contraction of the tensors. The above references employ the convention that every closed loop in the tensor network involves the quantum dimension, while we use the standard convention that the closed loop of a virtual bond labeled by a fixed object evaluates to 1.

- The dynamical variables on the plaquettes are labeled by elements of $G$. These dynamical variables must satisfy

$$g_i^{-1}g_j \in K \tag{7.46}$$

for any adjacent plaquettes $i$ and $j$.

- There are no dynamical variables on the edges and vertices.

The Levin-Wen plaquette operator is trivial due to the equivalence $\mathrm{End}_{2\mathsf{Vec}_G^\omega}(D_2^g) \cong \mathsf{Vec}$ for all $g \in G$. Therefore, the state space is given by the vector space spanned by all possible configurations of the dynamical variables. The state corresponding to the configuration $\{g_i \mid i \in P\}$ is denoted by $|\{g_i\}\rangle$. The symmetry operator $U_g$ acts on this state as

$$U_g |\{g_i\}\rangle = \prod_{[ijk] \in V} \omega(g, g_i, g_i^{-1}g_j, g_j^{-1}g_k)^{s_{ijk}} |\{gg_i\}\rangle, \tag{7.47}$$

where $s_{ijk}$ is defined by (5.90). As in the case of $2\mathsf{Vec}_G$ symmetry, the state space can be decomposed into a direct sum

$$\mathcal{H} = \bigoplus_{\mu=1,2,\cdots,|G/K|} \mathcal{H}^{(\mu)}, \tag{7.48}$$

where each sector $\mathcal{H}^{(\mu)}$ is defined by

$$\mathcal{H}^{(\mu)} := \mathrm{Span}\{|g_i\rangle \mid g_i \in g^{(\mu)}K\}. \tag{7.49}$$

Here, $g^{(\mu)}$ is a representative of the left $K$-coset $g^{(\mu)}K$. We note that the symmetry action permutes the above sectors.

The Hamiltonian is given by

$$H = -\sum_{i \in P} \widehat{\mathsf{h}}_i, \tag{7.50}$$

where $\widehat{\mathsf{h}}_i$ is defined by (4.16). When $B = \mathsf{Vec}_K^\lambda$, the action of $\widehat{\mathsf{h}}_i$ can be computed as

$$\widehat{\mathsf{h}}_i \left| \begin{array}{c} g_8 \\ g_6 \quad g_7 \\ g_4 \\ g_2 \quad g_3 \\ g_1 \end{array} \right\rangle = \frac{1}{|K|} \sum_{g_5 \in G} \delta_{g_{45} \in K} \frac{\lambda_{1245}^g \lambda_{2456}^g \lambda_{4568}^g \, \omega_{1245}^g \omega_{2456}^g \omega_{4568}^g}{\lambda_{1345}^g \lambda_{3457}^g \lambda_{4578}^g \, \omega_{1345}^g \omega_{3457}^g \omega_{4578}^g} \left| \begin{array}{c} g_8 \\ g_6 \quad g_7 \\ g_5 \\ g_2 \quad g_3 \\ g_1 \end{array} \right\rangle, \tag{7.51}$$

where $\lambda_{ijkl}^g := \lambda(g_{ij}, g_{jk}, g_{kl})$, $\omega_{ijkl}^g := \omega(g_i, g_{ij}, g_{jk}, g_{kl})$, $g_{ij} := g_i^{-1}g_j$, and $\delta_{g_{ij} \in K}$ is defined by (7.10). The above model can be embedded into a tensor product state space without breaking the symmetry, just as we did in the case of non-anomalous symmetries.

As in the gapped phases with $2\mathsf{Vec}_G$ symmetry, the ground states on an infinite plane are in one-to-one correspondence with left $K$-cosets in $G$. The ground state labeled by the left

$K$-coset $g^{(\mu)}K$ is given by

$$|\text{GS}; \mu\rangle = \prod_{i \in P} \widehat{\mathsf{h}}_i \, |\{g_i = g^{(\mu)}\}\rangle, \qquad (7.52)$$

where $|\{g_i = g^{(\mu)}\}\rangle$ is the state where all plaquettes are labeled by $g^{(\mu)}$. By a direct computation, we find

$$\boxed{|\text{GS}; \mu\rangle = \sum_{\{k_i \in K\}} \prod_{i \in P} \frac{1}{|K|} \prod_{[ijk] \in V} \omega(g^{(\mu)}, k_i, k_{ij}, k_{jk})^{s_{ijk}} \lambda(k_i, k_{ij}, k_{jk})^{s_{ijk}} |\{g^{(\mu)}k_i\}\rangle,} \qquad (7.53)$$

where $k_{ij} := k_i^{-1}k_j$. The derivation of the above equation will be explained in section 7.2.3. The symmetry operators act on the above ground states as

$$U_g \, |\text{GS}; \mu\rangle = |\text{GS}; \nu\rangle, \qquad (7.54)$$

where $g \in g^{(\nu)}K(g^{(\mu)})^{-1}$.

### 7.2.2 Tensor Network Representation for Minimal Gapped Phases

The ground state (7.53) can be represented by the following tensor network:

$$|\text{GS}; \mu\rangle = \quad  \quad . \qquad (7.55)$$

The physical leg on each plaquette takes values in $G$, while the virtual bond written in blue takes values in $K$. The non-zero components of the local tensors in (7.55) are given by

$$ = \frac{1}{|K|}\delta_{k,(g^{(\mu)})^{-1}g}, \qquad (7.56)$$

$$ = \omega(g^{(\mu)}, k_i, k_{ij}, k_{jk})\lambda(k_i, k_{ij}, k_{jk}), \qquad (7.57)$$

$$ = \omega(g^{(\mu)}, k_i, k_{ij}, k_{jk})^{-1}\lambda(k_i, k_{ij}, k_{jk})^{-1}. \qquad (7.58)$$

### 7.2.3 Non-Minimal Gapped Phases

A non-minimal (non-chiral) gapped phase with $2\mathsf{Vec}_G^\omega$ symmetry is obtained by choosing the separable algebra $B \in 2\mathsf{Vec}_G^\omega$ to be a general $G$-graded $\omega$-twisted unitary fusion category

$$B = \bigoplus_{g \in G} B_g. \tag{7.59}$$

We suppose that $B$ is faithfully graded by a subgroup $K \subset G$. The non-minimal gapped phase reduces to a minimal gapped phase when $B = \mathsf{Vec}_K^\lambda$.

For a general $B$, possible configurations of the dynamical variables on the lattice can be described as follows:

- The dynamical variable $g_i$ on each plaquette $i$ takes values in $G$. For any pair of adjacent plaquettes $i, j \in P$, we have a constraint

$$g_i^{-1} g_j \in K. \tag{7.60}$$

- For a given configuration $\{g_i \mid i \in P\}$, the dynamical variable $b_{ij}$ on each edge $[ij]$ takes values in the set of simple objects of $B_{g_{ij}}$, where $g_{ij} := g_i^{-1} g_j$.

- For a given configuration $\{g_i, b_{ij} \mid i \in P, [ij] \in E\}$, the dynamical variable on each vertex $[ijk]$ takes values in $\mathrm{Hom}_B(b_{ik}, b_{ij} \otimes b_{jk})$ or $\mathrm{Hom}_B(b_{ij} \otimes b_{jk}, b_{ik})$ depending on whether $[ijk]$ is in the $A$-sublattice or $B$-sublattice.

Since the Levin-Wen plaquette operator is trivial, the state space is given by the vector space spanned by all possible configurations of the above dynamical variables. The state corresponding to a configuration $\{g_i, b_{ij}, \beta_{ijk} \mid i \in P, \ [ij] \in E, \ [ijk] \in V\}$ is written as

$$|\{g_i, b_{ij}, \beta_{ijk}\}\rangle = \left| \begin{array}{c} \end{array} \right\rangle. \tag{7.61}$$

The symmetry operator $U_g$ acts on this state as

$$U_g |g_i, b_{ij}, \beta_{ijk}\rangle = \prod_{[ijk] \in V} \omega(g, g_i, g_i^{-1} g_j, g_j^{-1} g_k)^{s_{ijk}} |\{gg_i, b_{ij}, \beta_{ijk}\}\rangle. \tag{7.62}$$

As in the case of $2\mathsf{Vec}_G$ symmetry, the state space can be decomposed into a direct sum

$$\mathcal{H} = \bigoplus_{\mu = 1, 2, \cdots, |G/K|} \mathcal{H}^{(\mu)}, \tag{7.63}$$

where each sector $\mathcal{H}^{(\mu)}$ is defined by

$$\mathcal{H}^{(\mu)} := \mathrm{Span}\{|g_i, b_{ij}, \beta_{ijk}\rangle \mid g_i \in g^{(\mu)}K, \ b_{ij} \in B_{g_{ij}}, \ \beta_{ijk} \in \mathrm{Hom}_B(b_{ij} \otimes b_{jk}, b_{ik})\}. \tag{7.64}$$

These sectors are permuted by the symmetry action.

The Hamiltonian is again given by

$$H = -\sum_{i \in P} \widehat{\mathsf{h}}_i, \tag{7.65}$$

where $\widehat{\mathsf{h}}_i$ is defined by (4.16). For a general $B$, the action of $\widehat{\mathsf{h}}_i$ can be computed as

$$\widehat{\mathsf{h}}_i \left| \begin{array}{c} \text{(hexagon diagram with } g_4 \text{ and labels } b_{68}, b_{78}, b_{48}, b_{46}, b_{47}, b_{37}, b_{26}, b_{24}, b_{34}, b_{14}, b_{12}, b_{13}) \end{array} \right\rangle$$

$$= \sum_{g_5 \in G} \delta_{g_{45} \in K} \sum_{b_{45} \in B_{g_{45}}} \frac{d_{45}^b}{\mathcal{D}_B} \sum_{b_{15}, \cdots, b_{58}} \sum_{\beta_{145}, \cdots, \beta_{458}} \sum_{\beta_{125}, \cdots, \beta_{578}} \sqrt{\frac{d_{15}^b d_{56}^b d_{57}^b}{d_{14}^b d_{46}^b d_{47}^b}} \sqrt{\frac{d_{24}^b d_{34}^b d_{48}^b}{d_{25}^b d_{35}^b d_{58}^b}} \tag{7.66}$$

$$\frac{\omega_{1245}^g \omega_{2456}^g \omega_{4568}^g}{\omega_{1345}^g \omega_{3457}^g \omega_{4578}^g} F_{1245}^{b;\beta} F_{2456}^{b;\beta} F_{4568}^{b;\beta} \overline{F}_{1345}^{b;\beta} \overline{F}_{3457}^{b;\beta} \overline{F}_{4578}^{b;\beta} \left| \begin{array}{c} \text{(hexagon diagram with } g_5 \text{ and labels } b_{68}, b_{78}, b_{58}, b_{56}, b_{57}, b_{37}, b_{26}, b_{25}, b_{35}, b_{15}, b_{12}, b_{13}) \end{array} \right\rangle,$$

where $\omega_{ijkl}^g := \omega(g_i, g_{ij}, g_{jk}, g_{kl})$. More concisely, the above Hamiltonian can be expressed as

$$\widehat{\mathsf{h}}_i = \sum_{k \in K} \sum_{b \in B_k} \frac{\dim(b)}{\mathcal{D}_B} \widehat{L}_i^{(b)}, \tag{7.67}$$

where the loop operator $\widehat{L}_i^{(b)}$ for $b \in B_k$ is defined by

$$\widehat{L}_i^{(b)} \left| \begin{array}{c} \text{(hexagon diagram with } g_4) \end{array} \right\rangle = \left| \begin{array}{c} \text{(hexagon diagram with } g_4 k \text{ and loop } b) \end{array} \right\rangle. \tag{7.68}$$

On the right-hand side, the fusion of the loop and the nearby edges is computed by using the following modified $F$-move

$$\boxed{\begin{array}{c} \text{(tree diagram with } b_{ij}, b_{jk}, b_{kl}, g_j, g_k, \overline{\beta}_{ijk}, b_{ik}, \overline{\beta}_{ikl}, g_i, g_l, b_{il}) \end{array} = \omega_{ijkl}^g \sum_{b_{jl}} \sum_{\beta_{ijl}, \beta_{jkl}} F_{ijkl}^{b;\beta} \begin{array}{c} \text{(tree diagram with } b_{ij}, b_{jk}, b_{kl}, g_j, g_k, \overline{\beta}_{jkl}, \overline{\beta}_{ijl}, b_{jl}, g_i, g_l, b_{il}) \end{array}} \tag{7.69}$$

where the 2d regions are labeled by $g_i, g_j, g_k, g_l \in G$. We note that the modified $F$-symbols $\omega^g_{ijkl} F^{b;\beta}_{ijkl}$ satisfy the ordinary pentagon equation due to the twisted pentagon equation for the $F$-symbols. The above model can be embedded into a tensor product state space without breaking the symmetry, as in the case of gapped phases with non-anomalous symmetries.

We expect that the ground state of the above model on an infinite plane is unique in each sector $\mathcal{H}^{(\mu)}$. Thus, the ground states on an infinite plane are in one-to-one correspondence with left $K$-cosets in $G$. The ground state labeled by the left $K$-coset $g^{(\mu)}K$ is given by

$$|\mathrm{GS}; \mu\rangle = \prod_{i \in P} \widehat{\mathsf{h}}_i \, |\{g_i = g^{(\mu)}, b_{ij} = \mathbb{1}, \beta_{ijk} = \mathrm{id}\}\rangle, \tag{7.70}$$

where $g^{(\mu)}$ is a representative of $g^{(\mu)}K$. The right-hand side is computed in the same way as (7.37). Recalling that the $F$-symbols are modified as in (7.69), we find

$$\boxed{\begin{aligned}
|\mathrm{GS}; \mu\rangle = & \sum_{\{k_i, b_i, b_{ij}, \beta_{ij}, \beta_{ijk}\}} \prod_{i \in P} \frac{1}{\mathcal{D}_B} \prod_{[ijk] \in V} \left( \frac{d^b_{ij} d^b_{jk}}{d^b_{ik}} \right)^{\frac{1}{4}} \\
& \prod_{[ijk] \in V} \omega(g^{(\mu)}, k_i, k_{ij}, k_{jk})^{s_{ijk}} (F^{b;\beta}_{ijk})^{s_{ijk}} \, |\{g^{(\mu)} k_i, b_{ij}, \beta_{ijk}\}\rangle,
\end{aligned}} \tag{7.71}$$

where $k_{ij} := k_i^{-1} k_j$. The summation on the right-hand side is taken over $k_i \in K$, $b_i \in B_{k_i}$, $b_{ij} \in \overline{b_i} \otimes b_j$, $\beta_{ij} \in \mathrm{Hom}_B(\overline{b_i} \otimes b_j, b_{ij})$, and $\beta_{ijk} \in \mathrm{Hom}_B(b_{ij} \otimes b_{jk}, b_{ik})$. The above equation reduces to the ground state (7.53) of the minimal gapped phase when $B = \mathsf{Vec}^\lambda_K$. The symmetry operators act on the above ground states as

$$U_g \, |\mathrm{GS}; \mu\rangle = |\mathrm{GS}; \nu\rangle, \tag{7.72}$$

where $g \in g^{(\nu)} K (g^{(\mu)})^{-1}$.

### 7.2.4  Tensor Network Representation for Non-Minimal Gapped Phases

When $B$ is multiplicity-free, the ground state (7.71) can be represented by the following tensor network:

$$|\mathrm{GS}; \mu\rangle = \quad \text{} \quad . \tag{7.73}$$

The physical leg on each plaquette takes values in $G$, while the physical leg on each edge takes values in the set of simple objects of $B$. On the other hand, the blue and orange bonds take

values in $K$ and the set of simple objects of $B$, respectively. The non-zero components of the local tensors in (7.73) are given by

$$\text{[diagram]} = \frac{1}{\mathcal{D}_B}\delta_{k,(g^{(\mu)})^{-1}g}, \qquad \text{[diagram]} = 1, \qquad \text{[diagram]} = \delta_{b \in B_k}, \qquad (7.74)$$

$$\text{[diagram]} = \left(\frac{d_{ij}^b d_{jk}^b}{d_{ik}^b}\right)^{\frac{1}{4}} \omega(g^{(\mu)}, k_i, k_{ij}, k_{jk}) F_{ijk}^b, \qquad (7.75)$$

$$\text{[diagram]} = \left(\frac{d_{ij}^b d_{jk}^b}{d_{ik}^b}\right)^{\frac{1}{4}} \omega(g^{(\mu)}, k_i, k_{ij}, k_{jk})^{-1} \overline{F}_{ijk}^b. \qquad (7.76)$$

## 7.3 Gapped Phases with All-Boson Fusion 2-Category Symmetries

In this subsection, we consider the gapped phases obtained by the generalized gauging of the gapped phases with $2\mathsf{Vec}_G^\omega$ symmetry. We will provide concrete lattice Hamiltonians for these gapped phases and write down their ground states explicitly using tensor networks.

### 7.3.1 Minimal Gapped Phases

Let us first discuss the non-minimal gauging of minimal gapped phases with $2\mathsf{Vec}_G^\omega$ symmetry. A minimal gapped phase is specified by a separable algebra of the form[86]

$$B = \mathsf{Vec}_K^\lambda \in 2\mathsf{Vec}_G^\omega. \qquad (7.77)$$

On the other hand, the generalized gauging is specified by an arbitrary $G$-graded $\omega$-twisted unitary fusion category $A \in 2\mathsf{Vec}_G^\omega$.

In the gauged model, possible configurations of the dynamical variables can be described as follows:

- The dynamical variables on the plaquettes are labeled by the representatives of the connected components of $_A(2\mathsf{Vec}_G^\omega)$, which are given by

$$M^g = A\square D_2^g, \qquad g \in S_{H\backslash G}. \qquad (7.78)$$

  We recall that $S_{H\backslash G}$ denotes the set of representatives of right $H$-cosets in $G$. A configuration of these dynamical variables is denoted by $\{g_i \mid i \in P\}$, where $g_i \in S_{H\backslash G}$.

---

[86]A minimal gapped phase of the gauged model generally has topological order, which is in contrast to the minimal gapped phases with $2\mathsf{Vec}_G^\omega$ symmetry.

- The dynamical variables on the edges are labeled by simple objects of $A^{\mathrm{rev}}$. The dynamical variable $a_{ij}$ on edge $[ij]$ must obey the constraint

$$g_i^{-1} h_{ij} g_j \in K, \tag{7.79}$$

where $h_{ij} \in H$ is the grading of $a_{ij}$, that is, $a_{ij} \in A_{h_{ij}}^{\mathrm{rev}}$. This constraint comes from the fact that the vertical surface below edge $[ij]$ is labeled by $D_2^{g_i^{-1} h_{ij} g_j} \in 2\mathsf{Vec}_G^\omega$, which needs to be contained in $B = \mathsf{Vec}_K^\lambda$.

- For a given configuration $\{g_i, a_{ij} \mid i \in P, [ij] \in E\}$, the dynamical variable on vertex $[ijk]$ takes values in $\mathrm{Hom}_{A^{\mathrm{rev}}}(a_{ik}, a_{ij} \otimes a_{jk})$ or $\mathrm{Hom}_{A^{\mathrm{rev}}}(a_{ij} \otimes a_{jk}, a_{ik})$ depending on whether $[ijk]$ is in the A-sublattice or B-sublattice.

A configuration of all dynamical variables on the lattice is denoted by

$$\{g_i, a_{ij}, \alpha_{ijk} \mid i \in P, \ [ij] \in E, \ [ijk] \in V\}. \tag{7.80}$$

The state corresponding to the above configuration is written as

$$|\{g_i, a_{ij}, \alpha_{ijk}\}\rangle = \left| \begin{array}{c} \end{array} \right\rangle. \tag{7.81}$$

The state space of the gauged model is then given by

$$\mathcal{H}_{\mathrm{gauged}} = \widehat{\pi}_{\mathrm{LW}} \mathcal{H}'_{\mathrm{gauged}}, \tag{7.82}$$

where $\mathcal{H}'_{\mathrm{gauged}}$ is the vector space spanned by all possible configurations of the dynamical variables, and $\widehat{\pi}_{\mathrm{LW}} := \prod_{i \in P} \widehat{B}_i$ is the product of the Levin-Wen plaquette operators (6.62) based on $A_e^{\mathrm{rev}}$. The state space (7.82) can be decomposed into the direct sum

$$\mathcal{H}_{\mathrm{gauged}} = \bigoplus_{\mu = 1, 2, \cdots, |H \backslash G / K|} \mathcal{H}_{\mathrm{gauged}}^{(\mu)}, \tag{7.83}$$

where each sector $\mathcal{H}_{\mathrm{gauged}}^{(\mu)}$ consists of states such that all plaquettes are labeled by elements of the same $(H, K)$-double coset $H g^{(\mu)} K$. Here, $g^{(\mu)} \in G$ is a representative of the double coset $H g^{(\mu)} K$.

The Hamiltonian of the gauged model is given by

$$H_{\mathrm{gauged}} = -\sum_i \widehat{\mathsf{h}}_i, \tag{7.84}$$

where each term $\widehat{\mathsf{h}}_i$ is defined by (4.16). A direct computation shows that $\widehat{\mathsf{h}}_i$ acts on a state as

$$
\widehat{\mathsf{h}}_i \left| \begin{array}{c} \\ M^{g_4} \\ \end{array} \right\rangle = \frac{1}{|K|} \sum_{g_5 \in S_{H\backslash G}} \sum_{h_{45} \in H} \delta_{g_{45}^h \in K} \frac{\lambda_{1245}^{g;h} \lambda_{2456}^{g;h} \lambda_{4568}^{g;h}}{\lambda_{1345}^{g;h} \lambda_{3457}^{g;h} \lambda_{4578}^{g;h}} \frac{\Omega_{1345}^{g;h} \Omega_{3457}^{g;h} \Omega_{4578}^{g;h}}{\Omega_{1245}^{g;h} \Omega_{2456}^{g;h} \Omega_{4568}^{g;h}}
$$

$$
\sum_{a_{45} \in A_{h_{45}}^{\mathrm{rev}}} \frac{d_{45}^a}{\mathcal{D}_{A_{h_{45}}}} \sum_{a_{15}, \cdots a_{58}} \sum_{\alpha_{145}, \cdots, \alpha_{458}} \sum_{\alpha_{125}, \cdots, \alpha_{578}} \overline{F}_{1245}^{a;\alpha} \overline{F}_{2456}^{a;\alpha} \overline{F}_{4568}^{a;\alpha}
$$

$$
F_{1345}^{a;\alpha} F_{3457}^{a;\alpha} F_{4578}^{a;\alpha} \sqrt{\frac{d_{15}^a d_{56}^a d_{57}^a}{d_{14}^a d_{46}^a d_{47}^a}} \sqrt{\frac{d_{24}^a d_{34}^a d_{48}^a}{d_{25}^a d_{35}^a d_{58}^a}} \left| \begin{array}{c} \\ M^{g_5} \\ \end{array} \right\rangle ,
$$

(7.85)

where $g_{ij}^h := g_i^{-1} h_{ij} g_j$, $\lambda_{ijkl}^{g;h} := \lambda(g_{ij}^h, g_{jk}^h, g_{kl}^h)$, and $\Omega_{ijkl}^{g;h}$ is defined by (6.32). We recall that $g_{ij}^h \in K$ due to the constraint (7.79).

On an infinite plane, we expect that the Hamiltonian (7.85) has a unique ground state $|\mathrm{GS}; \mu\rangle$ in each sector $\mathcal{H}_{\mathrm{gauged}}^{(\mu)}$. The ground state $|\mathrm{GS}; \mu\rangle$ is obtained by applying the projector $\prod_i \widehat{\mathsf{h}}_i$ to a generic state in the same sector. Concretely, we have

$$
|\mathrm{GS}; \mu\rangle = \prod_i \widehat{\mathsf{h}}_i \left| \{ g_i = g^{(\mu)}, a_{ij} = \mathbb{1}_A, \alpha_{ijk} = \mathrm{id}_{\mathbb{1}_A} \} \right\rangle ,
$$

(7.86)

where $\{ g_i = g^{(\mu)}, a_{ij} = \mathbb{1}_A, \alpha_{ijk} = \mathrm{id}_{\mathbb{1}_A} \}$ denotes the configuration where all plaquettes are labeled by $M^{g^{(\mu)}}$ and all edges and vertices are labeled by the unit object of $A$ and its identity morphism, respectively. A direct computation shows that $|\mathrm{GS}; \mu\rangle$ can be written explicitly as

$$
\boxed{
\begin{array}{c}
|\mathrm{GS}; \mu\rangle = \displaystyle\sum_{\{g_i \in S_{H\backslash G}\}} \sum_{\{k_i \in K\}} \sum_{\{h_i \in H\}} \prod_i \frac{1}{|K|} \delta_{k_i, (g^{(\mu)})^{-1} h_i g_i} \prod_{[ijk]} \lambda(k_i, k_{ij}, k_{jk})^{s_{ijk}} (\overline{\Omega}_{ijk;\mu}^{g;h})^{s_{ijk}} \\[3mm]
\displaystyle\sum_{\{a_i \in A_{h_i}^{\mathrm{rev}}\}} \sum_{\{a_{ij}, \alpha_{ij}, \alpha_{ijk}\}} \prod_i \frac{1}{\mathcal{D}_{A_{h_i}}} \prod_{[ijk]} \left( \frac{d_{ij}^a d_{jk}^a}{d_{ik}^a} \right)^{\frac{1}{4}} (\overline{F}_{ijk}^{a;\alpha})^{s_{ijk}} | \{ g_i, a_{ij}, \alpha_{ijk} \} \rangle ,
\end{array}
}
$$

(7.87)

where $k_{ij} := k_i^{-1} k_j$, $h_{ij} := h_i^{-1} h_j$, and $\overline{\Omega}_{ijk;\mu}^{g;h}$ is defined by

$$
\boxed{
\overline{\Omega}_{ijk;\mu}^{g;h} := (\Omega_{ijk;\mu}^{g;h})^{-1}, \qquad \Omega_{ijk;\mu}^{g;h} := \frac{\omega(h_i, h_{ij}, h_{jk}, g_k)\omega(h_i, g_i, g_{ij}^h, g_{jk}^h)}{\omega(h_i, h_{ij}, g_j, g_{jk}^h)\omega(g^{(\mu)}, (g^{(\mu)})^{-1} h_i g_i, g_{ij}^h, g_{jk}^h)} .
}
$$

(7.88)

We note that $\Omega_{ijk;\mu}^{g;h}$ reduces to $\Omega_{ijk}^{g;h}$ defined by (6.40) when $g^{(\mu)} = e$. The last summation

on the right-hand side of (7.87) is taken over $a_{ij} \in \overline{a_i} \otimes a_j$, $\alpha_{ij} \in \mathrm{Hom}_A(\overline{a_i} \otimes a_j, a_{ij})$, and $\alpha_{ijk} \in \mathrm{Hom}_A(a_{ij} \otimes a_{jk}, a_{ik})$. The derivation of (7.87) will be explained in section 7.3.4.

### 7.3.2 Tensor Network Representation for Minimal Gapped Phases

The ground state (7.87) can be written in terms of a tensor network. In particular, when $A$ is multiplicity-free, the tensor network representation of the ground state (7.87) is given by

$$|\mathrm{GS};\mu\rangle = \quad\text{}\quad . \tag{7.89}$$

The physical leg on each plaquette takes values in $S_{H\backslash G}$, and the physical leg on each edge takes values in the set of simple objects of $A$. On the other hand, the virtual bonds written in green, blue, and orange take values in $H$, $K$, and the set of simple objects of $A$, respectively. The non-zero components of the local tensors in (7.89) are given by

$$\text{} = \frac{1}{|K|} \frac{1}{\mathcal{D}_{A_h}} \delta_{k,(g^{(\mu)})^{-1}hg}, \qquad \text{} = 1, \qquad \text{} = \delta_{a \in A_h}, \tag{7.90}$$

$$\text{} = \lambda(k_i, k_{ij}, k_{jk}) \, \overline{\Omega}_{ijk;\mu}^{g;h}\Big|_{g=h^{-1}g^{(\mu)}k} \left(\frac{d_{ij}^a d_{jk}^a}{d_{ik}^a}\right)^{\frac{1}{4}} \overline{F}_{ijk}^{a;\alpha}, \tag{7.91}$$

$$\text{} = \lambda(k_i, k_{ij}, k_{jk})^{-1} \, \Omega_{ijk;\mu}^{g;h}\Big|_{g=h^{-1}g^{(\mu)}k} \left(\frac{d_{ij}^a d_{jk}^a}{d_{ik}^a}\right)^{\frac{1}{4}} F_{ijk}^{a;\alpha}, \tag{7.92}$$

where

$$\Omega_{ijk;\mu}^{g;h}\Big|_{g=h^{-1}g^{(\mu)}k} = \frac{\omega(h_i, h_{ij}, h_{jk}, h_k^{-1}g^{(\mu)}k_k)\omega(h_i, h_i^{-1}g^{(\mu)}k_i, k_{ij}, k_{jk})}{\omega(h_i, h_{ij}, h_j^{-1}g^{(\mu)}k_j, k_{jk})\omega(g^{(\mu)}, k_i, k_{ij}, k_{jk})}. \tag{7.93}$$

We recall that $h_{ij} := h_i^{-1}h_j$, $k_{ij} := k_i^{-1}k_j$, and $h_i$ is the grading of $a_i$. When $G$ is the trivial group, (7.89) reduces to the tensor network representation of the ground state of the Levin-Wen model based on $A^{\mathrm{rev}}$.[87]

---

[87]When $G$ is trivial, $\omega$ is also trivial and hence $A^{\mathrm{rev}}$ is an ordinary fusion category.

**Minimal Gauging.** In the case of the minimal gauging $A = \mathsf{Vec}_H^\nu$, the ground state (7.87) reduces to

$$\boxed{|\text{GS}; \mu\rangle = \sum_{\{g_i, h_i, k_i\}} \prod_i \frac{1}{|K|} \delta_{k_i, (g^{(\mu)})^{-1} h_i g_i} \prod_{[ijk]} \left( \frac{\lambda(k_i, k_{ij}, k_{jk})}{\Omega_{ijk;\mu}^{g;h} \nu(h_i, h_{ij}, h_{jk})} \right)^{s_{ijk}} |\{g_i, h_{ij}\}\rangle \, ,} \quad (7.94)$$

where $g_i \in S_{H\backslash G}$, $h_i \in H$, $k_i \in K$, and $h_{ij} := h_i^{-1} h_j$. Accordingly, the tensor network representation (7.89) is simplified as

$$|\text{GS}; \mu\rangle = \qquad\qquad . \qquad (7.95)$$

The non-zero components of the local tensors are given by

$$= \frac{1}{|K|} \delta_{k, (g^{(\mu)})^{-1} hg}, \qquad\qquad = 1, \qquad (7.96)$$

$$= \frac{\lambda(k_i, k_{ij}, k_{jk})}{\nu(h_i, h_{ij}, h_{jk})} \frac{\omega(h_i, h_{ij}, h_j^{-1} g^{(\mu)} k_j, k_{jk}) \omega(g^{(\mu)}, k_i, k_{ij}, k_{jk})}{\omega(h_i, h_{ij}, h_{jk}, h_k^{-1} g^{(\mu)} k_k) \omega(h_i, h_i^{-1} g^{(\mu)} k_i, k_{ij}, k_{jk})}, \quad (7.97)$$

$$= \frac{\nu(h_i, h_{ij}, h_{jk})}{\lambda(k_i, k_{ij}, k_{jk})} \frac{\omega(h_i, h_{ij}, h_{jk}, h_k^{-1} g^{(\mu)} k_k) \omega(h_i, h_i^{-1} g^{(\mu)} k_i, k_{ij}, k_{jk})}{\omega(h_i, h_{ij}, h_j^{-1} g^{(\mu)} k_j, k_{jk}) \omega(g^{(\mu)}, k_i, k_{ij}, k_{jk})}, \quad (7.98)$$

where $h_{ij} = h_i^{-1} h_j$ and $k_{ij} = k_i^{-1} k_j$. When $H$ is trivial, the above tensor network representation reduces to (7.55).

### 7.3.3 SPT Phases as Minimal Gapped Phases

Let us consider SPT phases as an example of minimal gapped phases of the gauged model. In general, 2+1d bosonic SPT phases with fusion 2-category symmetry $\mathcal{C}$ are expected to be classified by fiber 2-functors of $\mathcal{C}$,[88] which generalizes the classification of 1+1d SPT phases with fusion 1-category symmetries [26,116]. It is known that a fusion 2-category $\mathcal{C}$ that admits

---

[88] A fiber 2-functor of $\mathcal{C}$ is a tensor 2-functor from $\mathcal{C}$ to 2Vec.

a fiber 2-functor is group-theoretical [115], that is, $\mathcal{C}$ is obtained by the minimal gauging of $2\mathsf{Vec}_G^\omega$. Thus, without loss of generality, $\mathcal{C}$ can be written as

$$\mathcal{C} = C(G, \omega; H, \nu) := {}_{\mathsf{Vec}_H^\nu}(2\mathsf{Vec}_G^\omega)_{\mathsf{Vec}_H^\nu}, \tag{7.99}$$

where $H$ is a subgroup of $G$ and $\nu$ is a 3-cochain on $H$ such that $d\nu = \omega|_H^{-1}$. A fiber 2-functor of $\mathcal{C}(G, \omega; H, \nu)$ is labeled by a pair $(K, \lambda)$, where $K$ is a subgroup of $G$ such that

$$H \cap K = \{e\}, \qquad HK = G \tag{7.100}$$

and $\lambda$ is a 3-cochain on $K$ such that $d\lambda = \omega|_K^{-1}$ [115]. Correspondingly, a 2+1d SPT phase with $\mathcal{C}(G, \omega; H, \nu)$ symmetry should also be labeled by a pair $(K, \lambda)$ that satisfies the above conditions. The SPT phase labeled by such a pair is obtained by choosing

$$B = \mathsf{Vec}_K^\lambda. \tag{7.101}$$

In particular, any SPT phase with fusion 2-category symmetry can be obtained by the minimal gauging of a minimal gapped phase with $2\mathsf{Vec}_G^\omega$ symmetry.

Let us write down the ground state of the SPT phase labeled by $(K, \lambda)$. To this end, we first choose a representative $g^{(\mu)}$ of an $(H, K)$-double coset in $G$. The condition (7.100) implies that the $(H, K)$-double coset in $G$ is unique, whose representative can be chosen to be

$$g^{(\mu)} = e. \tag{7.102}$$

For this choice of the representative, the ground state (7.94) reduces to

$$|\text{GS}\rangle = \sum_{\{g_i \in S_{H\backslash G}\}} \sum_{\{k_i \in K\}} \sum_{\{h_i \in H\}} \prod_i \frac{1}{|K|} \delta_{k_i, h_i g_i} \prod_{[ijk]} \left( \frac{\lambda(k_i, k_{ij}, k_{jk})}{\Omega_{ijk}^{g;h} \nu(h_i, h_{ij}, h_{jk})} \right)^{s_{ijk}} |\{g_i, h_{ij}\}\rangle, \tag{7.103}$$

where $\Omega_{ijk}^{g;h}$ is defined by (6.40). Furthermore, the condition (7.100) also implies that any right $H$-coset in $G$ contains a unique element of $K$. Thus, any element $g \in S_{H\backslash G}$ can be uniquely decomposed as

$$g = h(k)^{-1} k, \qquad k \in K, \tag{7.104}$$

where $h(k) \in H$.[89] Using the above decomposition, one can compute the right-hand side of (7.103) as

$$\boxed{|\text{GS}\rangle = \sum_{\{k_i \in K\}} \prod_i \frac{1}{|K|} \prod_{[ijk]} \left( \frac{\lambda(k_i, k_{ij}, k_{jk})}{\Omega_{ijk}^{g;h} \nu(h_i, h_{ij}, h_{jk})} \right)^{s_{ijk}} |\{g_i, h_{ij}\}\rangle,} \tag{7.105}$$

---

[89]The map $h : K \to H$ is determined by the choice of representatives of right $H$-cosets in $G$. We emphasize that the representatives of right $H$-cosets in $G$ are chosen irrespective of $K$. In other words, the set $S_{H\backslash G}$ is independent of $K$.

where $g_i := h_i^{-1} k_i$ and $h_i := h(k_i)$.

The ground state (7.105) can be represented by the following tensor network:

$$|\text{GS}\rangle = \quad \text{} \quad . \tag{7.106}$$

The physical leg on each plaquette takes values in $S_{H\backslash G}$, while the physical leg on each edge takes values in $H$. On the other hand, the virtual bonds written in red and blue take values in $H$ and $K$, respectively. The non-zero components of the local tensors in (7.106) are given by

$$\text{} = \frac{1}{|K|}, \qquad \text{} = 1, \tag{7.107}$$

$$\text{} = \frac{\lambda(k_i, k_{ij}, k_{jk})}{\nu(h_i, h_{ij}, h_{jk})} \frac{\omega(h_i, h_{ij}, h_j^{-1}k_j, k_{jk})}{\omega(h_i, h_{ij}, h_{jk}, h_k^{-1}k_k)\omega(h_i, h_i^{-1}k_i, k_{ij}, k_{jk})}, \tag{7.108}$$

$$\text{} = \frac{\nu(h_i, h_{ij}, h_{jk})}{\lambda(k_i, k_{ij}, k_{jk})} \frac{\omega(h_i, h_{ij}, h_{jk}, h_k^{-1}k_k)\omega(h_i, h_i^{-1}k_i, k_{ij}, k_{jk})}{\omega(h_i, h_{ij}, h_j^{-1}k_j, k_{jk})}, \tag{7.109}$$

where $h_i = h(k_i)$, $h_{ij} = h_i^{-1}h_j$, and $k_{ij} = k_i^{-1}k_j$.

### 7.3.4 Non-Minimal Gapped Phases

We now move on to the non-minimal gauging of the non-minimal gapped phases with $2\mathsf{Vec}_G^\omega$ symmetry. The separable algebra specifying a non-minimal gapped phase is denoted by $B$, which is a $G$-graded $\omega$-twisted unitary fusion category faithfully graded by $K \subset G$. On the other hand, the separable algebra specifying the generalized gauging is denoted by $A$, which is a $G$-graded $\omega$-twisted unitary fusion category faithfully graded by $H \subset G$.

In the gauged model, possible configurations of the dynamical variables can be described as follows:

- The dynamical variables on the plaquettes are labeled by the representatives (7.78) of the connected components of $_A(2\mathsf{Vec}_G^\omega)$. A configuration of these dynamical variables is denoted by $\{g_i \mid i \in P\}$, where $g_i \in S_{H\backslash G}$.

- The dynamical variable on each edge $[ij]$ is labeled by a pair of simple objects $a_{ij} \in A^{\text{rev}}$ and $b_{ij} \in B$. These objects must obey the constraint

$$g_i^{-1} h_{ij} g_j = k_{ij}, \tag{7.110}$$

where $h_{ij} \in H$ is the grading of $a_{ij} \in A_{h_{ij}}^{\text{rev}}$ and $k_{ij} \in K$ is the grading of $b_{ij} \in B_{k_{ij}}$.

- The dynamical variable on each vertex $[ijk]$ takes values in $\text{Hom}_{A^{\text{rev}}}(a_{ik}, a_{ij} \otimes a_{jk}) \otimes \text{Hom}_B(b_{ik}, b_{ij} \otimes b_{jk})$ or its dual depending on whether $[ijk]$ is in the A-sublattice or B-sublattice.

A configuration of all dynamical variables on the lattice is denoted by

$$\{g_i, (a,b)_{ij}, (\alpha,\beta)_{ijk} \mid i \in P, \ [ij] \in E, \ [ijk] \in V\}. \tag{7.111}$$

The state corresponding to the above configuration is written as

$$|\{g_i, (a,b)_{ij}, (\alpha,\beta)_{ijk}\}\rangle = \tag{7.112}$$

where $(a,b)_{ij}$ is a pair of $a_{ij}$ and $b_{ij}$, and $(\overline{\alpha}^{\text{rev}}, \overline{\beta})_{ijk}$ is a pair of $\overline{\alpha}_{ijk}^{\text{rev}}$ and $\overline{\beta}_{ijk}$. The state space of the gauged model is given by

$$\mathcal{H}_{\text{gauged}} = \widehat{\pi}_{\text{LW}} \mathcal{H}'_{\text{gauged}}, \tag{7.113}$$

where $\mathcal{H}'_{\text{gauged}}$ is the vector space spanned by all possible configurations of the dynamical variables, and $\widehat{\pi}_{\text{LW}} := \prod_{i \in P} \widehat{B}_i$ is the product of the Levin-Wen plaquette operators (6.62) based on $A_e^{\text{rev}}$.[90]

The Hamiltonian of the gauged model is given by

$$H_{\text{gauged}} = -\sum_i \widehat{\mathsf{h}}_i, \tag{7.114}$$

where the local term $\widehat{\mathsf{h}}_i$ is defined by (4.16). Using the concrete data of the separable algebras

---

[90]We note that the projector $\widehat{\pi}_{\text{LW}}$ acts only on $a_{ij}$'s and $\alpha_{ijk}$'s.

$A, B \in 2\mathsf{Vec}_G^\omega$, we find

$$
\widehat{\mathsf{h}}_i \left| \begin{array}{c} {\scriptstyle (a,b)_{68}} \quad {\scriptstyle (a,b)_{78}} \\ {\scriptstyle (a,b)_{48}} \\ {\scriptstyle (a,b)_{46}} \quad {\scriptstyle (a,b)_{47}} \\ {\scriptstyle (a,b)_{26}} \longleftarrow \bullet \quad M^{g_4} \quad \bullet \longleftarrow {\scriptstyle (a,b)_{37}} \\ {\scriptstyle (a,b)_{24}} \quad {\scriptstyle (a,b)_{34}} \\ {\scriptstyle (a,b)_{14}} \\ {\scriptstyle (a,b)_{12}} \quad {\scriptstyle (a,b)_{13}} \end{array} \right\rangle = \sum_{g_5 \in S_{H\backslash G}} \sum_{h_{45} \in H} \sum_{k_{45} \in K} \delta_{k_{45}, g_{45}^h} \sum_{a_{45} \in A_{h_{45}}^{\mathrm{rev}}} \frac{d_{45}^a}{\mathcal{D}_{A_{h_{45}}}} \sum_{b_{45} \in B_{k_{45}}} \frac{d_{45}^b}{\mathcal{D}_B}
$$

$$
\sum_{a_{15},\cdots,a_{58}} \sum_{\alpha_{145},\cdots,\alpha_{458}} \sum_{\alpha_{125},\cdots,\alpha_{578}} \sum_{b_{15},\cdots,b_{58}} \sum_{\beta_{145},\cdots,\beta_{458}} \sum_{\beta_{125},\cdots,\beta_{578}}
$$

$$
\sqrt{\frac{d_{15}^a d_{56}^a d_{57}^a}{d_{14}^a d_{46}^a d_{47}^a}} \sqrt{\frac{d_{24}^a d_{34}^a d_{48}^a}{d_{25}^a d_{35}^a d_{58}^a}} \, {}^A\overline{F}_{1245}^{a;\alpha} \, {}^A\overline{F}_{2456}^{a;\alpha} \, {}^A\overline{F}_{4568}^{a;\alpha} \, {}^A F_{1345}^{a;\alpha} \, {}^A F_{3457}^{a;\alpha} \, {}^A F_{4578}^{a;\alpha}
$$

$$
\sqrt{\frac{d_{15}^b d_{56}^b d_{57}^b}{d_{14}^b d_{46}^b d_{47}^b}} \sqrt{\frac{d_{24}^b d_{34}^b d_{48}^b}{d_{25}^b d_{35}^b d_{58}^b}} \, {}^B F_{1245}^{b;\beta} \, {}^B F_{2456}^{b;\beta} \, {}^B F_{4568}^{b;\beta} \, {}^B\overline{F}_{1345}^{b;\beta} \, {}^B\overline{F}_{3457}^{b;\beta} \, {}^B\overline{F}_{4578}^{b;\beta}
$$

$$
\frac{\Omega_{1345}^{g;h} \Omega_{3457}^{g;h} \Omega_{4578}^{g;h}}{\Omega_{1245}^{g;h} \Omega_{2456}^{g;h} \Omega_{4568}^{g;h}} \left| \begin{array}{c} {\scriptstyle (a,b)_{68}} \quad {\scriptstyle (a,b)_{78}} \\ {\scriptstyle (a,b)_{58}} \\ {\scriptstyle (a,b)_{56}} \quad {\scriptstyle (a,b)_{57}} \\ {\scriptstyle (a,b)_{26}} \longrightarrow \bullet \quad M^{g_5} \quad \bullet \longleftarrow {\scriptstyle (a,b)_{37}} \\ {\scriptstyle (a,b)_{25}} \quad {\scriptstyle (a,b)_{35}} \\ {\scriptstyle (a,b)_{15}} \\ {\scriptstyle (a,b)_{12}} \quad {\scriptstyle (a,b)_{13}} \end{array} \right\rangle ,
$$

$$(7.115)$$

where $g_{ij}^h := g_i^{-1} h_{ij} g_j$, and ${}^A F$ and ${}^B F$ are the $F$-symbols of $A$ and $B$. More concisely, the above Hamiltonian can also be expressed as

$$
\widehat{\mathsf{h}}_i \left| \begin{array}{c} M^{g_4} \end{array} \right\rangle = \sum_{g_5 \in S_{H\backslash G}} \sum_{h_{45} \in H} \sum_{a_{45} \in A_{h_{45}}^{\mathrm{rev}}} \frac{d_{45}^a}{\mathcal{D}_{A_{h_{45}}}} \sum_{k_{45} \in K} \sum_{b_{45} \in B_{k_{45}}} \frac{d_{45}^b}{\mathcal{D}_B} \delta_{k_{45}, g_{45}^h} \left| \begin{array}{c} M^{g_5} \\ {\scriptstyle (a,b)_{45}} \end{array} \right\rangle .
$$

$$(7.116)$$

The action of the loop operator on the right-hand side is computed by using the modified $F$-move

$$
\boxed{
\begin{array}{c} {\scriptstyle (a,b)_{ij}} \; {\scriptstyle (a,b)_{jk}} \; {\scriptstyle (a,b)_{kl}} \\ g_j \\ {\scriptstyle (\overline{\alpha}^{\mathrm{rev}}, \overline{\beta})_{ijk}} \quad g_k \\ {\scriptstyle (a,b)_{ik}} \quad {\scriptstyle (\overline{\alpha}^{\mathrm{rev}}, \overline{\beta})_{ikl}} \\ g_i \qquad g_l \\ {\scriptstyle (a,b)_{il}} \end{array}
= \sum_{(a,b)_{jl}} \sum_{(\alpha,\beta)_{ijl}} \sum_{(\alpha,\beta)_{jkl}} \overline{\Omega}_{ijkl}^{g;h} \, {}^A\overline{F}_{ijkl}^{a;\alpha} \, {}^B F_{ijkl}^{b;\beta}
\begin{array}{c} {\scriptstyle (a,b)_{ij}} \; {\scriptstyle (a,b)_{jk}} \; {\scriptstyle (a,b)_{kl}} \\ g_k \\ g_j \quad {\scriptstyle (\overline{\alpha}^{\mathrm{rev}}, \overline{\beta})_{jkl}} \\ {\scriptstyle (\overline{\alpha}^{\mathrm{rev}}, \overline{\beta})_{ijl}} \quad {\scriptstyle (a,b)_{jl}} \\ g_i \qquad g_l \\ {\scriptstyle (a,b)_{il}} \end{array} ,
} \quad (7.117)
$$

where $g_i \in S_{H\backslash G}$, $a_{ij} \in A_{h_{ij}}^{\mathrm{rev}}$, $b_{ij} \in B_{k_{ij}}$, and $k_{ij} = g_{ij}^h$. The modified $F$-symbols $\overline{\Omega}_{ijkl}^{g;h} \, {}^A\overline{F}_{ijkl}^{a;\alpha} \, {}^B F_{ijkl}^{b;\beta}$ obey the ordinary pentagon equation due to the twisted pentagon equation (6.13) and (6.33).

As in the case of the minimal gapped phases, we expect that the ground states of the above Hamiltonian on an infinite plane are in one-to-one correspondence with $(H, K)$-double

cosets in $G$. The ground state labeled by the double coset $Hg^{(\mu)}K$ is given by

$$|\text{GS};\mu\rangle = \prod_{i \in P} \widehat{\mathsf{h}}_i \, |\{g_i = g^{(\mu)}, (a,b)_{ij} = (\mathbb{1}_A, \mathbb{1}_B), (\alpha,\beta)_{ijk} = (\text{id}_{\mathbb{1}_A}, \text{id}_{\mathbb{1}_B})\}\rangle, \qquad (7.118)$$

where $g^{(\mu)}$ is the representative of $Hg^{(\mu)}K$. The right-hand side can be computed in the same way as (7.37). Using the modified $F$-symbols (7.117), we find

$$
\begin{aligned}
|\text{GS};\mu\rangle = & \sum_{\{g_i \in S_{H\backslash G}\}} \sum_{\{h_i \in H\}} \sum_{\{k_i \in K\}} \prod_i \delta_{k_i, (g^{(\mu)})^{-1}h_i g_i} \prod_i \frac{1}{\mathcal{D}_{A_{h_i}}} \prod_i \frac{1}{\mathcal{D}_B} \\
& \sum_{\{a_i\}} \sum_{\{a_{ij}\}} \sum_{\{\alpha_{ij}\}} \sum_{\{\alpha_{ijk}\}} \sum_{\{b_i\}} \sum_{\{b_{ij}\}} \sum_{\{\beta_{ij}\}} \sum_{\{\beta_{ijk}\}} \prod_{[ijk]} \left( \frac{d^a_{ij} d^a_{jk}}{d^a_{ik}} \right)^{\frac{1}{4}} \left( \frac{d^b_{ij} d^b_{jk}}{d^b_{ik}} \right)^{\frac{1}{4}} \\
& \prod_{[ijk]} (\overline{\Omega}^{g;h}_{ijk;\mu})^{s_{ijk}} (^A\overline{F}^{a;\alpha}_{ijk})^{s_{ijk}} (^B F^{b;\beta}_{ijk})^{s_{ijk}} \, |\{g_i, (a,b)_{ij}, (\alpha,\beta)_{ijk}\}\rangle,
\end{aligned}
\qquad (7.119)
$$

where $\Omega^{g;h}_{ijk;\mu}$ is defined by (7.88). The summation on the second line is taken over $a_i \in A^{\text{rev}}_{h_i}$, $a_{ij} \in \overline{a_i} \otimes a_j$, $\alpha_{ij} \in \text{Hom}_A(\overline{a_i} \otimes a_j, a_{ij})$, $\alpha_{ijk} \in \text{Hom}_A(a_{ij} \otimes a_{jk}, a_{ik})$, $b_i \in B_{k_i}$, $b_{ij} \in \overline{b_i} \otimes b_j$, $\beta_{ij} \in \text{Hom}_B(\overline{b_i} \otimes b_j, b_{ij})$, and $\beta_{ijk} \in \text{Hom}_B(b_{ij} \otimes b_{jk}, b_{ik})$. The above equation reduces to (7.87) when $B = \mathsf{Vec}^\lambda_K$.[91]

### 7.3.5 Tensor Network Representation for Non-Minimal Gapped Phases

The ground state (7.119) can be written in terms of a tensor network. In particular, when both $A$ and $B$ are multiplicity-free and the twist $\omega$ is trivial, the ground state $|\text{GS};\mu\rangle$ can be represented by the following double-layered tensor network:

$$|\text{GS};\mu\rangle = \quad$$ 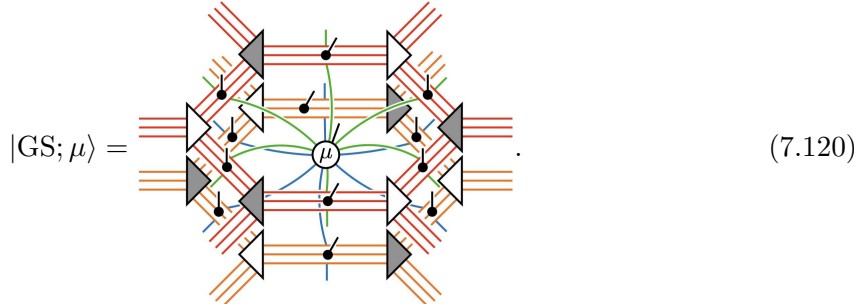 $$. \qquad (7.120)$$

Here, the physical leg on each plaquette takes values in $S_{H\backslash G}$, the physical leg on each edge of the top layer takes values in the set of simple objects of $A$, and the physical leg on each edge of the bottom layer takes values in the set of simple objects of $B$. On the other hand, the virtual bonds written in green, blue, red, and orange take values in $H$, $K$, the set of simple

---

[91]When $B = \mathsf{Vec}^\lambda_K$, the dynamical variable $b_{ij}$ on each edge is uniquely determined by the constraint (7.110).

objects of $A$, and the set of simple objects of $B$, respectively. The non-zero components of each tensor in (7.120) are given as follows:

$$\vcenter{\hbox{\includegraphics{fig1}}} = \frac{1}{\mathcal{D}_{A_h}}\frac{1}{\mathcal{D}_B}\delta_{k,(g^{(\mu)})^{-1}hg}, \qquad \vcenter{\hbox{\includegraphics{fig2}}} = \vcenter{\hbox{\includegraphics{fig3}}} = 1, \tag{7.121}$$

$$\vcenter{\hbox{\includegraphics{fig4}}} = \delta_{a\in A_h}, \qquad \vcenter{\hbox{\includegraphics{fig5}}} = \delta_{b\in B_k}, \tag{7.122}$$

$$\vcenter{\hbox{\includegraphics{fig6}}} = \left(\frac{d^a_{ij}d^a_{jk}}{d^a_{ik}}\right)^{\frac{1}{4}} {}^A\overline{F}^a_{ijk}, \qquad \vcenter{\hbox{\includegraphics{fig7}}} = \left(\frac{d^a_{ij}d^a_{jk}}{d^a_{ik}}\right)^{\frac{1}{4}} {}^A F^a_{ijk}, \tag{7.123}$$

$$\vcenter{\hbox{\includegraphics{fig8}}} = \left(\frac{d^b_{ij}d^b_{jk}}{d^b_{ik}}\right)^{\frac{1}{4}} {}^B F^b_{ijk}, \qquad \vcenter{\hbox{\includegraphics{fig9}}} = \left(\frac{d^b_{ij}d^b_{jk}}{d^b_{ik}}\right)^{\frac{1}{4}} {}^B\overline{F}^b_{ijk}. \tag{7.124}$$

When $\omega$ is non-trivial, the above tensor network representation is slightly modified. Specifically, when $\omega$ is non-trivial, the vertex tensors of the top and bottom layers carry additional $H$-valued and $K$-valued legs as follows:

$$\vcenter{\hbox{\includegraphics{fig10}}} = \left(\frac{d^a_{ij}d^a_{jk}}{d^a_{ik}}\right)^{\frac{1}{4}} {}^A\overline{F}^{a;\alpha}_{ijk}, \qquad \vcenter{\hbox{\includegraphics{fig11}}} = \left(\frac{d^a_{ij}d^a_{jk}}{d^a_{ik}}\right)^{\frac{1}{4}} {}^A F^{a;\alpha}_{ijk}, \tag{7.125}$$

$$\vcenter{\hbox{\includegraphics{fig12}}} = \left(\frac{d^b_{ij}d^b_{jk}}{d^b_{ik}}\right)^{\frac{1}{4}} {}^B F^{b;\beta}_{ijk}, \qquad \vcenter{\hbox{\includegraphics{fig13}}} = \left(\frac{d^a_{ij}d^a_{jk}}{d^a_{ik}}\right)^{\frac{1}{4}} {}^B\overline{F}^{b;\beta}_{ijk}. \tag{7.126}$$

Here, $h_i$ is the grading of $a_i$ and $k_i$ is the grading of $b_i$.[92] The additional legs on the top and bottom layers are then connected to each other via the following tensors, which we put in the middle of the two layers:

$$\vcenter{\hbox{\includegraphics{fig14}}} = \overline{\Omega}^{g;h}_{ijk;\mu}\Big|_{g=h^{-1}g^{(\mu)}k} = \frac{\omega(h_i,h_{ij},h_j^{-1}g^{(\mu)}k_j,k_{jk})\omega(g^{(\mu)},k_i,k_{ij},k_{jk})}{\omega(h_i,h_{ij},h_{jk},h_k^{-1}g^{(\mu)}k_k)\omega(h_i,h_i^{-1}g^{(\mu)}k_i,k_{ij},k_{jk})}, \tag{7.127}$$

---

[92]The tensors evaluate to zero when $h_i$ and $k_i$ do not agree with the grading of $a_i$ and that of $b_i$.

$$\begin{gathered}\raisebox{-1.5em}{\begin{array}{c}h_k\\k_k\ \raisebox{0.3em}{$\triangleright$}\ h_j\\h_i\ \ \ k_j\\k_i\end{array}} = \left.\Omega^{g;h}_{ijk;\mu}\right|_{g=h^{-1}g^{(\mu)}k} = \frac{\omega(h_i, h_{ij}, h_{jk}, h_k^{-1}g^{(\mu)}k_k)\omega(h_i, h_i^{-1}g^{(\mu)}k_i, k_{ij}, k_{jk})}{\omega(h_i, h_{ij}, h_j^{-1}g^{(\mu)}k_j, k_{jk})\omega(g^{(\mu)}, k_i, k_{ij}, k_{jk})},\end{gathered} \quad (7.128)$$

where $h_{ij} = h_i^{-1}h_j$ and $k_{ij} = k_i^{-1}k_j$. With the above modification, we obtain the tensor network representation of the ground state (7.119) for general $\omega$.

The above tensor network reduces to (7.89) when $B = \mathsf{Vec}_K^\lambda$. Similarly, when $A = \mathsf{Vec}$, the above tensor network reduces to (7.73).

## Acknowledgments

We thank Lakshya Bhardwaj for several initial discussions, in particular on section 5.3.4 and collaborations on related projects. We thank Maissam Barkeshli, Thibault Décoppet, Luisa Eck, Alison Warman and Matt Yu for discussions. SH, SSN, and AT thank the KITP for hospitality during the completion of this work. The work of SH and SSN is supported by the UKRI Frontier Research Grant, underwriting the ERC Advanced Grant "Generalized Symmetries in Quantum Field Theory and Quantum Gravity". The work of KI and SSN is supported in part by the EPSRC Open Fellowship EP/X01276X/1 (Schafer-Nameki). KI acknowledges support through the Leverhulme-Peierls Fellowship funded by the Leverhulme Trust. The work of AT is funded by Villum Fonden Grant no. VIL60714. This research was supported in part by grant NSF PHY-2309135 to the Kavli Institute for Theoretical Physics (KITP).

## A    Summary of Notations

Throughout the paper, we use the following notations:

- $G$: a finite group

- $\omega$: a 4-cocycle on $G$, which we choose to be trivial in sections 2, 5, and 7.1

- $H$, $K$: subgroups of $G$

- $\nu$: a 3-cochain on $H$ such that $d\nu = \omega|_H^{-1}$, where $\omega|_H$ is the restriction of $\omega$ to $H$

- $\lambda$: a 3-cochain on $K$ such that $d\lambda = \omega|_K^{-1}$

- $S_{H\backslash G}$: the set of representatives of right $H$-cosets in $G$

- $g_{ij}^h$: an element of $G$ defined by $g_i^{-1}h_{ij}g_j$, where $g_i, g_j \in S_{H\backslash G}$ and $h_{ij} \in H$

- $\Omega_{ijkl}^{g;h}$, $\Omega_{ijk}^{g;h}$: the phase factors defined by (6.32) and (6.40), respectively

- $A$: a $G$-graded $\omega$-twisted fusion category, which is faithfully graded by $H$

- $B$: a $G$-graded $\omega$-twisted fusion category, which is faithfully graded by $K$[93]

- $A^{\mathrm{rev}}$: the reverse category of $A$, i.e., $A$ with the reversed morphisms

- $A^{\mathrm{op}}$: the opposite category of $A$, i.e., $A$ with the opposite tensor product

- $\sum_{a \in A}$: the sum over all (isomorphism classes of) simple objects of $A$

- $\sum_{\alpha \in \mathrm{Hom}_A(a,a')}$: the sum over all basis elements of $\mathrm{Hom}_A(a, a')$

- $F_{ijkl}^{a;\alpha}$, $F_{ijk}^{a;\alpha}$: the $F$-symbols of $A$ defined by (6.11) and (6.39), respectively

- $d_{ij}^a$: the quantum dimension of $a_{ij} \in A$

- $\mathcal{D}_{A_h}$: the total dimension of $A_h$ defined by $\sum_{a \in A_h} \dim(a)^2$

- $s_{ijk}$: the sign defined by (5.90)

- $P$: the set of plaquettes of the honeycomb lattice

- $E$: the set of edges of the honeycomb lattice

- $V$: the set of vertices of the honeycomb lattice

- $\widehat{\pi}_{\mathrm{LW}}$: the product of Levin-Wen projectors (2.15) on all plaquettes

- $\widehat{\pi}_{\mathrm{Gauss}}$: the product of generalized Gauss law operators (2.24) on all plaquettes

- $\widehat{\pi}_{\mathrm{flat}}$: the projector that imposes the flatness condition on gauge fields

- $\widehat{\pi}_{\mathrm{fusion}}$: the projector that imposes the fusion constraint on the edge degrees of freedom

- $2\mathsf{Vec}_G^\omega$: the 2-category of finite semisimple $G$-graded categories with the 10-j symbol $\omega$

- ${}_A(2\mathsf{Vec}_G^\omega)$: the 2-category of left $A$-modules in $2\mathsf{Vec}_G^\omega$

- ${}_A(2\mathsf{Vec}_G^\omega)_A$: the 2-category of $(A, A)$-bimodules in $2\mathsf{Vec}_G^\omega$

- $D_2^g$: a simple object of $2\mathsf{Vec}_G^\omega$, where $g \in G$

- $M^g = A \square D_2^g$: a representative of a connected component of ${}_A(2\mathsf{Vec}_G^\omega)$, where $g \in S_{H \backslash G}$

---

[93] In sections 3 and 4, $A$ and $B$ denote separable algebras in a general fusion 2-category $\mathcal{C}$.

# B    More on $_A(2\mathsf{Vec}_G^\omega)$

In this appendix, we describe some details of the module 2-category $_A(2\mathsf{Vec}_G^\omega)$. We will follow the notations and conventions employed in sections 5.1 and 6.1.

## B.1    Category of 1- and 2-Morphisms

We show the monoidal equivalence

$$\mathrm{End}_{A(2\mathsf{Vec}_G^\omega)}(A\square D_2^g) \cong A_e^{\mathrm{op}} \tag{B.1}$$

and the equivalence of $(A_e^{\mathrm{op}}, A_e^{\mathrm{op}})$-bimodule categories

$$\mathrm{Hom}_{A(2\mathsf{Vec}_G^\omega)}(A\square D_2^g, A\square D_2^{g'}) \cong A_{g(g')^{-1}}^{\mathrm{op}}. \tag{B.2}$$

Here, the monoidal structure on $\mathrm{End}_{A(2\mathsf{Vec}_G^\omega)}(A\square D_2^g)$ is defined by the composition of 1-morphisms. Similarly, the $(A_e^{\mathrm{op}}, A_e^{\mathrm{op}})$-bimodule structure on $\mathrm{Hom}_{A(2\mathsf{Vec}_G^\omega)}(A\square D_2^g, A\square D_2^{g'})$ is defined by using the composition of 1-morphisms together with the monoidal equivalence $A_e^{\mathrm{op}} \cong \mathrm{End}_{A(2\mathsf{Vec}_G^\omega)}(A\square D_2^g) \cong \mathrm{End}_{A(2\mathsf{Vec}_G^\omega)}(A\square D_2^{g'})$.

Let us begin with the case where $\omega = 1$. We first show (B.1) with $g = e$, i.e., we show the equivalence of monoidal categories

$$\mathrm{End}_{A(2\mathsf{Vec}_G)}(A) \cong A_e^{\mathrm{op}}. \tag{B.3}$$

To this end, we recall that $A^{\mathrm{op}}$ is monoidally equivalent to the category $\mathrm{End}_{A(2\mathsf{Vec})}(A)$ of left $A$-module endofunctors of $A$. The monoidal equivalence between $A^{\mathrm{op}}$ and $\mathrm{End}_{A(2\mathsf{Vec})}(A)$ is given by [159, Example 7.12.3]

$$\begin{aligned} F : A^{\mathrm{op}} &\xrightarrow{\cong} \mathrm{End}_{A(2\mathsf{Vec})}(A) \\ a &\mapsto F(a) := - \otimes a. \end{aligned} \tag{B.4}$$

The functor $F(a) : A \to A$ preserves the $G$-grading if and only if $a \in A_e^{\mathrm{op}}$. Thus, $F$ gives a monoidal equivalence between the subcategories $A_e^{\mathrm{op}} \subset A^{\mathrm{op}}$ and $\mathrm{End}_{A(2\mathsf{Vec}_G)}(A) \subset \mathrm{End}_{A(2\mathsf{Vec})}(A)$. This shows (B.3) with $\omega = 1$.

When $g \neq e$, the 1-category $A\square D_2^g$ is not equivalent to $A$ as a left $A$-module in $2\mathsf{Vec}_G$. Nevertheless, $A\square D_2^g$ is still equivalent to $A$ as a left $A$-module category, i.e., they are equivalent as left $A$-modules in $2\mathsf{Vec}$. Therefore, the endomorphism 1-category $\mathrm{End}_{A(2\mathsf{Vec})}(A\square D_2^g)$ is monoidally equivalent to $\mathrm{End}_{A(2\mathsf{Vec})}(A)$, which is also monoidally equivalent to $A^{\mathrm{op}}$. The monoidal equivalence between $A^{\mathrm{op}}$ and $\mathrm{End}_{A(2\mathsf{Vec})}(A\square D_2^g)$ is given by

$$\begin{aligned} F_g : A^{\mathrm{op}} &\xrightarrow{\cong} \mathrm{End}_{A(2\mathsf{Vec})}(A\square D_2^g) \\ a &\mapsto F_g(a) := \Phi_g(\Phi_g^{-1}(-) \otimes a). \end{aligned} \tag{B.5}$$

Here, $\Phi_g : A \to A\square D_2^g$ is the functor that shifts the grading by $g \in G$, and $\Phi_g^{-1}$ is its inverse. We note that when $a \in A$ has the grading $h \in H$, its image $\Phi_g(a) \in A\square D_2^g$ has the grading $hg \in G$. The functor $F_g(a) : A\square D_2^g \to A\square D_2^g$ preserves the $G$-grading if and only if $a \in A_e^{\mathrm{op}}$. Thus, $F_g$ gives a monoidal equivalence between subcategories $A_e^{\mathrm{op}} \subset A^{\mathrm{op}}$ and $\mathrm{End}_{A(2\mathsf{Vec}_G)}(A\square D_2^g) \subset \mathrm{End}_{A(2\mathsf{Vec})}(A\square D_2^g)$. This shows (B.1) with $\omega = 1$.

Finally, to show the equivalence (B.2), we note that there is an equivalence of $(A^{\mathrm{op}}, A^{\mathrm{op}})$-bimodule categories

$$
\begin{array}{rcl}
F_{g,g'} : A^{\mathrm{op}} & \overset{\cong}{\Rightarrow} & \mathrm{Hom}_{A(2\mathsf{Vec})}(A\square D_2^g, A\square D_2^{g'}) \\
a & \mapsto & F_{g,g'}(a) := \Phi_{g'}(\Phi_g^{-1}(-) \otimes a).
\end{array}
\tag{B.6}
$$

The functor $F_{g,g'}(a)$ preserves the $G$-grading if and only if $a \in A_{g(g')^{-1}}^{\mathrm{op}}$. Therefore, $F_{g,g'}$ restricted to the subcategory $A_{g(g')^{-1}}^{\mathrm{op}} \subset A^{\mathrm{op}}$ gives an $(A_e^{\mathrm{op}}, A_e^{\mathrm{op}})$-bimodule equivalence between $A_{g(g')^{-1}}^{\mathrm{op}}$ and $\mathrm{Hom}_{A(2\mathsf{Vec}_G)}(A\square D_2^g, A\square D_2^{g'})$, which shows (B.2) with $\omega = 1$.

The above equivalences can be generalized to the case where $\omega \neq 1$. Specifically, the monoidal equivalence (B.1) for a general $\omega$ is given by

$$
\begin{array}{rcl}
A_e^{\mathrm{op}} & \overset{\cong}{\Rightarrow} & \mathrm{End}_{A(2\mathsf{Vec}_G^\omega)}(A\square D_2^g) \\
a & \mapsto & \Phi_g(\Phi_g^{-1}(-) \otimes a),
\end{array}
\tag{B.7}
$$

where $\Phi_g : A \to A\square D_2^g$ is again the functor that shifts the grading by $g \in G$. The inverse of the equivalence (B.7) is given by the functor that maps $f \in \mathrm{End}_{A(2\mathsf{Vec}_G^\omega)}(A\square D_2^g)$ to $\Phi_g^{-1}(f(\Phi_g(\mathbb{1}))) \in A_e^{\mathrm{op}}$, where $\mathbb{1}$ is the unit object of $A$. Similarly, the equivalence (B.2) of $(A_e^{\mathrm{op}}, A_e^{\mathrm{op}})$-bimodule categories is given by

$$
\begin{array}{rcl}
A_{g(g')^{-1}}^{\mathrm{op}} & \overset{\cong}{\Rightarrow} & \mathrm{Hom}_{A(2\mathsf{Vec}_G^\omega)}(A\square D_2^g, A\square D_2^{g'}) \\
a & \mapsto & \Phi_{g'}(\Phi_g^{-1}(-) \otimes a).
\end{array}
\tag{B.8}
$$

The inverse of this equivalence is given by the functor that maps $f \in \mathrm{Hom}_{A(2\mathsf{Vec}_G^\omega)}(A\square D_2^g, A\square D_2^{g'})$ to $\Phi_{g'}^{-1}(f(\Phi_g(\mathbb{1}))) \in A_{g(g')^{-1}}^{\mathrm{op}}$.

## B.2 Natural Isomorphisms Associated with Objects and Morphisms

We describe the natural isomorphisms associated with the object (6.35) and 1-morphism (6.36) of $_A(2\mathsf{Vec}_G^\omega)$. In addition, using these natural isomorphisms, we show that the 2-morphism (6.38) is indeed a left $A$-module 2-morphism in $2\mathsf{Vec}_G^\omega$.

Let us first consider an object $M^g \in {}_A(2\mathsf{Vec}_G^\omega)$ defined by (6.35):

$$
M^g = A\square D_2^g = \bigoplus_{h \in H} \bigoplus_{a_h \in A_h} D_2^{hg}[a_h].
\tag{B.9}
$$

The left $A$-action on $M^g$ is defined by using the multiplication 1-morphism $m : A \square A \to A$ as follows:

$$A \square M = A \square A \square D_2^g \xrightarrow{m \square \mathrm{id}_{D_2^g}} A \square D_2^g = M^g. \tag{B.10}$$

We denote this 1-morphism by $m^g : A \square M^g \to M^g$. By definition, the underlying 1-morphism of $m^g$ is given by

$$D_2^{h_{ik}g}[a_{ik}] \begin{array}{c} \boxed{m^g} \\ D_2^{h_{jk}g}[a_{jk}] \\ D_2^{h_{ij}}[a_{ij}] \end{array} = \bigoplus_{\alpha_{ijk} \in \mathrm{Hom}_A(a_{ij} \otimes a_{jk}, a_{ik})} 1_{h_{ij} \cdot h_{jk}g}[\alpha_{ijk}], \tag{B.11}$$

where $h_{ij}, h_{jk}, h_{ik} \in H$, $a_{ij} \in A_{h_{ij}}$, $a_{jk} \in A_{h_{jk}}$, $a_{ik} \in A_{h_{ik}}$, and $1_{h_{ij} \cdot h_{jk}g}[\alpha_{ijk}]$ is a simple 1-morphism $1_{h_{ij} \cdot h_{jk}g}$ associated with a basis morphism $\alpha_{ijk}$. The summation on the right-hand side is taken over all basis morphisms. In the above equation, subobjects of $M^g$ and $A$ are represented by the red and blue surfaces, respectively. Since $M^g$ is a left $A$-module in $2\mathsf{Vec}_G^\omega$, it is equipped with a natural isomorphism

$$l_{a,b,n} : (a \otimes b) \rhd n \to a \rhd (b \rhd n), \qquad \forall a, b \in A, \ \forall n \in M^g, \tag{B.12}$$

which satisfies the coherence condition (6.23). As a 2-morphism in $2\mathsf{Vec}_G^\omega$, the natural isomorphism $l$ is given component-wise as

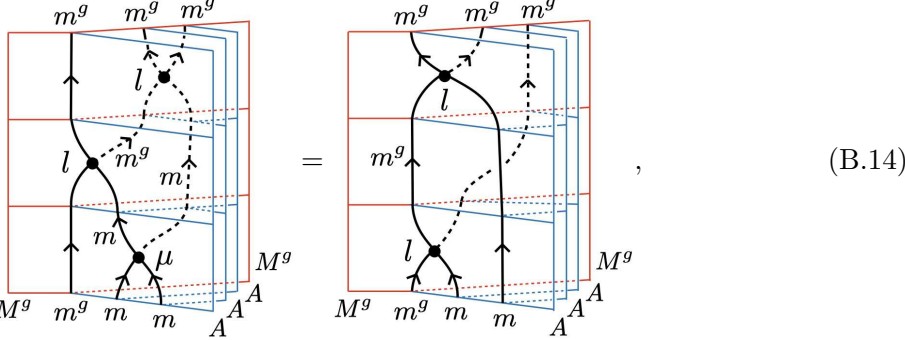

where $\Omega_{ijkl}^{g;h}$ is defined by (6.32), and the white dot on the right-hand side is the basis 2-morphism of $2\mathsf{Vec}_G^\omega$. The coherence condition (6.23) reduces to

$$\tag{B.14}$$

which follows from the twisted pentagon equation (6.13).

We next consider the 1-morphisms of $_A(2\mathsf{Vec}_G^\omega)$. Due to the equivalence (6.27), simple 1-morphisms from $M^{g_j} \lhd D_2^{g_{jk}^h}$ to $M^{g_k}$ are labeled by simple objects of $A_{h_{jk}}^{\mathrm{op}}$, where $g_{jk}^h :=$ $g_j^{-1} h_{jk} g_k$. The simple 1-morphism labeled by $a_{jk} \in A_{h_{jk}}^{\mathrm{op}}$ is denoted by

$$f_{a_{jk}} : M^{g_j} \lhd D_2^{g_{jk}^h} \to M^{g_k}, \tag{B.15}$$

which is given component-wise as in (6.36):

$$D_2^{h_{ik}g_k}[a_{ik}] \boxed{\begin{array}{c} a_{jk} \end{array}} \begin{array}{c} D_2^{g_{jk}^h} \\ D_2^{h_{ij}g_j}[a_{ij}] \end{array} = \bigoplus_{\alpha_{ijk}^{\mathrm{op}} \in \mathrm{Hom}_{A^{\mathrm{op}}}(a_{jk} \otimes^{\mathrm{op}} \overline{a_{ij}}, a_{jk})} 1_{h_{ij}g_j \cdot g_{jk}^h}[\alpha_{ijk}^{\mathrm{op}}]. \tag{B.16}$$

On the left-hand side, $f_{a_{jk}}$ is simply written as $a_{jk}$. Since $f_{a_{jk}}$ is a left $A$-module 1-morphism in $2\mathsf{Vec}_G^\omega$, it is equipped with a natural isomorphism

$$\xi_{b,n} : f_{a_{jk}}(b \rhd n) \to b \rhd f_{a_{jk}}(n), \qquad \forall b \in A, \ \forall n \in M^{g_j} \lhd D_2^{g_{jk}^h}, \tag{B.17}$$

which satisfies the coherence condition (5.31). As a 2-morphism in $2\mathsf{Vec}_G^\omega$, the natural isomorphism $\xi$ is given component-wise as

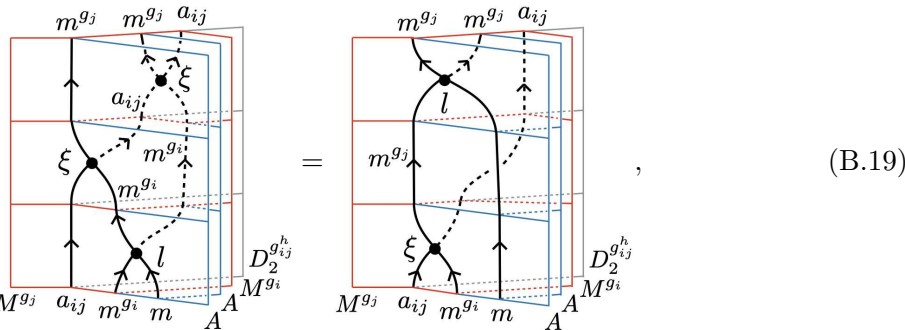

$$= \Omega_{ijkl}^{g;h}\Big|_{g_i=g_j=e} F_{ijkl}^{a;\alpha} \tag{B.18}$$

The coherence condition (5.31) reduces to

$$= \tag{B.19}$$

which follows from the twisted pentagon equation (6.13).

Finally, we consider the 2-morphisms of $2\mathsf{Vec}_G^\omega$. The equivalence (6.27) implies that the 2-morphisms from $F_{a_{kl}} \circ (F_{a_{jk}} \lhd 1_{g_{kl}^h})$ to $F_{a_{jl}} \circ (1_{M^{g_j}} \lhd 1_{g_{jk}^h \cdot g_{kl}^h})$ are labeled by elements of $\mathrm{Hom}_{A^{\mathrm{op}}}(a_{kl} \otimes^{\mathrm{op}} a_{jk}, a_{jl})$, see figure 12. The 2-morphisms labeled by a basis morphism $\alpha_{jkl}^{\mathrm{op}}$ is

given component-wise as in (6.38):

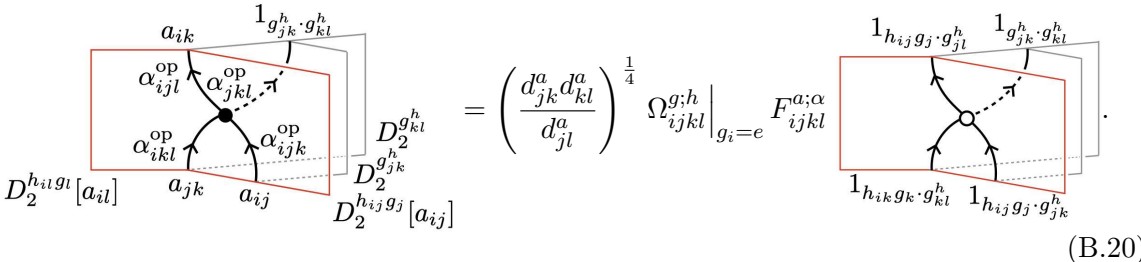

$$(\text{B.20})$$

The coherence condition (5.32) reduces to

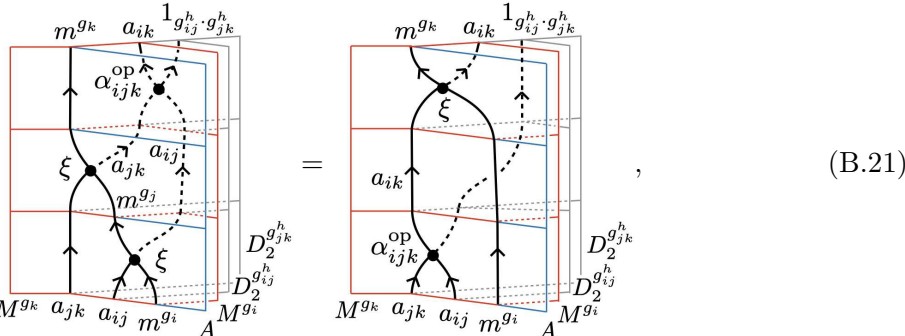

$$(\text{B.21})$$

which follows from the twisted pentagon equation (6.5). We note that the overall factor $(d_{ij}^a d_{jk}^a / d_{ik}^a)^{\frac{1}{4}}$ in (B.20) is just for the sake of normalization, which does not play any role in the coherence condition (B.21).

## C  Derivation of (5.97)

In this appendix, we show (5.97), i.e.,

$$\overline{\mathsf{D}}_A \mathsf{D}_A = \sum_{h \in H} \text{rank}(A_h) U_h, \qquad (\text{C.1})$$

where $\mathsf{D}_A$ and $\overline{\mathsf{D}}_A$ are the generalized gauging operators for the generalized gauging of $2\mathsf{Vec}_G$ symmetry. Using the explicit form of $\mathsf{D}_A$ and $\overline{\mathsf{D}}_A$ (see (5.89) and (5.94)), we can compute the

action of $\overline{\mathsf{D}}_A \mathsf{D}_A$ as follows:

$$
\overline{\mathsf{D}}_A \mathsf{D}_A \left| \{h_i g_i\} \right\rangle = \sum_{\{a_i \in A_{h_i}^{\text{rev}}\}} \sum_{\{a_{ij}\}} \sum_{\{\alpha_{ij}\}} \sum_{\{\alpha_{ijk}\}} \sum_{\{h_i' \in H\}} \sum_{\{a_i' \in A_{h_i'}^{\text{rev}}\}} \sum_{\{\alpha_{ij}'\}} \prod_i \frac{1}{\mathcal{D}_{A_{h_i}}} \prod_{[ijk]} \sqrt{\frac{\dim(a_{ij})\dim(a_{jk})}{\dim(a_{ik})}}
$$

$$
\prod_{[ijk]} (\overline{F}_{a_k}^{a_i a_{ij} a_{jk}})^{s_{ijk}}_{(a_j;\alpha_{ij},\alpha_{jk}),(a_{ik};\alpha_{ik},\alpha_{ijk})} (F_{a_k'}^{a_i' a_{ij} a_{jk}})^{s_{ijk}}_{(a_j';\alpha_{ij}',\alpha_{jk}'),(a_{ik};\alpha_{ik}',\alpha_{ijk})} \left| \{h_i' g_i\} \right\rangle
$$

$$
= \sum_{\{a_i \in A_{h_i}^{\text{rev}}\}} \sum_{\{a_{ij}\}} \sum_{\{\alpha_{ij}\}} \sum_{\{h_i' \in H\}} \sum_{\{a_i' \in A_{h_i'}^{\text{rev}}\}} \sum_{\{\alpha_{ij}'\}} \prod_i \frac{1}{\mathcal{D}_{A_{h_i}}} \prod_{[ijk]} \left( \sum_{a_{ijk}} \sum_{\alpha_{ii}} \sum_{\alpha_{jj}} \sum_{\alpha_{kk}} \right.
$$

$$
(\overline{F}_{a_j'}^{a_i' \overline{a_i} a_j})^{s_{ijk}}_{(a_{ijk};\alpha_{ii},\alpha_{jj}),(a_{ij};\alpha_{ij}',\alpha_{ij})} (\overline{F}_{a_k'}^{a_j' \overline{a_j} a_k})^{s_{ijk}}_{(a_{ijk};\alpha_{jj},\alpha_{kk}),(a_{jk};\alpha_{jk}',\alpha_{jk})}
$$

$$
\left. (F_{a_k'}^{a_i' \overline{a_i} a_k})^{s_{ijk}}_{(a_{ijk};\alpha_{ii},\alpha_{kk}),(a_{ik};\alpha_{ik}',\alpha_{ik})} \sqrt{\frac{\dim(a_j)\dim(a_j')}{\dim(a_{ijk})}} \right) \left| \{h_i' g_i\} \right\rangle
$$

$$
= \sum_{a \in A} \sum_{\{a_i \in A_{h_i}^{\text{rev}}\}} \sum_{\{h_i' \in H\}} \sum_{\{a_i' \in A_{h_i'}^{\text{rev}}\}} \sum_{\{\alpha_{ii} \in \text{Hom}_A(a_i' \otimes \overline{a_i}, a)\}} \prod_i \frac{1}{\mathcal{D}_{A_{h_i}}} \prod_i \frac{\dim(a_i)\dim(a_i')}{\dim(a)} \left| \{h_i' g_i\} \right\rangle
$$

$$
= \sum_{h \in H} \sum_{a_h \in A_h} \sum_{\{a_i \in A_{h_i}^{\text{rev}}\}} \prod_i \frac{\dim(a_i)^2}{\mathcal{D}_{A_{h_i}}} \left| \{hh_i g_i\} \right\rangle
$$

$$
= \sum_{h \in H} \text{rank}(A_h) U_h \left| \{h_i g_i\} \right\rangle .
$$

$$(C.2)$$

Here, the first equality follows from the definitions of $\mathsf{D}_A$ and $\overline{\mathsf{D}}_A$. In the second equality, we used the identity

$$
\sum_{\alpha_{ijk}} (\overline{F}_{a_k}^{a_i a_{ij} a_{jk}})_{(a_j;\alpha_{ij},\alpha_{jk}),(a_{ik};\alpha_{ik},\alpha_{ijk})} (F_{a_k'}^{a_i' a_{ij} a_{jk}})_{(a_j';\alpha_{ij}',\alpha_{jk}'),(a_{ik};\alpha_{ik}',\alpha_{ijk})}
$$

$$
= \sum_{a_{ijk}} \sum_{\alpha_{ii},\alpha_{jj},\alpha_{kk}} (\overline{F}_{a_j'}^{a_i' \overline{a_i} a_j})_{(a_{ijk};\alpha_{ii},\alpha_{jj}),(a_{ij};\alpha_{ij}',\alpha_{ij})} (\overline{F}_{a_k'}^{a_j' \overline{a_j} a_k})_{(a_{ijk};\alpha_{jj},\alpha_{kk}),(a_{jk};\alpha_{jk}',\alpha_{jk})} \qquad (C.3)
$$

$$
(F_{a_k'}^{a_i' \overline{a_i} a_k})_{(a_{ijk};\alpha_{ii},\alpha_{kk}),(a_{ik};\alpha_{ik}',\alpha_{ik})} \sqrt{\frac{\dim(a_{ik})}{\dim(a_{ij})\dim(a_{jk})}} \sqrt{\frac{\dim(a_j)\dim(a_j')}{\dim(a_{ijk})}},
$$

which can be verified by evaluating the following diagram in two different ways:

$$(C.4)$$

In the third equality of (C.2), we used the following identity for all edges $[ij] \in E$:

$$
\sum_{a_{ij}} \sum_{\alpha_{ij}} \sum_{\alpha_{ij}'} (F_{a_j'}^{a_i' \overline{a_i} a_j})^{s_{ijk}}_{(a_{ijk};\alpha_{ii},\alpha_{jj}),(a_{ij};\alpha_{ij}',\alpha_{ij})} (F_{a_j'}^{a_i' \overline{a_i} a_j})^{s_{ijk}}_{(b_{ijk};\beta_{ii},\beta_{jj}),(a_{ij};\alpha_{ij}',\alpha_{ij})} = \delta_{a_{ijk},b_{ijk}} \delta_{\alpha_{ii},\beta_{ii}} \delta_{\alpha_{jj},\beta_{jj}}.
$$

In the fourth equality of (C.2), we used the following identity for all plaquettes $i \in P$:

$$\sum_{\alpha_{ii} \in \mathrm{Hom}_A(a_i' \otimes \overline{a_i}, a)} \dim(a_i') = \delta_{h_i' h_i^{-1}, h} \dim(a_i) \dim(a), \tag{C.5}$$

where $a_i \in A_{h_i}$, $a_i' \in A_{h_i'}$, and $a \in A_h$. In the last equality of (C.2), we used the definition of $\mathcal{D}_{A_{h_i}}$, i.e.,

$$\mathcal{D}_{A_{h_i}} = \sum_{a_i \in A_{h_i}} \dim(a_i)^2. \tag{C.6}$$

Equation (C.2) shows that $\overline{\mathsf{D}}_A \mathsf{D}_A$ is given by (5.97).

# D   SymTFT Perspective on Generalized Gauging

For the 3+1d SymTFT $\mathcal{Z}(2\mathsf{Vec}_G)$, we denote by $\mathfrak{B}_A$ the topological boundary condition corresponding to a $2\mathsf{Vec}_G$-module 2-category $_A(2\mathsf{Vec}_G)$, where $A$ is a $G$-graded fusion category. In this appendix, we argue that $\mathfrak{B}_A$ is obtained by stacking a 3d $G$-symmetric TFT on the Dirichlet boundary $\mathfrak{B}_{\mathrm{Dir}}$ and gauging the diagonal subgroup $G^{\mathrm{diag}}$. Namely, we argue that the following equation holds:

$$\mathfrak{B}_A = (\mathfrak{B}_{\mathrm{Dir}} \boxtimes \mathfrak{T}^G_{A^{\mathrm{rev}}})/G^{\mathrm{diag}}. \tag{D.1}$$

Here, $\mathfrak{T}^G_{A^{\mathrm{rev}}}$ is a $G$-symmetric TFT constructed from the reverse category $A^{\mathrm{rev}}$. See section D.3 for more details of this TFT. Throughout this appendix, we suppose that $G$ is non-anomalous and $A$ is faithfully graded by a subgroup $H \subset G$. The generalization to the case of anomalous $G$ is straightforward, see [20, Section 3].

Before moving on, we note that the stacked 3d TFT $\mathfrak{T}^G_{A^{\mathrm{rev}}}$ is non-chiral. In general, one could also consider stacking a 3d chiral TFT with $G$ symmetry and gauging the diagonal subgroup $G^{\mathrm{diag}}$ as discussed in [1]. However, such an operation cannot be described as the generalized gauging by a separable algebra $A \in 2\mathsf{Vec}_G$ and is thus beyond the scope of this manuscript.

## D.1   Topological Boundaries of SymTFT $\mathcal{Z}(2\mathsf{Vec}_G)$

The defining property of the topological boundary $\mathfrak{B}_A$ is that the topological interfaces between $\mathfrak{B}_A$ and $\mathfrak{B}_{\mathrm{Dir}}$ form a right $2\mathsf{Vec}_G$-module 2-category $_A(2\mathsf{Vec}_G)$, see figure 17. In particular, two-dimensional interfaces between $\mathfrak{B}_A$ and $\mathfrak{B}_{\mathrm{Dir}}$ are labeled by objects of $_A(2\mathsf{Vec}_G)$. Similarly, one-dimensional interfaces between two-dimensional interfaces are labeled by 1-morphisms of $_A(2\mathsf{Vec}_G)$, and zero-dimensional interfaces between one-dimensional interfaces

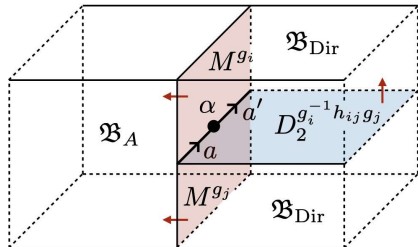

Figure 17: An interface between topological boundaries $\mathfrak{B}_A$ and $\mathfrak{B}_{\mathrm{Dir}}$ of the 4d Dijkgraaf-Witten theory. In the above figure (showing only the 3d boundary), $M^{g_i}$ and $M^{g_j}$ are objects of $_A(2\mathsf{Vec}_G)$, $a, a' : M^{g_i} \lhd D_2^{g_i^{-1}h_{ij}g_j} \to M^{g_j}$ are 1-morphisms of $_A(2\mathsf{Vec}_G)$, and $\alpha : a \Rightarrow a'$ is a 2-morphism of $_A(2\mathsf{Vec}_G)$.

are labeled by 2-morphisms of $_A(2\mathsf{Vec}_G)$. The right $2\mathsf{Vec}_G$-module action on $_A(2\mathsf{Vec}_G)$ is defined by the fusion of an interface and a topological defect supported on $\mathfrak{B}_{\mathrm{Dir}}$.

As discussed in section 5.1.2, the connected components of $_A(2\mathsf{Vec}_G)$ are labeled by right $H$-cosets in $G$. The representative of each connected component can be chosen to be $M^g = A \square D_2^g$, where $g \in S_{H \backslash G}$.[94] Therefore, the connected components of two-dimensional topological interfaces between $\mathfrak{B}_A$ and $\mathfrak{B}_{\mathrm{Dir}}$ are labeled by $\{M^g \mid g \in S_{H \backslash G}\}$. Similarly, one-dimensional and zero-dimensional topological interfaces between $M^{g_i} \lhd D_2^{g_i^{-1}h_{ij}g_j}$ and $M^{g_j}$ are labeled by objects and morphisms of the 1-category

$$\mathrm{Hom}_{A(2\mathsf{Vec}_G)}\left(M^{g_i} \lhd D_2^{g_i^{-1}h_{ij}g_j}, M^{g_j}\right) \cong A_{h_{ij}}^{\mathrm{op}}. \tag{D.2}$$

The above is an equivalence of $(A_e^{\mathrm{op}}, A_e^{\mathrm{op}})$-bimodule categories. This means that topological lines at the junction of $M^{g_i}$, $M^{g_j}$, and $D_2^{g_i^{-1}h_{ij}g_j}$ form an $(A_e^{\mathrm{op}}, A_e^{\mathrm{op}})$-bimodule category $A_{h_{ij}}^{\mathrm{op}}$. In particular, topological lines on a single interface $M^g$ form a fusion category $A_e^{\mathrm{op}}$.

In what follows, we will see that the topological boundary obtained by stacking and gauging has the same property as above, which implies eq. (D.1).

## D.2   $H$-symmetric TFT $\mathfrak{T}_{A^{\mathrm{rev}}}^H$

We first define a 3d $H$-symmetric TFT $\mathfrak{T}_{A^{\mathrm{rev}}}^H$, which is the building block of the $G$-symmetric TFT $\mathfrak{T}_{A^{\mathrm{rev}}}^G$. The 3d TFT $\mathfrak{T}_{A^{\mathrm{rev}}}^H$ is defined as the $H$-symmetric non-chiral TFT whose topological boundary conditions form a left $2\mathsf{Vec}_H$-module 2-category $(2\mathsf{Vec}_H)_{A^{\mathrm{rev}}}$. Due to the bulk-boundary relation, this 2-category of topological boundary conditions should uniquely determine the $H$-symmetric TFT $\mathfrak{T}_{A^{\mathrm{rev}}}^H$.

Let us describe the 2-category of topological boundary conditions in more detail. First of all, the indecomposable topological boundaries of $\mathfrak{T}_{A^{\mathrm{rev}}}^H$ are labeled by simple objects of

---

[94]We recall that $S_{H \backslash G}$ is the set of representatives of right $H$-cosets in $G$.

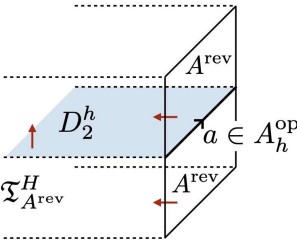

Figure 18: A topological boundary of the 3d TFT $\mathfrak{T}_{A^{\mathrm{rev}}}^{H}$. Boundary topological lines attached to the bulk surface $D_2^h$ form an $(A_e^{\mathrm{op}}, A_e^{\mathrm{op}})$-bimodule category $A_h^{\mathrm{op}}$.

$(2\mathsf{Vec}_H)_{A^{\mathrm{rev}}}$. Since $A^{\mathrm{rev}}$ is faithfully graded by $H$, all simple objects of $(2\mathsf{Vec}_H)_{A^{\mathrm{rev}}}$ are connected to each other.[95] The representative of the unique connected component can be chosen to be $A^{\mathrm{rev}}$. Similarly, the topological lines on the boundary and topological interfaces between them are labeled by 1-morphisms and 2-morphisms of $(2\mathsf{Vec}_H)_{A^{\mathrm{rev}}}$. In particular, the boundary topological lines attached to the bulk symmetry defect $D_2^h$ form a 1-category

$$\mathrm{Hom}_{(2\mathsf{Vec}_H)_{A^{\mathrm{rev}}}}(A^{\mathrm{rev}}, D_2^h \rhd A^{\mathrm{rev}}) \cong A_{h^{-1}}^{\mathrm{rev}} \cong A_h^{\mathrm{op}}. \tag{D.3}$$

See figure 18 for a schematic picture. The above is an equivalence of $(A_e^{\mathrm{op}}, A_e^{\mathrm{op}})$-bimodule categories. We note that boundary topological lines attached to no bulk surfaces form a fusion category $A_e^{\mathrm{op}} \cong A_e^{\mathrm{rev}}$

## D.3 $G$-symmetric TFT $\mathfrak{T}_{A^{\mathrm{rev}}}^{G}$

We now define the 3d $G$-symmetric TFT $\mathfrak{T}_{A^{\mathrm{rev}}}^{G}$ as

$$\mathfrak{T}_{A^{\mathrm{rev}}}^{G} = \bigoplus_{l_i \in S_{G/H}} \mathfrak{T}_{l_i A^{\mathrm{rev}}}^{l_i H l_i^{-1}}, \tag{D.4}$$

where $S_{G/H}$ is the set of representatives of left $H$-cosets in $G$,[96] and $\mathfrak{T}_{l_i A^{\mathrm{rev}}}^{l_i H l_i^{-1}}$ is the $l_i H l_i^{-1}$-symmetric TFT constructed from the following $l_i H l_i^{-1}$-graded fusion category:

$$l_i A^{\mathrm{rev}} := \bigoplus_{x \in l_i H l_i^{-1}} A_{l_i^{-1} x l_i}^{\mathrm{rev}}. \tag{D.5}$$

The action of the $G$-symmetry on $\mathfrak{T}_{A^{\mathrm{rev}}}^{G}$ permutes the direct sum components on the right-hand side, while each component $\mathfrak{T}_{l_i A^{\mathrm{rev}}}^{l_i H l_i^{-1}}$ is stabilized by its symmetry group $l_i H l_i^{-1}$. In other words, the 3d TFT $\mathfrak{T}_{A^{\mathrm{rev}}}^{G}$ spontaneously breaks the $G$ symmetry down to $H$, and each vacuum

---

[95]In general, the connected components of $(2\mathsf{Vec}_G)_{A^{\mathrm{rev}}}$ are in one-to-one correspondence with right $H$-cosets in $G$.

[96]We note that $S_{G/H}$ is not necessarily the same as $S_{H\backslash G}$ in general.

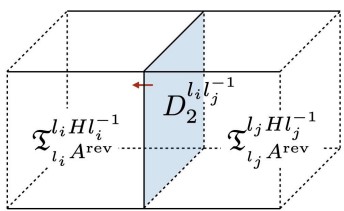

Figure 19: A topological interface $D_2^{l_i l_j^{-1}}$ between $\mathfrak{T}_{l_i A^{\mathrm{rev}}}^{l_i H l_i^{-1}}$ and $\mathfrak{T}_{l_j A^{\mathrm{rev}}}^{l_j H l_j^{-1}}$.

labeled by $l_i \in S_{G/H}$ realizes an $l_i H l_i^{-1}$-symmetric TFT $\mathfrak{T}_{l_i A^{\mathrm{rev}}}^{l_i H l_i^{-1}}$. The vacuum $|\Omega; l_i\rangle$ of $\mathfrak{T}_{l_i A^{\mathrm{rev}}}^{l_i H l_i^{-1}}$ is obtained by applying the symmetry operator $U_{l_i l_j^{-1}}$ to the vacuum $|\Omega; l_j\rangle$ of $\mathfrak{T}_{l_j A^{\mathrm{rev}}}^{l_j H l_j^{-1}}$, i.e.,

$$|\Omega; l_i\rangle = U_{l_i l_j^{-1}} |\Omega; l_j\rangle. \tag{D.6}$$

Equivalently, the topological surface $D_2^{l_i l_j^{-1}} \in 2\mathsf{Vec}_G$ becomes a topological interface between the 3d TFTs $\mathfrak{T}_{l_i A^{\mathrm{rev}}}^{l_i H l_i^{-1}}$ and $\mathfrak{T}_{l_j A^{\mathrm{rev}}}^{l_j H l_j^{-1}}$ as shown in figure 19.

The direct sum decomposition (D.4) implies that the 2-category of topological boundary conditions of $\mathfrak{T}_{A^{\mathrm{rev}}}^G$ is given by

$$\mathcal{M}_{\mathfrak{T}_{A^{\mathrm{rev}}}^G} = \bigoplus_{l_i \in S_{G/H}} \mathcal{M}_{\mathfrak{T}_{l_i A^{\mathrm{rev}}}^{l_i H l_i^{-1}}}, \tag{D.7}$$

where $\mathcal{M}_{\mathfrak{T}_{l_i A^{\mathrm{rev}}}^{l_i H l_i^{-1}}}$ is the 2-category of topological boundary conditions of $\mathfrak{T}_{l_i A^{\mathrm{rev}}}^{l_i H l_i^{-1}}$. We note that $\mathcal{M}_{\mathfrak{T}_{A^{\mathrm{rev}}}^G}$ is naturally equipped with a left $2\mathsf{Vec}_G$-module structure, which originates from the $2\mathsf{Vec}_{l_i H l_i^{-1}}$-module structure on each component. Since each direct summand in eq. (D.7) is connected, the connected components of $\mathcal{M}_{\mathfrak{T}_{A^{\mathrm{rev}}}^G}$ are in one-to-one correspondence with (the representatives of) left $H$-cosets in $G$. We choose $_{l_i} A^{\mathrm{rev}}$ as a representative of each connected component. Then, the boundary topological lines attached to the bulk symmetry defects form a category

$$\mathrm{Hom}_{\mathcal{M}_{\mathfrak{T}_{A^{\mathrm{rev}}}^G}} \left( _{l_i} A^{\mathrm{rev}}, \; D_2^{l_i h_{ij} l_j^{-1}} \vartriangleright \; _{l_j} A^{\mathrm{rev}} \right) \cong A_{h_{ij}}^{\mathrm{op}}, \tag{D.8}$$

where $\vartriangleright$ denotes the left $2\mathsf{Vec}_G$ action on $\mathcal{M}_{\mathfrak{T}_{A^{\mathrm{rev}}}^G}$. See figure 20 for a schematic picture. The above is an equivalence of $(A_e^{\mathrm{op}}, A_e^{\mathrm{op}})$-bimodule categories.

The equivalence (D.8) implies

$$\mathcal{M}_{\mathfrak{T}_{A^{\mathrm{rev}}}^G} \cong (2\mathsf{Vec}_G)_{A^{\mathrm{rev}}} \tag{D.9}$$

as left $2\mathsf{Vec}_G$-module 2-categories. Indeed, if we choose the representative of each connected component of $(2\mathsf{Vec}_G)_{A^{\mathrm{rev}}}$ to be $N^{l_i} := D_2^{l_i} \square A^{\mathrm{rev}}$, we have an equivalence of $(A_e^{\mathrm{op}}, A_e^{\mathrm{op}})$-bimodule categories

$$\mathrm{Hom}_{(2\mathsf{Vec}_G)_{A^{\mathrm{rev}}}} (N^{l_i}, D_2^{l_i h_{ij} l_j^{-1}} \vartriangleright N^{l_j}) \cong A_{h_{ij}}^{\mathrm{op}}, \tag{D.10}$$

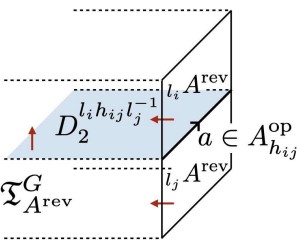

Figure 20: A topological bondary of the 3d TFT $\mathfrak{T}^G_{A^{\mathrm{rev}}}$. Boundary topological lines attached to the bulk surface $D_2^{l_i h_{ij} l_j^{-1}}$ form an $(A_e^{\mathrm{op}}, A_e^{\mathrm{op}})$-bimodule category $A_{h_{ij}}^{\mathrm{op}}$.

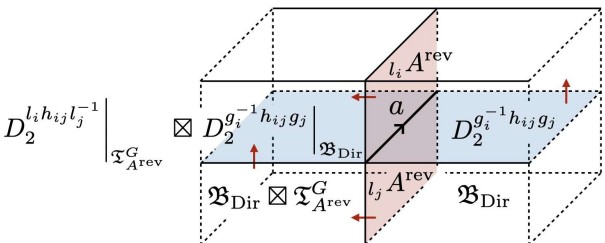

Figure 21: A topological interface between $\mathfrak{T}^G_{A^{\mathrm{rev}}} \boxtimes \mathfrak{B}_{\mathrm{Dir}}$ and $\mathfrak{B}_{\mathrm{Dir}}$. The topological surface on the left side of the interface is the stacking of $D_2^{l_i h_{ij} l_j^{-1}}$ on $\mathfrak{T}^G_A$ and $D_2^{g_i^{-1} h_{ij} g_j}$ on $\mathfrak{B}_{\mathrm{Dir}}$, where $g_i \in S_{H \backslash G}$, $l_i \in S_{G/H}$, and $h_{ij} \in H$. The topological line $a$ at the intersection is labeled by an object of $A_{h_{ij}}^{\mathrm{op}}$. The figure only shows the 3d boundary of the 4d TFT $\mathcal{Z}(2\mathsf{Vec}_G)$.

which agrees with eq. (D.8) upon indentifying $_{l_i} A^{\mathrm{rev}} \in \mathcal{M}_{\mathfrak{T}^G_{A^{\mathrm{rev}}}}$ and $N^{l_i} \in (2\mathsf{Vec}_G)_{A^{\mathrm{rev}}}$.

## D.4    Stacking and Gauging

Let us now show the equality

$$\mathfrak{B}_A = (\mathfrak{B}_{\mathrm{Dir}} \boxtimes \mathfrak{T}^G_{A^{\mathrm{rev}}})/G^{\mathrm{diag}}. \tag{D.11}$$

To this end, we first consider a topological interface between $\mathfrak{B}_{\mathrm{Dir}} \boxtimes \mathfrak{T}^G_{A^{\mathrm{rev}}}$ and $\mathfrak{B}_{\mathrm{Dir}}$ in the presence of a topological surface intersecting the interface, see figure 21. Here, the topological interface between $\mathfrak{B}_{\mathrm{Dir}} \boxtimes \mathfrak{T}^G_{A^{\mathrm{rev}}}$ and $\mathfrak{B}_{\mathrm{Dir}}$ is given by the stacking of the topological boundary of $\mathfrak{T}^G_{A^{\mathrm{rev}}}$ and the identity surface on $\mathfrak{B}_{\mathrm{Dir}}$. In particular, a topological line on the interface between $\mathfrak{B}_{\mathrm{Dir}} \boxtimes \mathfrak{T}^G_{A^{\mathrm{rev}}}$ and $\mathfrak{B}_{\mathrm{Dir}}$ comes from a topological line on the boundary of $\mathfrak{T}^G_{A^{\mathrm{rev}}}$ because $\mathfrak{B}_{\mathrm{Dir}}$ has no non-trivial topological lines. Therefore, the category of topological lines at the junction illustrated in figure 21 is given by

$$\mathrm{Hom}_{\mathcal{M}_{\mathfrak{T}^G_{A^{\mathrm{rev}}}}} \left( _{l_i} A^{\mathrm{rev}}, \ D_2^{l_i h_{ij} l_j^{-1}} \rhd \, _{l_j} A^{\mathrm{rev}} \right) \cong A_{h_{ij}}^{\mathrm{op}} \tag{D.12}$$

as an $(A_e^{\mathrm{op}}, A_e^{\mathrm{op}})$-bimodule category. If we choose the representative $l_i \in S_{G/H}$ to be

$$l_i = g_i^{-1}, \qquad g_i \in S_{H \backslash G}, \tag{D.13}$$

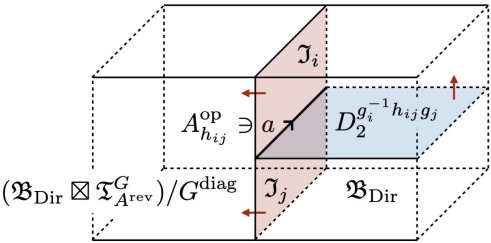

Figure 22: A topological interface between $(\mathfrak{B}_{\text{Dir}} \boxtimes \mathfrak{T}^G_{A^{\text{rev}}})/G^{\text{diag}}$ and $\mathfrak{B}_{\text{Dir}}$ is obtained by gauging the diagonal $G$ symmetry on the left side of the interface in figure 22. Here, $\mathfrak{I}_i$ is the topological interface that descends from the topological boundary $_{l_i}A^{\text{rev}}$ of $\mathfrak{T}^G_{A^{\text{rev}}}$. The figure again shows the 3d boundary of the 4d TFT.

the topological surface on the left side of the interface in figure 21 becomes

$$D_2^{g_i^{-1} h_{ij} g_j}\bigg|_{\mathfrak{B}_{\text{Dir}}} \boxtimes \ D_2^{g_i^{-1} h_{ij} g_j}\bigg|_{\mathfrak{T}^G_{A^{\text{rev}}}} \in 2\mathsf{Vec}_{G^{\text{diag}}}. \tag{D.14}$$

This surface becomes invisible if we gauge the diagonal symmetry $G^{\text{diag}}$ on $\mathfrak{B}_{\text{Dir}} \boxtimes \mathfrak{T}^G_{A^{\text{rev}}}$. Therefore, after the diagonal gauging, the topological surface $D_2^{g_i^{-1} h_{ij} g_j}$ on $\mathfrak{B}_{\text{Dir}}$ ends on the interface between $(\mathfrak{B}_{\text{Dir}} \boxtimes \mathfrak{T}^G_{A^{\text{rev}}})/G^{\text{diag}}$ and $\mathfrak{B}_{\text{Dir}}$, see figure 22. The category of topological lines at the junction in figure 22 is given by (D.12). Namely, if we denote the 2-category of topological interfaces between $(\mathfrak{B}_{\text{Dir}} \boxtimes \mathfrak{T}^G_{A^{\text{rev}}})/G^{\text{diag}}$ and $\mathfrak{B}_{\text{Dir}}$ by $\mathcal{I}_A$, we have

$$\text{Hom}_{\mathcal{I}_A}\big(\mathfrak{I}_i \lhd D_2^{g_i^{-1} h_{ij} g_j}, \mathfrak{I}_j\big) \cong A^{\text{op}}_{h_{ij}}, \tag{D.15}$$

where $\mathfrak{I}_i$ and $\mathfrak{I}_j$ are simple objects of $\mathcal{I}_A$. Comparing (D.15) with (D.2), we conclude that $\mathcal{I}_A$ is equivalent to $_A(2\mathsf{Vec}_G)$ as a right $2\mathsf{Vec}_G$-module 2-category. This implies (D.11).

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
