# Peer review of "(2+1)d Lattice Models and Tensor Networks for Gapped Phases with Categorical Symmetry"

_SciPost Physics_

## Round 1 · Referee Report · Anonymous (Referee 1) · 2025-10-22

Report

This manuscript addresses the problem of providing explicit lattice-model and tensor-network constructions for the non-chiral gapped phases in (2+1) dimensions that are characterized by so-called “all-boson type” fusion 2-category symmetries. The authors build on a previously developed continuum classification (via the Symmetry TFT / SymTFT framework) and then show how that classification can be realized microscopically via commuting-projector Hamiltonians and tensor network ground states. Specifically, one begins from a 0-form symmetry group which can have ’t Hooft anomaly, stacks with a TFT symmetric under a subgroup H, then gauges H to obtain the categorical symmetry in question; this input then dictates the data for the lattice model. The authors also present a number of concrete examples.

The topic is timely and of broad interest in condensed-matter, mathematical physics and high-energy theory alike. The notion of categorical (higher) symmetries is increasingly important for the classification of phases of matter, and the gap between abstract classification and microscopic (lattice/tensor-network) realisation remains a bottleneck. This work helps bridge that gap. The manuscript also proposes a systematic construction, rather than just isolated case studies.

Requested changes

  1. Given the technical depth and the likely breadth of the audience (condensed matter, mathematical physics, lattice quantum models), the authors give some simple examples early on in the summary sectio, which helps nonexperts follow the logic before diving into the full generality. But the presentation is a bit dense. It might help to provide a table or schematic summarising the key phases, symmetries in one place for reader convenience.

Recommendation

Publish (easily meets expectations and criteria for this Journal; among top 50%)

---

## Round 1 · Referee Report · Anonymous (Referee 3) · 2025-11-17

Strengths

The paper gives a very solid discussion on lattice realization of model that has a fusion 2-category symmetry.

Weaknesses

The paper involves numerous mathematical concepts from higher category theory, making it difficult to access for readers who are not familiar with the subject.

Report

The non-invertible symmetries of two-dimensional quantum field theories and lattice models are widely believed to be characterized by fusion $2$-categories. This paper offers a systematic two-dimensional lattice construction of quantum models exhibiting non-invertible symmetry for what the authors describe as “all-boson-type’’ fusion $2$-categories.

In my understanding, the “all-boson-type’’ condition makes the corresponding fusion $2$-category easier to control, since in this case it can be regarded as a boundary of $\mathcal{Z}(2\mathsf{Vec}_G^{\omega})$, which corresponds to a $(3+1)$-dimensional Dijkgraaf–Witten theory. Using the SymTFT framework, this provides a general way to understand the associated gapped SPT and SSB phases with the given fusion $2$-category symmetry.

A key difficulty lies in determining all topological boundary conditions. The authors propose using symmetry gauging to construct them, starting from the smooth boundary and then gauging a subgroup $H$ to obtain various topological boundary conditions. This approach is also commonly used in the field. They further introduce two major classes of topological boundary conditions: minimal and non-minimal.

The main contribution of this work lies in constructing the lattice model (based on a generalization of the fusion surface model), introducing the corresponding symmetry-gauging operation at the lattice level, and providing a tensor-network representation.

The results presented in this paper are interesting and constitute a significant contribution to the community. I therefore recommend its publication.

The following are some of my questions:

— In Eq. (4.1), where the basic setup of the fusion surface model is introduced, I found the labeling somewhat confusing since many labels are omitted. For instance, the authors use $\Gamma_i$, $\Gamma_{ij}$, and $\Gamma_{ijk}$, but additional labels such as $f$ and $\sigma$ also appear, which may be unclear to the reader. Moreover, the model involves three colors, yet the precise roles played by these colors are not explicitly explained (one need to go back the the fusion surface model paper to figure it out). For instance, are all faces of the same color labeled by the same object throughout the entire lattice? Why are these colors essential, and why are the edges connecting these colored faces to the white plaquettes of the honeycomb lattice labeled by $\Gamma_{ij}$, etc.?
A clearer presentation of these conventions would be helpful, as the current description leaves the construction of the Hilbert space somewhat opaque. Although the authors seem to assume familiarity with the fusion surface model, I believe a more explicit explanation is necessary, since this setup forms the foundation for the subsequent lattice construction developed in the paper.

— I did not find a clear discussion of the topological excitations of the model (although some remarks are provided in the paper, they still seem too vague to form a coherent picture of the excitation structure) or of the corresponding string operators. Is there a fundamental difficulty in developing the excitation theory for this construction?

— If I understand correctly, a topological boundary of the model corresponds to a module $2$-category over $2\mathsf{Vec}_G^{\omega}$ (equivalently, to a Lagrangian algebra in $\mathcal{Z}(2\mathsf{Vec}_G^{\omega})$). I have several questions about this correspondence:

(a) What algebraic feature distinguishes “minimal’’ from “non-minimal’’ boundaries? Concretely, does minimality mean that the module $2$-category $\mathcal{M}$ is indecomposable (i.e.\ cannot be written as a Deligne tensor product), or is a different criterion intended?

(b) Can the gauging operation starting from the smooth boundary produce \emph{all} possible topological boundaries? If so, is there a rigorous proof (or a reference) demonstrating that every Lagrangian algebra in $\mathcal{Z}(2\mathsf{Vec}_G^{\omega})$ arises from gauging a suitable subgroup of the smooth boundary?

(c) Given the algebraic data of a module $2$-category, is there a direct, systematic procedure to construct the corresponding lattice model (rather than obtaining it only indirectly via gauging)?

The following are some minor issues:

— The term “all boson type” is used in most places, but some places use “all-boson type”. Also, in the Abstract, “All boson type” should not be capitalized.

— P 25, bottom footnote: “an anadditional” should be “an additional”.

— P 43: “gappped”.

— P 52: “i..e”.

— P 55: “single vacua TFT” should be “single vacuum TFT”.

— P 61: “anologue” should be “analogue”.

— P 67: “particualr” should be “particular”.

— P 132: “becasue” should be “because”.

Recommendation

Ask for minor revision

---

## Round 1 · Referee Report · Anonymous (Referee 2) · 2025-11-17

Disclosure of Generative AI use

The referee discloses that the following generative AI tools have been used in the preparation of this report:

Use AI to smooth my sentences and correct typos.

Strengths

1- The paper gives a very general framework to study the 2+1d non-invertible symmetry and its gapped phases. The paper also provides concrete lattice realizations of these gapped phases. 2- The paper provides the generalized gauging on a lattice, which is useful to generate a large family of gapped phases. 3- The paper proposes a new phase -- Spontaneously Nonuniform Entangled Phase (SNEP) as a phase exhibiting spontaneous symmetry breaking in which distinct ground states possess inequivalent entanglement structures.

Weaknesses

1- The paper focuses on non-chiral gapped phases. This is also due to the commuting projector Hamiltonian formalism. 2- Lack of discussion of order parameters.

Report

As listed in the strenghts part, this paper provides a general framework to study the bosonic non-invertible non-chiral gapped phases on lattice, using the generalized gauging. The manuscript is self-contained and well-written. The manuscript meets the criteria for publication in SciPost Physics.

Recommendation

Publish (surpasses expectations and criteria for this Journal; among top 10%)

---

## Round 2 · Referee Report · Anonymous (Referee 3) · 2025-12-27

Report

The authors have satisfactorily addressed all the questions and concerns raised regarding the previous version of the manuscript. This revised version is considerably clearer in its presentation, and I therefore recommend its acceptance for publication.

Recommendation

Publish (easily meets expectations and criteria for this Journal; among top 50%)

---

## Round 2 · Author Response

We thank the referees for their in-depth reading and comments on our paper. Let us reply to each referee in turn and explain what changes we have implemented in the revised version.

Referee 1:

We have taken the suggestion on board and have added Table 1 in Section 2.3 to summarize the key phases that we discussed.

Referee 2:

No changes requested

Referee 3:

  1. Below (4.1), we have added explanations regarding the conventions.

  2. Regarding the topological excitations, we believe that it should be possible to write down the string operators as in the Levin-Wen model. However, computing the symmetry action (including the symmetry fractionalization) on the quasi-particle excitations may involve a technical difficulty because such a computation technique has not been well-established for general non-invertible symmetries as far as we are aware. As such, developing the excitation theory for our models may not be straightforward in practice. It would indeed be an interesting direction to explore in the future.

  3. (a) The minimal boundaries correspond to $2\text{Vec}_G^{\omega}$-module 2-categories of the form ${}_{\text{Vec}_H^{\nu}} (2\text{Vec}_G^{\omega})$, i.e., the 2-category of left $\text{Vec}_H^{\nu}$-modules in $2\text{Vec}_G^{\omega}$. The non-minimal boundaries correspond to $2\text{Vec}_G^{\omega}$-module 2-categories ${}_A (2\text{Vec}_G^{\omega})$ for more general $G$-graded $\omega$-twisted fusion categories $A$. In both cases, the module 2-categories are indecomposable. The relation between topological boundaries and module 2-categories is briefly mentioned in Section 2.5 (on page 58) and discussed in more detail in Appendix D.

(b) Indeed, all topological boundary conditions can be obtained by generalized gauging (meaning stacking with an $H$-symmetric TFT and gauging $H$). In fact, Lagrangian algebras of $\mathcal{Z}(2\text{Vec}_G^{\omega})$ are classified rigorously in arXiv:2411.13367, and the classification data are consistent with the data of the generalized gauging. A more physical proof can be found in Section 5 of arXiv:2408.05266 (see also the sequel arXiv:2502.20440). For topological boundaries labeled by module 2-categories (i.e., non-chiral topological boundaries), we have provided a detailed physical argument supporting this fact in Appendix D.

(c) We have described a systematic way to construct lattice models directly using the data of a module 2-category in Section 4.2 (for a general fusion 2-category), Section 5.2 (for $2\text{Vec}_G$), and Section 6.2 (for $2\text{Vec}_G^{\omega}$). A module 2-category provides a mathematical way of defining generalized gauging on the lattice. The generalized gauging defined via module 2-categories is equivalent to the one defined in terms of stacking and gauging, as argued in Section 4.2.

We thank the referee for finding all these typos. We have corrected them.

---

## Round 2 · List of Changes

1. We added Table 1 in Section 2.3.
  2. We expanded the explanations below (4.1).
  3. We fixed minor typos and added references.

---

## Editorial Decision

in_voting